# In-Context Universal Approximation, Compositional Generalization, and Algorithm Emulation

Jerry Yao-Chieh Hu [* 1]   Hong-Yu Chen [* 1]   Po-Chiao Lin [* 2]   Maojiang Su [1]   Han Liu [1 3]

## Abstract

We study in-context universal approximation and compositional generalization in frozen softmax Transformers as prompt-programmable computation. We prove in-context universality via in-context emulation: a fixed-weight Transformer emulates target computations specified by the prompt and hence approximates a broad class of continuous sequence-to-sequence functions. Building on this view, we establish one-pass and multi-pass composition theorems: prompts associated with simple "subprograms" let the same fixed Transformer execute their composition and thereby synthesize more complex programs on-the-fly. These results support a principled view of prompts as programs and fixed-weight Transformers as program interpreters. They also provide concrete mechanisms by which GPT-style models execute and assemble algorithms in context.[1]

## 1  Introduction

We study the *general-purpose* behavior of foundation models through in-context algorithm emulation and compositional generalization in frozen-weight softmax Transformers. We prove two main results: (i) an *in-context universal approximation* theorem for softmax Transformers, and (ii) an *in-context function composition* theorem showing prompt-level program assembly and execution of the composite program in the same frozen model. These results support a principled view of *prompts as programs* and *Transformers as program executors*. We provide a concrete mechanism for foundation models to both execute and assemble algorithms in-context.

Foundation models (large pretrained Transformers) show a striking generality: a single pretrained model does many tasks through prompt changes alone, with no gradient updates (Radford et al., 2019; Brown et al., 2020; Wei et al., 2022; Chowdhery et al., 2023). The community often calls this *In-Context Learning* (Brown et al., 2020; Min et al., 2022), but the term conflates weight-space learning with inference-time computation. Pretraining learns the model weights. Let $\theta^\star$ denote the frozen weights after pretraining. Inference runs the fixed maps defined by $\theta^\star$ on prompt inputs. Task specification therefore enters through the input context, not through task-specific weight updates. The prompt thus becomes the task interface. There is no explicit learning at all.

To understand better, we define Transformer ICL as prompt-conditioned computations and present a spectrum organized by where the task procedure resides. This spectrum captures much of the observed general-purpose behavior of foundation models (Figure 1):

- **Parametric recall** sits on one end. The prompt acts as a query. The relevant skill or fact resides in $\theta^\star$. Several statistical analyses formalize this regime as *task identification*: the model matches the prompt to a latent pretrained task and activates its predictor (Wies et al., 2023; Lin & Lee, 2024).

- **Amortized in-context inference** sits in the middle. The prompt supplies examples. The forward pass applies an implicit inference rule encoded in $\theta^\star$. Several analyses formalize this regime as *implicit inference*: the model treats the prompt as a dataset and implements an estimator (e.g., implicit Bayes, least squares, or gradient-style updates) (Xie et al., 2022; Garg et al., 2022; Akyürek et al., 2023; von Oswald et al., 2023; Bai et al., 2023; Li et al., 2023).

- **Prompt-programmed execution** sits on the other end. The prompt specifies a program: a procedure and a con-

---

[1]Center for Foundation Models and Generative AI & Department of Computer Science, Northwestern University & ; NSF–Simons AI Institute for the Sky (SkAI) [2]Department of Physics, National Taiwan University [3]Department of Statistics and Data Science, Northwestern University.  Correspondence to:  Jerry Yao-Cheih Hu <jhu@u.northwestern.edu>, Hong-Yu Chen <charlie.chen@u.northwestern.edu>, Po-Chiao Lin <r13b44015@ntu.edu.tw>, Maojiang Su <smj@u.northwestern.edu>, Han Liu <han-liu@northwestern.edu>.

[1]Please see arXiv for the full version.

text layout for inputs, tokens that store intermediate results, and outputs. Several programmability constructions formalize this regime as *prompt-as-program* (Giannou et al., 2023; Qiu et al., 2025; Hu et al., 2026b).

The ultimate goal is to explain the full spectrum of general-purpose behavior in Transformer foundation models. However, most existing ICL theory explains only the first two regimes. It explains how a frozen model $\theta^\star$ retrieves pre-trained knowledge (i.e., parametric recall) or applies a fixed inference rule to in-context examples (i.e., amortized inference). In both cases, the procedure template remains fixed in $\theta^\star$. The prompt changes only the queried content or the in-context data. So they explain generality only within a task family. They do not explain prompt changes that specify new procedures under one fixed $\theta^\star$. They also do not explain prompt-level modular assembly of such procedures. Consequently, they leave the pure *prompt-programmed* end under-specified.[2]

To combat this, recent work makes progress toward the prompt-programmed regime and formalizes prompt-as-program behavior (Wang et al., 2023; Giannou et al., 2023; Hu et al., 2025a; Qiu et al., 2025). However, most of them takes an existential form and leave the constructions implicit[3]. They do not yield a constructive, reusable prompt compiler or modular composition guarantees for one frozen softmax Transformer. Moreover, a shared limitation cuts across these lines: many results replace softmax attention with simplified surrogates (e.g., linear, ReLU or hardmax attention) (von Oswald et al., 2023; Bai et al., 2023; Ahn et al., 2023; Qiu et al., 2025). Collectively, these gaps leave open systematic prompt programming of a frozen softmax Transformer for diverse computations and their modular assembly. We target this gap in this paper.

We now define our scope and problem setup. We study in-context algorithm emulation and prompt-level function composition in softmax Transformer models. An algorithm induces one function or a composition of functions. We work with target functions $f \in \mathcal{F}_L$, where $\mathcal{F}_L$ contains $L$-Lipschitz continuous sequence-to-sequence maps on a compact domain $\mathcal{C}$. Let $T_{\theta^\star}$ denote the sequence-to-sequence

map computed by the frozen Transformer.[4]

- We formalize *in-context algorithm emulation* through function realization. We study prompts $(P_f, X)$ that induce the frozen model $T_{\theta^\star}$ to implement a target function $f(X)$: $T_{\theta^\star}(P_f, X) \simeq f(X)$. This function-level view gives a more general formulation of algorithm emulation (Hu et al., 2026b). Moreover, it also implies in-context universal approximation: after fixing the target class and accuracy, one frozen Transformer approximates any $f \in \mathcal{F}_L$ by changing only prompt $P_f$.

- We formalize *in-context compositional generalization* as a prompt-level construction that maps instructions for subfunctions (e.g., $P_f, P_g$) to an instruction for the composite function (e.g., $P_{g \circ f}$), so that the same frozen model executes the composite program (algoritm + data) with controlled error. This serve as the fundamental mechanism for compositional generalization with fixed-weight Transformers.

**Contributions.** Our contributions are three-fold:

- **In-Context Universal Approximation.** We prove that one fixed softmax Transformer approximates every target function in a broad Lipschitz sequence-to-sequence class by changing only the prompt. Specifically, we show that for any $L$-Lipschitz $f$ on a compact domain, there exists a specific prompt $P_f$ that induces the frozen model to approximate $f$ to arbitrary precision (Theorem 3.1 and Theorem 3.2). This establishes prompt-programmable universal approximation with one fixed softmax Transformer.

- **General Prompt-Programmable Algorithm Emulation.** We develop an algorithm-emulation framework in which the prompt encodes the target computation and a frozen Transformer executes the encoded instruction. This grounds the view of prompts as programs and fixed-weight Transformers as program interpreters. Concretely, our framework provides explicit prompt encodings for feed-forward and multi-head attention layers, allowing one fixed softmax Transformer to execute these prompt-specified computations in context (Lemmas 3.1 and 3.4).

- **In-Context Function Composition.** We establish in-context function composition mechanisms for assembling subroutine prompts to execute composite functions in context. This extends the view of prompts as programs from single computations to program assembly. We analyze two regimes: *one-pass composition* (Theorem 4.1), which

---

[2] A satisfactory theory of this prompt-programmed regime should meet three requirements:

- It should use standard softmax attention.

- It should use one fixed set of model weights across tasks.

- It should provide a constructive and reusable prompt interface for procedure programming and composition. Namely, an effective way to "instruct" the model to execute the desired procedures (i.e., tasks or algorithms.)

Yet, existing theory satisfies only subsets as discussed above (please also see Section A for more detailed discussion.)

[3] The only exception is (Giannou et al., 2023). It gives an explicit construction and establishes a Turing-completeness-style programmability result.

[4] To clarify more, a function specifies an input-output map. An algorithm specifies a procedure for computing it. A program encodes the procedure for a fixed executor. In our setting, the prompt is the full input sequence. It contains an instruction block $P_f$ and a data block $X$. The statement $T_{\theta^\star}(P_f, X) \approx f(X)$ means that the frozen Transformer executes the instruction $P_f$ on data $X$. Please see Section C for linear regression example.

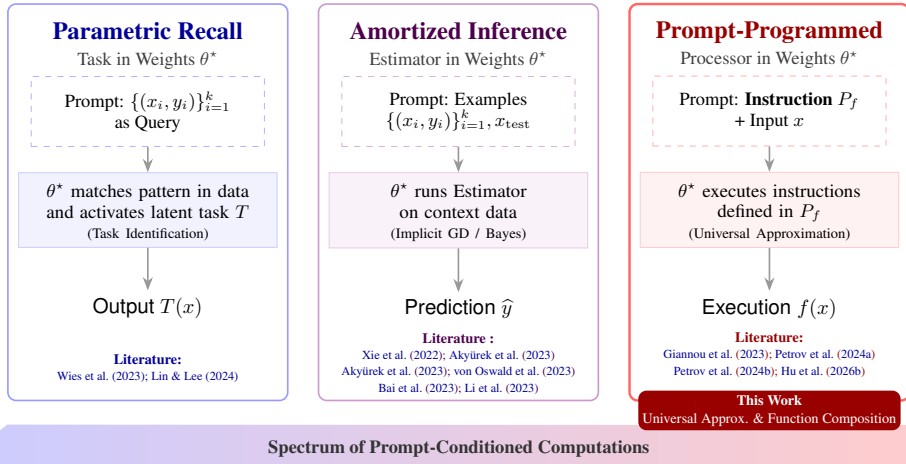

*Figure 1.* **Spectrum of Prompt-Conditioned Computations.** We organize general-purpose behaviors by where the task procedure resides. (1) **Parametric Recall** and (2) **Amortized Inference** (blue/purple) rely on fixed tasks or estimators encoded in the frozen weights $\theta^\star$, the prompt serves only as a query or dataset. In contrast, (3) **Prompt-Programmed Execution** (red) encodes the algorithm *in the prompt* $(P_f, X)$, treating $\theta^\star$ as a program executor. We address the theoretical gap in (3) by providing constructive proofs for in-context universal approximation and function composition.

executes an $m$-fold composition in one forward pass with depth linear in $m$; and *multi-pass* (Theorem 4.2), which reuses the same fixed-depth model across $m$ calls to execute the composition iteratively.

Together, these results extend the scope of provable general-purpose in-context learning. We move beyond the "attention-implementable" algorithms of (Hu et al., 2026b) to broad classes of continuous sequence-to-sequence functions, and beyond single-task prompting to multi-step function composition. By showing how one frozen Transformer serves as a universal and composable in-context approximator, our work takes a step toward explaining the general-purpose behavior of foundation models. This line of inquiry gives a theoretical basis for viewing foundation models not only as pattern-matching engines, but also as prompt-programmable executors for broad classes of computations.

**Organizations.** Section 2 covers preliminaries. Section 3 demonstrates how the *Prompt-Program Execution* perspective leads to the in-context universality. Section 4 presents two results of function composition. Section A summarizes the theoretical results and contextualizes the contributions to in-context learning in more detail. We include the related work in Section B and detailed proofs in Section E.

**Notations.** We use $\mathbb{N}^+$ to denote the set of positive integers, and for $n \in \mathbb{N}^+$, we let $[n] := \{1, 2, \ldots, n\}$. We use lowercase letters for vectors, and uppercase letters for matrices. For $X \in \mathbb{R}^{d \times n}$, $X_{i,:}$ denotes the $i$-th row, and $X_{:, i:j}$ denotes the sub-matrix consisting of columns $i$ through $j$, where $i \le j$ and $i, j \in [n]$. We assume $n > 2$.[5] We write

---

[5]For $n \le 2$, one can pad the sequence with dummy tokens to length 3 and ignore the padded outputs to satisfy technical requirements of (Hu et al., 2026a;b).

$\mathbb{1}_d \in \mathbb{R}^d$ for the all-ones vector. For $X \in \mathbb{R}^{d \times n}$, we define $\|X\|_\infty := \max_{i \in [d], j \in [n]} |X_{ij}|$, and $\|X\|_p$ denotes the entry-wise $\ell_p$ norm. We denote matrices concatenation $\begin{bmatrix} A \\ B \end{bmatrix}$ as $[A; B]$. Lastly, we denote an $\ell$-layer multi-head attention as $\mathrm{Attn}_\ell$, and an attention layer with a residual connection as $\mathrm{Attn}^{\mathrm{Res}}(X) := X + \mathrm{Attn}(X)$.

## 2 Preliminaries

In this section, we introduce the ideas we built on.

### 2.1 Attention and Feed-Forward Layer

This section introduces the components of Transformer we consider in this paper.

We first define attention and the feed-Forward layers. For $x = [x_1; x_2; \ldots; x_d] \in \mathbb{R}^d$, let $\sigma_\beta(\cdot)$ be the softmax function with inverse temperature $\beta$

$$\sigma_\beta(x) := \left[\frac{\exp(\beta x_1)}{\sum_{j=1}^d \exp(\beta x_j)}, \cdots, \frac{\exp(\beta x_d)}{\sum_{j=1}^d \exp(\beta x_j)}\right]. \quad (1)$$

**Definition 2.1** (Attention Layer). *Let $X \in \mathbb{R}^{d \times n}$ be the input sequence. We denote an $H$-head self-attention layer $\mathrm{Attn}(\cdot)$ as*

$$\mathrm{Attn}(X) := \sum_{h=1}^H W_O^h \cdot W_V^h X \cdot \sigma_\beta((W_K^h X)^\top W_Q^h X),$$

*where $\sigma_\beta$ applies column-wise softmax as defined in (1), $W_O^h \in \mathbb{R}^{d_O \times d_V}$, $W_V^h \in \mathbb{R}^{d_V \times d}$, and $W_K^h, W_Q^h \in \mathbb{R}^{d_h \times d}$, are weight matrices. We denote an $\ell$-layer multi-head attention as $\mathrm{Attn}_\ell$. Finally, we denote an attention layer with a residual connection as $\mathrm{Attn}^{\mathrm{Res}}(X) := X + \mathrm{Attn}(X)$.*

**Definition 2.2** (Feed-Forward Layer). *Let $X \in \mathbb{R}^{d \times n}$ be the input sequence. Then, we define a two-layer feed-forward*

*neural network as*

$$\mathrm{FF}(X) := W_2 \cdot \mathrm{ReLU}(W_1 X + b_1 \mathbb{1}_n^\top) + b_2 \mathbb{1}_n^\top,$$

*where $W_1 \in \mathbb{R}^{r \times d}$, $W_2 \in \mathbb{R}^{d \times r}$ are weight matrices, and $b_1 \in \mathbb{R}^r$, $b_2 \in \mathbb{R}^d$ are bias vectors.*

## 2.2 Two Routes of Transformer Universal Approximation

This work builds on two universal approximation routes for softmax Transformers. We review them here. They provide the surrogate approximators used later.

- **Contextual Mapping.** This route views attention as a contextual map (Yun et al., 2020a; Kajitsuka & Sato, 2024; Hu et al., 2025a). Attention assigns each token a code that depends on the whole sequence. A tokenwise FFN then maps this code to the target output. Without attention, a tokenwise FFN acts on each token separately. It therefore does not capture global sequence dependence.

- **Attention-as-Selector.** This route views softmax attention through interpolation (Hu et al., 2026a). Softmax acts as a selector over interpolation points. With a large inverse temperature, it approximates hard selection. This construction proves universal approximation with attention layers alone. It does not use feed-forward networks.

The two routes use different softmax regimes. The contextual-mapping route uses finite $\beta$. It stays away from the hardmax limit. The selector route uses large $\beta$. It makes softmax close to hard selection. Section 3 turns both surrogate constructions into in-context emulation results. Other Transformer universal approximation results also exist. They require stronger assumptions than the compact-domain setting used here (Takakura & Suzuki, 2023; Jiang & Li, 2024). See Section B for a more detailed discussion.

**Distance Metric for Universal Approximation.** We use an integrated distance between sequence-to-sequence maps. Let $1 \le p < \infty$, and let $\| \cdot \|_p$ denote the entrywise $\ell_p$ matrix norm. Let $\mathcal{C} \subset \mathbb{R}^{d \times n}$ be compact. For maps $f_1, f_2 : \mathcal{C} \to \mathbb{R}^{d \times n}$, define

$$\mathrm{d}_p(f_1, f_2) := \left( \int_{\mathcal{C}} \| f_1(X) - f_2(X) \|_p^p \mathrm{d}X \right)^{1/p}. \quad (2)$$

Small $\mathrm{d}_p(f_1, f_2)$ means that the two maps have small average entrywise $\ell_p$ error over $\mathcal{C}$.

## 3 In-Context Universal Approximation

This section proves in-context universality with frozen softmax-attention Transformers. We work over a compact domain $\mathcal{C} \subset \mathbb{R}^{d \times n}$ and a target family $\mathcal{F}_L$ of $L$-Lipschitz continuous sequence-to-sequence functions $f : \mathcal{C} \to \mathbb{R}^{d \times n}$. Our goal is to construct one fixed-weight softmax Transformer $T_{\theta^\star}$ such that, for every target $f \in \mathcal{F}_L$, we are able to construct a prompt $P_f$ satisfying

$$T_{\theta^\star}(P_f, X) \approx f(X), \quad \text{for all} \quad X \in \mathcal{C}.$$

Thus the target function $f$ changes through the prompt $P_f$, while the emulator $T_{\theta^\star}$ stays fixed.

Our strategy is to reduce existing universal approximation results to in-context emulation: given a target $f \in \mathcal{F}_L$, construct a surrogate universal approximator $A_{\phi_f}$, encode $\phi_f$ into an instruction $P_f$, and show that one fixed-weight softmax Transformer $T_{\theta^\star}$ emulates $A_{\phi_f}$ on input $X$. Thus, $T_{\theta^\star}(P_f, X) \approx A_{\phi_f}(X) \approx f(X)$. Since the same emulator $T_{\theta^\star}$ works for every $f \in \mathcal{F}_L$ through its corresponding prompt $P_f$, this gives in-context universal approximation.

We present two constructive proofs based on two different universal approximation results:

- Section 3.2 follows this reduction through the *Contextual Mapping* route (Kajitsuka & Sato, 2024).

- Section 3.3 follows this reduction through the *Attention-as-Selector* route (Hu et al., 2026a).

### 3.1 Proof Strategy Overview

To prove the class-level statement, we fix an arbitrary target $f \in \mathcal{F}_L$ and an arbitrary input sequence $X \in \mathcal{C}$. Since $f$ and $X$ are arbitrary, the same argument applies to every $f \in \mathcal{F}_L$ and every $X \in \mathcal{C}$. For readability, we keep the norm $\| \cdot \|$ generic in this overview. The later theorem statements specify the precise metric and input format.

- **Step 1: Surrogate Approximator $A_{\phi_f}$.** Approximate $f(\cdot)$ with a sequence-to-sequence surrogate approximator $A_{\phi_f}(\cdot)$ and then realize it at $X$:

$$\| \underbrace{A_{\phi_f}(X)}_{\text{Surrogate Approximator}} - \underbrace{f(X)}_{\text{Target}} \| \le \epsilon_{\mathrm{Sur}}. \quad (3)$$

- **Step 2: In-Context Emulation of $A_{\phi_f}$.** Encode the surrogate parameters $\phi_f$ into a prompt $P_f$. Then prompt the fixed-weight emulator $T_{\theta^\star}$ with $(P_f, X)$ to emulate the surrogate computation:

$$\sup_{X \in \mathcal{C}} \| \underbrace{T_{\theta^\star} \overbrace{(P_f, X)}^{\text{Prompt Program}}}_{\text{Emulator (Fixed-Weight Transformer)}} - \underbrace{A_{\phi_f}(X)}_{\text{Surrogate Approximator}} \|$$

$$\le \epsilon_{\mathrm{Emu}}. \quad (4)$$

We call $(P_f, X)$ the "prompt program." The $P_f$ part specifies the instruction associated with the target function $f$. The $X$ part specifies the data the function is applied to.

- **Step 3: Error Reduction.** Combine the surrogate approximation and the emulator approximation:

$$\| A_{\phi_f}(X) - f(X) \| \le \epsilon_{\mathrm{Sur}},$$
$$\| T_{\theta^\star}(P_f, X) - A_{\phi_f}(X) \| \le \epsilon_{\mathrm{Emu}},$$

for any $X \in \mathcal{C}$. By triangle inequality, we obtain

$$\| \underbrace{T_{\theta^\star}(P_f, X)}_{\text{Emulator}} - \underbrace{f(X)}_{\text{Target}} \| \le \epsilon_{\mathrm{Sur}} + \epsilon_{\mathrm{Emu}}. \quad (5)$$

Since $f$ and $X$ are arbitrary, the same fixed emulator $T_{\theta^\star}$

approximates every target in $\mathcal{F}_L$ through the corresponding prompt $P_f$.

Equivalently, we may also view $f(X)$ as the target computation realized at $X$. Under this view, $P_f$ specifies the computation, and $X$ supplies the data. Thus, in-context universal approximation also gives a *prompt-programmable algorithm-emulation* interpretation (Hu et al., 2026b, Section 4).

We realize this strategy with two distinct surrogate approximators $A_{\phi_f}(\cdot)$ from (Kajitsuka & Sato, 2024; Hu et al., 2026a) and (Hu et al., 2026a) in Section 3.2 and Section 3.3, respectively. In both routes, $P_f$ encodes the algorithm instruction, and a frozen Transformer executes the encoded instruction on $X$ to approximate the target function.

## 3.2 Attention-as-Contextual-Mapping Route

We now realize the reduction in Section 3.1 through the contextual-mapping route (Yun et al., 2020a; Kajitsuka & Sato, 2024; Hu et al., 2025a). Fix a compact domain $\mathcal{C} \subset \mathbb{R}^{d \times n}$ and a target function $f \in \mathcal{F}_L$. To approximate $f$ in-context, we first construct a Transformer following Kajitsuka & Sato (2024, Proposition 1, or Theorem D.1) as the surrogate approximator in (3)

$$A_{\phi_f} = \mathrm{FF}_2 \circ \mathrm{Attn}^{\mathrm{Res}} \circ \mathrm{FF}_1, \qquad (6)$$

and choose its parameters $\phi_f$ such that $A_{\phi_f}$ approximates $f$ on $\mathcal{C}$. Here the surrogate architecture is fixed for the entire $\mathcal{F}_L$. Only the surrogate parameters $\phi_f$ depend on the target function $f$.

Then, we emulate this surrogate approximator $A_{\phi_f}$ with an emulator $T_{\theta^\star}$ in-context. Specifically, we encode the surrogate parameters $\phi_f$ into a prompt $P_f$ and construct a six-layer softmax-attention emulator $T_{\theta^\star} = \mathrm{Attn}_6$ such that

$$T_{\theta^\star}(P_f, X) \approx A_{\phi_f}(X).$$

Here $P_f$ stores the instruction for the target function $f$, and $X$ is the data on which the instruction is executed. The emulator weights $\theta^\star$ are fixed after the surrogate architecture, prompt shape, common norm bounds, and error budget are fixed. They do not depend on $f$, $X$, or the surrogate parameters $\phi_f$. Thus, the task-specific information is carried by the prompt $P_f$, while the emulator $T_{\theta^\star}$ is shared across the whole target family $\mathcal{F}_L$. Only the prompt $P_f$ changes with the target function $f$. Consequently, this route proves in-context universal approximation by reducing Transformer universal approximation to Transformer in-context emulation:

$$\underbrace{T_{\theta^\star}(P_f, X) \approx \overbrace{A_{\phi_f}(X)}^{\text{Universal Approximation}} \approx f(X).}_{\text{In-Context Emulation}}$$

Since $f, X$ are arbitrary and the same emulator $T_{\theta^\star}$ works for every $f \in \mathcal{F}_L$ through corresponding prompt $P_f$, this gives in-context universal approximation by Transformer in-context emulation.

We now define the weight encodings used to build $P_f$, and then prove that fixed attention layers executes the encoded feed-forward and attention modules. Specifically, we want the prompt $P_f$ to encode the surrogate weights. To do so, we use one elementary encoding throughout this route.

**Definition 3.1** (Token-Indexed Vector Encoding). *Let $X \in \mathbb{R}^{d \times n}$ be the input sequence. Let $w \in \mathbb{R}^d$ be the weight vector. Define the encoding function* $\mathrm{Enc} : \mathbb{R}^d \to \mathbb{R}^{2d \times n}$ *to map the vector $w$ to a length-$n$ matrix later append below $X$ as*

$$\mathrm{Enc}(w) := \begin{bmatrix} 0 \cdot w & 1 \cdot w & \cdots & (n-1) \cdot w \\ w & w & \cdots & w \end{bmatrix} \in \mathbb{R}^{2d \times n}.$$

This encoding records both the value of a vector and its token-indexed multiples. It allows fixed attention heads to recover linear forms specified by encoded weights. We use Definition 3.1 to turn the surrogate parameters $\phi_f$ into prompt blocks. For the contextual-mapping route, these parameters are the weights of $\mathrm{FF}_1$, $\mathrm{Attn}^{\mathrm{Res}}$, and $\mathrm{FF}_2$.

**Emulating** $\mathrm{FF}_1, \mathrm{FF}_2$. We first apply this encoding to feed-forward networks (FFNs).

**Definition 3.2** (FFN Weight Encoding for In-Context Emulation). *Consider a feed-forward layer*

$$\mathrm{FF}(X) = W_2 \mathrm{ReLU}(W_1 X + b_1 \mathbb{1}_n^\top) + b_2 \mathbb{1}_n^\top,$$

*where $W_1 \in \mathbb{R}^{r \times d}$, $b_1 \in \mathbb{R}^r$, $W_2 \in \mathbb{R}^{d \times r}$, and $b_2 \in \mathbb{R}^d$. For each hidden dimension $j \in [r]$ and each output dimension $k \in [d]$, define the augmented first-layer and second-layer weight vectors*

$$a_j := \begin{bmatrix} (W_1)_{j,:}^\top \\ (b_1)_j \end{bmatrix} \in \mathbb{R}^{d+1}, \quad c_k := \begin{bmatrix} (W_2)_{k,:}^\top \\ (b_2)_k \end{bmatrix} \in \mathbb{R}^{r+1},$$

*encode them by $W_{\mathrm{FF},j}^{(1)} := \mathrm{Enc}(a_j)$ and $W_{\mathrm{FF},k}^{(2)} := \mathrm{Enc}(c_k)$, and define the prompt block for FF by*

$$W_{\mathrm{FF}} := \underbrace{[W_{\mathrm{FF},1}^{(1)}; \cdots; W_{\mathrm{FF},r}^{(1)}; W_{\mathrm{FF},1}^{(2)}; \cdots; W_{\mathrm{FF},d}^{(2)}]}_{2(r(d+1)+d(r+1)) \times n}.$$

Definition 3.2 packages the two affine maps of an FFN into one prompt block $W_{\mathrm{FF}}$. In the surrogate $A_{\phi_f} = \mathrm{FF}_2 \circ \mathrm{Attn}^{\mathrm{Res}} \circ \mathrm{FF}_1$, we apply this encoding twice, producing $W_{\mathrm{FF}_1}$ and $W_{\mathrm{FF}_2}$.

We first prove the in-context emulation of $\mathrm{FF}_1$. Since $\mathrm{FF}_1$ is followed by $\mathrm{Attn}^{\mathrm{Res}}$ and $\mathrm{FF}_2$, the emulation must also preserve the prompt blocks used by later layers. We denote these preserved blocks by $Z$ and call $Z$ the flow-through block. Then next Lemma 3.1 then shows that, the same two-layer attention emulator approximates the FFN specified by $W_{\mathrm{FF}}$ and forwards $Z$. Changing $W_{\mathrm{FF}}$ changes the executed FFN, while the attention emulator weights remain fixed.

**Lemma 3.1** (Prompt-Encoded FFN Emulation with Flow-Through). *Fix dimensions $d, n, r, d_Z$ and bounds $B_X, B_Z, B_W > 0$. For any $\epsilon_1 > 0$, there exists a two-layer attention $\mathrm{Attn}_2$ such that the following holds. Let*

$X \in \mathbb{R}^{d \times n}$ *satisfy* $\|X\|_\infty \leq B_X$. *Let* $Z \in \mathbb{R}^{d_Z \times n}$ *be a flow-through block containing a fixed* $I_n$ *sub-block and satisfying* $\|Z\|_\infty \leq B_Z$. *For any feed-forward layer* FF *with hidden width* $r$ *and prompt block* $W_{\mathrm{FF}}$ *constructed from Definition 3.2 with* $\|W_{\mathrm{FF}}\|_\infty \leq B_W$, *then*

$$\|\mathrm{Attn}_2(\begin{bmatrix} X \\ W_{\mathrm{FF}} \\ Z \end{bmatrix}) - \begin{bmatrix} \mathrm{FF}(X) \\ Z \end{bmatrix}\|_\infty \leq \epsilon_1.$$

*Proof Sketch.* To approximate the first hidden layer of the ReLU neural network in context, we use a one-layer multi-head attention constructed from Corollary D.4.1. Moreover, we use an attention head to flow through the unchanged portion of input (flow-through component $Z$) to the next layer. Please see Section E.1 for a detailed proof. □

Notably, the attention $\mathrm{Attn}_2$'s weights depend only on the fixed dimensions, bounds, and $\epsilon_1$. They do not depend on $X, Z$, or the particular emulation target feed-forward layer FF.

For the second FFN $\mathrm{FF}_2$, no later prompt block needs to be preserved. Thus we use the same FFN-emulation idea without returning the flow-through block.

**Lemma 3.2** (Prompt-Encoded FFN Emulation). *Under the same notation and assumptions of Lemma 3.1, for any* $\epsilon_2 > 0$, *there exists a two-layer attention* $\mathrm{Attn}_2$ *such that*

$$\|\mathrm{Attn}_2(\begin{bmatrix} X \\ W_{\mathrm{FF}} \end{bmatrix}) - \mathrm{FF}(X)\|_\infty \leq \epsilon_2.$$

*Proof.* Please see Section E.2 for a detailed proof. □

Together, Lemmas 3.1 and 3.2 let fixed attention execute both FFNs in the surrogate $A_{\phi_f}$: it executes $\mathrm{FF}_1$ while forwarding the remaining prompt blocks, and then executes $\mathrm{FF}_2$.

**Remark 3.1.** *While recent work shows that attention approximates FFNs (Hu et al., 2026a), it does not generalize to in-context learning. The model must not merely approximate one fixed FFN, but must allow the prompt to define which FFN (hence target function) to emulate.*

**Emulating** $\mathrm{Attn}^{\mathrm{Res}}$. Next, we show frozen attention emulates single-head attention with a residual connection, i.e., $\mathrm{Attn}^{\mathrm{Res}}$. We first encode the attention block of the surrogate. This is the multi-head attention analogue of Definition 3.2: the prompt stores the target attention weights, and the fixed emulator executes the encoded attention operation. We begin by defining a row-vectorization convention, and then package the weights of each attention head into a prompt block.

**Definition 3.3** (Vectorization of a Matrix). *Given* $W \in \mathbb{R}^{p \times s}$, *we define* $\underline{W} \in \mathbb{R}^{ps}$ *such that for* $i \in [p]$ *and* $j \in [s]$, $\underline{W}_{(i-1)s+j} = W_{ij}$. *That is, we stack the row vectors of* $W$ *into a column vector.*

**Definition 3.4** (Multi-Head Attention Weight Encoding for In-Context Emulation). *Suppose the target attention has* $H$ *heads. For each head* $h \in [H]$, *let* $W_K^h, W_Q^h \in \mathbb{R}^{d_h \times d}$ *and* $W_V^h \in \mathbb{R}^{d_V \times d}$ *be the weight matrices*[6]. *Then, we define the weight encoding for each target head*

$$w^h := [\underline{W}_K^h; \underline{W}_Q^h; \underline{W}_V^h] \in \mathbb{R}^{d(2d_h + d_V)},$$
$$W^h := \mathrm{Enc}(w^h) \in \mathbb{R}^{2d(2d_h + d_V) \times n},$$

*where* $\mathrm{Enc}(\cdot)$ *is defined in Definition 3.1. Also, given an input sequence* $X$ *and an auxiliary block* $I_n$, *we use* $X_p := [X; W^1; \cdots; W^H; I_n]$ *as the* $H$*-head attention-emulation input.*

We now emulate the residual-attention block in the surrogate (6). For the contextual-mapping route, this block is single-head attention $\mathrm{Attn}^{\mathrm{Res}} := X + W_V X \sigma_\beta((W_K X)^\top W_Q X)$. Thus we use the single-head specialization of Definition 3.4 and write $W_{\mathrm{Attn}^{\mathrm{Res}}}$ as its weight encoding (Definition D.5 with $H = 1$). The next lemma shows that a fixed 2-layer attention executes the residual-attention layer specified by $W_{\mathrm{Attn}^{\mathrm{Res}}}$ and preserves the flow-through block $Z$.

**Lemma 3.3** (Prompt-Encoded Residual-Attention Emulation with Flow-Through). *Fix dimensions* $d, n, d_h, d_Z$ *and bounds* $B_X, B_Z, B_W, B_{KQV} > 0$. *For any* $\epsilon_3 > 0$, *there exists a two-layer attention* $\mathrm{Attn}_2$ *such that the following holds. Let* $X \in \mathbb{R}^{d \times n}$ *satisfy* $\|X\|_\infty \leq B_X$. *Let* $Z \in \mathbb{R}^{d_Z \times n}$ *be a flow-through block satisfying* $\|Z\|_\infty \leq B_Z$. *Consider any single-head residual-attention layer*

$$\mathrm{Attn}^{\mathrm{Res}}(X) := X + W_V X \sigma_\beta((W_K X)^\top W_Q X),$$

*where* $W_K, W_Q \in \mathbb{R}^{d_h \times d}$ *and* $W_V \in \mathbb{R}^{d \times d}$. *Let* $W_{\mathrm{Attn}^{\mathrm{Res}}} := \mathrm{Enc}([\underline{W}_K; \underline{W}_Q; \underline{W}_V])$ *and* $X_p = [X; W_{\mathrm{Attn}^{\mathrm{Res}}}; I_n] \in \mathbb{R}^{(d + 2d(2d_h + d) + n) \times n}$ *following Definition 3.4. Suppose* $\|W_{\mathrm{Attn}^{\mathrm{Res}}}\|_\infty \leq B_W$, $\|W_K X\|_\infty, \|W_Q X\|_\infty, \|W_V X\|_\infty \leq B_{KQV}$. *Then,*

$$\|\mathrm{Attn}_2(\begin{bmatrix} X_p \\ Z \end{bmatrix}) - \begin{bmatrix} \mathrm{Attn}^{\mathrm{Res}}(X) \\ Z \end{bmatrix}\|_\infty \leq \epsilon_3.$$

*Proof Sketch.* To propagate $Z$ through the attention module, we add an auxiliary head in each layer and exploit the identity matrix $I_n$ in $X_p$. Concretely, we choose the value projection to read $Z$, and the key/query projections to read $I_n$. With an appropriate choice of $\beta$, the resulting attention weights concentrate on the diagonal, so this head propagates $Z$. See Section E.3 for a detailed proof. □

**In-Context Universal Approximation.** We now realize the emulator $T_{\theta^\star}$ with six-layer attention $\mathrm{Attn}_6$ by stacking the three emulator modules in the same order as the surrogate $A_{\phi_f} = \mathrm{FF}_2 \circ \mathrm{Attn}^{\mathrm{Res}} \circ \mathrm{FF}_1$. The first two attention layers emulate $\mathrm{FF}_1$ and forward the remaining prompt blocks (Lemma 3.1). The next two layers emulate $\mathrm{Attn}^{\mathrm{Res}}$ and

---

[6]If the target attention uses an output projection $W_O^h$, we absorb it into the value matrix and write $\widetilde{W}_V^h := W_O^h W_V^h$. Otherwise, set $\widetilde{W}_V^h := W_V^h$.

forward the block needed by $\mathrm{FF}_2$ (Lemma 3.3). The last two layers emulate $\mathrm{FF}_2$ (Lemma 3.2). Thus a six-layer attention emulator executes the full Transformer surrogate in context. This yields an attention-only, in-context universal approximator for continuous sequence-to-sequence functions.

**Theorem 3.1** (In-Context Universality via Transformer-Surrogate Emulation). *Let $\mathcal{C} = [-b, b]^{d \times n} \subset \mathbb{R}^{d \times n}$ be compact, and let $\mathcal{F}_L := \{f : \mathcal{C} \to \mathcal{C} \mid \mathrm{Lip}_\infty(f) \leq L\}$ be a set of L-Lipschitz continuous sequence-to-sequence functions. Fix a deterministic $\rho$-separating positional encoding $E_p \in \mathbb{R}^{d \times n}$ (Definition D.3). Then, for any $\epsilon > 0$, there exists a fixed Transformer $T_{\theta^\star} = \mathrm{Attn}_6$ such that, for every $f \in \mathcal{F}_L$, there exists a prompt $P_f$ satisfying*

$$\mathrm{d}_p(T_{\theta^\star, P_f}, f) \leq \epsilon, \quad \text{where} \quad T_{\theta^\star, P_f}(X) := T_{\theta^\star}([X + E_p; P_f]).$$

*Here $P_f$ follows Definitions 3.2 and 3.4. The emulator weights $\theta^\star$ and the prompt shape depend only on $\mathcal{C}, L, p, d, n, E_p, \rho, \epsilon,$, not on $f$ or $X$. Only $P_f$ depend on $f$.*

*Proof.* Please see Section E.4 for a detailed proof. $\square$

Intuitively, the prompt $P_f$ stores information of the surrogate Transformer $\mathrm{FF}_2 \circ \mathrm{Attn}^{\mathrm{Res}} \circ \mathrm{FF}_1$. The frozen model $T_{\theta^\star}$ then reads this prompt and executes the encoded computation on $X$. Thus, this route shows that one fixed softmax Transformer $T_{\theta^\star}$ is capable of emulating another Transformer $A_{\phi_f}$ in context. Since $A_{\phi_f}$ is well-established sequence-to-sequence universal approximator (Kajitsuka & Sato, 2024; Hu et al., 2025a), this establishes in-context universal approximation with a fixed weight Transformer $T_{\theta^\star}$. Moreover, Theorem 3.1 gives an explicit construction of $\theta^\star$ and $P_f$ for any continuous $f : \mathbb{R}^{d \times n} \to \mathbb{R}^{d \times n}$ on a compact domain, see the following remark.

**Remark 3.2** (Explicit Construction of Prompt $P_f$). *A satisfactory theory of the prompt-programmable regime should provide a constructive and reusable prompt interface. Theorem 3.1 gives out the specific structure of $P_f$: the parameters of surrogate approximators are serialized into token embeddings through the weight blocks $[W_{\mathrm{FF}_1}; W_{\mathrm{Attn}^{\mathrm{Res}}}; I_n; W_{\mathrm{FF}_2}]$. We further give the explicit construction of $P_f$ that depends on function $f$ at Section D.2 following (Hu et al., 2025b).*

**Prompt-Programmable In-Context Emulation.** Theorem 3.1 also implies a generic in-context algorithm-emulation template. Once we encode a target computation as a sequence-to-sequence map $f$, the prompt $P_f$ serves as the instruction and $X$ serves as the data. The emulator $T_{\theta^\star}$, weights fixed independently of both $P_f, X$, then executes the instructed computation on $X$.[7]

**Corollary 3.1.1.** *Under the setup of Theorem 3.1, let $A_\phi(X) = \mathrm{FF}_2 \circ \mathrm{Attn}^{\mathrm{Res}} \circ \mathrm{FF}_1(X + E_p)$ be the fixed surrogate approximator and $P_f$ be the prompt encoding constructed from $f$. Suppose that, for every $f \in \mathcal{F}_L$ and every $X \in \mathcal{C}$, $\|A_{\phi_f}(X) - f(X)\|_\infty \leq \epsilon_{\mathrm{sur}}$. Then, for any $\epsilon_{\mathrm{emu}} > 0$, there exists a fixed Transformer $T_{\theta^\star} = \mathrm{Attn}_6$ such that for every $f \in \mathcal{F}$ and every $X \in \mathcal{C}$*

$$\| \underbrace{T_{\theta^\star}}_{\text{Emulator (Fixed Transformer)}} \overbrace{([X; P_f])}^{\text{Prompt Program}} - \underbrace{f(X)}_{\text{Target Algorithm } f \text{ specified on data } X} \|_\infty$$

$$\leq \underbrace{\epsilon_{\mathrm{emu}}}_{\text{Emulator Approximation}} + \underbrace{\epsilon_{\mathrm{sur}}}_{\text{Surrogate Approximation}} .$$

*Proof.* By Equation (94) and triangle inequality, we complete the proof. $\square$

This template subsumes prompt-programmable in-context algorithm emulation at the function-realization level. For example, linear regression, gradient-style updates, and attention-implementable procedures (Hu et al., 2026b) correspond to particular choices of the map $f$ and the encoded instance $X$. This differs from task-specific-weight emulation (Bai et al., 2023): here the emulator weights stay fixed, and only the program prompt $(P_f, X)$ changes.

### 3.3 Attention-as-Selector Route

This section provides another Transformer in-context universal approximation guarantee. We realize the same reduction in Section 3.1 through the attention-as-selector route (Hu et al., 2026a). This route uses an attention-only surrogate, so it avoids the feed-forward emulation in Section 3.2. Fix a compact domain $\mathcal{C} \subset \mathbb{R}^{d \times n}$ and a target function $f \in \mathcal{F}_L$. To approximate $f$ in context, we first choose a four-layer attention-only surrogate $A_{\phi_f} = \mathrm{Attn}_4$ from (Hu et al., 2026a) such that $A_{\phi_f}(X) \approx f(X)$ on $\mathcal{C}$. Then, we encode $\phi_f$ into a prompt $P_f$ and construct one fixed eight-layer attention emulator $T_{\theta^\star} = \mathrm{Attn}_8$ such that $T_{\theta^\star}([X; P_f]) \approx A_{\phi_f}(X)$. Since $f \in \mathcal{F}_L$ and $X \in \mathcal{C}$ are arbitrary, the same fixed emulator $T_{\theta^\star}$ works for the whole class $f \in \mathcal{F}_L$ through the corresponding prompt $P_f$. This gives in-context universal approximation through the selector route.

First we extend the single-head attention emulator (Lemma 3.3) to the multi-head setting.

**Lemma 3.4** (Prompt-Encoded Multi-Head Attention Emulation). *Fix dimensions $d, m, H, d_h, d_V$, a fixed inverse temperature $\beta > 0$, and bounds $B_X, B_W, B_{KQV} > 0$. For any $\epsilon_4 > 0$, there exists a two-layer attention $\mathrm{Attn}_2$ such that the following holds. Let $X \in \mathbb{R}^{d \times n}$ satisfy $\|X\|_\infty \leq B_X$.*

---

[7]We remark that, both $T_{\theta^\star}, P_f$ are constructive. Once we fix the surrogate architecture, prompt shape, uniform bounds, and error budget, the emulator weights $\theta^\star$ are fixed. They do not depend on the target function $f$, the input $X$, or the surrogate parameters $\phi_f$. Consequently, the emulator weights $\theta^\star$ do not depend on $f, X$, or the surrogate parameters $\phi_f$. We construct $P_f$ from $f$ by encoding the surrogate parameters $\phi_f$. Thus $P_f$ plays the role of the program, while $X$ plays the role of the data.

*Let* $\mathrm{Attn}$ *be any $H$-head attention layer in Definition 2.1, and $X_p$ the prompt encoding from Definition 3.4. Suppose $\|W_{\mathrm{Attn}}\|_\infty \le B_W$, $\|W_K^h X\|_\infty, \|W_Q^h X\|_\infty, \|W_V^h X\|_\infty \le B_{KQV}$ for all $h \in [H]$. Then*
$$\|\mathrm{Attn}_2(X_p) - \mathrm{Attn}(X)\|_\infty \le \epsilon_4.$$

*Proof.* Please see Section E.5 for a detailed proof. $\square$

For intermediate layers, the emulator must also preserve the prompt blocks used by later layers. We therefore use the following flow-through version.

**Lemma 3.5** (Prompt-Encoded Multi-Head Attention Emulation with Flow-Through). *Under the same assumptions as Lemma 3.4, fix a flow-through dimension $d_Z$ and a bound $B_Z > 0$. For any $\epsilon_5 > 0$, there exists a two-layer attention $\mathrm{Attn}_2$ such that the following holds. For any $Z \in \mathbb{R}^{d_Z \times n}$ satisfying $\|Z\|_\infty \le B_Z$,*
$$\left\| \mathrm{Attn}_2 \begin{bmatrix} X_p \\ Z \end{bmatrix} - \begin{bmatrix} \mathrm{Attn}(X) \\ Z \end{bmatrix} \right\|_\infty \le \epsilon_5.$$

*Proof.* Please see Section E.6 for a detailed proof. $\square$

We now construct the full emulator. The surrogate approximator $A_{\phi_f}$ now is a four-layers attention $\mathrm{Attn}_4$. We emulate the first three layers by Lemma 3.5 and the fourth layer by Lemma 3.4.

**Definition 3.5** (Attention-Only Surrogate Weight Encoding for In-Context Emulation). *Fix the uniform four-layer attention-only surrogate template from Theorem D.2. For each $f \in \mathcal{F}_L$, let $A_{\phi_f} := \mathrm{Attn}_{\phi_{f,4}} \circ \mathrm{Attn}_{\phi_{f,3}} \circ \mathrm{Attn}_{\phi_{f,2}} \circ \mathrm{Attn}_{\phi_{f,1}}$ with $\phi_f = (\phi_{f,1}, \phi_{f,2}, \phi_{f,3}, \phi_{f,4})$ be the parameter choice in this fixed template. For each layer $i \in [4]$, let $H_i$ denote the number of heads. For each $h \in [H_i]$, let $W_i^h$ be the weight encoding of the $h$-th head in the $i$-th layer, following the format of $W^h$ in Definition 3.4 with sequence length $n+1$. Define the weight encoding of $A_{\phi_f}$ as*
$$W_{\phi_f,i} := [W_i^1; \cdots; W_i^{H_i}; I_{n+1}],$$
$$P_f := [W_{\phi_f,1}; W_{\phi_f,2}; W_{\phi_f,3}; W_{\phi_f,4}].$$

*For input sequence $X \in \mathbb{R}^{d \times n}$, define the selector-route prompt program as $[X_a; P_f]$ where*
$$X_a := \begin{bmatrix} X & 0_{d \times 1} \\ I_n & 0_{n \times 1} \\ 0_{1 \times n} & 1 \end{bmatrix} \in \mathbb{R}^{(d+n+1) \times (n+1)}.$$

**In-Context Universal Approximation.** The next theorem states that a fixed Transformer $T_{\theta^\star} = \mathrm{Attn}_8$ is an in-context universal approximator for continuous sequence-to-sequence functions.

**Theorem 3.2** (In-Context Universality via Attention-Only Surrogate Emulation). *Let $\mathcal{C} = [-b, b]^{d \times n} \subset \mathbb{R}^{d \times n}$ be compact, and let $\mathcal{F}_L := \{ f : \mathcal{C} \to \mathcal{C} \mid \mathrm{Lip}_\infty(f) \le L \}$ be a set of $L$-Lipschitz continuous sequence-to-sequence functions. Then, for any $\epsilon > 0$, there exists a fixed Transformer*

$T_{\theta^\star} = \mathrm{Attn}_8$ *such that, for every $f \in \mathcal{F}_L$, there exists a prompt $P_f$ satisfying*
$$\mathrm{d}_p(\widetilde{T}_{\theta^\star, P_f}, f) \le \epsilon, \quad \widetilde{T}_{\theta^\star, P_f}(X) := T_{\theta^\star}([X_a; P_f])_{:,1:n},$$
*where $P_f, X_a$ follow Definition 3.5. The emulator weights $\theta^\star$ and the prompt shape depend only on $\mathcal{C}, L, p, d, n, \epsilon$, not on $f$ or $X$. Only $P_f$ depends on $f$.*

**Remark 3.3** (Explicit Construction of Prompt $P_f$). *Again, we recall the reusable and constructive prompt interface proposed in the introduction. Theorem 3.2 provides the reusable instruction format $P_f$. $P_f$ encodes the weights of surrogate approximator $\mathrm{Attn}_4$. In Section D.3, given a target function $f$, we restate the explicit weight construction provided by Hu et al. (2026a).*

**Prompt-Programmable In-Context Emulation.** Theorem 3.2 also implies a generic in-context algorithm-emulation template for attention-only models. Once we encode a target computation as a sequence-to-sequence map $f$, the prompt $P_f$ serves as the instruction and $X_a$ serves as the augmented data. The emulator $T_{\theta^\star}$, with weights fixed independently of $f$ and $X$, then executes the instructed computation on $X_a$. The same constructivity as in Corollary 3.1.1 applies: $T_{\theta^\star}$ is fixed after we fix the surrogate architecture, prompt shape, uniform bounds, and error budget, while $P_f$ is constructed from $f$ by encoding the surrogate parameters.

**Corollary 3.2.1.** *Under the setup of Theorem 3.2, fix $\epsilon_{\mathrm{sur}}, \epsilon_{\mathrm{emu}} > 0$ and set $\epsilon := \epsilon_{\mathrm{sur}} + \epsilon_{\mathrm{emu}}$. Then there exists a fixed eight-layer softmax-attention emulator $T_{\theta^\star} = \mathrm{Attn}_8$ such that, for every $f \in \mathcal{F}_L$, the prompt encoding $P_f$ constructed from $f$ as in Theorem 3.2 satisfies, for every $X \in \mathcal{C}$,*

$$\| \underbrace{T_{\theta^\star}(\overbrace{[X_a; P_f]}^{\text{Prompt Program}})_{:,1:n}}_{\text{Emulator (Fixed Transformer)}} - \underbrace{f(X)}_{\text{Target Algorithm } f \text{ specified on data } X} \|_p$$
$$\le \underbrace{\epsilon_{\mathrm{emu}}}_{\text{Emulator Approximation}} + \underbrace{\epsilon_{\mathrm{sur}}}_{\text{Surrogate Approximation}} = \epsilon.$$

*Proof.* Please see Section E.8 for a detailed proof. $\square$
This template gives the attention-only counterpart of Corollary 3.1.1. The first route stores an FFN-attention-FFN program in $P_f$, while the second (this) route stores a four-layer attention-only program in $P_f$. In both routes, the emulator weights stay fixed, and only the program prompt changes with the target computation.

We also summarize the emulators $T_{\theta^\star}$ complexity in Section C. The attention-only route requires a larger embedding dimension and head count. This is because surrogate attention requires larger model complexity to achieve universal approximation.

## 4 In-Context Function Composition

Section 3 shows that one fixed softmax Transformer executes a broad class of prompt-encoded functions. We now

show that the same mechanism also supports in-context function composition. Given subroutine prompts $P_{f_1}, \ldots, P_{f_m}$, we construct a composite prompt that makes a frozen Transformer execute $f_m \circ f_{m-1} \circ \cdots \circ f_1$. This turns in-context universality into prompt-level program assembly. Specifically, we consider two practical scenarios for the use of language models:

- **One-Pass Composition:** One prompt contains all subroutine instructions, and a deeper frozen Transformer executes the full composition in one forward pass.

- **Multi-Pass Composition:** We prompt the frozen model iteratively. Each call supplies one subroutine instruction, and the same fixed-depth Transformer is reused across turns.

These two mechanisms capture basic forms of in-context compositional generalization with Transformers: the prompts specify the subroutines, the frozen Transformer execute, and composition happens through prompt assembly and intermediate-state passing.

### 4.1 One-Pass Composition

One-pass composition models a single-call LLM workflow. In practice, a user gives a full multi-step instruction in one prompt, and the model returns the final output in one forward pass. For example, the prompt may ask the model to apply one rule, then another rule, and then report the final answer. We formalize this workflow as one-pass in-context function composition. Please see Figure 3 for illustration.

**Theorem 4.1** (One-Pass In-Context Composition). *Let $\mathcal{C} = [-b, b]^{d \times n} \subset \mathbb{R}^{d \times n}$, and let $\mathcal{F}_L := \{f : \mathcal{C} \to \mathcal{C} \mid \mathrm{Lip}(f) \leq L\}$. Fix $m \in \mathbb{N}^+$. Then, for any $\epsilon > 0$, there exists a fixed-weight Transformer $T_{\theta^\star} = \mathrm{Attn}_{8m}$ such that the following holds. For every $f_1, \ldots, f_m \in \mathcal{F}_L$, there exist prompts $P_{f_1}, \ldots, P_{f_m}$ in the format of Definition 3.5. Define the target composition function and the composite prompt as $g := f_m \circ f_{m-1} \circ \cdots \circ f_1$ and $P_g := [P_{f_1}; P_{f_2}; \cdots ; P_{f_m}]$. Then, the induced map $\widetilde{T}_{\theta^\star, P_g}(X) := T_{\theta^\star}([\begin{smallmatrix} X_a \\ P_g \end{smallmatrix}])_{1:d, 1:n}$ satisfies*

$$\mathrm{d}_p(\widetilde{T}_{\theta^\star, P_g}, g) \leq \epsilon.$$

*The emulator weights $\theta^\star$ and the composite prompt shape depend only on $\mathcal{C}, L, p, d, n, m, \epsilon$, not on $f_1, \ldots, f_m$ or $X$. Only the prompt entries in $P_g$ depend on the subroutines $f_1, \ldots, f_m$.*

*Proof.* Please see Section E.9 for a detailed proof. □

This gives a basic mechanism for in-context compositional generalization: the prompt assembles simple subprograms $P_{f_1}, \ldots, P_{f_m}$ into one composite program, and the frozen Transformer executes the assembled program on $X$. The depth grows linearly with the number of composed subroutines in this one-pass construction. Thus one-pass composition executes the full program in a single forward pass, but it pays for longer compositions with greater depth.

### 4.2 Multi-Pass In-Context Composition

Re-prompting models a multi-turn LLM workflow. In practice, a user may ask the model to complete one step, then feed the output into the next prompt. Each call uses the same model, but the instruction changes from step to step. We formalize this workflow as reprompting in-context function composition. Please see Figure 4 for illustration.

**Theorem 4.2** (Multi-Pass In-Context Composition). *Let $\mathcal{C} = [-b, b]^{d \times n} \subset \mathbb{R}^{d \times n}$, and let $\mathcal{F}_L := \{f : \mathcal{C} \to \mathcal{C} \mid \mathrm{Lip}(f) \leq L\}$. Fix $m \in \mathbb{N}^+$. Then, for any $\epsilon > 0$, there exists a fixed-weight Transformer $T_{\theta^\star} = \mathrm{Attn}_8$ such that the following holds. For every $f_1, \ldots, f_m \in \mathcal{F}_L$, there exist prompts $P_{f_1}, \ldots, P_{f_m}$ in the format of Definition 3.5. For every $X \in \mathcal{C}$, define the exact trajectory $c_i$ and the re-prompted trajectory $\widehat{c}_i$ recursively by:*

$$c_0(X) := X, \quad c_i(X) := f_i(c_{i-1}(X)), \quad i \in [m],$$

$$\widehat{c}_0(X) := X, \quad \widehat{c}_i(X) := T_{\theta^\star}\left(\begin{bmatrix} \widehat{c}_{i-1}(X) & 0_{d \times 1} \\ I_n & 0_{n \times 1} \\ 0_{1 \times n} & 1 \\ P_{f_i} \end{bmatrix}\right)_{:, 1:n},$$

*for $i \in [m]$. Then, for every $i \in [m]$,*

$$\mathrm{d}_p(\widehat{c}_i, c_i) \leq \epsilon.$$

*The emulator weights $\theta^\star$ and the prompt shape depend only on $\mathcal{C}, L, p, d, n, m, \epsilon$, not on $f_1, \ldots, f_m$ or $X$. Only the prompt entries $\{P_{f_i}\}_{i \in [m]}$ depend on the subroutines $f_1, \ldots, f_m$.*

*Proof.* Please see Section E.10 for a detailed proof. □

This gives the multi-pass counterpart of in-context compositional generalization. The same frozen Transformer executes each subprogram, passes its output to the next call, and builds the composite computation across turns. Compared with one-pass composition, multi-pass keeps the model depth fixed and pays through the number of model calls.

**Comparison with Existing Works.** Xiong et al. (2025) characterize Transformers as parallel processors that execute independent tasks in one prompt. In contrast, many algorithmic workflows involve sequential composition: the output of one subroutine becomes the input to the next. Our results target this sequential regime by establishing in-context function composition. Technically, Xiong et al. (2025) rely on the ReLU-Transformer framework of Bai et al. (2023) and define tasks through input-output datasets. Our results apply to standard softmax Transformers and use prompts that encode program-like instructions, such as rules, code, and algorithmic specifications. Our weight encoding provides a formal abstraction of such program descriptions. Thus, while Xiong et al. (2025) emphasize parallel multi-task behavior, our results show how a frozen softmax Transformer composes prompt-encoded subprograms in context.

**Concluding Discussions.** Please see Section A

## Impact Statement

By the theoretical nature of this work, we do not anticipate any negative social impact.

## Acknowledgments

JH thanks Yingyu Liang for drawing his attention to compositional generalization during ICML 2025. He also thanks Mimi Gallagher, Sara Sanchez, T.Y. Ball, Dino Feng and Andrew Chen for helpful conversations; and Mingcheng Lu, Venkat Sripad Ganti, and Yi-Chen Lee for collaborations on related problems. The authors thank the anonymous reviewers and program chairs for their careful reading and constructive comments.

JH is partially supported by Northwestern University's Walter P. Murphy Fellowship and Terminal Year Fellowship (Paul K. Richter Memorial Award). Han Liu is partially supported by NIH R01LM1372201, NSF AST-2421845, the Simons Foundation MPS-AI-00010513, AbbVie, Dolby, and the Chan Zuckerberg Biohub Chicago Spoke Award. This work was supported in part by the computational resources and staff contributions of the Quest high-performance computing facility at Northwestern University, jointly supported by the Office of the Provost, the Office for Research, and Northwestern University Information Technology. The content is solely the responsibility of the authors and does not necessarily represent the official views of the funding agencies.

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

# Appendix

# A  Conclusion and Discussion

Section 3 and Section 4 establish two capabilities of one frozen softmax Transformer.

- Section 3 proves prompt-programmable in-context universal approximation. Fix the compact domain, target class, and accuracy. One frozen Transformer model approximates every $f \in \mathcal{F}_L$ by changing only $P_f$. Here $P_f$ specifies the computation, and $X$ supplies the data.

- Section 4 proves prompt-level composition. Namely, prompts for subroutines (i.e., $P_{f_1}, \ldots, P_{f_m}$) are assembled so the frozen Transformer executes their function composition either through one-pass composition with depth growth or through multi-pass with fixed depth.

We now place these results within existing ICL theory. Recall the spectrum Figure 1. Existing ICL theory largely explains two regimes: parametric recall and amortized in-context inference. In both regimes, the procedure template lies in the frozen weights. The prompt selects content or supplies examples. Our results address the prompt-programmed regime. Here the prompt carries the target computation, and the frozen model executes it. To make this distinction precise, we state three desiderata for a *general-purpose in-context* theory of prompt-programmed execution:

1. **Standard Softmax Attention (D1).** The theory studies ordinary softmax attention. It does not replace attention with linearized, hardmax, or otherwise simplified attention dynamics.

2. **One Frozen Model Across Targets (D2).** The theory fixes one model $\theta^\star$ across target computations. Prompts carry target-specific information. The model weights remain fixed.

3. **Explicit Prompt Interface (D3).** The theory gives a target-to-prompt rule $f \mapsto P_f$. It also gives a modular rule for assembling prompts for composite functions.

We use these desiderata to compare three strands of ICL theory. This comparison identifies which part of prompt-conditioned computation each strand explains.

**Notation.** We write $T_\theta$ for the sequence-to-sequence map of the Transformer with parameters $\theta$. Given an instruction $P$ and an input $X$, write $T_\theta(P, X)$ for the model output on the concatenated input $(P, X)$. For a fixed prompt $P$, write $T_{\theta, P}(X) := T_\theta(P, X)$.

**Statistical-Learning/Pattern-Matching Analyses.** These works study a fixed learned predictor $T_\theta(\cdot)$ and analyze generalization and/or behavior under assumptions on a distribution over tasks, inputs, or prompts, often in stylized ICL settings (Li et al., 2023; Garg et al., 2022; Ahuja & Lopez-Paz, 2023; Bhattamishra et al., 2024). They emphasize the distributional generalization of the learned mapping rather than programmable re-targeting across a broad function family. As a result, they do not target the prompt-programmed form of (D2)-(D3): they do not exhibit one fixed $\theta^\star$ that re-targets across a broad function class through $P_f$. They also do not give an explicit target-to-prompt rule $f \mapsto P_f$. Moreover, for analytic tractability, some mechanistic analyses adopt simplified attention models (e.g., linear self-attention (von Oswald et al., 2023; Ahn et al., 2023)), so they address (D1) only partially.

**Task-Specific ICL as Forward-Pass Algorithm Execution.** This line formalizes in-context learning as algorithm execution in the forward pass. Typically, these works place the target algorithm in task-specific weights. The prompt supplies instances, such as training examples or datasets, rather than a program description (Bai et al., 2023; Akyürek et al., 2023; von Oswald et al., 2023; Cheng et al., 2024). Formally, their guarantees often take the form

$$\forall \text{ algorithm } a, \quad \exists \theta_a \quad \text{s.t.} \quad T_{\theta_a}(\mathcal{D}, x) \approx a(\mathcal{D}, x),$$

where $\mathcal{D}$ denotes the in-context data/examples. Because $\theta_a$ varies with $a$, this framework violates (D2). It also omits (D3): its prompts serve as data containers, not as program encodings with a reusable compiler interface. Moreover, many proofs again simplify attention, so those proofs address (D1) only partially (von Oswald et al., 2023; Bai et al., 2023).

**Expressivity, Turing Completeness, and Prompting.** Another line studies the computational power of Transformers. Expressivity and Turing-completeness results show that Transformers represent rich function classes or simulate computation (Pérez et al., 2021; Wang et al., 2023; Qiu et al., 2025; Jiang et al., 2025; Li & Wang, 2025). These results make important progress on general computation. However, their interface differs from ours. Many results use autoregressive generation, CoT steps, growing context length, or discrete string computation. They do not establish the continuous prompt-programmed approximation statement studied here:

$$\exists \theta^\star, \quad \forall f, \quad \exists P_f \quad \text{s.t.} \quad T_{\theta^\star}(P_f, X) \approx f(X).$$

Thus, they do not address (D2) in the same sense: one fixed model changes target computations through prompts over a broad continuous function class. They also do not address (D3) in our sense: they do not give a reusable compiler $f \mapsto P_f$ for continuous sequence-to-sequence maps, nor a modular rule that assembles $P_f$ and $P_g$ into a prompt for $g \circ f$. Some of these results replace softmax attention with hardmax or other variants, and therefore leave (D1) open (Pérez et al., 2021; Qiu et al., 2025). This quantifier order explains why computational expressivity alone does not imply prompt-programmable in-context approximation.

**Where Our Results Fit?** Our results meet all three desiderata. We operate with standard softmax attention (D1). We fix a single model $\theta^\star$ and show that prompts $P_f$ steer the model to realize broad function classes $f \in \mathcal{F}_L$ (D2). We also provide an explicit prompt interface through surrogate weight encodings (D3). For compositional generalization, we provide mechanisms to assemble a composite prompt from subfunction prompts and controls error propagation. Thus, our results shed light on prompt-programmable computations in Transformers beyond fixed inference rules, task-specific weights, and nonconstructive expressivity.

## B  Related Work

**From In-Context Algorithm Execution to Prompt-Programmed Emulation.** Much of ICL theory starts from a fixed inference rule. The prompt supplies examples or data, and the Transformer applies an estimator in the forward pass. Prior works show that Transformers implement or approximate least squares, gradient descent, preconditioned gradient descent, algorithm selection, and related learning rules (Akyürek et al., 2023; von Oswald et al., 2023; Bai et al., 2023; Ahn et al., 2023; Li et al., 2023; Cheng et al., 2024; Wu et al., 2025). In these works, the weights encode the procedure, while the prompt supplies the data.

A second line makes the program more explicit. Giannou et al. (2023) encode programs in input tokens and execute them with looped Transformers. Qiu et al. (2025) prove a Turing-completeness result for prompting. Closest to us, Hu et al. (2026b) give a minimal softmax-attention construction for fixed-weight in-context algorithm emulation. Their construction uses prompt-induced query-key margins to make one frozen Transformer follow an encoded attention computation. This gives a "one fixed model for many programs" mechanism for attention-implementable algorithms. Our work keeps the fixed-weight and softmax setting. It extends the scope from attention-implementable algorithms to continuous sequence-to-sequence functions, and from one prompt program to prompt-level composition.

**From Example-Conditioned ICL to In-Context Universal Approximation.** Another line asks whether one Transformer supports many functions in context. Here the context acts as data. Li et al. (2025) study the example-conditioned regime. Their prompt supplies input-output examples. A fixed Transformer then recovers coefficients over universal features through a Lasso-like computation. Thus, the prompt identifies a target function from examples. It does not provide an explicit procedure for the target function. Furuya et al. (2025) study continuous in-context mappings where the context appears as a token distribution, and prove universality even when the number of context tokens varies. These works develop approximation theory for example-conditioned or context-conditioned prediction. In contrast, we establish a different form of in-context universal approximation. Our full prompt may also contain data examples through $X$. But these examples play a different role. They are inputs to the computation, not the mechanism that specifies the target function. The target computation is specified by $P_f$. One frozen softmax Transformer reads $P_f$ and executes the encoded computation on $X$. Thus, this shifts in-context universal approximation from function identification to program execution. It also makes composition natural: prompts for simple maps become reusable blocks for prompts of composite maps.

**Prompting, Prefix Tuning, and Function Encoding.** A neighboring line studies how prompts or prefixes encode target functions more directly. Wang et al. (2023) study universality and limitations of prompt tuning. Petrov et al. (2024b) show that a pretrained Transformer becomes a universal approximator when one tunes the prefix. Their construction treats attention as kernel interpolation. The prefix stores reference inputs and target values $(u_i, f(u_i))$, and the model interpolates from these values at the query. Hu et al. (2025a) sharpen this direction by proving universality, capacity, and efficiency results for prompt tuning in simple Transformer architectures. Petrov et al. (2024a) extend this idea to fully recurrent models. Recent work also studies memory limitations of prompt tuning: Meyer et al. (2025) show that the information stored by a prompt has intrinsic limits, even as the context length grows.

This line is close in spirit because it changes the prompt or prefix while keeping the model fixed. It already shows that prompts carry rich function information. In contrast to this work, the interface differs. Their prompt or prefix acts as task-specific side information, often as trainable prompt tokens or an interpolation table. Our prompt acts as a program encoding for a surrogate computation. For a composite target $g \circ f$, an interpolation-table prompt typically stores values of

$g \circ f$. It does not assemble $P_f$ and $P_g$ as reusable subprograms. Our composition theorem does.

**From Multi-Task Superposition to Sequential Composition.** Xiong et al. (2025) study pretrained Transformers under mixed-task demonstrations. A single prompt contains examples from several tasks, and the model performs several in-context tasks in one inference call. Their theory builds on the ReLU-Transformer framework of Bai et al. (2023). They provide a parallel mechanism for performing multi-tasking ICL in superposition. In contrast, our composition theorem studies a sequential mechanism for provably composing sub-tasks sequentially. Each subroutine consumes the output of the previous subroutine. The prompt supplies procedure descriptions, such as rules, code, or algorithmic specifications. Our weight encoding abstracts this program text.

Recent chain-of-thought theory gives another view of serial computation (Merrill & Sabharwal, 2024; Li et al., 2024; Huang et al., 2025). These works study how generated intermediate tokens increase Transformer expressivity or length generalization. Their analyses focus on formal languages, circuit complexity, or synthetic state-tracking tasks. Our results give a continuous-function counterpart. Prompts specify reusable subroutines, and one frozen softmax Transformer executes their composition either in one completion or through re-prompting.

**Transformer Universal Approximation and the Two Routes.** The theory of Transformer universal approximation property (UAP) first develops through the contextual-mapping view. Yun et al. (2020a) prove that standard Transformers approximate continuous permutation-equivariant sequence-to-sequence functions on compact domains. With positional encodings, their results also extend the result beyond permutation equivariance. Kajitsuka & Sato (2024) refine this view by analyzing softmax attention directly. They show that one softmax-attention layer with low-rank weights gives a contextual map, while one-layer hardmax attention does not. This reduces the depth needed in UAP constructions. In the same vein, Hu et al. (2025a) show that single-head, single-layer softmax attention with any-rank weight matrices gives a contextual map in their UAP construction. They also characterize a width-depth tradeoff in the FFN part of the construction.

A second route views softmax attention as interpolation. Hu et al. (2026a) use large inverse temperature so that softmax acts as a near-argmax selector over interpolation points. This gives universal approximation with attention layers alone. Other recent works broaden Transformer UAP theory. Liu et al. (2026) prove UAP for single-layer, single-head self-attention and cross-attention with minimal attached structures. Cheng et al. (2026) give a unified framework for UAP in transformer-type architectures. These works show when a Transformer class contains an approximator with target-dependent weights. In contrast, our work adds a prompt-programming layer. We encode those target-dependent weights into a prompt and use one frozen emulator to execute the encoded computations. Consequently, this allows use to study in-context univeral approximation and in-context composition generalization.

**Approximation Rates and Other Expressivity Results.** Several works derive quantitative rates under stronger structural assumptions. Takakura & Suzuki (2023) study sequence-to-sequence functions on infinite-length inputs. They assume a shift-invariant input distribution and shift-equivariant targets with mixed or anisotropic smoothness. Jiang & Li (2024) establish Jackson-type approximation rates for single-layer, single-head Transformers under measured-complexity assumptions on the target function. Other works study related expressivity questions, including sparse Transformers (Yun et al., 2020b), probabilistic Transformers (Kratsios, 2023), attention-matrix limitations (Bhojanapalli et al., 2020), expressive flexibility of self-attention matrices (Likhosherstov et al., 2023), variable creation in self-attention (Edelman et al., 2022), and finite-precision output accessibility (Meyer et al., 2026). These results complement our work. They study what Transformer architectures represent or generate. We study how one fixed softmax Transformer changes the represented computation through the prompt.

# C  Notation and Complexity Tables, Illustration Figures and Function-Algorithm-Program Relation

## C.1  Notation and Complexity Tables

*Table 1.* Notations and Symbols

| Notation | Description |
|---|---|
| $a$ | Lowercase letters for vectors |
| $A$ | Uppercase letters for matrices |
| $A_{i,:}$ | $i$-th row of $A$ |
| $A_{:,i}$ | $i$-th column of $A$ |
| $A_{:,i:j}$ | Sub-matrix of columns $i$ through $j$ |
| $\mathbb{1}_d$ | $d$ dimensional all-ones vector |
| $\mathbb{R}$ | Set of real numbers |
| $\mathbb{N}^+$ | Set of positive integers |
| $[n]$ | $[n] \coloneqq \{1, 2, \dots, n\}$ |
| $\mathcal{C}$ | Compact domain |
| $d$ | Token dimension |
| $n$ | Length of input sequence |
| $H$ | Number of attention heads |
| $X = [x_1, \dots, x_n]$ | Matrix of input sequence |
| $x_i$ | $i$-th token (column) of $X$ |
| $\|x\|_2$ | $\ell_2$ norm of $x$ |
| $\|X\|_p$ | Entry-wise $\ell_p$ norm of $X$ |
| $\|X\|_\infty$ | $\|X\|_\infty \coloneqq \max_{i,j} |X_{ij}|$ |
| $\mathrm{d}_p(f_1, f_2)$ | $\left(\int_{\mathcal{C}} \|f_1(X) - f_2(X)\|_p^p \mathrm{d}X\right)^{\frac{1}{p}}$ |
| $\mathbb{1}_{\{\text{condition}\}}$ | The indicator |
| $\sigma_\beta$ | Softmax with inverse temperature $\beta$ |
| ReLU | ReLU activation |
| FF | Feed-forward layer |
| $\mathrm{Attn}_l$ | $l$-layer attention |
| $\mathrm{Attn}_{l\text{-}k}$ | The $k$-th layer of an $l$-layer attention |
| $\mathrm{Attn}_{l\text{-}k}^{(h)}$ | The $h$-th head of the $k$-th layer from an $l$-layer attention |
| $\mathrm{Attn}_{l,k}$ | The $k$-th $l$-layer attention |
| $\mathrm{Attn}^{\mathrm{Res}}$ | Attention with a residual connection |

## C.2  Illustration Figures

*Table 2.* **Attention Emulator Complexity in Theorems 3.1 and 3.2.** We report the worst-case model complexity as a function of the granularity $g$ (see Section D.2 for definition.) Here $d$ is the token dimension and $n$ is the sequence length of the input $X \in \mathbb{R}^{d \times n}$. We write $\epsilon_{\text{flow}}$ for the flow-through error (see (80) for definition). The scaling is similar to prior universal approximation results (Yun et al., 2020a; Kajitsuka & Sato, 2024; Hu et al., 2026a). We remark such exponential dependence reflects the generality of the continuous function class. Please see (Yun et al., 2020a, Section 4.4) for relevant discussion.

| | **Theorem 3.1** | **Theorem 3.2** |
|---|---|---|
| Context length | $n$ (independent of error) | $n+1$ (independent of error) |
| Embedding dimension | $O(dg^{dn})$ | $O(dng^{2dn})$ |
| Head count | $O(g^{dn}/n)$ | $O(g^{2dn})$ |
| Softmax sharpness ($\beta$) | Flow-through heads require $O(\log n + \log \frac{1}{\epsilon_{\text{flow}}})$; task heads inherit the target-attention $\beta$. | Same as Theorem 3.1. |

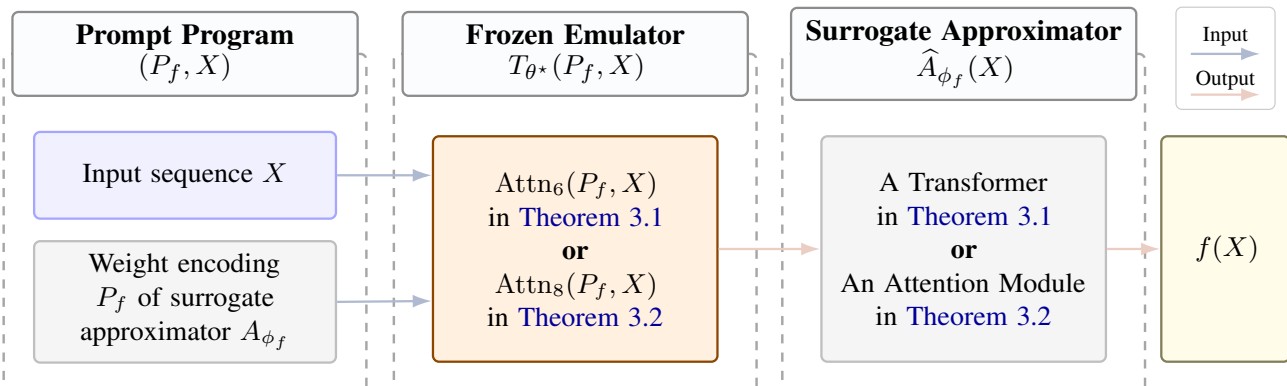

*Figure 2.* **In-Context Universal Approximation Overview (Section 3).** Our construction takes three steps. **Step 1.** We use a surrogate approximator $A_{\phi_f}$ to approximate $f$ at $X$ (see (3)). In Theorem 3.1, it's a Transformer. In Theorem 3.2, it's an attention block. **Step 2.** We approximate surrogate $A_{\phi_f}$ by another frozen Transformer $T_{\theta^\star}$ called emulator. Specifically, we encode the weights $\phi_f$ of the surrogate into $P_f$. The frozen emulator $T_{\theta^\star}$ receives $(P_f, X)$, and executes the encoded computation on the input data as $T_{\theta^\star}(P_f, X) = \widehat{A}_{\phi_f}(X)$. Its output realizes either an emulated Transformer surrogate $\widehat{\text{FF}}_2 \circ \widehat{\text{Attn}^{\text{Res}}} \circ \widehat{\text{FF}}_1(X)$ or an emulated attention surrogate $\widehat{\text{Attn}_4}(X)$. **Step 3.** Chain the above two steps we approximate the target function as in (5). The weights of the emulator $T_{\theta^\star}$ stay fixed across functions. Only the instruction $P_f$ changes with $f$. Since this procedure holds for all $f \in \mathcal{F}_L$, it is indeed in-context universal approximation.

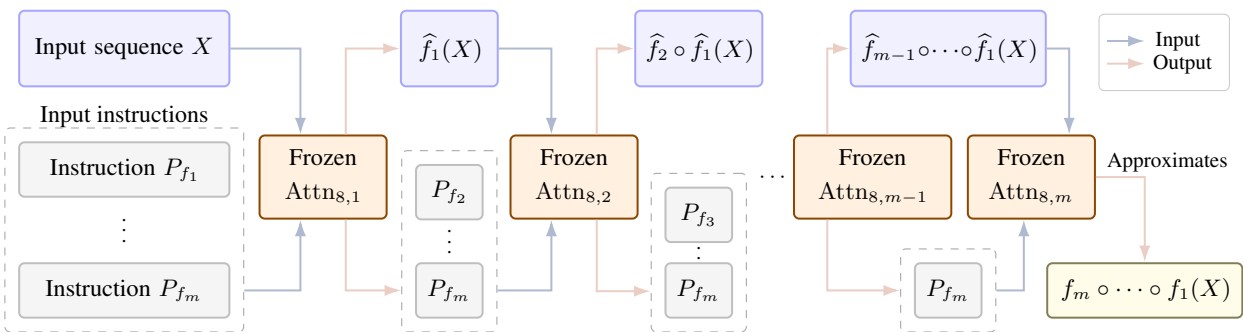

*Figure 3.* **Illustration of the Construction in Theorem 4.1.** Consider a function composition of length $m$, realized at $X$. We prepare the sequence $X$ and instructions $P_{f_1}, \ldots, P_{f_m}$ as initial inputs. For $i \in [m]$, $P_{f_i}$ encodes the computation instruction required to implement $f_i$. An $8m$-layer attention emulator $\text{Attn}_{8m}$ processes these inputs in a single forward pass. $\text{Attn}_{8m}$ consists of $m$ eight-layer attention, written as $\text{Attn}_{8,m} \circ \cdots \circ \text{Attn}_{8,1}$. The block $\text{Attn}_{8,i}$ uses the instruction $P_{f_i}$ to update the current data from $c_{i-1}(X)$ to an approximation of $c_i(X)$. Also, $\text{Attn}_{8,i}$ propagates $P_{f_{i+1}}, \ldots, P_{f_m}$. Thus, the full composition $f_m \circ \cdots \circ f_1(X)$ is realized in one forward pass.

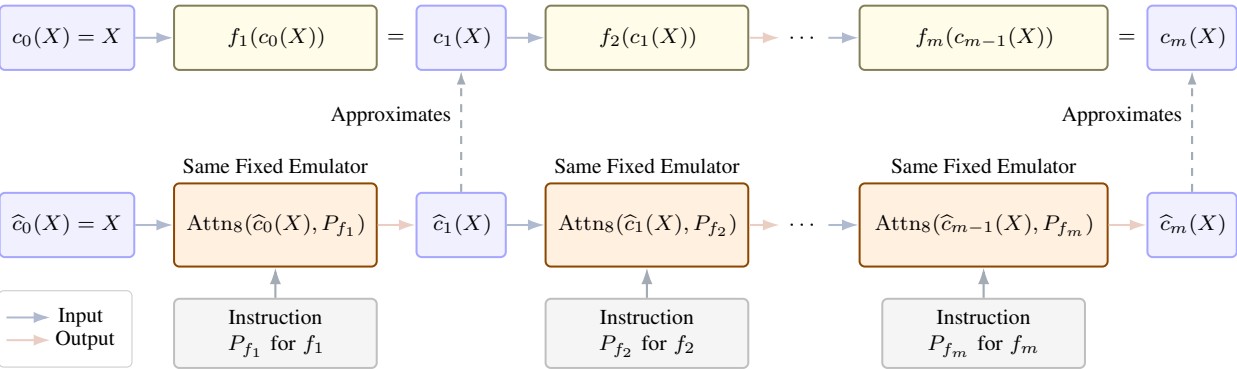

*Figure 4.* **Illustration of the Construction in Theorem 4.2** . Consider a function composition of length $m$, realized at $X$. For $i \in [m]$, the top row shows the target composition $c_i(X) = f_i(c_{i-1}(X))$, starting from $c_0(X) = X$. The bottom row shows its approximation $\widehat{c}_i(X)$. We obtain $\widehat{c}_i(X)$ by prompting the emulator with $\widehat{c}_{i-1}(X)$ and the instruction $P_{f_i}$. $P_{f_i}$ encodes the computation required to implement $f_i$. The emulator reads $P_{f_i}$, executes the computation on $\widehat{c}_{i-1}(X)$, and outputs $\widehat{c}_i(X)$. Repeating this procedure for $i = 1, \ldots, m$ realizes the full composition by re-prompting a single frozen emulator.

### C.3 Functions, Algorithms, and Programs

We now clarify the relation among *function*, *algorithm*, and *program*. A function specifies an input-output map. An algorithm specifies a procedure for computing this map. A program encodes the procedure for a fixed executor. An algorithm may induce one function or a composition of functions. This paper studies guarantees at the function level. Thus, our results apply to any procedure that realizes the same input-output map.

In our setting, a prompt is the full input sequence to the frozen Transformer. It contains an instruction part and a data part. We write these two parts as $(P_f, X)$. The block $P_f$ specifies the target computation. The block $X$ supplies the data. Thus, $T_{\theta^\star}(P_f, X) \approx f(X)$ means that the frozen Transformer $T_{\theta^\star}$ executes the instruction $P_f$ on the data $X$.

We illustrate the distinction with linear regression. Let $\mathcal{D} = \{(x_i, y_i)\}_{i=1}^n$ be a training set, and let $x_{\text{test}}$ be a test point. The algorithm fits a linear coefficient from $\mathcal{D}$ and predicts at $x_{\text{test}}$. We describe this computation by two maps. The first map sends the dataset to an estimator

$$f_{\text{est}}(\mathcal{D}) = \widehat{w}(\mathcal{D}),$$

where $\widehat{w}(\mathcal{D})$ denotes the estimated coefficient. The second map sends the estimator and test point to a prediction

$$f_{\text{pred}}(\widehat{w}, x_{\text{test}}) = x_{\text{test}}^\top \widehat{w}.$$

Their composition gives the final prediction

$$f_{\text{pred}}(f_{\text{est}}(\mathcal{D}), x_{\text{test}}).$$

Different algorithms may realize the estimator map $f_{\text{est}}$ differently:

- **Gradient Descent.** Define the one-step update

$$f_{\text{step}, \mathcal{D}}(w) = w - \eta \nabla \mathcal{L}(w; \mathcal{D}).$$

  Starting from $w_0$, $T$-step estimator gives

$$f_{\text{GD}}(\mathcal{D}) = f_{\text{step}, \mathcal{D}}^{\circ T}(w_0).$$

  Here $f_{\text{step}, \mathcal{D}}^{\circ T}$ denotes the $T$-fold composition of $f_{\text{step}, \mathcal{D}}$.

- **Closed-Form Estimator.** Let $X_{\mathcal{D}}$ and $y_{\mathcal{D}}$ denote the design matrix and response vector from $\mathcal{D}$. Assume $X_{\mathcal{D}}^\top X_{\mathcal{D}}$ is invertible. Define

$$f_{\text{CF}}(\mathcal{D}) = (X_{\mathcal{D}}^\top X_{\mathcal{D}})^{-1} X_{\mathcal{D}}^\top y_{\mathcal{D}}.$$

Both $f_{\text{GD}}$ and $f_{\text{CF}}$ produce a coefficient for $f_{\text{pred}}$. This example separates the three roles used throughout the paper. A function describes input-output behavior. An algorithm describes how to compute this behavior. A prompt program $P_f$ gives the algorithm instruction and data to the frozen Transformer. Section 3 studies how a prompt specifies a target computation. Section 4 studies how simple prompted computations chain together.

# D  Supplementary Theoretical Backgrounds

This section includes some known results we build on.

## D.1  Universal Approximation of Transformer and Attention-Only Model

We state the universal approximation result from (Kajitsuka & Sato, 2024; Hu et al., 2025a; 2026a).

**Theorem D.1** (Transformers Universal Approximation, Theorem B.1 of (Hu et al., 2025a)). *Let $\mathcal{T}_2$ denote the class of two-layer Transformers with self-attention and positional encoding:*

$$\mathcal{T}_2 := \{g : \mathbb{R}^{d\times n} \to \mathbb{R}^{d\times n} \mid g(X) = \mathrm{FF}_2 \circ \mathrm{Attn}^{\mathrm{Res}} \circ \mathrm{FF}_1(X + E_p),\ \ E_p \in \mathbb{R}^{d\times n}\},$$

*where $\mathrm{FF}_1, \mathrm{FF}_2$ and $\mathrm{Attn}^{\mathrm{Res}}$ are feed-forward neural network layers and a single-head self-attention layer as defined in Definitions 2.1 and 2.2, respectively. For $1 \le p < \infty$, let distance $\mathrm{d}_p(\cdot,\cdot)$ be defined as in (2). Then, for any continuous function $f$ defined on a compact domain $\mathcal{C} \in \mathbb{R}^{d\times n}$, and $\epsilon_{\mathrm{u}} > 0$, there exists a Transformer $g \in \mathcal{T}_2$ such that*

$$\mathrm{d}_p(f,g) \le \epsilon_{\mathrm{u}}.$$

**Theorem D.2** (Sequence-to-Sequence Universal Approximation of Multi-Head Softmax Attention, Theorem G.1 of (Hu et al., 2026a)). *Let $1 \le p < \infty$. Let $\mathcal{C} \subset \mathbb{R}^{d\times n}$ be a compact domain of input sequences. Let $f : \mathcal{C} \to \mathbb{R}^{d\times n}$ be a continuous sequence-to-sequence function. Let $X_a \in \mathbb{R}^{(d+n+1)\times(n+1)}$ be the augmented input*

$$X_a := \begin{bmatrix} X & 0_{d\times 1} \\ I_n & 0_{n\times 1} \\ 0_{1\times n} & 1 \end{bmatrix} \in \mathbb{R}^{(d+n+1)\times(n+1)}.$$

*Then, for any $\epsilon > 0$, there exists a network $A_{\phi_f}$ composed of four multi-head attention layers such that the induced sequence-to-sequence function $\widetilde{A}_{\phi_f}(X) := A_{\phi_f}(X_a)_{:,1:n}$ satisfies*

$$\mathrm{d}_p(\widetilde{A}_{\phi_f}, f) \le \epsilon.$$

## D.2  Explicit Weight Construction of Theorem D.1 Given Function $f$

In this section, we make the construction behind Theorem D.1 explicit and prepare the two ingredients for the proof of Theorem 3.1:

- **Sections D.2.1 to D.2.3:** an explicit construction of the Transformer universal approximator in Theorem D.1, i.e., $\mathrm{FF}_2 \circ \mathrm{Attn}^{\mathrm{Res}} \circ \mathrm{FF}_1(X + E_p)$. This construction follows (Hu et al., 2025b, Theorems G.1 and G.2);

- **Section D.2.4:** a uniform finite-grid version of this construction. This allows us to choose one surrogate architecture, one prompt shape, and common weight bounds for all target functions in the class.

These ingredients are handy for later parts in Section E.4 to prove Theorem 3.1.[8]

The surrogate Transformer in Theorem D.1 has the form

$$\mathrm{FF}_2 \circ \mathrm{Attn}^{\mathrm{Res}} \circ \mathrm{FF}_1.$$

Thus the input first passes through a feed-forward layer $\mathrm{FF}_1$, then through one residual self-attention layer $\mathrm{Attn}^{\mathrm{Res}}$, and finally through a second feed-forward layer $\mathrm{FF}_2$.

We describe these three pieces separately. In Section D.2.1 and Section D.2.3, we give the explicit constructions of $\mathrm{FF}_1$ and $\mathrm{FF}_2$, respectively. In Section D.2.2, we give the construction of $\mathrm{Attn}^{\mathrm{Res}}$. Finally, in Section D.2.4, we extract the finite-grid consequence needed later: after the target class and accuracy are fixed, the surrogate dimensions, prompt shape, and relevant bounds can be fixed uniformly, while only the encoded weights depend on the target function.

---

[8]Although the construction is established in (Hu et al., 2025b), we restate the needed parts here to make our in-context construction self-contained and explicit.

### D.2.1 GRID POINTS ON INPUT DOMAIN AND $\mathrm{FF}_1$ CONSTRUCTION

Let $\mathcal{C} := [-b, b]^{d \times n}$ be the compact domain. They first discretize the compact domain by forming a uniform grid with granularity $g$

$$\Gamma_g := -b + \frac{1}{g}\{0, 1, \ldots, 2bg\},$$

and

$$\mathcal{G}_g := \{C \in \mathbb{R}^{d \times n} \mid \forall i \in [d], j \in [n], \ C_{ij} \in \Gamma_g\}.$$

That is, the number of elements in $\mathcal{G}_g$ is $(2bg + 1)^{dn}$.

Then, for the input sequence $X \in \mathbb{R}^{d \times n}$, they define the following $\mathbb{R} \to \mathbb{R}$ function to map each entry of $X$ to discrete values

$$\mathrm{quant}_g(z) := \begin{cases} -b, & z < -b \\ -b + \frac{1}{g}, & -b \le z < -b + \frac{1}{g} \\ \vdots & \vdots \\ b, & b - \frac{1}{g} \le z. \end{cases}$$

They extend $\mathrm{quant}_g$ to an entry-wise map $\mathrm{quant}_g^{d \times n}(X) : \mathbb{R}^{d \times n} \to \mathbb{R}^{d \times n}$ by

$$\mathrm{quant}_g^{d \times n}(X)_{ij} := \mathrm{quant}_g(X_{ij}) \quad \text{for all } i \in [d], \ j \in [n].$$

That is, the function $\mathrm{quant}_g^{d \times n}$ maps $X$ to the grid points defined by $\mathcal{G}_g$.

Next, for $X \in \mathbb{R}^{d \times n} \setminus [-b, b]^{d \times n}$, they define the following $\mathbb{R} \to \mathbb{R}$ function to indicate out-of-domain inputs

$$\mathrm{penalty}(z) := \begin{cases} -2b, & z < -b \\ 0, & z \in [-b, b] \\ -2b, & z > b. \end{cases}$$

Again, they extend the penalty function to an entry-wise map $\mathrm{penalty}^{d \times n}(X) : \mathbb{R}^{d \times n} \to \mathbb{R}^{d \times n}$, where they apply $\mathrm{penalty}(z)$ in an entry-wise manner.

Lastly, they let $B \in \mathbb{R}^{d \times n}$ be a matrix with all entries equal to $b$, and they define $h_1 : \mathbb{R}^{d \times n} \to \mathbb{R}^{d \times n}$ by

$$h_1(X) := \underbrace{\mathrm{quant}_g^{d \times n}(X) + B}_{(A)} + \underbrace{dn \cdot \mathrm{penalty}^{d \times n}(X)}_{(B)} \tag{7}$$

To implement (7), they design the following functions to approximate (7). In addition, these functions are implementable with a feed-forward layer. Thus, they encode the weights of these functions into the first feed-forward layer to form the explicit construction of $\mathrm{FF}_1$.

For the first term in (A) of (7), they design

$$\widetilde{f}_1(z) := -b + \frac{1}{g} \sum_{t=-bg}^{bg-1} \left( \mathrm{ReLU}\left(\frac{z}{\delta} - \frac{t}{\delta g}\right) - \mathrm{ReLU}\left(\frac{z}{\delta} - \frac{t}{\delta g} - 1\right) \right), \tag{8}$$

where $\delta > 0$ determines how quickly each summand goes from 0 to 1, and then they design

$$f_1(z) := \widetilde{f}_1(z) - b \cdot \left( \mathrm{ReLU}\left(\frac{z}{\delta} - \frac{b}{\delta}\right) - \mathrm{ReLU}\left(\frac{z}{\delta} - \frac{b}{\delta} - 1\right) \right) + b \cdot \left( \mathrm{ReLU}\left(-\frac{z}{\delta} - \frac{b}{\delta}\right) - \mathrm{ReLU}\left(-\frac{z}{\delta} - \frac{b}{\delta} - 1\right) \right). \tag{9}$$

For term (B), they design

$$f_2(z) := -2b(\text{ReLU}(\frac{z-b}{\delta}) + \text{ReLU}(\frac{z-b}{\delta} - 1)) - 2b(\text{ReLU}(\frac{-z-b}{\delta}) + \text{ReLU}(\frac{-z-b}{\delta} - 1)). \tag{10}$$

Then, they implement (7) by encoding the weights of (9) and (10) into the following feed-forward neural network $\text{FF}_1$

$$\text{FF}_1 := W_{1,2} \cdot \text{ReLU}(W_{1,1} \cdot X + b_{1,1}\mathbb{1}_n^\top) + b_{1,2}\mathbb{1}_n^\top,$$

where

$$W_{1,1} := \underbrace{\begin{bmatrix} W_{1,1}^{(1)} \\ W_{1,1}^{(2)} \\ \vdots \\ W_{1,1}^{(d)} \end{bmatrix}}_{d(4bg+8)\times d}, \quad b_{1,1} := \underbrace{\begin{bmatrix} b_{1,1}^{(1)} \\ b_{1,1}^{(2)} \\ \vdots \\ b_{1,1}^{(d)} \end{bmatrix}}_{d(4bg+8)\times 1}, \quad W_{1,2} := \underbrace{\begin{bmatrix} W_{1,2}^{(1)} & W_{1,2}^{(2)} & \cdots & W_{1,2}^{(d)} \end{bmatrix}}_{d\times d(4bg+8)}, \quad b_{1,2} := \underbrace{\begin{bmatrix} 0 \\ 0 \\ \vdots \\ 0 \end{bmatrix}}_{d\times 1}, \tag{11}$$

and they set

$$W_{1,1}^{(1)} := \underbrace{\begin{bmatrix} \frac{1}{\delta} & 0 & 0 & \cdots & 0 \\ \frac{1}{\delta} & 0 & 0 & \cdots & 0 \\ \vdots & \vdots & \vdots & \ddots & \vdots \\ \frac{1}{\delta} & 0 & 0 & \cdots & 0 \\ \frac{1}{\delta} & 0 & 0 & \cdots & 0 \\ \frac{1}{\delta} & 0 & 0 & \cdots & 0 \\ \frac{1}{\delta} & 0 & 0 & \cdots & 0 \\ -\frac{1}{\delta} & 0 & 0 & \cdots & 0 \\ -\frac{1}{\delta} & 0 & 0 & \cdots & 0 \\ \frac{1}{\delta} & 0 & 0 & \cdots & 0 \\ \frac{1}{\delta} & 0 & 0 & \cdots & 0 \\ -\frac{1}{\delta} & 0 & 0 & \cdots & 0 \\ -\frac{1}{\delta} & 0 & 0 & \cdots & 0 \end{bmatrix}}_{(4bg+8)\times d}, \quad W_{1,1}^{(2)} := \underbrace{\begin{bmatrix} 0 & \frac{1}{\delta} & 0 & \cdots & 0 \\ 0 & \frac{1}{\delta} & 0 & \cdots & 0 \\ \vdots & \vdots & \vdots & \ddots & \vdots \\ 0 & \frac{1}{\delta} & 0 & \cdots & 0 \\ 0 & \frac{1}{\delta} & 0 & \cdots & 0 \\ 0 & \frac{1}{\delta} & 0 & \cdots & 0 \\ 0 & \frac{1}{\delta} & 0 & \cdots & 0 \\ 0 & -\frac{1}{\delta} & 0 & \cdots & 0 \\ 0 & -\frac{1}{\delta} & 0 & \cdots & 0 \\ 0 & \frac{1}{\delta} & 0 & \cdots & 0 \\ 0 & \frac{1}{\delta} & 0 & \cdots & 0 \\ 0 & -\frac{1}{\delta} & 0 & \cdots & 0 \\ 0 & -\frac{1}{\delta} & 0 & \cdots & 0 \end{bmatrix}}_{(4bg+8)\times d}, \quad \cdots, \quad W_{1,1}^{(d)} := \underbrace{\begin{bmatrix} 0 & 0 & 0 & \cdots & \frac{1}{\delta} \\ 0 & 0 & 0 & \cdots & \frac{1}{\delta} \\ \vdots & \vdots & \vdots & \ddots & \vdots \\ 0 & 0 & 0 & \cdots & \frac{1}{\delta} \\ 0 & 0 & 0 & \cdots & \frac{1}{\delta} \\ 0 & 0 & 0 & \cdots & \frac{1}{\delta} \\ 0 & 0 & 0 & \cdots & \frac{1}{\delta} \\ 0 & 0 & 0 & \cdots & -\frac{1}{\delta} \\ 0 & 0 & 0 & \cdots & -\frac{1}{\delta} \\ 0 & 0 & 0 & \cdots & \frac{1}{\delta} \\ 0 & 0 & 0 & \cdots & \frac{1}{\delta} \\ 0 & 0 & 0 & \cdots & -\frac{1}{\delta} \\ 0 & 0 & 0 & \cdots & -\frac{1}{\delta} \end{bmatrix}}_{(4bg+8)\times d}, \tag{12}$$

with

$$b_{1,1}^{(i)} := \underbrace{\begin{bmatrix} \frac{b}{\delta} \\ \frac{b}{\delta} - 1 \\ \vdots \\ -\frac{b}{\delta} + \frac{1}{\delta g} \\ -\frac{b}{\delta} - 1 + \frac{1}{\delta g} \\ -\frac{b}{\delta} \\ -\frac{b}{\delta} - 1 \\ -\frac{b}{\delta} \\ -\frac{b}{\delta} - 1 \\ -\frac{b}{\delta} \\ -\frac{b}{\delta} - 1 \\ -\frac{b}{\delta} \\ -\frac{b}{\delta} - 1 \end{bmatrix}}_{(4bg+8)\times 1}. \tag{13}$$

That is, $W_{1,1}^{(i)}$ has all-zero columns except for the $i$-th column. The entries are $1/\delta$ in the rows corresponding to $\text{ReLU}(\frac{z}{\delta} - \cdot)$ terms in (9) and (10), and $-1/\delta$ in the rows corresponding to $\text{ReLU}(-\frac{z}{\delta} - \cdot)$ terms.

Then, for all $i \in [d]$, they construct $W_{1,2}^{(i)}$ as

$$W_{1,2}^{(i)} := \underbrace{e_i^{(d)}}_{d \times 1} \cdot \underbrace{\begin{bmatrix} \frac{1}{g} & -\frac{1}{g} & \cdots & \frac{1}{g} & -\frac{1}{g} & -b & b & b & -b & -2bdn & -2bdn & -2bdn & -2bdn \end{bmatrix}}_{1 \times (4bg+8)}.$$

That is, $W_{1,2}^{(i)}$ has only one non-zero row $i$.

### D.2.2 Contextual-Mapping Attention Construction

Next, to prepare for the construction of the attention layer, we need to describe the properties of the output from term (A) in (7). For this purpose, we restate two helper definitions from (Hu et al., 2025b).

**Definition D.1** (Vocabulary, Definition G.1 of (Hu et al., 2025b)). *For each $i \in [N]$, define the $i$-th vocabulary set by*

$$\mathcal{V}^{(i)} := \bigcup_{k \in [n]} X_{:,k}^{(i)} \subset \mathbb{R}^d,$$

*and define the whole vocabulary set by*

$$\mathcal{V} := \bigcup_{i \in [N]} \mathcal{V}^{(i)} \subset \mathbb{R}^d.$$

**Definition D.2** (Tokenwise Separateness, Definition G.2 of (Hu et al., 2025b)). *Let $X^{(1)}, \ldots, X^{(N)} \in \mathbb{R}^{d \times n}$ be embeddings. We say that $X^{(1)}, \ldots, X^{(N)}$ are tokenwise $(\gamma_{\min}, \gamma_{\max}, \epsilon)$-separated if the following three conditions hold:*

(i) *For any $i \in [N]$ and $k \in [n]$, $\|X_{:,k}^{(i)}\|_2 > \gamma_{\min}$.*

(ii) *For any $i \in [N]$ and $k \in [n]$, $\|X_{:,k}^{(i)}\|_2 < \gamma_{\max}$.*

(iii) *For any $i, j \in [N]$ and $k, l \in [n]$, if $X_{:,k}^{(i)} \neq X_{:,l}^{(j)}$ holds, then $\|X_{:,k}^{(i)} - X_{:,l}^{(j)}\|_2 > \epsilon$.*

*Note that when only conditions (ii) and (iii) hold, we denote this as $(\gamma, \epsilon)$-separateness. Moreover, if only condition (iii) holds, we denote it as $(\epsilon)$-separateness.*

They use $\mathcal{G}_g^{\circ}$ to denote the set of all possible output sequences from term (A) in (7). Also, let $\widetilde{\mathcal{G}}_g$ be

$$\widetilde{\mathcal{G}}_g := \{G \in \mathcal{G}_g^{\circ} \mid G_{:,i} \neq G_{:,j} \text{ for all } i, j \in [n] \text{ and } i \neq j\}.$$

By construction, $\widetilde{\mathcal{G}}_g$ is finite, and the sequences in $\widetilde{\mathcal{G}}_g$ are tokenwise $(1/g, 2b\sqrt{d}, 1/g)$-separated.

Besides, let $\mathcal{V}$ be the set of all tokens from $\widetilde{\mathcal{G}}_g$. Since $\widetilde{\mathcal{G}}_g$ is finite, $\mathcal{V}$ is also finite.

Again, to prepare for the construction of the attention layer, we restate a helper lemma from (Park et al., 2021).

**Lemma D.1** (Lemma 13 of (Park et al., 2021)). *For any finite subset $\mathcal{X} \subset \mathbb{R}^d$, there exists at least one unit vector $u \in \mathbb{R}^d$ such that*

$$\frac{1}{|\mathcal{X}|^2} \sqrt{\frac{8}{\pi d}} \|x - x'\|_2 \leq |u^\top (x - x')| \leq \|x - x'\|_2 \tag{14}$$

*for any $x, x' \in \mathcal{X}$.*

Since $\mathcal{V}$ is a finite set, by Lemma D.1, there exists at least one unit vector $u \in \mathbb{R}^d$ such that (14) holds for any two tokens from $\mathcal{V}$.

Now, we restate their construction of the attention layer.

1. We state the dimension first. They let $W_K, W_Q, W_V \in \mathbb{R}^{s \times d}$ and $W_O \in \mathbb{R}^{d \times s}$.

2. Then, set the rank of the weight matrices to be $\rho$. Any choice of $\rho$ that satisfies $1 \leq \rho \leq \min\{d, s\}$ works.

3. Next, for each $i \in [\rho]$, pick vectors $a_i, b_i \in \mathbb{R}^s$ such that

$$|a_i^\top b_i| = (|\mathcal{V}| + 1)^4 \frac{d\kappa}{\epsilon\gamma_{\min}},$$

where they set $\kappa = (4 \ln n)/\beta$. Also, $\epsilon = \gamma_{\min} = 1/g$ here.

4. Then, for each $i \in [\rho]$, choose $u_i, w_i \in \mathbb{R}^d$ to be unit vectors. Moreover, they require that there be at least one index $i^\star \in [\rho]$ such that $u_{i^\star} = u$ and $w_{i^\star} = u$. For $[\rho] \setminus \{i^\star\}$, any unit vector in $\mathbb{R}^d$ works.

5. Then, they construct the key and query weight matrices as follows

$$W_K := \sum_{i=1}^{\rho} a_i u_i^\top \in \mathbb{R}^{s \times d}, \quad W_Q := \sum_{i=1}^{\rho} b_i w_i^\top \in \mathbb{R}^{s \times d}.$$

6. Next, for each $i \in [\rho]$, pick an arbitrary non-zero vector $c_i \in \mathbb{R}^s$ and $o_i \in \mathbb{R}^d$, except for $o_1$, where they set $o_1 = u$. Then, they construct the value weight matrix as

$$W_V := \sum_{i=1}^{\rho} c_i o_i^\top \in \mathbb{R}^{s \times d}.$$

7. Finally, they require $W_O$ to satisfy the following constraint

$$\|W_O c_i\|_2 = \frac{\epsilon}{4\rho\gamma_{\max}}, \tag{15}$$

where $\epsilon = 1/g$ and $\gamma_{\max} = 2b\sqrt{g}$. They provide one explicit way to achieve (15). For each $i \in [\rho]$, they pick a vector $d_i \in \mathbb{R}^d$ such that

$$\|d_i\|_2 = \frac{\epsilon}{4\rho^2\gamma_{\max}\|c_i\|_2^2},$$

with the same $\epsilon = 1/g$ and $\gamma_{\max} = 2b\sqrt{g}$. Then, they construct

$$W_O := \sum_{i=1}^{\rho} d_i c_i^\top \in \mathbb{R}^{d \times s}.$$

This completes the construction of the attention layer.

### D.2.3 BUMP FUNCTION AND FF$_2$ CONSTRUCTION

Regarding the construction of FF$_2$, they first consider a fixed $\bar{C} \in \widetilde{\mathcal{G}}_g$, and they define

$$U := \mathrm{Attn}^{\mathrm{Res}}(\bar{C}),$$

where $\mathrm{Attn}^{\mathrm{Res}}$ is the single-head attention with a residual connection, specified by the weight matrices constructed in the previous steps.

Additionally, they define

$$S := \mathrm{Attn}^{\mathrm{Res}} \circ \mathrm{FF}_1(X).$$

For a fixed $\bar{C}$, they define $h_2 : \mathbb{R}^{d \times n} \to \mathbb{R}^{d \times n}$, where

$$h_2(S)_{ij} := f(\bar{C} - B)_{ij} \cdot \mathbb{1}_{U_{ij} = S_{ij}} \quad \text{for all } i \in [d], j \in [n],$$

and $f$ is the target function.

Then, they use a bump function $\mathrm{bump}_R : \mathbb{R}^{d \times n} \to \mathbb{R}^{d \times n}$ to achieve the indicator in $h_2$:

$$\mathrm{bump}_R(S)_{ij} := \mathrm{ReLU}(R_{\mathrm{FF}}(S_{ij} - U_{ij}) - 1) - 2\mathrm{ReLU}(R_{\mathrm{FF}}(S_{ij} - U_{ij})) + \mathrm{ReLU}(R_{\mathrm{FF}}(S_{ij} - U_{ij}) + 1),$$

and they choose $R_{\text{FF}}$ to scale like $R_{\text{FF}} = O(\exp(640\beta(bg)^{4d+2}d^2\ln n))$. This bump function is implementable by a feed-forward layer.

Next, for a fixed $\bar{C}$ and $U = \text{Attn}^{\text{Res}}(\bar{C})$, they use part of $\text{FF}_2$ to implement $h_2$. Specifically, they construct

$$
W_{2,1}^{(i)} := \underbrace{\begin{bmatrix}
R_{\text{FF}} & 0 & 0 & \cdots & 0 \\
R_{\text{FF}} & 0 & 0 & \cdots & 0 \\
R_{\text{FF}} & 0 & 0 & \cdots & 0 \\
0 & R_{\text{FF}} & 0 & \cdots & 0 \\
0 & R_{\text{FF}} & 0 & \cdots & 0 \\
0 & R_{\text{FF}} & 0 & \cdots & 0 \\
0 & 0 & R_{\text{FF}} & \cdots & 0 \\
0 & 0 & R_{\text{FF}} & \cdots & 0 \\
0 & 0 & R_{\text{FF}} & \cdots & 0 \\
\vdots & \vdots & \vdots & \ddots & \vdots \\
0 & 0 & 0 & \cdots & R_{\text{FF}} \\
0 & 0 & 0 & \cdots & R_{\text{FF}} \\
0 & 0 & 0 & \cdots & R_{\text{FF}} \\
\vdots & \vdots & \vdots & & \vdots \\
R_{\text{FF}} & 0 & 0 & \cdots & 0 \\
R_{\text{FF}} & 0 & 0 & \cdots & 0 \\
R_{\text{FF}} & 0 & 0 & \cdots & 0 \\
0 & R_{\text{FF}} & 0 & \cdots & 0 \\
0 & R_{\text{FF}} & 0 & \cdots & 0 \\
0 & R_{\text{FF}} & 0 & \cdots & 0 \\
0 & 0 & R_{\text{FF}} & \cdots & 0 \\
0 & 0 & R_{\text{FF}} & \cdots & 0 \\
0 & 0 & R_{\text{FF}} & \cdots & 0 \\
\vdots & \vdots & \vdots & \ddots & \vdots \\
0 & 0 & 0 & \cdots & R_{\text{FF}} \\
0 & 0 & 0 & \cdots & R_{\text{FF}} \\
0 & 0 & 0 & \cdots & R_{\text{FF}}
\end{bmatrix}}_{3dn \times d}, \quad
b_{2,1}^{(i)} := \underbrace{\begin{bmatrix}
-R_{\text{FF}} \cdot U_{11} \\
-R_{\text{FF}} \cdot U_{11} - 1 \\
-R_{\text{FF}} \cdot U_{11} + 1 \\
-R_{\text{FF}} \cdot U_{21} \\
-R_{\text{FF}} \cdot U_{21} - 1 \\
-R_{\text{FF}} \cdot U_{21} + 1 \\
-R_{\text{FF}} \cdot U_{31} \\
-R_{\text{FF}} \cdot U_{31} - 1 \\
-R_{\text{FF}} \cdot U_{31} + 1 \\
\vdots \\
-R_{\text{FF}} \cdot U_{d1} \\
-R_{\text{FF}} \cdot U_{d1} - 1 \\
-R_{\text{FF}} \cdot U_{d1} + 1 \\
\vdots \\
-R_{\text{FF}} \cdot U_{1n} \\
-R_{\text{FF}} \cdot U_{1n} - 1 \\
-R_{\text{FF}} \cdot U_{1n} + 1 \\
-R_{\text{FF}} \cdot U_{2n} \\
-R_{\text{FF}} \cdot U_{2n} - 1 \\
-R_{\text{FF}} \cdot U_{2n} + 1 \\
-R_{\text{FF}} \cdot U_{3n} \\
-R_{\text{FF}} \cdot U_{3n} - 1 \\
-R_{\text{FF}} \cdot U_{3n} + 1 \\
\vdots \\
-R_{\text{FF}} \cdot U_{dn} \\
-R_{\text{FF}} \cdot U_{dn} - 1 \\
-R_{\text{FF}} \cdot U_{dn} + 1
\end{bmatrix}}_{3dn}.
$$

That is, they repeat the first $3d$ rows of $W_1^{(i)}$ for $n$ times.

Then, they construct $W_{2,2}^{(i)}$ to be

$$
W_{2,2}^{(i)} := \underbrace{W_{2,2}''^{(i)} \cdot W_{2,2}'^{(i)}}_{d \times 3dn},
$$

where

$$
W_{2,2}'^{(i)} := \underbrace{\begin{bmatrix}
-2f(\bar{C} - B)_{11} & f(\bar{C} - B)_{11} & f(\bar{C} - B)_{11} & 0 & \cdots & 0 \\
\vdots & \vdots & \vdots & \vdots & \ddots & \vdots \\
0 & \cdots & 0 & -2f(\bar{C} - B)_{dn} & f(\bar{C} - B)_{dn} & f(\bar{C} - B)_{dn}
\end{bmatrix}}_{dn \times 3dn},
$$

and

$$
W_{2,2}''^{(i)} := \underbrace{\begin{bmatrix} I_d & I_d & \cdots & I_d \end{bmatrix}}_{d \times dn}.
$$

Note that the above construction is for a specific $\bar{C}$, so it stacks the construction for all $\bar{C} \in \widetilde{\mathcal{G}}_g$ to obtain the final $\mathrm{FF}_2$

$$\mathrm{FF}_2(S) := W_{2,2} \cdot \mathrm{ReLU}(W_{2,1} \cdot S + b_{2,1} \mathbb{1}_n^\top),$$

where

$$W_{2,1} := \underbrace{\begin{bmatrix} W_{2,1}^1 \\ W_{2,1}^2 \\ \vdots \\ W_{2,1}^{q_2} \end{bmatrix}}_{3dnq_2 \times d}, \quad b_{2,1} := \underbrace{\begin{bmatrix} b_{2,1}^1 \\ b_{2,1}^2 \\ \vdots \\ b_{2,1}^{q_2} \end{bmatrix}}_{3dnq_2 \times 1}, \quad W_{2,2} := \underbrace{\begin{bmatrix} W_{2,2}^1 & W_{2,2}^2 & \cdots & W_{2,2}^{q_2} \end{bmatrix}}_{d \times 3dnq_2}, \tag{16}$$

and they set $q_2 = (2bg)^{dn}/(n!)$. That is, they set $b_{2,2}$ as a zero vector.

This completes the construction of the entire network.

**Remark D.1.** *In the constructions (11) and (16), the dimensions of the feed-forward weight matrices depend on the size of the compact domain and the discretization granularity. Thus, achieving a prescribed uniform approximation accuracy requiring finer granularity $g$ or larger domain size $b$. This leads to a larger hidden width. Hence, the matrix dimensions (and therefore their weight encodings) vary across target functions. This appears to challenge the "one fixed model across tasks" interpretation, since a fixed model requires a fixed shape of input weight encoding. However, for smaller matrices, we are always able to pad them with zeros*

$$\widetilde{W}_2 = \underbrace{\begin{bmatrix} W_2 & 0_{d \times \widetilde{r}} \end{bmatrix}}_{d \times (r + \widetilde{r})}, \quad \widetilde{W}_1 = \underbrace{\begin{bmatrix} W_1 \\ 0_{\widetilde{r} \times d} \end{bmatrix}}_{(r + \widetilde{r}) \times d}, \quad \widetilde{b}_1 = \underbrace{\begin{bmatrix} b_1 \\ 0_{\widetilde{r} \times 1} \end{bmatrix}}_{(r + \widetilde{r}) \times 1},$$

*where $\widetilde{r} \in \mathbb{N}^+$. Then, by Definition 2.2, the padded network realizes the same map as the original one*

$$W_2 \cdot \mathrm{ReLU}(W_1 X + b_1 \mathbb{1}_n^\top) + b_2 \mathbb{1}_n^\top = \widetilde{W}_2 \cdot \mathrm{ReLU}(\widetilde{W}_1 X + \widetilde{b}_1 \mathbb{1}_n^\top) + b_2 \mathbb{1}_n^\top.$$

*Also, our approximation technique is applicable to these zero-padded weights. This resolves the shape mismatch whenever a common finite architecture width $\widetilde{r}$ is fixed in advance.*

### D.2.4 A UNIFORM FINITE-GRID CONSEQUENCE

So far the construction is target-specific: the output layer stores the grid values of the target function. For the in-context theorem, we need a stronger form. After fixing the target class and the approximation accuracy, the surrogate architecture, prompt shape, and weight bounds must be chosen uniformly over all targets. We now isolate this consequence.

Since the setup considered in this paper includes positional encoding $E_p$. We introduce below definition to ensure above stated constructions hold with $E_p$.

**Definition D.3** (Separating Positional Encoding). *Let $\mathcal{C} = [-b, b]^{d \times n}$. We say that $E_p \in \mathbb{R}^{d \times n}$ is $\rho$-separating on $\mathcal{C}$ if*

$$\inf_{X \in \mathcal{C}} \min_{i \in [n]} \|(X + E_p)_{:,i}\|_\infty \geq \rho,$$

*and*

$$\inf_{X \in \mathcal{C}} \min_{i \neq j} \|(X + E_p)_{:,i} - (X + E_p)_{:,j}\|_\infty \geq \rho,$$

*for some $\rho > 0$.*

Such an encoding exists. For example, fix one row index $r \in [d]$ and choose the entries $(E_p)_{r,1}, \ldots, (E_p)_{r,n}$ so that

$$|(E_p)_{r,i}| \geq 2b + \rho, \quad |(E_p)_{r,i} - (E_p)_{r,j}| \geq 2b + \rho, \quad (i \neq j).$$

Then both inequalities in Definition D.3 hold for every $X \in \mathcal{C}$.

Then, we show that such $\rho$-separating $E_p$ does not disrupt the quantization steps.

**Lemma D.2** (Quantized Separation). *Let $\mathcal{C} = [-b,b]^{d\times n}$, and let $E_p$ be $\rho$-separating on $\mathcal{C}$. Let $\mathcal{D} := \mathcal{C} + E_p$. Let $\pi_g : \mathcal{D} \to \mathbb{R}^{d\times n}$ be an entrywise quantization map satisfying*

$$\|Y - \pi_g(Y)\|_\infty \le g^{-1}, \quad \text{for all} \quad Y \in \mathcal{D}.$$

*If $g^{-1} \le \rho/4$, then for every $X \in \mathcal{C}$ and every $i \ne j$,*

$$\|\pi_g(X + E_p)_{:,i}\|_\infty \ge 3\rho/4,$$

*and*

$$\|\pi_g(X + E_p)_{:,i} - \pi_g(X + E_p)_{:,j}\|_\infty \ge \rho/2.$$

*In particular, the quantized sequences used by the contextual-mapping construction are tokenwise separated with constants depending only on $\rho$, $d$, and the enclosing box.*

*Proof.* For every $X \in \mathcal{C}$ and $i \in [n]$,

$$
\begin{aligned}
&\|\pi_g(X + E_p)_{:,i}\|_\infty \\
&\ge \|(X + E_p)_{:,i}\|_\infty - \|X + E_p - \pi_g(X + E_p)\|_\infty && \text{(By reverse triangle inequality)} \\
&\ge \rho - g^{-1} && \text{(By } E_p \text{ being } \rho\text{-separating and } \|Y - \pi_g(Y)\|_\infty \le g^{-1}) \\
&\ge 3\rho/4. && (g^{-1} \le \rho/4)
\end{aligned}
$$

For any $X \in \mathcal{C}$ and $i \ne j$, we have

$$
\begin{aligned}
&\|\pi_g(X + E_p)_{:,i} - \pi_g(X + E_p)_{:,j}\|_\infty \\
&= \|((X + E_p)_{:,i} - (X + E_p)_{:,j}) + (\pi_g(X + E_p)_{:,i} - (X + E_p)_{:,i}) + ((X + E_p)_{:,j} - \pi_g(X + E_p)_{:,j})\|_\infty \\
&\ge \|(X + E_p)_{:,i} - (X + E_p)_{:,j}\|_\infty - \|\pi_g(X + E_p)_{:,i} - (X + E_p)_{:,i}\|_\infty - \|(X + E_p)_{:,j} - \pi_g(X + E_p)_{:,j}\|_\infty \\
&&& \text{(By reverse triangle inequality)} \\
&\ge \rho - g^{-1} - g^{-1} && \text{(By } E_p \text{ being } \rho\text{-separating and } \|Y - \pi_g(Y)\|_\infty \le g^{-1}) \\
&\ge \rho/2. && (\text{By } g^{-1} \le \rho/4)
\end{aligned}
$$

This completes the proof. $\square$

We now package the finite-grid part of the construction into a uniform statement. The proposition below is the only property of Section D.2 needed later to choose a common surrogate template for all targets.

**Proposition D.1** (Uniform Bounded Transformer Approximation of Finite-Grid Maps). *Let $\mathcal{C} = [-b,b]^{d\times n}$, and let $E_p$ be $\rho$-separating on $\mathcal{C}$. Let $\mathcal{D} := \mathcal{C} + E_p$ Fix $1 \le p < \infty$ and $M, \eta > 0$. Let $\pi_g : \mathcal{D} \to \mathcal{G}_g$ be an entrywise finite-grid quantization map with finite image $\mathcal{G}_g := \pi_g(\mathcal{D})$ and mesh size at most $g^{-1}$, such that*

$$\|Y - \pi_g(Y)\|_\infty \le g^{-1}, \quad \text{for all} \quad Y \in \mathcal{D}.$$

*Assume $g^{-1} \le \rho/4$, and define $\mathcal{H}_{g,M}$ be the class of grid-value maps*

$$\mathcal{H}_{g,M} := \{F : \mathcal{D} \to \mathbb{R}^{d\times n} \mid F(Y) = a_{\pi_g(Y)}, \|a_v\|_\infty \le M \text{ for all } v \in \mathcal{G}_g\}.$$

*Then there exists a ($\mathcal{T}_2$-type) Transformer*

$$B_\phi(Y) = \mathrm{FF}_{2,\phi} \circ \mathrm{Attn}_\phi^{\mathrm{Res}} \circ \mathrm{FF}_{1,\phi}(Y),$$

*with dimensions depending only on*

$$\mathcal{D}, g, M, p, d, n, \eta,$$

*such that, for every $F \in \mathcal{H}_{g,M}$, there exists a parameter choice $\phi_F$ satisfying*

$$\left(\int_{\mathcal{D}} \|B_{\phi_F}(Y) - F(Y)\|_p^p \mathrm{d}Y\right)^{1/p} \le \eta.$$

*Moreover, all weights in $\phi_F$, and hence their encoded prompt blocks, are bounded by constants depending only on $\mathcal{D}, g, M, p, d, n, \eta$, not on $F$.*

*Proof.* Choose $\bar{b} > 0$ such that

$$\mathcal{D} \subset \mathcal{Q} := [-\bar{b}, \bar{b}]^{d \times n}.$$

We apply the explicit construction of Section D.2 on the fixed symmetric box $\mathcal{Q}$ and then restrict the resulting approximation to $\mathcal{D}$.

First, the $\mathrm{FF}_1$ construction in Equations (7) to (10) implements the shifted quantization code used by the construction. On $\mathcal{D} \subset \mathcal{Q}$, the penalty term vanishes. Thus the $\mathrm{FF}_1$ weights depend only on the fixed box $\mathcal{Q}$, the grid granularity $g$, and the quantization parameter $\delta$. They do not depend on $F$.

Second, by Lemma D.2, every sequence in $\mathcal{G}_g$ has distinct token columns. Hence the contextual-mapping attention construction stated above applies to this finite set. It gives a single softmax attention layer whose output codes separate the finitely many quantized sequences in $\mathcal{G}_g$. Once $\mathcal{G}_g$ and the softmax sharpness parameter $\beta$ are fixed, the attention dimensions and weights are fixed independently of $F$.

Third, the $\mathrm{FF}_2$ bump-function construction leading to Equation (16) maps each separated code to the prescribed grid value $a_v$. The only $F$-dependent entries appear in the output coefficients $a_v$. Since

$$\|a_v\|_\infty \le M, \quad v \in \mathcal{G}_g,$$

these entries are uniformly bounded over $F \in \mathcal{H}_{g,M}$.

Choose $\delta > 0$ sufficiently small and choose $\beta$ and $R_{\mathrm{FF}}$ sufficiently large so that the finite-grid approximation error on $\mathcal{D}$ is at most $\eta$. These choices depend only on

$$\mathcal{C}, E_p, \rho, g, M, p, d, n, \eta,$$

not on $F$.

Therefore,

$$\left(\int_{\mathcal{D}} \|B_{\phi_F}(Y) - F(Y)\|_p^p \mathrm{d}Y\right)^{1/p} \le \eta.$$

All dimensions are fixed because $\mathcal{Q}$, $\mathcal{G}_g$, and the construction parameters are fixed. Since $\mathcal{Q}$ is chosen from $\mathcal{D}$, these dimensions depend only on $\mathcal{D}, \rho, g, M, p, d, n, \eta$. If a realization uses fewer hidden units, the zero-padding argument in Remark D.1 embeds it into the common template without changing the realized map. Thus the template dimensions, prompt shape, and weight bounds are uniform over $F \in \mathcal{H}_{g,M}$.

This completes the proof. $\square$

### D.3 Explicit Weight Construction of Theorem D.2 Given Function $f$

In this subsection, we isolate the explicit weight construction behind Theorem D.2. The construction follows (Hu et al., 2026a, Theorem G.1). We use it later in the proof of Theorem 3.2.

The proof of (Hu et al., 2026a, Theorem G.1) has two steps. First, it builds a two-hidden-layer ReLU grid network. Second, it converts this ReLU network into a four-layer attention-only Transformer. We spell out the first step below because it shows exactly where the target function enters the weights. For the second step, we use the attention-only conversion in the proof of (Hu et al., 2026a, Theorem G.1).

**Input Quantization.** Let

$$\mathcal{C} = [-b, b]^{d \times n}$$

and set $N := dn$. We identify each $X \in \mathcal{C}$ with a vector $x = \mathrm{vec}(X) \in [-b, b]^N$ using a fixed vectorization order. Fix an integer $g \ge 2$ and a margin parameter $\delta \in (0, 1)$. Define the one-dimensional grid

$$\Gamma_g := \left\{ -b + \frac{(2r-1)b}{g} \mid r \in [g] \right\}.$$

Define the product grid

$$\mathcal{G}_g := \Gamma_g^N.$$

The grid spacing equals $2b/g$, so every $x \in [-b, b]^N$ lies within $\ell_\infty$ distance $b/g$ of some grid point in $\mathcal{G}_g$. Let

$$\pi_g : \mathcal{C} \to \mathcal{G}_g$$

be the standard nearest-center quantization map, with ties broken by a fixed rule. Then

$$\|X - \pi_g(X)\|_\infty \le b/g, \quad \text{for all } X \in \mathcal{C}.$$

We now construct grid gates. Each gate checks whether $X$ lies in the grid cell around one grid point $v \in \mathcal{G}_g$.

**Layer 1: Bump Functions.** For each grid point $v \in \mathcal{G}_g$ and each coordinate $i \in [N]$, introduce four ReLU units

$$h_{v,i}^+(x) = \text{ReLU}(\frac{g}{\delta b}x_i + \frac{1}{\delta}(1 - \frac{g}{b}v_i)),$$

$$\widetilde{h}_{v,i}^+(x) = \text{ReLU}(\frac{g}{\delta b}x_i + \frac{1}{\delta}(1 - \delta - \frac{g}{b}v_i)),$$

$$h_{v,i}^-(x) = \text{ReLU}(-\frac{g}{\delta b}x_i + \frac{1}{\delta}(1 + \frac{g}{b}v_i)),$$

$$\widetilde{h}_{v,i}^-(x) = \text{ReLU}(-\frac{g}{\delta b}x_i + \frac{1}{\delta}(1 - \delta + \frac{g}{b}v_i)).$$

Define

$$\psi_{v,i}(x) := h_{v,i}^+(x) - \widetilde{h}_{v,i}^+(x) + h_{v,i}^-(x) - \widetilde{h}_{v,i}^-(x) - 1,$$

$$R_v(x) := \sum_{i=1}^{N} \psi_{v,i}(x).$$

These units depend only on $b, g, \delta, d, n$ and the grid point $v$. They do not depend on the target function.

The function $\psi_{v,i}$ has the following elementary properties:

$$0 \le \psi_{v,i}(x) \le 1, \quad \text{for all } x \in [-b, b]^N,$$
$$\psi_{v,i}(x) = 1, \quad \text{if } |x_i - v_i| \le (1 - \delta)b/g,$$
$$\psi_{v,i}(x) = 0, \quad \text{if } |x_i - v_i| \ge b/g.$$

Thus, $\psi_{v,i}$ acts as a softened interval indicator for the $i$-th coordinate.

**Layer 2: Grid Gating.** For each $v \in \mathcal{G}_g$, introduce one ReLU unit

$$u_{v,\delta}(x) := \text{ReLU}(R_v(x) - N + 1).$$

Since $0 \le \psi_{v,i} \le 1$, we have

$$0 \le u_{v,\delta}(x) \le 1.$$

Moreover,

$$u_{v,\delta}(x) = 1, \quad \text{if } |x_i - v_i| \le (1 - \delta)b/g \text{ for every } i \in [N],$$
$$u_{v,\delta}(x) = 0, \quad \text{if } |x_i - v_i| \ge b/g \text{ for some } i \in [N].$$

Thus $u_{v,\delta}$ acts as a softened indicator for the grid cell around $v$.

**Transition Set.** The ReLU gates act like grid-cell indicators, but they cannot jump discontinuously at grid-cell boundaries. Hence they may be ambiguous near the boundaries. We collect all such ambiguous inputs into one small set.

Define the one-dimensional grid-cell boundaries

$$\mathcal{T}_g := \{-b + \frac{2rb}{g} \mid r = 0, 1, \dots, g\}.$$

Define the transition set

$$\mathcal{B}_\delta := \{X \in \mathcal{C} \mid \exists i \in [N], \ \exists t \in \mathcal{T}_g \text{ such that } |x_i - t| \leq \delta b/g\}.$$

Thus $\mathcal{B}_\delta$ contains exactly the inputs whose some coordinate lies within $\delta b/g$ of a grid-cell boundary.

For $X \in \mathcal{C} \setminus \mathcal{B}_\delta$, every coordinate lies safely inside one grid cell. Hence exactly one grid gate turns on:

$$u_{\pi_g(X),\delta}(x) = 1, \quad u_{v,\delta}(x) = 0 \quad \text{for all } v \in \mathcal{G}_g \setminus \{\pi_g(X)\}.$$

Therefore, the ReLU grid network matches the grid-value map exactly outside $\mathcal{B}_\delta$.

The set $\mathcal{B}_\delta$ is a union of finitely many thin slabs. Hence there exists a constant $C_{\mathrm{bd}} < \infty$, depending only on $b, g, d, n$, such that

$$|\mathcal{B}_\delta| \leq C_{\mathrm{bd}}\delta.$$

This volume bound lets us make the total $L_p$ error small by choosing $\delta$ small.

**Network Size.** The construction uses two ReLU hidden layers. Layer 1 has $4N|\mathcal{G}_g|$ ReLU units. Layer 2 has $|\mathcal{G}_g|$ ReLU units. The output layer is linear and uses the coefficients $\{a_v\}_{v \in \mathcal{G}_g}$.

**Uniform Attention-Only Conversion.** We use the following consequence of the proof of (Hu et al., 2026a, Theorem G.1). If a two-hidden-layer ReLU grid template has fixed dimensions and all its weights are bounded by a constant $B_{\mathrm{ReLU}}$, then, for any conversion accuracy $\eta_{\mathrm{att}} > 0$, the proof of (Hu et al., 2026a, Theorem G.1) gives a four-layer attention-only Transformer template

$$B_\phi := \mathrm{Attn}_{\phi_4} \circ \mathrm{Attn}_{\phi_3} \circ \mathrm{Attn}_{\phi_2} \circ \mathrm{Attn}_{\phi_1}, \quad \phi = (\phi_1, \phi_2, \phi_3, \phi_4),$$

with fixed layer dimensions, fixed sequence length, fixed head numbers, fixed head dimensions, and fixed inverse temperatures. The template and a uniform attention-weight bound depend only on the ReLU template, $B_{\mathrm{ReLU}}$, and $\eta_{\mathrm{att}}$. They do not depend on the particular grid-value map $F$. If a realization uses fewer heads, zero-padding embeds it into the fixed template without changing the realized map. This gives a fixed prompt shape for the encoded attention weights.

**Proposition D.2** (Uniform Bounded Attention-Only Approximation of Finite-Grid Maps). *Let $\mathcal{C} = [-b, b]^{d \times n} \subset \mathbb{R}^{d \times n}$. Fix an integer $g \geq 2$, $1 \leq p < \infty$, and $M, \eta > 0$. Let $\mathcal{G}_g$ and $\pi_g : \mathcal{C} \to \mathcal{G}_g$ be the standard finite-grid quantization map above. Define $\mathcal{H}_{g,M}$ be the class of grid-value maps*

$$\mathcal{H}_{g,M} := \{F : \mathcal{C} \to \mathbb{R}^{d \times n} \mid F(X) = a_{\pi_g(X)}, \|a_v\|_\infty \leq M \text{ for all } v \in \mathcal{G}_g\}.$$

*Set $m := n + 1$ and define*

$$X_a := \begin{bmatrix} X & 0_{d \times 1} \\ I_n & 0_{n \times 1} \\ 0_{1 \times n} & 1 \end{bmatrix} \in \mathbb{R}^{(d+n+1) \times m}.$$

*Then there exists a four-layer attention-only Transformer*

$$B_\phi := \mathrm{Attn}_{\phi_4} \circ \mathrm{Attn}_{\phi_3} \circ \mathrm{Attn}_{\phi_2} \circ \mathrm{Attn}_{\phi_1}, \quad \phi = (\phi_1, \phi_2, \phi_3, \phi_4),$$

*with fixed layer dimensions, fixed sequence length $m$, fixed head numbers, fixed head dimensions, fixed inverse temperatures, and fixed prompt shape, such that, for every $F \in \mathcal{H}_{g,M}$, there exists a parameter choice $\phi_F = (\phi_{F,1}, \dots, \phi_{F,4})$ satisfying*

$$\left(\int_{\mathcal{C}} \|B_{\phi_F}(X_a)_{:,1:n} - F(X)\|_p^p \mathrm{d}X\right)^{1/p} \leq \eta.$$

*Moreover, all weights in $\phi_F$, and hence their encoded prompt blocks $W_{\phi_F,1}, \dots, W_{\phi_F,4}$ from Definition 3.5, are bounded by constants depending only on $b, g, M, p, d, n, \eta$ not on $F$.*

*Proof.* Let $N := dn$ and fix an arbitrary $F \in \mathcal{H}_{g,M}$. By the definition of $\mathcal{H}_{g,M}$, there exist coefficients $\{a_v\}_{v \in \mathcal{G}_g} \subset \mathbb{R}^{d \times n}$ such that

$$F(X) = a_{\pi_g(X)}, \quad \|a_v\|_\infty \leq M, \quad v \in \mathcal{G}_g.$$

We use the two-hidden-layer ReLU grid network from this subsection:

$$R_{F,g,\delta}(X) := \sum_{v \in \mathcal{G}_g} a_v u_{v,\delta}(\mathrm{vec}(X)).$$

The hidden-layer weights depend only on

$$b, g, \delta, d, n,$$

not on $F$. The target map $F$ enters only through the output coefficients $\{a_v\}_{v \in \mathcal{G}_g}$.

For every $X \in \mathcal{C}$, we have

$$\|R_{F,g,\delta}(X)\|_\infty = \|\sum_{v \in \mathcal{G}_g} a_v u_{v,\delta}(\mathrm{vec}(X))\|_\infty$$
$$\leq \sum_{v \in \mathcal{G}_g} \|a_v\|_\infty |u_{v,\delta}(\mathrm{vec}(X))| \qquad \text{(By triangle inequality)}$$
$$\leq |\mathcal{G}_g| M, \tag{17}$$

where the last line is by $\|a_v\|_\infty \leq M$ and $0 \leq u_{v,\delta} \leq 1$.

Also, for every $X \in \mathcal{C}$, we have

$$\|F(X)\|_\infty = \|a_{\pi_g(X)}\|_\infty \qquad \text{(By defnitinio of } \mathcal{H}_{g,M})$$
$$\leq M. \tag{18}$$

Outside the transition set $\mathcal{B}_\delta$, exactly one grid gate turns on. Hence

$$R_{F,g,\delta}(X) = \sum_{v \in \mathcal{G}_g} a_v u_{v,\delta}(\mathrm{vec}(X))$$
$$= a_{\pi_g(X)} \qquad \text{(By } u_{\pi_g(X),\delta} = 1 \text{ and } u_{v,\delta} = 0 \text{ for } v \neq \pi_g(X))$$
$$= F(X), \quad X \in \mathcal{C} \setminus \mathcal{B}_\delta. \qquad \text{(By } F(X) = a_{\pi_g(X)})$$

On the transition set $\mathcal{B}_\delta$, we use the uniform bound

$$\|R_{F,g,\delta}(X) - F(X)\|_\infty \leq \|R_{F,g,\delta}(X)\|_\infty + \|F(X)\|_\infty \qquad \text{(By triangle inequality)}$$
$$\leq |\mathcal{G}_g| M + M \qquad \text{(By Equations (17) and (18))}$$
$$= (|\mathcal{G}_g| + 1) M. \tag{19}$$

Therefore,

$$(\int_{\mathcal{C}} \|R_{F,g,\delta}(X) - F(X)\|_p^p \mathrm{d}X)^{1/p}$$
$$= (\int_{\mathcal{B}_\delta} \|R_{F,g,\delta}(X) - F(X)\|_p^p \mathrm{d}X)^{1/p} \qquad \text{(By } R_{F,g,\delta} = F \text{ on } \mathcal{C} \setminus \mathcal{B}_\delta)$$
$$\leq (\int_{\mathcal{B}_\delta} (N^{1/p} \|R_{F,g,\delta}(X) - F(X)\|_\infty)^p \mathrm{d}X)^{1/p} \qquad \text{(By } \|A\|_p \leq N^{1/p} \|A\|_\infty)$$
$$\leq N^{1/p} (|\mathcal{G}_g| + 1) M |\mathcal{B}_\delta|^{1/p} \qquad \text{(By Equation (19))}$$
$$\leq N^{1/p} (|\mathcal{G}_g| + 1) M (C_{\mathrm{bd}} \delta)^{1/p}. \qquad \text{(By } |\mathcal{B}_\delta| \leq C_{\mathrm{bd}} \delta)$$

Choose $\delta \in (0,1)$ such that

$$N^{1/p}(|\mathcal{G}_g| + 1)M(C_{\mathrm{bd}}\delta)^{1/p} \leq \eta/2.$$

This choice depends only on

$$b, g, M, p, d, n, \eta,$$

not on $F$. Then

$$\left(\int_{\mathcal{C}} \|R_{F,g,\delta}(X) - F(X)\|_p^p \mathrm{d}X\right)^{1/p} \leq \eta/2.$$

We next check the uniform ReLU weight bound. The hidden-layer weights contain only the constants from the gate construction, so they depend only on

$$b, g, \delta, d, n.$$

The output-layer weights equal the coefficients $\{a_v\}_{v \in \mathcal{G}_g}$, and

$$\|a_v\|_\infty \leq M, \quad v \in \mathcal{G}_g.$$

Hence all weights in the ReLU grid network have a common bound

$$B_{\mathrm{ReLU}} < \infty$$

depending only on

$$b, g, M, p, d, n, \eta,$$

not on $F$.

Apply the uniform attention-only conversion paragraph above to this fixed bounded ReLU template with conversion accuracy $\eta/2$. Then there exists a fixed four-layer attention-only Transformer

$$B_\phi := \mathrm{Attn}_{\phi_4} \circ \mathrm{Attn}_{\phi_3} \circ \mathrm{Attn}_{\phi_2} \circ \mathrm{Attn}_{\phi_1}, \quad \phi = (\phi_1, \phi_2, \phi_3, \phi_4),$$

with fixed layer dimensions, fixed sequence length $n + 1$, fixed head numbers, fixed head dimensions, fixed inverse temperatures, and fixed prompt shape. Moreover, for this $F$, there exists

$$\phi_F = (\phi_{F,1}, \phi_{F,2}, \phi_{F,3}, \phi_{F,4}),$$

such that

$$\left(\int_{\mathcal{C}} \|B_{\phi_F}(X_a)_{:,1:n} - R_{F,g,\delta}(X)\|_p^p \mathrm{d}X\right)^{1/p} \leq \eta/2.$$

The conversion gives a common attention-weight bound

$$B_{\mathrm{Attn}} < \infty$$

depending only on

$$b, g, M, p, d, n, \eta,$$

not on $F$.

Now combine the ReLU grid approximation with the attention-only conversion:

$$\left(\int_{\mathcal{C}} \|B_{\phi_F}(X_a)_{:,1:n} - F(X)\|_p^p \mathrm{d}X\right)^{1/p}$$

$$\leq (\int_{\mathcal{C}} \|B_{\phi_F}(X_a)_{:,1:n} - R_{F,g,\delta}(X)\|_p^p \mathrm{d}X)^{1/p} + (\int_{\mathcal{C}} \|R_{F,g,\delta}(X) - F(X)\|_p^p \mathrm{d}X)^{1/p} \qquad \text{(By Minkowski inequality)}$$

$$\leq \eta/2 + \eta/2 \qquad \text{(By the two bounds above)}$$

$$= \eta.$$

It remains to check the prompt-shape and prompt-boundedness claims. The attention-only template has fixed layer dimensions and fixed head numbers. Therefore, Definition 3.5 gives prompt blocks

$$W_{\phi_F,1}, \ldots, W_{\phi_F,4},$$

with fixed shape. Also, Definition 3.5 builds these prompt blocks from the head encodings in Definition 3.4. Each head encoding stacks attention weights and their token-indexed multiples $0, 1, \ldots, m-1$. Thus, for each layer $i \in [4]$, there exists a constant $B_P < \infty$ such that

$$\|W_{\phi_F,i}\|_\infty \leq B_P, \quad i \in [4]. \qquad \text{(By } \|\phi_F\|_\infty \leq B_{\text{Attn}} \text{ and fixed } m\text{)}$$

The constant $B_P$ depends only on

$$b, g, M, p, d, n, \eta,$$

not on $F$.

Since $F \in \mathcal{H}_{g,M}$ was arbitrary, the same fixed attention-only template works for every $F \in \mathcal{H}_{g,M}$ through its corresponding parameter choice $\phi_F$. This completes the proof. $\qquad\square$

## D.4   In-Context Emulation

We state the approximation of a single-head self-attention from (Hu et al., 2026b).

**Definition D.4** (Vectorization, Definition 4.1 of (Hu et al., 2026b)). *For any matrix $W \in \mathbb{R}^{p \times s}$, we define $\underline{W} := \mathrm{vec}(W) \in \mathbb{R}^{ps}$ such that $\underline{W}_{(i-1)s+j} = W_{ij}$ for all $i \in [p]$ and $j \in [s]$.*

**Definition D.5** (Input Prompt for In-Context Emulation of Attention, Definition 4.2 of (Hu et al., 2026b)). *Let $X \in \mathbb{R}^{d \times n}$ be the input sequence, and let $W_K, W_Q \in \mathbb{R}^{d_h \times d}$, $W_V \in \mathbb{R}^{d_V \times d}$ be the weight matrices of the target attention head to be emulated. Define the vectorizations*

$$\underline{W}_K := \mathrm{vec}(W_K) \in \mathbb{R}^{dd_h}, \ \underline{W}_Q := \mathrm{vec}(W_Q) \in \mathbb{R}^{dd_h}, \ \underline{W}_V := \mathrm{vec}(W_V) \in \mathbb{R}^{dd_V},$$

*and $w := [\underline{W}_K; \underline{W}_Q; \underline{W}_V] \in \mathbb{R}^{d(2d_h + d_V)}$, where $w$ is the concatenation of $\underline{W}_K, \underline{W}_Q, \underline{W}_V$. Finally, define the extended input $X_p$ for the in-context emulation of the attention head specified by $W_K, W_Q, W_V$ as*

$$X_p := \begin{bmatrix} X \\ W_{\text{in}} \\ I_n \end{bmatrix} \quad \text{with} \quad W_{\text{in}} := \begin{bmatrix} 0 \cdot w & 1 \cdot w & 2 \cdot w & \cdots & (n-1) \cdot w \\ w & w & w & \cdots & w \end{bmatrix} \in \mathbb{R}^{2d(2d_h + d_V) \times n}.$$

**Theorem D.3** (In-Context Emulation of Attention, Theorem 4.1 of (Hu et al., 2026b)). *Let $X \in \mathbb{R}^{d \times n}$ be an input sequence, and let $W_K \in \mathbb{R}^{d_h \times d}$, $W_Q \in \mathbb{R}^{d_h \times d}$, $W_V \in \mathbb{R}^{d_V \times d}$ be the weight matrices of the target attention head we wish to emulate in-context. Assume $\|W_K X\|_\infty, \|W_Q X\|_\infty, \|W_V X\|_\infty \leq B_{KQV}$ with $B_{KQV} > 0$. Then, for any $\epsilon_e > 0$, there exists a two-layer attention network - a multi-head attention layer $\mathrm{Attn}$ followed by a single-head attention layer $\mathrm{Attn}_s$ - such that*

$$\|\mathrm{Attn}_s \circ \mathrm{Attn}(X_p) - W_V X \cdot \sigma_\beta((W_K X)^\top W_Q X)\|_\infty \leq \epsilon_e,$$

*where $X_p$ is the prompt defined in Definition D.5.*

## D.5   In-Context Approximation of Truncated Linear Model

We first introduce the truncated linear function and the truncated linear model.

**Definition D.6** (Truncated Linear Function, Definition B.1 of Hu et al. (2026b)). *We define the truncated linear function as follows:*

$$\mathrm{Range}_{[a,b]}(x) = \begin{cases} a & x \leq a, \\ x & a \leq x \leq b, \\ b & b \leq x. \end{cases}$$

Intuitively, the truncated linear function is a segment of a linear function, with the output value ranging from $a$ to $b$.

**Definition D.7** (Truncated Linear Model). *Define a truncated linear model as* $\text{Range}_{[a,b]}(w^\top x + t)$, *where* $w \in \mathbb{R}^d$ *is a learnable weight and* $t \in \mathbb{R}$ *is a bias.*

Then, we restate the in-context approximation of the truncated linear model from Hu et al. (2026b).

**Theorem D.4** (Multi-Head Attention Approximates Truncated Linear Models In Context, Theorem B.1 of (Hu et al., 2026b)).
*Let $X \in \mathbb{R}^{d \times n}$ be the input. Fix real numbers $a < b$, and let the truncation operator $\text{Range}_{[a,b]}(\cdot)$ follow Definition D.6. Let $w_s$ denote the linear coefficient of the in-context truncated linear model. Define $W_s$ as*

$$W_s := \begin{bmatrix} 0 \cdot w_s & 1 \cdot w_s & \cdots & (n-1) \cdot w_s \\ w_s & w_s & \cdots & w_s \end{bmatrix} \in \mathbb{R}^{2d \times n}.$$

*For a precision parameter $p > n$ with $\epsilon = O(1/p)$, number of head $H = p/(n-2)$ there exists a single-layer, $H$-head attention $\text{Attn}$ with a linear transformation $A : \mathbb{R}^{d \times n} \to \mathbb{R}^{(3d+n) \times n}$, such that $\text{Attn} \circ A : \mathbb{R}^{d \times n} \to \mathbb{R}^{d_o \times n}$ satisfies, for any $i \in [n]$,*

$$\|\text{Attn} \circ A(\begin{bmatrix} X \\ W_s \end{bmatrix})_{:,i} - \text{Range}_{[a,b]}(w_s^\top x_i)e_{\widetilde{k}_i}\|_\infty \leq \underbrace{\max\{|a|, |b|\} \cdot \epsilon_0}_{\text{finite-}\beta \text{ softmax error}} + \underbrace{\frac{b-a}{(n-2)H}}_{\text{interpolation error}} .$$

*Here $e_{\widetilde{k}_i}$ is the one-hot vector with a $1$ at position $\widetilde{k}_i$-th index and $0$ elsewhere, and*

$$\widetilde{k}_i = G(k_i) \in [d_o], \quad \text{with} \quad k_i = \underset{k \in \{0,1,\cdots,p-1\}}{\text{argmin}} \, (-2x_i^\top w_i - 2t_i + \widetilde{L}_0 + \widetilde{L}_k) \cdot k,$$

*where $G : [p] \to [d_o]$ denotes any set-to-constant function sending each selected interpolation index $k_i$ into an appropriate integer $\widetilde{k}_i \in [d_o]$ for $i \in [n]$.*

*Proof.* For completeness, we restate the necessary step for later modifications. Define $A : \mathbb{R}^{d \times n} \to \mathbb{R}^{(3d+n) \times n}$ for the input sequence $X$ as

$$A(X) := \underbrace{\begin{bmatrix} I_{3d} \\ 0_{n \times 3d} \end{bmatrix}}_{\text{token-wise linear}} \begin{bmatrix} X \\ W_s \end{bmatrix} + \underbrace{\begin{bmatrix} 0_{3d \times n} \\ I_n \end{bmatrix}}_{\text{positional encoding}} = \begin{bmatrix} X \\ W_s \\ I_n \end{bmatrix} \in \mathbb{R}^{(3d+n) \times n}.$$

Thus, A is a token-wise linear layer augmented with positional encoding. It applies a linear projection to each token and then adds a unique per-token bias. Please refer to the original proof for the remaining details. □

We extend the above theorem as follows to prepare for the later proof in Section E.1 for Lemma 3.1.

**Corollary D.4.1** (Extension of Theorem D.4 to Multiple Truncated Linear Models In Context). *Let $X \in \mathbb{R}^{d \times n}$ be the input. Fix real numbers $a < b$, and let the truncation operator $\text{Range}_{[a,b]}(\cdot)$ follow Definition D.6. For $j \in [l]$, let $w_j$ denote the linear coefficient of the $j$-th in-context truncated linear model. Define $W_j$ as*

$$W_j := \begin{bmatrix} 0 \cdot w_j & 1 \cdot w_j & \cdots & (n-1) \cdot w_j \\ w_j & w_j & \cdots & w_j \end{bmatrix} \in \mathbb{R}^{2d \times n},$$

*where $j \in [l]$. For a precision parameter $p > n$ with $\epsilon = O(1/p)$, number of head $H = p/(n-2)$ there exists a single-layer, $H$-head attention $\text{Attn}^{[j]}$ such that $\text{Attn}^{[j]} : \mathbb{R}^{(d(1+2l)+n) \times n} \to \mathbb{R}^{d_o \times n}$ satisfies, for any $i \in [n]$ and $j \in [l]$,*

$$\|\text{Attn}^{[j]}(\begin{bmatrix} X \\ W_1 \\ W_2 \\ \vdots \\ W_l \\ I_n \end{bmatrix})_{:,i} - \text{Range}_{[a,b]}(w_j^\top x_i)e_{\widetilde{k}_i}\|_\infty \leq \underbrace{\max\{|a|, |b|\} \cdot \epsilon_0}_{\text{finite-}\beta \text{ softmax error}} + \underbrace{\frac{b-a}{(n-2)H}}_{\text{interpolation error}} .$$

*Here $e_{\widetilde{k}_i}$ is the one-hot vector with a $1$ at position $\widetilde{k}_i$-th index and $0$ elsewhere, and*

$$\widetilde{k}_i = G(k_i) \in [d_o], \quad with \quad k_i = \underset{k \in \{0,1,\cdots,p-1\}}{\arg\min} \; (-2x_i^\top w_i - 2t_i + \widetilde{L}_0 + \widetilde{L}_k) \cdot k,$$

*where $G : [p] \to [d_o]$ denotes any set-to-constant function sending each selected interpolation index $k_i$ into an appropriate integer $\widetilde{k}_i \in [d_o]$ for $i \in [n]$.*

*Proof.* Modify the affine linear map $A$ in the original proof to a linear one $A_j : \mathbb{R}^{(d(1+2l)+n)\times n} \to \mathbb{R}^{(3d+n)\times n}$ such that

$$A_j \cdot \begin{bmatrix} X \\ W_1 \\ W_2 \\ \vdots \\ W_l \\ I_n \end{bmatrix} := \begin{bmatrix} I_d & 0_{d\times(2dl+n)} \\ 0_{d\times(2d(j-1)+d)} & I_{2d} & 0_{d\times(2d(l-j)+n)} \\ & 0_{n\times d(2l+1)} & I_n \end{bmatrix} \cdot \begin{bmatrix} X \\ W_1 \\ W_2 \\ \vdots \\ W_l \\ I_n \end{bmatrix} = \begin{bmatrix} X \\ W_j \\ I_n \end{bmatrix} \in \mathbb{R}^{(3d+n)\times n}.$$

That is, $A_j$ picks out $W_j$.

Since the input already contains $I_n$, the linear transformation $A_j$ is just a linear projection instead of an affine map used to add $I_n$ to the output. We combine $A_j$ with the linear projection of the key, query, and value matrices of attention, denoted as $\text{Attn}^{[j]}$.

Then, we follow the same proof of Theorem D.4 to complete the proof. $\qquad\square$

# E    Proofs of Main Text

## E.1    Proof of Lemma 3.1

**Lemma E.1** (Lemma 3.1 Restated: Prompt-Encoded FFN Emulation with Flow-Through). *Let $X \in \mathbb{R}^{d \times n}$ be the input sequence. Consider a feed-forward layer* FF *defined in Definition 2.2. Let $W_{\mathrm{FF}}$ be the weight encoding defined in Definition 3.2. Let $Z \in \mathbb{R}^{d_Z \times n}$ be a matrix containing an identity matrix $I_n$ with $n \le d_Z$. We assume $\|X\|_\infty \le B_X, \|W_{\mathrm{FF}}\|_\infty \le B_W, \|Z\|_\infty \le B_Z$ for $B_X, B_W, B_Z > 0$. Then, for any $\epsilon_1 > 0$, there exists a two-layer attention $\mathrm{Attn}_2$ such that*

$$\|\mathrm{Attn}_2(\begin{bmatrix} X \\ W_{\mathrm{FF}} \\ Z \end{bmatrix}) - \begin{bmatrix} \mathrm{FF}(X) \\ Z \end{bmatrix}\|_\infty \le \epsilon_1.$$

*Proof.* The proof consists of three steps:

- **Step 1.** We rewrite $W_1 x_i + b_1$ as an inner product by augmenting each token, and we fix a truncation range using $(B_X, B_W)$ to obtain ReLU activation.

- **Step 2.** We apply Corollary D.4.1 and sum over $j \in [r]$ to obtain an in-context approximation of the first hidden layer of the ReLU neural network $h_i := \mathrm{ReLU}(W_1 x_i + b_1)$. We further use extra heads to achieve the flow-through operation.

- **Step 3.** We repeat the same in-context construction for the affine map $W_2 h_i + b_2$ (with another augmentation), then we propagate the step-1 error through $W_2$.

For the convenience of presentation, we use $\mathrm{Attn}_{2\text{-}2} \circ \mathrm{Attn}_{2\text{-}1}$ to denote the two layers of $\mathrm{Attn}_2$.

**Step 1.** As in Definition 2.2, the feed-forward layer is defined as

$$\mathrm{FF}(X) = W_2 \mathrm{ReLU}(W_1 X + b_1 \mathbb{1}_n^\top) + b_2 \mathbb{1}_n^\top,$$

where $W_1 \in \mathbb{R}^{r \times d}$, $b_1 \in \mathbb{R}^r$, $W_2 \in \mathbb{R}^{d \times r}$, and $b_2 \in \mathbb{R}^d$.

For proof simplicity, we absorb biases in FF by augmenting each token with a constant coordinate. For each token $i \in [n]$, define the augmented input [9]

$$\widetilde{x}_i := \begin{bmatrix} x_i \\ 1 \end{bmatrix} \in \mathbb{R}^{d+1}, \quad \widetilde{X} := \begin{bmatrix} X \\ \mathbb{1}_n^\top \end{bmatrix} \in \mathbb{R}^{(d+1) \times n}. \tag{20}$$

For each hidden row $j \in [r]$ in $W_1$, define $a_j$ same as Definition 3.2

$$a_j := \begin{bmatrix} (W_1)_{j,:}^\top \\ (b_1)_j \end{bmatrix} \in \mathbb{R}^{d+1}.$$

Then $a_j^\top \widetilde{x}_i = (W_1)_{j,:} x_i + (b_1)_j$. Later proof we sum over $j \in [r]$ to recover the first practivation in FFN: $\sum_{j=1}^r a_j^\top \widetilde{x}_i \cdot e_j = W_1 x_i + b_1$. Here $e_j$ is the one-hot vector with only $j$-th entry equal to 1 and zero entries elsewhere.

Same for $W_2 \in \mathbb{R}^{d \times r}$, for each output row $k \in [d]$, define

$$c_k := \begin{bmatrix} (W_2)_{k,:}^\top \\ (b_2)_k \end{bmatrix} \in \mathbb{R}^{r+1}.$$

Now we move to determine the bound for $|a_j^\top \widetilde{x}_i|$. One needs such bound so we know how to choose the truncated range $a, b$ of $\mathrm{Range}_{a,b}(\cdot)$ in Corollary D.4.1 to create the ReLU activation.

We use elementwise $\|\cdot\|_\infty$. Hence for $u, v \in \mathbb{R}^m$ we have $|u^\top v| \le m \|u\|_\infty \|v\|_\infty$. With $\|x_i\|_\infty \le B_X$, $\|\widetilde{x}_i\|_\infty \le \widetilde{B}_X := \max\{B_X, 1\}$, and $\|a_j\|_\infty \le B_W$, we get for all $i, j$,

$$|a_j^\top \widetilde{x}_i| \le (d+1)\|a_j\|_\infty \|\widetilde{x}_i\|_\infty \le (d+1)\widetilde{B}_X B_W.$$

---

[9] The extra row $\mathbb{1}_n^\top$ can be absorbed into $W_{\mathrm{FF}}$. Specifically, we abuse the notation of $W_{\mathrm{FF},j}^{(1)}$ by redefining $W_{\mathrm{FF},1}^{(1)}$ of Definition 3.2 to have an extra one row $\mathbb{1}_n^\top$ on top of the matrix $W_{\mathrm{FF},1}^{(1)} := \begin{bmatrix} 1 & 1 & \cdots & 1 \\ 0 \cdot a_1 & 1 \cdot a_1 & \cdots & (n-1) \cdot a_1 \\ a_1 & a_1 & \cdots & a_1 \end{bmatrix} \in \mathbb{R}^{(2d+3) \times n}$.

Define

$$B_1 := (d+1)\widetilde{B}_X B_W.$$

Then $\mathrm{Range}_{[0,B_1]}(a_j^\top \widetilde{x}_i) = \mathrm{ReLU}(a_j^\top \widetilde{x}_i)$ for all $i, j$.

**Step 2: Approximate the Hidden Activation** $\mathrm{ReLU}(W_1 x_i + b_1)$ **In-Context.** For each $j \in [r]$, we form $W_{\mathrm{FF},j}^{(1)}$ follows Definition 3.2:

$$W_{\mathrm{FF},j}^{(1)} := \mathrm{Enc}(a_j) = \begin{bmatrix} 0 \cdot a_j & 1 \cdot a_j & \cdots & (n-1) \cdot a_j \\ a_j & a_j & \cdots & a_j \end{bmatrix} \in \mathbb{R}^{2(d+1) \times n}.$$

By Corollary D.4.1, with $a = 0$ and $b = B_1$, for each $j \in [r]$ there exists an attention module $\mathrm{Attn}_{2\text{-}1}^{[j]}$ using $H_1$ heads such that for all $i \in [n]$,

$$\left\| \mathrm{Attn}_{2\text{-}1}^{[j]} \left( \begin{bmatrix} \widetilde{X} \\ W_{\mathrm{FF},1}^{(1)} \\ \vdots \\ W_{\mathrm{FF},r}^{(1)} \\ W_{\mathrm{FF},1}^{(2)} \\ \vdots \\ W_{\mathrm{FF},d}^{(2)} \\ Z \end{bmatrix} \right)_{:,i} - \mathrm{Range}_{[0,B_1]}(a_j^\top \widetilde{x}_i) e_j \right\|_\infty \leq B_1 \epsilon_0 + \frac{B_1}{(n-2)H_1}.$$

Here $\mathrm{Attn}_{2\text{-}1}^{[j]}$ denotes a group of heads inside the first attention $\mathrm{Attn}_{2\text{-}1}$.

Define

$$\mathrm{Attn}_{2\text{-}1}^{\mathrm{hid}} := \sum_{j=1}^r \mathrm{Attn}_{2\text{-}1}^{[j]}.$$

Since the groups $\mathrm{Attn}_{2\text{-}1}^{[j]}$ write to disjoint hidden coordinates through one-hot vector $e_j$, summing over $j \in [r]$ yields one multi-head attention module that satisfies

$$\left\| \mathrm{Attn}_{2\text{-}1}^{\mathrm{hid}} \left( \begin{bmatrix} \widetilde{X} \\ W_{\mathrm{FF},1}^{(1)} \\ \vdots \\ W_{\mathrm{FF},r}^{(1)} \\ W_{\mathrm{FF},1}^{(2)} \\ \vdots \\ W_{\mathrm{FF},d}^{(2)} \\ Z \end{bmatrix} \right)_{:,i} - \begin{bmatrix} \mathrm{ReLU}(W_1 x_i + b_1) \\ 0_{(2d(r+1)+d_Z)} \end{bmatrix} \right\|_\infty \leq \eta_1 \quad \text{with} \quad \eta_1 := B_1 \epsilon_0 + \frac{B_1}{(n-2)H_1}. \tag{21}$$

This completes the in-context approximation of the first hidden layer of FF.

However, the output of (21) doesn't contain the weight of $W_2$ in FF and the flow-through component $Z$. We need to construct another attention module to flow the weight encoding blocks $[W_{\mathrm{FF},1}^{(2)}, \cdots, W_{\mathrm{FF},d}^{(2)}, Z]$ for later-layer use.

In this theorem, $Z$ contains $I_n$. We hence decompose $Z$ into $Z = [Z_1, I_n, Z_2]$. We use one head to copy $I_n$, another head to flow $[W_{\mathrm{FF},1}^{(2)}, \cdots, W_{\mathrm{FF},d}^{(2)}, Z_1]$, and another head to flow $Z_2$. We briefly state the construction of such attention here, and defer the details to the proof of Lemma 3.3 in Section E.3.

- **Attention for Exact Copy of $I_n$.** In Section E.3, we show there exists an attention layer that exactly copies the identity matrix $I_n$ (see Equations (37) to (44)).

- **Attention for Flow-Through Operation.** Furthermore, there also exists an attention layer that approximates the flow-through operation (see Equations (45) to (48)).

Combining these three attentions into one multi-head attention $\text{Attn}_{2\text{-}1}^{\text{flow}}$ we have

$$\text{Attn}_{2\text{-}1}^{\text{flow}}\left(\begin{bmatrix} \widetilde{X} \\ W_{\text{FF},r}^{(1)} \\ \vdots \\ W_{\text{FF},r}^{(1)} \\ W_{\text{FF},1}^{(2)} \\ \vdots \\ W_{\text{FF},d}^{(2)} \\ Z \end{bmatrix}\right) = \begin{bmatrix} 0_{r\times n} \\ W_{\text{FF},1}^{(2)}\sigma_{\beta_2}(I_n) \\ \vdots \\ W_{\text{FF},d}^{(2)}\sigma_{\beta_2}(I_n) \\ Z_1\sigma_{\beta_2}(I_n) \\ I_n \\ Z_2\sigma_{\beta_2}(I_n) \end{bmatrix}. \tag{22}$$

The error is, by (73),

$$\left\| \text{Attn}_{2\text{-}1}^{\text{flow}}\left(\begin{bmatrix} \widetilde{X} \\ W_{\text{FF},1}^{(1)} \\ \vdots \\ W_{\text{FF},r}^{(1)} \\ W_{\text{FF},1}^{(2)} \\ \vdots \\ W_{\text{FF},d}^{(2)} \\ Z \end{bmatrix}\right) - \begin{bmatrix} 0_{r\times n} \\ W_{\text{FF},1}^{(2)} \\ \vdots \\ W_{\text{FF},d}^{(2)} \\ Z \end{bmatrix} \right\|_\infty \le n \cdot \max\{B_Z, B_W\}\frac{(n-1)}{e^{\beta_2}+(n-1)} =: \eta_{\text{flow}}. \tag{23}$$

Define the first attention layer by

$$\text{Attn}_{2\text{-}1} := \text{Attn}_{2\text{-}1}^{\text{hid}} + \text{Attn}_{2\text{-}1}^{\text{flow}}.$$

Combining (21) with (23), there exists an attention layer $\text{Attn}_{2\text{-}1}$ such that

$$\left\| \text{Attn}_{2\text{-}1}\left(\begin{bmatrix} X \\ W_{\text{FF}} \\ Z \end{bmatrix}\right)_{:,i} - \begin{bmatrix} \text{ReLU}(W_1 x_i + b_1) \\ (W_{\text{FF},1}^{(2)})_{:,i} \\ \vdots \\ (W_{\text{FF},d}^{(2)})_{:,i} \\ Z_{:,i} \end{bmatrix} \right\|_\infty \le \max\{\eta_1, \eta_{\text{flow}}\}.$$

**Step 3: Second-Layer Attention for $W_2\text{ReLU}(W_1 x_i + b_1) + b_2$.** We write $h_i := \text{ReLU}(W_1 x_i + b_1)$ and denote $\widehat{h}_i \in \mathbb{R}^r$ as the output of the ReLU-approximation heads from Step 2. We construct $\text{Attn}_{2\text{-}2}$ to implement the in-context affine map on input $\widehat{h}_i$.

Denote the output of the first-layer $\text{Attn}_{2\text{-}1}$ as $U_{\text{input}}$ and define the ideal input as $U_{\text{ideal}}$:

$$U_{\text{input}} := \begin{bmatrix} \widehat{h} \\ W_{\text{FF},1}^{(2)}\sigma_{\beta_2}(I_n) \\ \vdots \\ W_{\text{FF},d}^{(2)}\sigma_{\beta_2}(I_n) \\ Z\sigma_{\beta_2}(I_n) \end{bmatrix}, \quad U_{\text{ideal}} := \begin{bmatrix} \widehat{h} \\ W_{\text{FF},1}^{(2)} \\ \vdots \\ W_{\text{FF},d}^{(2)} \\ Z \end{bmatrix}.$$

From (23), we have

$$\|U_{\text{input}} - U_{\text{ideal}}\|_\infty \le n \max\{B_W, B_Z\} \frac{n-1}{e^{\beta_2} + (n-1)} =: \eta_{\text{flow}}. \tag{24}$$

Then, combining Lemma E.11 and (24), with $X = U_{\text{input}}$ and $\widehat{X} = U_{\text{ideal}}$, we have

$$\|\text{Attn}_{\text{2-2}}(U_{\text{input}}) - \text{Attn}_{\text{2-2}}(U_{\text{ideal}})\|_\infty \le L\eta_{\text{flow}}, \tag{25}$$

where the exact Lipschitz constant is in Lemma E.11.

Following Step 2, we implement $b_2$ by augmenting the hidden state with a constant 1. Define the augmented hidden vector

$$\widetilde{h}_i := \begin{bmatrix} \widehat{h}_i \\ 1 \end{bmatrix} \in \mathbb{R}^{r+1}.$$

Now we bound the linear model to choose the truncation range $[a, b]$ properly. From Step 1 we have $\|h_i\|_\infty \le B_1$, and from (21) we have $\|\widehat{h}_i - h_i\|_\infty \le \eta_1$. Hence

$$\|\widehat{h}_i\|_\infty \le B_1 + \eta_1 \quad \text{and} \quad \|\widetilde{h}_i\|_\infty \le \max\{B_1 + \eta_1, 1\} =: \widetilde{B}_1.$$

For each $k \in [d]$, define $c_k := [(W_2)_{k,:}^\top; (b_2)_k] \in \mathbb{R}^{r+1}$, so $c_k^\top \widetilde{h}_i = (W_2 \widehat{h}_i + b_2)_k$. With elementwise $\|\cdot\|_\infty$ and $\|c_k\|_\infty \le B_W$,

$$|c_k^\top \widetilde{h}_i| \le (r+1)\|c_k\|_\infty \|\widetilde{h}_i\|_\infty \le (r+1) B_W \widetilde{B}_1.$$

Set

$$B_2 := (r+1) B_W \widetilde{B}_1.$$

Apply Corollary D.4.1 with dimension $(r+1)$, $l = d$, and $[a, b] = [-B_2, B_2]$ to the input $U_{\text{ideal}}$. We obtain a multi-head attention module $\text{Attn}_{\text{2-2}}$ using $H_2$ heads per output coordinate such that for all $i \in [n]$,

$$\|\text{Attn}_{\text{2-2}}(U_{\text{ideal}})_{:,i} - \begin{bmatrix} (W_2 \widehat{h}_i + b_2) \\ 0_{d_Z} \end{bmatrix}\|_\infty \le \eta_2, \quad \eta_2 := B_2 \epsilon_0 + \frac{2 B_2}{(n-2) H_2}. \tag{26}$$

For each $i \in [n]$,

$$\|W_2(\widehat{h}_i - h_i)\|_\infty \le r\|W_2\|_\infty \|\widehat{h}_i - h_i\|_\infty \le r B_W \eta_1. \tag{27}$$

Using triangle inequality and (25), (26), (27), we get for all $i \in [n]$,

$$\left\| \text{Attn}_{\text{2-2}} \circ \text{Attn}_{\text{2-1}}\left( \begin{bmatrix} X \\ W_{\text{FF}} \\ Z \end{bmatrix} \right)_{:,i} - \begin{bmatrix} (W_2 h_i + b_2) \\ 0_{d_Z} \end{bmatrix} \right\|_\infty$$

$$\le \|\text{Attn}_{\text{2-2}}(U_{\text{input}})_{:,i} - \text{Attn}_{\text{2-2}}(U_{\text{ideal}})_{:,i}\|_\infty$$

$$+ \left\| \text{Attn}_{\text{2-2}}(U_{\text{ideal}})_{:,i} - \begin{bmatrix} (W_2 \widehat{h}_i + b_2) \\ 0_{d_Z} \end{bmatrix} \right\|_\infty + \left\| \begin{bmatrix} W_2(\widehat{h}_i - h_i) \\ 0_{d_Z} \end{bmatrix} \right\|_\infty$$

$$\le L\eta_{\text{flow}} + \eta_2 + r B_W \eta_1. \tag{28}$$

Finally, as in Step 2, we construct extra heads into $\text{Attn}_{\text{2-2}}$ that copy the $Z$ block into the last $d_Z$ coordinates of the output in (28). Following from (22), the $Z$-block approximation after two-layer attention equals

$$\begin{bmatrix} Z_1 \sigma_{\beta_1}(I_n) \sigma_{\beta_2}(I_n) \\ I_n \\ Z_2 \sigma_{\beta_1}(I_n) \sigma_{\beta_2}(I_n) \end{bmatrix},$$

where $I_n$ can be exactly copied.

We invoke the two-layer flow analysis (see (70)-(79)). For any $\varepsilon_1 > 0$, choose $\beta_1, \beta_2$ as in (79). Then

$$\left\| Z_{\{1,2\}}\big(\sigma_{\beta_1}(I_n)\sigma_{\beta_2}(I_n) - I_n\big) \right\|_\infty \leq 2\varepsilon_1 + \frac{\varepsilon_1^2}{B_Z}. \tag{29}$$

Combined with (28), we finally have

$$\left\| \mathrm{Attn}_{2\text{-}2} \circ \mathrm{Attn}_{2\text{-}1}\left( \begin{bmatrix} X \\ W_{\mathrm{FF}} \\ Z \end{bmatrix} \right) - \begin{bmatrix} \mathrm{FF}(X) \\ Z \end{bmatrix} \right\|_\infty \leq \max\left\{ L\eta_{\mathrm{flow}} + \eta_2 + rB_W\eta_1, 2\varepsilon_1 + \frac{\varepsilon_1^2}{B_Z} \right\}.$$

Choose parameters so that the right-hand side is at most $\epsilon_1$. This completes the proof. $\qquad\square$

## E.2 Proof of Lemma 3.2

**Lemma E.2** (Lemma 3.2 Restated: In-Context Emulation of Feed-Forward Layer). *Under the same notation and assumptions of Lemma 3.1, for any $\epsilon_2 > 0$, there exists a two-layer self-attention $\mathrm{Attn}_2$ such that*

$$\left\| \mathrm{Attn}_2\left( \begin{bmatrix} X \\ W_{\mathrm{FF}} \end{bmatrix} \right) - \mathrm{FF}(X) \right\|_\infty \leq \epsilon_2.$$

*Proof.* We follow the proof of Lemma 3.1 and keep the same constructions and notation for the two attention layers. The only change is that we remove the auxiliary heads that copy $Z$.

After this removal, the two-layer attention output contains only the $\mathrm{FF}(X)$ block. Hence, the final error bound contains only the FF approximation error terms and does not include any $Z$-flow term. Therefore, for any $\epsilon_2$, by choosing proper parameters, we obtain

$$\left\| \mathrm{Attn}_2\left( \begin{bmatrix} X \\ W_{\mathrm{FF}} \end{bmatrix} \right) - \mathrm{FF}(X) \right\|_\infty \leq L\eta_{\mathrm{flow}} + \eta_2 + rB_W\eta_1 \leq \epsilon_2.$$

This completes the proof. $\qquad\square$

## E.3 Proof of Lemma 3.3

**Lemma E.3** (Lemma 3.3 Restated: In-Context Emulation of Single-Head Attention with a Residual Connection and a Flow-Through Component). *Let $X \in \mathbb{R}^{d \times n}$ be an input sequence. Let $\mathrm{Attn}^{\mathrm{Res}}$ be a single-head attention with a residual connection, as introduced in Definition 2.1. Let $X_p \in \mathbb{R}^{(d+2d(2d_h+d)+n)\times n}$ be the prompt from Definition 3.4 with $H = 1$, and the free parameter $d_V = d$. Let $Z \in \mathbb{R}^{d_Z \times n}$. Assume $\|X\|_\infty \leq B_X, \|Z\|_\infty \leq B_Z$ and $\|W_K X\|_\infty, \|W_Q X\|_\infty, \|W_V X\|_\infty \leq B_{KQV}$ for some $B_X, B_Z, B_{KQV} > 0$. Then, for any $\epsilon_3 > 0$, there exists a two-layer self-attention $\mathrm{Attn}_2$ such that*

$$\left\| \mathrm{Attn}_2\left( \begin{bmatrix} X_p \\ Z \end{bmatrix} \right) - \begin{bmatrix} \mathrm{Attn}^{\mathrm{Res}}(X) \\ Z \end{bmatrix} \right\|_\infty \leq \epsilon_3.$$

*Proof.* We first simplify the notation of the input dimension to keep our proof clean. We define $d_p := (d + 2d(2d_h + d) + n)$ and express the input dimension as

$$\underbrace{\begin{bmatrix} X_p \\ Z \end{bmatrix}}_{(d_p + d_Z) \times n}.$$

Also, we use $\mathrm{Attn}_{2\text{-}2} \circ \mathrm{Attn}_{2\text{-}1}$ to denote the two layers of $\mathrm{Attn}_2$.

Then, we first present our high-level proof sketch. Our proof consists of three major steps:

- **Step 1.** In this part of our proof, our goal is to construct an $\mathrm{Attn}_{2\text{-}1}$ such that

$$\mathrm{Attn}_{2\text{-}1}(\ \underbrace{\begin{bmatrix} X_p \\ Z \end{bmatrix}}_{(d_p+d_Z)\times n}\ ) \approx \underbrace{\begin{bmatrix} \mathrm{Attn}(X_p) \\ I_n \\ Z \\ X \end{bmatrix}}_{((2d_h+d)+n+d_Z+d)\times n}\ , \tag{30}$$

where $\mathrm{Attn}$ is the first layer emulator from Theorem D.3, and the $(2d_h + d)$ dimension follows from the proof of Theorem D.3 in Hu et al. (2026b). We achieve this by incorporating the first-layer emulator heads into $\mathrm{Attn}_{2\text{-}1}$.

Next, to propagate $Z$ through $\mathrm{Attn}_{2\text{-}1}$, we add an auxiliary head to exploit the identity $I_n$ in $X_p$. Concretely, we choose the value projection to read $Z$, and the key/query projections to read $I_n$. With an appropriate choice of $\beta$, the resulting softmax scores concentrate on the diagonal, so this head simulates $Z \cdot I_n$ and thus propagates $Z$. We propagate $X$ with the same technique.

- **Step 2.** In this part of our proof, our goal is to construct an $\mathrm{Attn}_{2\text{-}2}$ such that

$$\mathrm{Attn}_{2\text{-}2} \circ \mathrm{Attn}_{2\text{-}1}(\begin{bmatrix} X_p \\ Z \end{bmatrix}) \approx \underbrace{\begin{bmatrix} X + \mathrm{Attn_s} \circ \mathrm{Attn}(X_p) \\ Z \end{bmatrix}}_{(d+d_Z)\times n}, \tag{31}$$

where $\mathrm{Attn_s}$ is the second-layer emulator from Theorem D.3. We achieve the full emulator by incorporating the second-layer emulator head into $\mathrm{Attn}_{2\text{-}2}$. Also, we propagate $X, Z$ with the same technique as in $\mathrm{Attn}_{2\text{-}1}$ (utilize the identity), and route them to the desired row with output projection $W_O$. Summing all heads gives us the residual connection.

- **Step 3.** In this part of our proof, we analyze the error between

$$\mathrm{Attn}_{2\text{-}2} \circ \mathrm{Attn}_{2\text{-}1}(\begin{bmatrix} X_p \\ Z \end{bmatrix}) \approx \begin{bmatrix} X + \mathrm{Attn_s} \circ \mathrm{Attn}(X_p) \\ Z \end{bmatrix} \approx \begin{bmatrix} \mathrm{Attn}^{\mathrm{Res}}(X) \\ Z \end{bmatrix}.$$

There are two source of error. The first one comes from simulating identity $I_n$ with softmax scores. The other one comes from attention emulation. The first part of error is bounded by the error from diagonal terms, and this can be controlled by a large enough $\beta$. The second part of error (emulation error) is controlled by Theorem D.3. With these two parts being controlled, we complete the proof.

**Step 1.** To achieve (30), we construct $\mathrm{Attn}_{2\text{-}1}$ to be an $(H + 3)$-head attention. The first $H$ heads reconstruct $\mathrm{Attn}(X_p)$. The $(H + 1)$-th head outputs the middle $I_n$, and the $(H + 2)$-th head outputs the approximation of $Z$. The last 1 head outputs an approximation of $X$.

To keep our proof clear, we denote the $h$-th head from $\mathrm{Attn}_{2\text{-}1}$ as $\mathrm{Attn}_{2\text{-}1}^{(h)}$ and the weight matrices from the $h$-th head as $W_{1,K}^h, W_{1,Q}^h, W_{1,V}^h$ and $W_{1,O}^h$.

Now we construct the heads in $\mathrm{Attn}_{2\text{-}1}$.

We construct the weight matrices of the first $H$ heads as follows:

$$W_{1,O}^h := \underbrace{\begin{bmatrix} I_{2d_h+d} \\ 0_{n\times(2d_h+d)} \\ 0_{d_Z\times(2d_h+d)} \\ 0_{d\times(2d_h+d)} \end{bmatrix}}_{((2d_h+d)+n+d_Z+d)\times(2d_h+d)} \tag{32}$$

$$W_{1,V}^h := \underbrace{\begin{bmatrix} W_V^h & 0_{(2d_h+d)\times d_Z} \end{bmatrix}}_{(2d_h+d)\times(d_p+d_Z)} \tag{33}$$

$$W_{1,K}^h := \underbrace{\begin{bmatrix} W_K^h & 0_{(d+1)\times d_Z} \end{bmatrix}}_{(d+1)\times(d_p+d_Z)} \tag{34}$$

$$W_{1,Q}^h := \underbrace{\begin{bmatrix} W_Q^h & 0_{(d+1)\times d_Z} \end{bmatrix}}_{(d+1)\times(d_p+d_Z)}, \tag{35}$$

where $W_K^h, W_Q^h$ and $W_V^h$ denote the weight matrices of the first-layer emulator, and their dimensions follow from the proof of Theorem D.3, and we set $d_V \to d$.

The construction provides us with

$$\underbrace{W_{1,V}^h}_{(2d_h+d)\times(d_p+d_Z)} \cdot \underbrace{\begin{bmatrix} X_p \\ Z \end{bmatrix}}_{(d_p+d_Z)\times n} = \underbrace{\begin{bmatrix} W_V^h & 0_{(2d_h+d)\times d_Z} \end{bmatrix}}_{(2d_h+d)\times(d_p+d_Z)} \cdot \underbrace{\begin{bmatrix} X_p \\ Z \end{bmatrix}}_{(d_p+d_Z)\times n} = \underbrace{W_V^h}_{(2d_h+d)\times d_p} \cdot \underbrace{X_p}_{d_p\times n} \qquad \text{(By (33))}$$

$$\underbrace{W_{1,K}^h}_{(d+1)\times(d_p+d_Z)} \cdot \underbrace{\begin{bmatrix} X_p \\ Z \end{bmatrix}}_{(d_p+d_Z)\times n} = \underbrace{\begin{bmatrix} W_K^h & 0_{(d+1)\times d_Z} \end{bmatrix}}_{(d+1)\times(d_p+d_Z)} \cdot \underbrace{\begin{bmatrix} X_p \\ Z \end{bmatrix}}_{(d_p+d_Z)\times n} = \underbrace{W_K^h}_{(d+1)\times d_p} \cdot \underbrace{X_p}_{d_p\times n} \qquad \text{(By (34))}$$

$$\underbrace{W_{1,Q}^h}_{(d+1)\times(d_p+d_Z)} \cdot \underbrace{\begin{bmatrix} X_p \\ Z \end{bmatrix}}_{(d_p+d_Z)\times n} = \underbrace{\begin{bmatrix} W_Q^h & 0_{(d+1)\times d_Z} \end{bmatrix}}_{(d+1)\times(d_p+d_Z)} \cdot \underbrace{\begin{bmatrix} X_p \\ Z \end{bmatrix}}_{(d_p+d_Z)\times n} = \underbrace{W_Q^h}_{(d+1)\times d_p} \cdot \underbrace{X_p}_{d_p\times n}. \qquad \text{(By (35))}$$

Thus, for $h \in [H]$, we have

$$\text{Attn}_{2\text{-}1}^{(h)}(\begin{bmatrix} X_p \\ Z \end{bmatrix}) = W_{1,O}^h \cdot W_{1,V}^h \begin{bmatrix} X_p \\ Z \end{bmatrix} \cdot \sigma_{\beta_h}((W_{1,K}^h \begin{bmatrix} X_p \\ Z \end{bmatrix})^\top W_{1,Q}^h \begin{bmatrix} X_p \\ Z \end{bmatrix})$$

$$= \underbrace{\begin{bmatrix} I_{2d_h+d} \\ 0_{n\times(2d_h+d)} \\ 0_{d_Z\times(2d_h+d)} \\ 0_{d\times(2d_h+d)} \end{bmatrix}}_{((2d_h+d)+n+d_Z+d)\times(2d_h+d)} \cdot \underbrace{W_V^h X_p}_{(2d_h+d)\times n} \cdot \underbrace{\sigma_{\beta_h}((W_K^h X_p)^\top W_Q^h X_p)}_{n\times n} \qquad \text{(By (32))}$$

$$= \underbrace{\begin{bmatrix} W_V^h X_p \cdot \sigma_{\beta_h}((W_K^h X_p)^\top W_Q^h X_p) \\ 0_{n\times n} \\ 0_{d_Z\times n} \\ 0_{d\times n} \end{bmatrix}}_{((2d_h+d)+n+d_Z+d)\times n}, \qquad \text{(By matrix multiplication)}$$

where $\beta_h$ denotes the temperature of the $h$-th head in the first-layer emulator.

Then, we have

$$\sum_{h=1}^H \text{Attn}_{2\text{-}1}^{(h)}(\begin{bmatrix} X_p \\ Z \end{bmatrix})$$

$$= \begin{bmatrix} \sum_{h=1}^H W_V^h X_p \cdot \sigma_{\beta_h}((W_K^h X_p)^\top W_Q^h X_p) \\ 0_{n\times n} \\ 0_{d_Z\times n} \\ 0_{d\times n} \end{bmatrix}$$

$$= \underbrace{\begin{bmatrix} \text{Attn}(X_p) \\ 0_{n\times n} \\ 0_{d_Z\times n} \\ 0_{d\times n} \end{bmatrix}}_{((2d_h+d)+n+d_Z+d)\times n}. \qquad (36)$$

Secondly, we construct the weight matrices of the $(H+1)$-th head of $\text{Attn}_{2\text{-}1}$ as

$$W_{1,K}^{H+1} := \underbrace{\begin{bmatrix} 0_{n\times d} & 0_{n\times 2d(2d_h+d)} & I_n & 0_{n\times d_Z} \end{bmatrix}}_{n\times(d_p+d_Z)}, \qquad (37)$$

$$W_{1,Q}^{H+1} := \underbrace{\begin{bmatrix} 0_{n\times d} & 0_{n\times 2d(2d_h+d)} & M_{n\times n} & 0_{n\times d_Z} \end{bmatrix}}_{n\times(d_p+d_Z)}, \tag{38}$$

$$W_{1,V}^{H+1} := \underbrace{\begin{bmatrix} 0_{n\times d} & 0_{n\times 2d(2d_h+d)} & A_{n\times n}^{-1} & 0_{n\times d_Z} \end{bmatrix}}_{n\times(d_p+d_Z)}, \tag{39}$$

$$W_{1,O}^{H+1} := \underbrace{\begin{bmatrix} 0_{(2d_h+d)\times n} \\ I_n \\ 0_{d_Z\times n} \\ 0_{d\times n} \end{bmatrix}}_{((2d_h+d)+n+d_Z+d)\times n}, \tag{40}$$

where

$$M := \frac{1}{\beta_1}\ln(A), \quad \text{for some } \beta_1 > 0, \tag{41}$$

and $\ln(\cdot)$ denotes the element-wise log. Also, we define

$$A := (1-\alpha)I_n + \frac{\alpha}{n}J, \quad \text{for some } 0 < \alpha < 1, \tag{42}$$

$$J := \underbrace{\begin{bmatrix} 1 \\ 1 \\ \vdots \\ 1 \end{bmatrix}}_{n\times 1} \underbrace{\begin{bmatrix} 1 & 1\cdots & 1 \end{bmatrix}}_{1\times n} = \underbrace{\begin{bmatrix} 1 & 1 & \cdots & 1 \\ 1 & 1 & \cdots & 1 \\ \vdots & \vdots & \ddots & \vdots \\ 1 & 1 & \cdots & 1 \end{bmatrix}}_{n\times n}, \tag{43}$$

and $A^{-1}$ is

$$A^{-1} = \frac{1}{1-\alpha}I_n - \frac{\alpha}{n(1-\alpha)}J.$$

The construction provides us with

$$W_{1,K}^{H+1}\begin{bmatrix} X \\ W_{\text{in}} \\ I_n \\ Z \end{bmatrix} = \underbrace{\begin{bmatrix} 0_{n\times d} & 0_{n\times 2d(2d_h+d)} & I_n & 0_{n\times d_Z} \end{bmatrix}}_{n\times(d_p+d_Z)} \underbrace{\begin{bmatrix} X \\ W_{\text{in}} \\ I_n \\ Z \end{bmatrix}}_{(d_p+d_Z)\times n} = I_n \qquad \text{(By (37))}$$

$$W_{1,Q}^{H+1}\begin{bmatrix} X \\ W_{\text{in}} \\ I_n \\ Z \end{bmatrix} = \underbrace{\begin{bmatrix} 0_{n\times d} & 0_{n\times 2d(2d_h+d)} & M_{n\times n} & 0_{n\times d_Z} \end{bmatrix}}_{n\times(d_p+d_Z)} \underbrace{\begin{bmatrix} X \\ W_{\text{in}} \\ I_n \\ Z \end{bmatrix}}_{(d_p+d_Z)\times n} = M_{n\times n} \qquad \text{(By (38))}$$

$$W_{1,V}^{H+1}\begin{bmatrix} X \\ W_{\text{in}} \\ I_n \\ Z \end{bmatrix} = \underbrace{\begin{bmatrix} 0_{n\times d} & 0_{n\times 2d(2d_h+d)} & A_{n\times n}^{-1} & 0_{n\times d_Z} \end{bmatrix}}_{n\times(d_p+d_Z)} \underbrace{\begin{bmatrix} X \\ W_{\text{in}} \\ I_n \\ Z \end{bmatrix}}_{(d_p+d_Z)\times n} = A_{n\times n}^{-1}, \qquad \text{(By (39))}$$

such that

$$\sigma_{\beta_1}\left(\left(W_{1,K}^{H+1}\begin{bmatrix} X_p \\ Z \end{bmatrix}\right)^\top \left(W_{1,Q}^{H+1}\begin{bmatrix} X_p \\ Z \end{bmatrix}\right)\right)$$

$$= \sigma_{\beta_1}(I_n^\top M)$$

$$= \sigma_{\beta_1}(M) \hspace{4cm} (\text{By } I_n^\top M = I_n M = M)$$

and each entry in $\sigma_{\beta_1}(M)$ is

$$\sigma_{\beta_1}(M)_{ii} = \frac{\exp\left\{\beta_1 \cdot \frac{1}{\beta_1} \ln\left(1 - \alpha + \frac{\alpha}{n}\right)\right\}}{\exp\left\{\beta_1 \cdot \frac{1}{\beta_1} \ln(1 - \alpha)\right\} + n \cdot \exp\left\{\beta_1 \cdot \frac{1}{\beta_1} \ln\left(\frac{\alpha}{n}\right)\right\}} = 1 - \alpha + \frac{\alpha}{n}, \hspace{1cm} (\text{By (1) and (41)})$$

$$\sigma_{\beta}(M)_{ij} = \frac{\exp\left\{\beta_1 \cdot \frac{1}{\beta_1} \ln\left(\frac{\alpha}{n}\right)\right\}}{\exp\left\{\beta_1 \cdot \frac{1}{\beta_1} \ln(1 - \alpha)\right\} + n \cdot \exp\left\{\beta_1 \cdot \frac{1}{\beta} \ln\left(\frac{\alpha}{n}\right)\right\}} = \frac{\alpha}{n}. \hspace{1cm} (\text{By (1) and (41)})$$

Thus,

$$\sigma_{\beta_1}(M) = (1 - \alpha)I_n + \frac{\alpha}{n}J = A,$$

and

$$W_{1,O}^{H+1} \cdot W_{1,V}^{H+1} \begin{bmatrix} X_p \\ Z \end{bmatrix} \cdot \sigma_{\beta_1}\left(\left(W_{1,K}^{H+1} \begin{bmatrix} X_p \\ Z \end{bmatrix}\right)^\top \left(W_{1,Q}^{H+1} \begin{bmatrix} X_p \\ Z \end{bmatrix}\right)\right)$$

$$= \underbrace{\begin{bmatrix} 0_{(2d_h+d)\times n} \\ I_n \\ 0_{d_Z \times n} \\ 0_{d \times n} \end{bmatrix}}_{((2d_h+d)+n+d_Z+d)\times n} \cdot \underbrace{A^{-1}A}_{n \times n} = \begin{bmatrix} 0_{(2d_h+d)\times n} \\ I_n \\ 0_{d_Z \times n} \\ 0_{d \times n} \end{bmatrix} \cdot I_n = \begin{bmatrix} 0_{(2d_h+d)\times n} \\ I_n \\ 0_{d_Z \times n} \\ 0_{d \times n} \end{bmatrix}, \hspace{1cm} (44)$$

where the last equality follows from (40).

Thirdly, we construct the weight matrices of the $(H + 2)$-th head as follows:

$$W_{1,O}^{H+2} := \underbrace{\begin{bmatrix} 0_{(2d_h+d)\times d_Z} \\ 0_{n \times d_Z} \\ I_{d_Z} \\ 0_{d \times d_Z} \end{bmatrix}}_{((2d_h+d)+n+d_Z+d)\times d_Z} \hspace{3cm} (45)$$

$$W_{1,V}^{H+2} := \underbrace{\begin{bmatrix} 0_{d_Z \times d_p} & I_{d_Z} \end{bmatrix}}_{d_Z \times (d_p+d_Z)} \hspace{3cm} (46)$$

$$W_{1,K}^{H+2}, W_{1,Q}^{H+2} := \underbrace{\begin{bmatrix} 0_{n \times d} & 0_{n \times 2d(2d_h+d)} & I_n & 0_{n \times d_Z} \end{bmatrix}}_{n \times (d_p+d_Z)}. \hspace{2cm} (47)$$

The construction provides us with

$$W_{1,V}^{H+2} \begin{bmatrix} X_p \\ Z \end{bmatrix} = \underbrace{\begin{bmatrix} 0_{d_Z \times d_p} & I_{d_Z} \end{bmatrix}}_{d_Z \times (d_p+d_Z)} \cdot \underbrace{\begin{bmatrix} X_p \\ Z \end{bmatrix}}_{(d_p+d_Z)\times n} = \underbrace{Z}_{d_Z \times n} \hspace{2cm} (\text{By (46)})$$

$$W_{1,K}^{H+2} \begin{bmatrix} X \\ W_{\text{in}} \\ I_n \\ Z \end{bmatrix} = \underbrace{\begin{bmatrix} 0_{n \times d} & 0_{n \times 2d(2d_h+d)} & I_n & 0_{n \times d_Z} \end{bmatrix}}_{n \times (d_p+d_Z)} \cdot \underbrace{\begin{bmatrix} X \\ W_{\text{in}} \\ I_n \\ Z \end{bmatrix}}_{(d_p+d_Z)\times n} = I_n \hspace{1cm} (\text{By (47)})$$

$$W_{1,Q}^{H+2} \begin{bmatrix} X \\ W_{\text{in}} \\ I_n \\ Z \end{bmatrix} = \underbrace{\begin{bmatrix} 0_{n \times d} & 0_{n \times 2d(2d_h+d)} & I_n & 0_{n \times d_Z} \end{bmatrix}}_{n \times (d_p+d_Z)} \cdot \underbrace{\begin{bmatrix} X \\ W_{\text{in}} \\ I_n \\ Z \end{bmatrix}}_{(d_p+d_Z)\times n} = I_n. \hspace{1cm} (\text{By (47)})$$

Thus, we have

$$
\begin{aligned}
\mathrm{Attn}_{2\text{-}1}^{(H+2)}\left(\begin{bmatrix} X_p \\ Z \end{bmatrix}\right) &= W_{1,O}^{H+2} \cdot W_{1,V}^{H+2} \begin{bmatrix} X_p \\ Z \end{bmatrix} \cdot \sigma_{\beta_2}\left((W_{1,K}^{H+2}\begin{bmatrix} X_p \\ Z \end{bmatrix})^\top W_{1,Q}^{H+2}\begin{bmatrix} X_p \\ Z \end{bmatrix}\right) \\
&= \underbrace{\begin{bmatrix} 0_{(2d_h+d)\times d_Z} \\ 0_{n\times d_Z} \\ I_{d_Z} \\ 0_{d\times d_Z} \end{bmatrix}}_{((2d_h+d)+n+d_Z+d)\times d_Z} \cdot \underbrace{Z}_{d_Z\times n} \cdot \underbrace{\sigma_{\beta_2}((I_n)^\top I_n)}_{n\times n} \\
&= \underbrace{\begin{bmatrix} 0_{(2d_h+d)\times n} \\ 0_{n\times n} \\ Z\sigma_{\beta_2}(I_n) \\ 0_{d\times n} \end{bmatrix}}_{((2d_h+d)+n+d_Z+d)\times n} .
\end{aligned} \tag{48}
$$

Lastly, we construct the weight matrices of the $(H+3)$-th head as follows:

$$
W_{1,O}^{H+3} := \underbrace{\begin{bmatrix} 0_{(2d_h+d)\times d} \\ 0_{n\times d} \\ 0_{d_Z\times d} \\ I_d \end{bmatrix}}_{((2d_h+d)+n+d_Z+d)\times d} \tag{49}
$$

$$
W_{1,V}^{H+3} := \underbrace{\begin{bmatrix} I_d & 0_{d\times 2d(2d_h+d)} & 0_{d\times n} & 0_{d\times d_Z} \end{bmatrix}}_{d\times(d_p+d_Z)} \tag{50}
$$

$$
W_{1,K}^{H+3}, W_{1,Q}^{H+3} := \underbrace{\begin{bmatrix} 0_{n\times d} & 0_{n\times 2d(2d_h+d)} & I_n & 0_{n\times d_Z} \end{bmatrix}}_{n\times(d_p+d_Z)} . \tag{51}
$$

The construction provides us with

$$
W_{1,V}^{H+3}\begin{bmatrix} X \\ W_{\mathrm{in}} \\ I_n \\ Z \end{bmatrix} = \underbrace{\begin{bmatrix} I_d & 0_{d\times 2d(2d_h+d)} & 0_{d\times n} & 0_{d\times d_Z} \end{bmatrix}}_{d\times(d_p+d_Z)} \cdot \underbrace{\begin{bmatrix} X \\ W_{\mathrm{in}} \\ I_n \\ Z \end{bmatrix}}_{(d_p+d_Z)\times n} = \underbrace{X}_{d\times n} \qquad \text{(By (50))}
$$

$$
W_{1,K}^{H+3}\begin{bmatrix} X \\ W_{\mathrm{in}} \\ I_n \\ Z \end{bmatrix} = \underbrace{\begin{bmatrix} 0_{n\times d} & 0_{n\times 2d(2d_h+d)} & I_n & 0_{n\times d_Z} \end{bmatrix}}_{n\times(d_p+d_Z)} \cdot \underbrace{\begin{bmatrix} X \\ W_{\mathrm{in}} \\ I_n \\ Z \end{bmatrix}}_{(d_p+d_Z)\times n} = I_n \qquad \text{(By (47))}
$$

$$
W_{1,Q}^{H+3}\begin{bmatrix} X \\ W_{\mathrm{in}} \\ I_n \\ Z \end{bmatrix} = \underbrace{\begin{bmatrix} 0_{n\times d} & 0_{n\times 2d(2d_h+d)} & I_n & 0_{n\times d_Z} \end{bmatrix}}_{n\times(d_p+d_Z)} \cdot \underbrace{\begin{bmatrix} X \\ W_{\mathrm{in}} \\ I_n \\ Z \end{bmatrix}}_{(d_p+d_Z)\times n} = I_n . \qquad \text{(By (47))}
$$

Thus, we have

$$
\mathrm{Attn}_{2\text{-}1}^{(H+3)}\left(\begin{bmatrix} X_p \\ Z \end{bmatrix}\right) = W_{1,O}^{H+3} \cdot W_{1,V}^{H+3} \begin{bmatrix} X_p \\ Z \end{bmatrix} \cdot \sigma_{\beta_3}\left((W_{1,K}^{H+3}\begin{bmatrix} X_p \\ Z \end{bmatrix})^\top W_{1,Q}^{H+3}\begin{bmatrix} X_p \\ Z \end{bmatrix}\right)
$$

$$
= \underbrace{\begin{bmatrix} 0_{(2d_h+d)\times d} \\ 0_{n\times d} \\ 0_{d_Z\times d} \\ I_d \end{bmatrix}}_{((2d_h+d)+n+d_Z+d)\times d} \cdot \underbrace{X}_{d\times n} \cdot \underbrace{\sigma_{\beta_3}(I_n^\top I_n)}_{n\times n}
$$

$$
= \underbrace{\begin{bmatrix} 0_{(2d_h+d)\times n} \\ 0_{n\times n} \\ 0_{d_Z\times n} \\ X\sigma_{\beta_3}(I_n) \end{bmatrix}}_{((2d_h+d)+n+d_Z+d)\times n} . \tag{52}
$$

Finally, all of the heads from $\mathrm{Attn}_{2\text{-}1}$ give us

$$
\mathrm{Attn}_{2\text{-}1}\left(\begin{bmatrix} X_p \\ Z \end{bmatrix}\right) = \sum_{h=1}^{H} \mathrm{Attn}_{2\text{-}1}^{(h)}\left(\begin{bmatrix} X_p \\ Z \end{bmatrix}\right) + \mathrm{Attn}_{2\text{-}1}^{(H+1)}\left(\begin{bmatrix} X_p \\ Z \end{bmatrix}\right) + \mathrm{Attn}_{2\text{-}1}^{(H+2)}\left(\begin{bmatrix} X_p \\ Z \end{bmatrix}\right) + \mathrm{Attn}_{2\text{-}1}^{(H+3)}\left(\begin{bmatrix} X_p \\ Z \end{bmatrix}\right)
$$

$$
= \underbrace{\begin{bmatrix} \mathrm{Attn}_m(X_p) \\ 0_{n\times n} \\ 0_{d_Z\times n} \\ 0_{d\times n} \end{bmatrix}}_{((2d_h+d)+n+d_Z+d)\times n} + \underbrace{\begin{bmatrix} 0_{(2d_h+d)\times n} \\ I_n \\ 0_{d_Z\times n} \\ 0_{d\times n} \end{bmatrix}}_{((2d_h+d)+n+d_Z+d)\times n} + \underbrace{\begin{bmatrix} 0_{(2d_h+d)\times n} \\ 0_{n\times n} \\ Z\sigma_{\beta_2}(I_n) \\ 0_{d\times n} \end{bmatrix}}_{((2d_h+d)+n+d_Z+d)\times n} + \underbrace{\begin{bmatrix} 0_{(2d_h+d)\times n} \\ 0_{n\times n} \\ 0_{d_Z\times n} \\ X\sigma_{\beta_3}(I_n) \end{bmatrix}}_{((2d_h+d)+n+d_Z+d)\times n}
$$

$$
\text{(By (36), (44), (48) and (52))}
$$

$$
= \underbrace{\begin{bmatrix} \mathrm{Attn}_m(X_p) \\ I_n \\ Z\sigma_{\beta_2}(I_n) \\ X\sigma_{\beta_3}(I_n) \end{bmatrix}}_{((2d_h+d)+n+d_Z+d)\times n} .
$$

**Step 2.** To achieve (31), we construct $\mathrm{Attn}_{2\text{-}2}$ as a three-head attention layer. The first head reconstructs $\mathrm{Attn}_s \circ \mathrm{Attn}(X_p)$. The second head outputs an approximation of $Z$. The third head outputs an approximation of $X$ to the first row block. Summing these three heads realizes the residual connection and the flow-through component.

Again, to keep our proof clear, we denote the $h$-th head from $\mathrm{Attn}_{2\text{-}2}$ as $\mathrm{Attn}_{2\text{-}2}^{(h)}$ and the weight matrices from the $h$-th head as $W_{2,K}^h, W_{2,Q}^h, W_{2,V}^h$ and $W_{2,O}^h$.

We construct the first head of $\mathrm{Attn}_{2\text{-}2}$ as

$$
W_{2,O}^1 := \underbrace{\begin{bmatrix} I_d \\ 0_{d_Z\times d} \end{bmatrix}}_{(d+d_Z)\times d} \tag{53}
$$

$$
W_{2,V}^1 := \underbrace{\begin{bmatrix} W_V^s & 0_{d\times n} & 0_{d\times d_Z} & 0_{d\times d} \end{bmatrix}}_{d\times((2d_h+d)+n+d_Z+d)} \tag{54}
$$

$$
W_{2,K}^1 := \underbrace{\begin{bmatrix} W_K^s & 0_{d\times n} & 0_{d\times d_Z} & 0_{d\times d} \end{bmatrix}}_{d_h\times((2d_h+d)+n+d_Z+d)} \tag{55}
$$

$$
W_{2,Q}^1 := \underbrace{\begin{bmatrix} W_Q^s & 0_{d\times n} & 0_{d\times d_Z} & 0_{d\times d} \end{bmatrix}}_{d_h\times((2d_h+d)+n+d_Z+d)}, \tag{56}
$$

where the dimensions of $W_K^s, W_Q^s$ and $W_V^s$ follow from the proof of Theorem D.3, and we set $d_V \to d$.

The construction provides us with

$$
W_{2,V}^1 \begin{bmatrix} \mathrm{Attn}(X_p) \\ I_n \\ Z\sigma_{\beta_2}(I_n) \\ X\sigma_{\beta_3}(I_n) \end{bmatrix} = \underbrace{\begin{bmatrix} W_V^{\mathrm{s}} & 0_{d\times n} & 0_{d\times d_Z} & 0_{d\times d} \end{bmatrix}}_{d\times((2d_h+d)+n+d_Z+d)} \underbrace{\begin{bmatrix} \mathrm{Attn}(X_p) \\ I_n \\ Z\sigma_{\beta_2}(I_n) \\ X\sigma_{\beta_3}(I_n) \end{bmatrix}}_{((2d_h+d)+n+d_Z+d)\times n} = \underbrace{W_V^{\mathrm{s}}}_{d\times(2d_h+d)} \cdot \underbrace{\mathrm{Attn}(X_p)}_{(2d_h+d)\times n} \qquad (\text{By } (54))
$$

$$
W_{2,K}^1 \begin{bmatrix} \mathrm{Attn}(X_p) \\ I_n \\ Z\sigma_{\beta_2}(I_n) \\ X\sigma_{\beta_3}(I_n) \end{bmatrix} = \underbrace{\begin{bmatrix} W_K^{\mathrm{s}} & 0_{d\times n} & 0_{d\times d_Z} & 0_{d\times d} \end{bmatrix}}_{d_h\times((2d_h+d)+n+d_Z+d)} \underbrace{\begin{bmatrix} \mathrm{Attn}(X_p) \\ I_n \\ Z\sigma_{\beta_2}(I_n) \\ X\sigma_{\beta_3}(I_n) \end{bmatrix}}_{((2d_h+d)+n+d_Z+d)\times n} = \underbrace{W_K^{\mathrm{s}}}_{d_h\times(2d_h+d)} \cdot \underbrace{\mathrm{Attn}(X_p)}_{(2d_h+d)\times n} \qquad (\text{By } (55))
$$

$$
W_{2,Q}^1 \begin{bmatrix} \mathrm{Attn}(X_p) \\ I_n \\ Z\sigma_{\beta_2}(I_n) \\ X\sigma_{\beta_3}(I_n) \end{bmatrix} = \underbrace{\begin{bmatrix} W_Q^{\mathrm{s}} & 0_{d\times n} & 0_{d\times d_Z} & 0_{d\times d} \end{bmatrix}}_{d_h\times((2d_h+d)+n+d_Z+d)} \underbrace{\begin{bmatrix} \mathrm{Attn}(X_p) \\ I_n \\ Z\sigma_{\beta_2}(I_n) \\ X\sigma_{\beta_3}(I_n) \end{bmatrix}}_{((2d_h+d)+n+d_Z+d)\times n} = \underbrace{W_Q^{\mathrm{s}}}_{d_h\times(2d_h+d)} \cdot \underbrace{\mathrm{Attn}(X_p)}_{(2d_h+d)\times n}. \qquad (\text{By } (56))
$$

Thus, we have

$$
\mathrm{Attn}_{2\text{-}2}^{(1)}\left( \begin{bmatrix} \mathrm{Attn}(X_p) \\ I_n \\ Z\sigma_{\beta_2}(I_n) \\ X\sigma_{\beta_3}(I_n) \end{bmatrix} \right)
$$

$$
= W_{2,O}^1 \cdot W_{2,V}^1 \begin{bmatrix} \mathrm{Attn}(X_p) \\ I_n \\ Z\sigma_{\beta_2}(I_n) \\ X\sigma_{\beta_3}(I_n) \end{bmatrix} \cdot \sigma_\beta\left( \left( W_{2,K}^1 \begin{bmatrix} \mathrm{Attn}(X_p) \\ I_n \\ Z\sigma_{\beta_2}(I_n) \\ X\sigma_{\beta_3}(I_n) \end{bmatrix} \right)^\top W_{2,Q}^1 \begin{bmatrix} \mathrm{Attn}(X_p) \\ I_n \\ Z\sigma_{\beta_2}(I_n) \\ X\sigma_{\beta_3}(I_n) \end{bmatrix} \right)
$$

$$
= \underbrace{\begin{bmatrix} I_d \\ 0_{d_Z\times d} \end{bmatrix}}_{(d+d_Z)\times d} \cdot \underbrace{W_V^{\mathrm{s}}\mathrm{Attn}(X_p)}_{d\times n} \cdot \underbrace{\sigma_\beta\left( (W_K^{\mathrm{s}}\mathrm{Attn}(X_p))^\top W_Q^{\mathrm{s}}\mathrm{Attn}(X_p) \right)}_{n\times n} \qquad (\text{By } (53))
$$

$$
= \begin{bmatrix} W_V^{\mathrm{s}}\mathrm{Attn}(X_p) \cdot \sigma_\beta\left( (W_K^{\mathrm{s}}\mathrm{Attn}(X_p))^\top W_Q^{\mathrm{s}}\mathrm{Attn}(X_p) \right) \\ 0_{d_Z\times n} \end{bmatrix}
$$

$$
= \begin{bmatrix} \mathrm{Attn_s} \circ \mathrm{Attn}(X_p) \\ 0_{d_Z\times n} \end{bmatrix}, \qquad (57)
$$

where $\beta$ denotes the softmax temperature of the first layer emulator in Theorem D.3.

Next, we construct the second head of $\mathrm{Attn}_{2\text{-}2}$ as

$$
W_{2,O}^2 := \underbrace{\begin{bmatrix} 0_{d\times d_Z} \\ I_{d_Z} \end{bmatrix}}_{(d+d_Z)\times d_Z} \qquad (58)
$$

$$
W_{2,V}^2 := \underbrace{\begin{bmatrix} 0_{d_Z\times(2d_h+d)} & 0_{d_Z\times n} & I_{d_Z} & 0_{d_Z\times d} \end{bmatrix}}_{d_Z\times((2d_h+d)+n+d_Z+d)} \qquad (59)
$$

$$
W_{2,K}^2 := \underbrace{\begin{bmatrix} 0_{n\times(2d_h+d)} & I_n & 0_{n\times d_Z} & 0_{n\times d} \end{bmatrix}}_{n\times((2d_h+d)+n+d_Z+d)} \qquad (60)
$$

$$
W_{2,Q}^2 := \underbrace{\begin{bmatrix} 0_{n\times(2d_h+d)} & I_n & 0_{n\times d_Z} & 0_{n\times d} \end{bmatrix}}_{n\times((2d_h+d)+n+d_Z+d)}, \qquad (61)
$$

such that

$$W_{2,V}^2 \begin{bmatrix} \text{Attn}(X_p) \\ I_n \\ Z\sigma_{\beta_2}(I_n) \\ X\sigma_{\beta_3}(I_n) \end{bmatrix} = \underbrace{\begin{bmatrix} 0_{d_Z \times (2d_h+d)} & 0_{d_Z \times n} & I_{d_Z} & 0_{d_Z \times d} \end{bmatrix}}_{d_Z \times ((2d_h+d)+n+d_Z+d)} \underbrace{\begin{bmatrix} \text{Attn}(X_p) \\ I_n \\ Z\sigma_{\beta_2}(I_n) \\ X\sigma_{\beta_3}(I_n) \end{bmatrix}}_{((2d_h+d)+n+d_Z+d)\times n} = \underbrace{Z}_{d_Z \times n}\underbrace{\sigma_{\beta_2}(I_n)}_{n \times n} \qquad \text{(By (59))}$$

$$W_{2,K}^2 \begin{bmatrix} \text{Attn}(X_p) \\ I_n \\ Z\sigma_{\beta_2}(I_n) \\ X\sigma_{\beta_3}(I_n) \end{bmatrix} = \underbrace{\begin{bmatrix} 0_{n \times (2d_h+d)} & I_n & 0_{n \times d_Z} & 0_{n \times d} \end{bmatrix}}_{n \times ((2d_h+d)+n+d_Z+d)} \underbrace{\begin{bmatrix} \text{Attn}(X_p) \\ I_n \\ Z\sigma_{\beta_2}(I_n) \\ X\sigma_{\beta_3}(I_n) \end{bmatrix}}_{((2d_h+d)+n+d_Z+d)\times n} = I_n \qquad \text{(By (60))}$$

$$W_{2,Q}^2 \begin{bmatrix} \text{Attn}(X_p) \\ I_n \\ Z\sigma_{\beta_2}(I_n) \\ X\sigma_{\beta_3}(I_n) \end{bmatrix} = \underbrace{\begin{bmatrix} 0_{n \times (2d_h+d)} & I_n & 0_{n \times d_Z} & 0_{n \times d} \end{bmatrix}}_{n \times ((2d_h+d)+n+d_Z+d)} \underbrace{\begin{bmatrix} \text{Attn}(X_p) \\ I_n \\ Z\sigma_{\beta_2}(I_n) \\ X\sigma_{\beta_3}(I_n) \end{bmatrix}}_{((2d_h+d)+n+d_Z+d)\times n} = I_n. \qquad \text{(By (60))}$$

Thus, we have

$$\text{Attn}_{2\text{-}2}^{(2)}\left(\begin{bmatrix} \text{Attn}(X_p) \\ I_n \\ Z\sigma_{\beta_2}(I_n) \\ X\sigma_{\beta_3}(I_n) \end{bmatrix}\right)$$

$$= W_{2,O}^2 \cdot W_{2,V}^2 \begin{bmatrix} \text{Attn}(X_p) \\ I_n \\ Z\sigma_{\beta_2}(I_n) \\ X\sigma_{\beta_3}(I_n) \end{bmatrix} \cdot \sigma_{\beta_4}\left(\left(W_{2,K}^2 \begin{bmatrix} \text{Attn}(X_p) \\ I_n \\ Z\sigma_{\beta_2}(I_n) \\ X\sigma_{\beta_3}(I_n) \end{bmatrix}\right)^\top W_{2,Q}^2 \begin{bmatrix} \text{Attn}(X_p) \\ I_n \\ Z\sigma_{\beta_2}(I_n) \\ X\sigma_{\beta_3}(I_n) \end{bmatrix}\right)$$

$$= \underbrace{\begin{bmatrix} 0_{d \times d_Z} \\ I_{d_Z} \end{bmatrix}}_{(d+d_Z)\times d_Z} \cdot \underbrace{Z\sigma_{\beta_2}(I_n)}_{d_Z \times n} \cdot \underbrace{\sigma_{\beta_4}(I_n^\top I_n)}_{n \times n} \qquad \text{(By (58))}$$

$$= \underbrace{\begin{bmatrix} 0_{d \times n} \\ Z\sigma_{\beta_2}(I_n)\sigma_{\beta_4}(I_n) \end{bmatrix}}_{(d+d_Z)\times n}. \qquad (62)$$

Thirdly, we construct the third head of $\text{Attn}_{2\text{-}2}$ as

$$W_{2,O}^3 := \underbrace{\begin{bmatrix} I_d \\ 0_{d_Z \times d} \end{bmatrix}}_{(d+d_Z)\times d} \qquad (63)$$

$$W_{2,V}^3 := \underbrace{\begin{bmatrix} 0_{d \times (2d_h+d)} & 0_{d \times n} & 0_{d \times d_Z} & I_d \end{bmatrix}}_{d \times ((2d_h+d)+n+d_Z+d)} \qquad (64)$$

$$W_{2,K}^3 := \underbrace{\begin{bmatrix} 0_{n \times (2d_h+d)} & I_n & 0_{n \times d_Z} & 0_{n \times d} \end{bmatrix}}_{n \times ((2d_h+d)+n+d_Z+d)} \qquad (65)$$

$$W_{2,Q}^3 := \underbrace{\begin{bmatrix} 0_{n \times (2d_h+d)} & I_n & 0_{n \times d_Z} & 0_{n \times d} \end{bmatrix}}_{n \times ((2d_h+d)+n+d_Z+d)}, \qquad (66)$$

such that

$$W_{2,V}^3 \begin{bmatrix} \text{Attn}(X_p) \\ I_n \\ Z\sigma_{\beta_2}(I_n) \\ X\sigma_{\beta_3}(I_n) \end{bmatrix} = \underbrace{\begin{bmatrix} 0_{d\times(2d_h+d)} & 0_{d\times n} & 0_{d\times d_Z} & I_d \end{bmatrix}}_{d\times((2d_h+d)+n+d_Z+d)} \underbrace{\begin{bmatrix} \text{Attn}(X_p) \\ I_n \\ Z\sigma_{\beta_2}(I_n) \\ X\sigma_{\beta_3}(I_n) \end{bmatrix}}_{((2d_h+d)+n+d_Z+d)\times n} = \underbrace{X}_{d\times n}\underbrace{\sigma_{\beta_3}(I_n)}_{n\times n} \qquad (\text{By } (64))$$

$$W_{2,K}^3 \begin{bmatrix} \text{Attn}(X_p) \\ I_n \\ Z\sigma_{\beta_2}(I_n) \\ X\sigma_{\beta_3}(I_n) \end{bmatrix} = \underbrace{\begin{bmatrix} 0_{n\times(2d_h+d)} & I_n & 0_{n\times d_Z} & 0_{n\times d} \end{bmatrix}}_{n\times((2d_h+d)+n+d_Z+d)} \underbrace{\begin{bmatrix} \text{Attn}(X_p) \\ I_n \\ Z\sigma_{\beta_2}(I_n) \\ X\sigma_{\beta_3}(I_n) \end{bmatrix}}_{((2d_h+d)+n+d_Z+d)\times n} = I_n \qquad (\text{By } (65))$$

$$W_{2,Q}^3 \begin{bmatrix} \text{Attn}(X_p) \\ I_n \\ Z\sigma_{\beta_2}(I_n) \\ X\sigma_{\beta_3}(I_n) \end{bmatrix} = \underbrace{\begin{bmatrix} 0_{n\times(2d_h+d)} & I_n & 0_{n\times d_Z} & 0_{n\times d} \end{bmatrix}}_{n\times((2d_h+d)+n+d_Z+d)} \underbrace{\begin{bmatrix} \text{Attn}(X_p) \\ I_n \\ Z\sigma_{\beta_2}(I_n) \\ X\sigma_{\beta_3}(I_n) \end{bmatrix}}_{((2d_h+d)+n+d_Z+d)\times n} = I_n. \qquad (\text{By } (66))$$

Thus, we have

$$\text{Attn}_{2\text{-}2}^{(3)}\left(\begin{bmatrix} \text{Attn}(X_p) \\ I_n \\ Z\sigma_{\beta_2}(I_n) \\ X\sigma_{\beta_3}(I_n) \end{bmatrix}\right)$$

$$= W_{2,O}^3 \cdot W_{2,V}^3 \begin{bmatrix} \text{Attn}(X_p) \\ I_n \\ Z\sigma_{\beta_2}(I_n) \\ X\sigma_{\beta_3}(I_n) \end{bmatrix} \cdot \sigma_{\beta_5}\left(\left(W_{2,K}^3 \begin{bmatrix} \text{Attn}(X_p) \\ I_n \\ Z\sigma_{\beta_2}(I_n) \\ X\sigma_{\beta_3}(I_n) \end{bmatrix}\right)^\top W_{2,Q}^3 \begin{bmatrix} \text{Attn}(X_p) \\ I_n \\ Z\sigma_{\beta_2}(I_n) \\ X\sigma_{\beta_3}(I_n) \end{bmatrix}\right)$$

$$= \underbrace{\begin{bmatrix} I_d \\ 0_{d_Z\times d} \end{bmatrix}}_{(d+d_Z)\times d} \cdot \underbrace{X\sigma_{\beta_3}(I_n)}_{d\times n} \cdot \underbrace{\sigma_{\beta_5}(I_n^\top I_n)}_{n\times n} \qquad (\text{By } (63))$$

$$= \underbrace{\begin{bmatrix} X\sigma_{\beta_3}(I_n)\sigma_{\beta_5}(I_n) \\ 0_{d_Z\times n} \end{bmatrix}}_{(d+d_Z)\times n}. \qquad (67)$$

Finally, all of the heads from $\text{Attn}_{2\text{-}2}$ give us

$$\text{Attn}_{2\text{-}2}\left(\begin{bmatrix} \text{Attn}(X_p) \\ I_n \\ Z\sigma_{\beta_2}(I_n) \\ X\sigma_{\beta_3}(I_n) \end{bmatrix}\right)$$

$$= \sum_{i=1}^3 \text{Attn}_{2\text{-}2}^{(h)}\left(\begin{bmatrix} \text{Attn}(X_p) \\ I_n \\ Z\sigma_{\beta_2}(I_n) \\ X\sigma_{\beta_3}(I_n) \end{bmatrix}\right)$$

$$= \underbrace{\begin{bmatrix} \text{Attn}_s \circ \text{Attn}(X_p) \\ 0_{d_Z\times n} \end{bmatrix}}_{(d+d_Z)\times n} + \underbrace{\begin{bmatrix} 0_{d\times n} \\ Z\sigma_{\beta_2}(I_n)\sigma_{\beta_4}(I_n) \end{bmatrix}}_{(d+d_Z)\times n} + \underbrace{\begin{bmatrix} X\sigma_{\beta_3}(I_n)\sigma_{\beta_5}(I_n) \\ 0_{d_Z\times n} \end{bmatrix}}_{(d+d_Z)\times n} \qquad (\text{By } (57), (62) \text{ and } (67))$$

$$= \underbrace{\begin{bmatrix} X\sigma_{\beta_3}(I_n)\sigma_{\beta_5}(I_n) + \text{Attn}_s \circ \text{Attn}(X_p) \\ Z\sigma_{\beta_2}(I_n)\sigma_{\beta_4}(I_n) \end{bmatrix}}_{(d+d_Z)\times n}$$

**Step 3.** In this part of our proof, our goal is to compute the error between the outputs of $\text{Attn}_{2\text{-}2} \circ \text{Attn}_{2\text{-}1}$ and the target. Specifically, we compute

$$\|\text{Attn}_{2\text{-}2} \circ \text{Attn}_{2\text{-}1}(\begin{bmatrix} X_p \\ Z \end{bmatrix}) - \begin{bmatrix} X + W_V X \cdot \sigma_\beta((W_K X)^\top W_Q X) \\ Z \end{bmatrix}\|_\infty$$

$$= \|\begin{bmatrix} X\sigma_{\beta_3}(I_n)\sigma_{\beta_5}(I_n) + \text{Attn}_s \circ \text{Attn}(X_p) \\ Z\sigma_{\beta_2}(I_n)\sigma_{\beta_4}(I_n) \end{bmatrix} - \begin{bmatrix} X + W_V X \cdot \sigma_\beta((W_K X)^\top W_Q X) \\ Z \end{bmatrix}\|_\infty$$

$$\leq \|\begin{bmatrix} X\sigma_{\beta_3}(I_n)\sigma_{\beta_5}(I_n) + \text{Attn}_s \circ \text{Attn}(X_p) \\ Z\sigma_{\beta_2}(I_n)\sigma_{\beta_4}(I_n) \end{bmatrix} - \begin{bmatrix} X + \text{Attn}_s \circ \text{Attn}(X_p) \\ Z \end{bmatrix}\|_\infty$$

$$+ \|\begin{bmatrix} X + \text{Attn}_s \circ \text{Attn}(X_p) \\ Z \end{bmatrix} - \begin{bmatrix} X + W_V X \cdot \sigma_\beta((W_K X)^\top W_Q X) \\ Z \end{bmatrix}\|_\infty \qquad \text{(By triangle inequality)}$$

$$= \|\begin{bmatrix} X\sigma_{\beta_3}(I_n)\sigma_{\beta_5}(I_n) - X \\ Z\sigma_{\beta_2}(I_n)\sigma_{\beta_4}(I_n) - Z \end{bmatrix}\|_\infty + \epsilon_e \qquad \text{(By Theorem D.3)}$$

$$\leq \|Z(\sigma_{\beta_2}(I_n)\sigma_{\beta_4}(I_n) - I_n)\|_\infty + \|X(\sigma_{\beta_3}(I_n)\sigma_{\beta_5}(I_n) - I_n)\|_\infty + \epsilon_e, \qquad (68)$$

and (68) follows from the triangle inequality.

In the following proof, we treat the two error terms associated with $X$ and $Z$ using identical computations. We analyze the one associated with $Z$ to demonstrate our procedure and provide the result of the one associated with $X$ directly.

To proceed with our proof, we define

$$E_1 := \sigma_{\beta_2}(I_n) - I_n,$$
$$E_2 := \sigma_{\beta_4}(I_n) - I_n,$$

such that

$$\sigma_{\beta_2}(I_n)\sigma_{\beta_4}(I_n) - I_n$$
$$= (E_1 + I_n) \cdot (E_2 + I_n) - I_n$$
$$= E_1 \cdot E_2 + E_1 + E_2. \qquad (69)$$

Thus, we have

$$\|Z(\sigma_{\beta_2}(I_n)\sigma_{\beta_4}(I_n) - I_n)\|_\infty$$
$$= \|Z(E_1 \cdot E_2 + E_1 + E_2)\|_\infty \qquad \text{(By (69))}$$
$$\leq \underbrace{\|ZE_1E_2\|_\infty}_{:=(I)} + \underbrace{\|ZE_1\|_\infty}_{:=(II)} + \underbrace{\|ZE_2\|_\infty}_{:=(III)}, \qquad (70)$$

where the last line follows from triangle inequality.

As a result, our goal is to bound each of $(I), (II)$ and $(III)$, and then aggregate the results to obtain the error bound associated with $Z$. We treat term $(II)$ and term $(III)$ with identical computation, and our analysis of term $(I)$ relies on the results from $(II)$ and $(III)$.

We start with the term $(II)$. To bound the term $(II)$, we analyze each entry $(i, j)$ from $(II)$

$$|(ZE_1)_{ij}|$$
$$\leq \sum_{k=1}^n |Z_{ik}| \cdot |(E_1)_{kj}| \qquad \text{(By triangle inequality)}$$
$$\leq \sum_{k=1}^n \|Z\|_\infty \|E_1\|_\infty \qquad \text{(By } |Z_{ik}| \leq \|Z\|_\infty \text{ and } |(E_1)_{kj}| \leq \|E_1\|_\infty\text{)}$$
$$= n\|Z\|_\infty \|E_1\|_\infty$$
$$\leq nB_Z\|E_1\|_\infty. \qquad \text{(By } \|Z\|_\infty \leq B_Z\text{)}$$

Therefore, $nB_Z\|E_1\|_\infty$ bounds every entry from $(II)$ and gives us

$$(II) \le nB_Z\|E_1\|_\infty. \tag{71}$$

To further analyze (71), we examine each entry from $E_1$. By definition, $E_1$ has all diagonal entries equal to the same value, and similarly for off-diagonal entries. Therefore, to examine each entry from $E_1$, we consider two cases: the diagonal and off-diagonal entries. We start with the diagonal ones.

$$\begin{aligned}
& |(E_1)_{ii}| \\
&= |(\sigma_{\beta_2}(I_n) - I_n)_{ii}| \\
&= |\frac{e^{\beta_2}}{e^{\beta_2} + (n-1)} - 1| && \text{(By the definition of } \sigma_\beta(\cdot). \text{ i.e. (1))} \\
&= |\frac{-(n-1)}{e^{\beta_2} + (n-1)}| \\
&= \frac{(n-1)}{e^{\beta_2} + (n-1)}.
\end{aligned}$$

Concerning off-diagonal terms, we have

$$\begin{aligned}
& |(\sigma_{\beta_2}(I_n) - I_n)_{ij}| \\
&= |\frac{1}{e^{\beta_2} + (n-1)} - 0| && \text{(By the definition of } \sigma_\beta(\cdot). \text{ i.e. (1))} \\
&= |\frac{1}{e^{\beta_2} + (n-1)}| \\
&= \frac{1}{e^{\beta_2} + (n-1)} \\
&\le \frac{(n-1)}{e^{\beta_2} + (n-1)} \\
&= |(\sigma_{\beta_2}(I_n) - I_n)_{jj}|.
\end{aligned}$$

Therefore, the magnitudes of all $E_1$ entries are no greater than the diagonal ones, and we obtain

$$\|E_1\|_\infty \le \frac{(n-1)}{e^{\beta_2} + (n-1)}. \tag{72}$$

As a result,

$$(II) \le \frac{n(n-1)B_Z}{e^{\beta_2} + (n-1)}, \tag{73}$$

where (73) follows from (71) and (72).

Next, we treat term $(III)$ with the same proof steps as those for $(II)$, and we have

$$(III) = \|ZE_2\|_\infty \le \frac{n(n-1)B_Z}{e^{\beta_4} + (n-1)}. \tag{74}$$

As for term $(I)$, we analyze each entry $(i,j)$ from $(I)$

$$\begin{aligned}
& |(ZE_1E_2)_{ij}| \\
&\le \sum_{k=1}^n |Z_{ik}| \cdot |(E_1E_2)_{kj}| && \text{(By triangle inequality)} \\
&\le \sum_{k=1}^n \|Z\|_\infty \|E_1E_2\|_\infty && \text{(By } |Z_{ik}| \le \|Z\|_\infty \text{ and } |(E_1E_2)_{kj}| \le \|E_1E_2\|_\infty)
\end{aligned}$$

$$= n\|Z\|_\infty \|E_1 E_2\|_\infty$$
$$\leq n B_Z \|E_1 E_2\|_\infty. \tag{By $\|Z\|_\infty \leq B_Z$}$$

Therefore, $n B_Z \|E_1 E_2\|_\infty$ bounds every entry from $(I)$ and gives us

$$(I) \leq n B_Z \|E_1 E_2\|_\infty. \tag{75}$$

To further analyze (75), we examine each entry from $E_1 E_2$

$$|(E_1 E_2)_{ij}|$$
$$\leq \sum_{k=1}^{n} |(E_1)_{ik}| \cdot |(E_2)_{kj}| \tag{By triangle inequality}$$
$$\leq \sum_{k=1}^{n} \|E_1\|_\infty \|E_2\|_\infty \tag{By $|(E_1)_{ik}| \leq \|E_1\|_\infty$ and $|(E_2)_{kj}| \leq \|E_2\|_\infty$}$$
$$\leq n \cdot \frac{(n-1)}{e^{\beta_2} + (n-1)} \cdot \frac{(n-1)}{e^{\beta_4} + (n-1)}. \tag{By (72)}$$

The above bound on $|(E_1 E_2)_{ij}|$ holds for all $(i, j)$. Therefore, we have

$$\|E_1 E_2\|_\infty \leq n \cdot \frac{(n-1)}{e^{\beta_2} + (n-1)} \cdot \frac{(n-1)}{e^{\beta_4} + (n-1)}, \tag{76}$$

and

$$(I) \leq B_Z \cdot \frac{n(n-1)}{e^{\beta_2} + (n-1)} \cdot \frac{n(n-1)}{e^{\beta_4} + (n-1)}, \tag{77}$$

where (77) follows from (75) and (76).

Combining the analysis results of $(I)$, $(II)$, and $(III)$, we have

$$(I) + (II) + (III)$$
$$\leq \underbrace{B_Z \cdot \frac{n(n-1)}{e^{\beta_2} + (n-1)} \cdot \frac{n(n-1)}{e^{\beta_4} + (n-1)}}_{\text{from } (I)} + \underbrace{\frac{n(n-1)B_Z}{e^{\beta_2} + (n-1)}}_{\text{from } (II)} + \underbrace{\frac{n(n-1)B_Z}{e^{\beta_4} + (n-1)}}_{\text{from } (III)}, \tag{78}$$

and (78) follows from (73), (74) and (77).

For any $\widehat{\epsilon}_1 > 0$, we take

$$\beta_2, \beta_4 \geq \ln\left( (n-1)(B_Z \frac{n}{\widehat{\epsilon}_1} - 1) \right), \tag{79}$$

then we have

$$(II) \leq \widehat{\epsilon}_1,$$
$$(III) \leq \widehat{\epsilon}_1,$$

and

$$(I)$$
$$\leq B_Z \cdot \frac{n(n-1)}{e^{\beta_2} + (n-1)} \cdot \frac{n(n-1)}{e^{\beta_4} + (n-1)}$$
$$\leq B_Z \cdot \frac{n(n-1)}{(n-1)(B_Z \frac{n}{\widehat{\epsilon}_1} - 1) + (n-1)} \cdot \frac{n(n-1)}{(n-1)(B_Z \frac{n}{\widehat{\epsilon}_1} - 1) + (n-1)} \tag{By (79)}$$

$$= B_Z \cdot \frac{\widehat{\epsilon}_1}{B_Z} \cdot \frac{\widehat{\epsilon}_1}{B_Z}$$

$$= \frac{\widehat{\epsilon}_1^2}{B_Z}.$$

Thus,

$$\|Z(\sigma_{\beta_2}(I_n)\sigma_{\beta_4}(I_n) - I)\|_\infty \le 2\widehat{\epsilon}_1 + \frac{\widehat{\epsilon}_1^2}{B_Z}. \tag{80}$$

As for the error associated with $X$, we treat the error using the same analysis steps as those for $Z$. For any $\widehat{\epsilon}_2 > 0$, we take

$$\beta_3, \beta_5 \ge \ln\left((n-1)(B_X\frac{n}{\epsilon_2} - 1)\right),$$

and we have

$$\|X(\sigma_{\beta_3}(I_n)\sigma_{\beta_5}(I_n) - I)\|_\infty \le 2\widehat{\epsilon}_2 + \frac{\widehat{\epsilon}_2^2}{B_X}. \tag{81}$$

Finally, we have

$$\|\text{Attn}_{\text{2-2}} \circ \text{Attn}_{\text{2-1}}\left(\begin{bmatrix} X_p \\ Z \end{bmatrix}\right) - \begin{bmatrix} X + W_V X \cdot \sigma_\beta((W_K X)^\top W_Q X) \\ Z \end{bmatrix}\|_\infty$$

$$\le 2\widehat{\epsilon}_1 + 2\widehat{\epsilon}_2 + \frac{\widehat{\epsilon}_1^2}{B_Z} + \frac{\widehat{\epsilon}_2^2}{B_X} + \epsilon_e. \qquad \text{(By (68), (80) and (81))}$$

Since we are able to make each $\epsilon$ arbitrarily small, this completes the proof. □

### E.4 Proof of Theorem 3.1

We obtain Theorem 3.1 by stacking the three emulator modules in the same order as the surrogate $A_{\phi_f} = \text{FF}_2 \circ \text{Attn}^{\text{Res}} \circ \text{FF}_1$. In order to do that, we need the next lemma to ensure controlled error propagation.

**Lemma E.4** (Uniform Bounded Surrogate Template). *Let $1 \le p < \infty$ and $L > 0$. Let $\mathcal{C} = [-b, b]^{d\times n} \subset \mathbb{R}^{d\times n}$ be compact, and let $\mathcal{F}_L := \{f : \mathcal{C} \to \mathcal{C} \mid \text{Lip}(f) \le L\}$ be a set of $L$-Lipschitz continuous sequence-to-sequence functions on $\mathcal{C}$. Fix $\epsilon_{\text{sur}} > 0$ and a $\rho$-separating positional encoding $E_p \in \mathbb{R}^{d\times n}$ (Definition D.3). Then there exists a fixed surrogate template*

$$A_\phi(X) = \text{FF}_2 \circ \text{Attn}^{\text{Res}} \circ \text{FF}_1(X + E_p),$$

*with fixed dimensions and fixed prompt shape such that, for every $f \in \mathcal{F}_L$, there exists $\phi_f$ satisfying*

$$d_p(A_{\phi_f}, f) \le \epsilon_{\text{sur}}.$$

*Moreover, the corresponding encoded prompt blocks $W_{\text{FF}_1}, W_{\text{FF}_2}, W_{\text{Attn}^{\text{Res}}}$ and all intermediate quantities required by Lemmas 3.1 to 3.3*

$$X + E_p, \quad \text{FF}_{1,\phi_f}(X + E_p), \quad \text{Attn}^{\text{Res}}_{\phi_f} \circ \text{FF}_{1,\phi_f}(X + E_p),$$
$$W_K \text{FF}_{1,\phi_f}(X + E_p), \quad W_Q \text{FF}_{1,\phi_f}(X + E_p), \quad W_V \text{FF}_{1,\phi_f}(X + E_p),$$

*are uniformly bounded over $f \in \mathcal{F}_L$ and $X \in \mathcal{C}$. In addition, All constants in these bounds, as well as the template dimensions and prompt shape, depend only on $\mathcal{C}, L, p, d, n, E_p, \epsilon_{\text{sur}}$, not on $f$.*

*Proof of Lemma E.4.* Let $N := dn$ and define the shifted domain

$$\mathcal{D} := \mathcal{C} + E_p.$$

The surrogate takes $X + E_p$ as input. Thus, we first rewrite the target function on this shifted domain. For each $f \in \mathcal{F}_L$, define

$$\bar{f}(Y) := f(Y - E_p), \quad Y \in \mathcal{D}. \tag{82}$$

Then $\bar{f} : \mathcal{D} \to \mathcal{C}$. Since $\mathcal{C} = [-b, b]^{d \times n}$, we have

$$\|\bar{f}(Y)\|_\infty \leq b, \qquad\qquad\qquad \text{(By } \bar{f}(Y) \in \mathcal{C} = [-b, b]^{d \times n}\text{)}$$

for all $Y \in \mathcal{D}$.

Moreover, shifting the input by $E_p$ does not change distances. Hence, for any $Y, Y' \in \mathcal{D}$,

$$
\begin{aligned}
\|\bar{f}(Y) - \bar{f}(Y')\|_\infty &= \|f(Y - E_p) - f(Y' - E_p)\|_\infty && \text{(By Equation (82))} \\
&\leq L\|(Y - E_p) - (Y' - E_p)\|_\infty && \text{(By } \mathrm{Lip}_\infty(f) \leq L\text{)} \\
&= L\|Y - Y'\|_\infty. && \text{(Cancel the same shift } E_p\text{)}
\end{aligned}
$$

Thus $\bar{f}$ is also $L$-Lipschitz on $\mathcal{D}$.

We now approximate $\bar{f}$ in two steps. First, we approximate $\bar{f}$ by a function that is constant on each grid cell. Then we approximate this grid-value function by a fixed Transformer surrogate.

Choose a grid granularity $g$ large enough so that

$$g^{-1} \leq \rho/4, \quad 2|\mathcal{D}|^{1/p} N^{1/p} L g^{-1} \leq \epsilon_{\mathrm{sur}}/3. \tag{83}$$

Such a choice exists because both left-hand sides go to zero as $g \to \infty$.

Let $\pi_g : \mathcal{D} \to \mathcal{G}_g$ be the finite-grid quantization map in Proposition D.1. Thus

$$\|Y - \pi_g(Y)\|_\infty \leq g^{-1}, \quad \text{for all} \quad Y \in \mathcal{D}.$$

For each grid point $v \in \mathcal{G}_g$, choose one representative $Y_v \in \mathcal{D}$ such that

$$\pi_g(Y_v) = v.$$

This is possible because $\mathcal{G}_g = \pi_g(\mathcal{D})$.

Define

$$a_v := \bar{f}(Y_v), \quad F_{f,g}(Y) := a_{\pi_g(Y)}. \tag{84}$$

Thus $F_{f,g}$ assigns one fixed value to each grid cell.

Recall from Proposition D.1 that $\mathcal{H}_{g,M}$ is the class of grid-value maps

$$\mathcal{H}_{g,M} := \big\{ F : \mathcal{D} \to \mathbb{R}^{d \times n} \mid F(Y) = a_{\pi_g(Y)}, \; \|a_v\|_\infty \leq M \text{ for all } v \in \mathcal{G}_g \big\}.$$

Then $F_{f,g} \in \mathcal{H}_{g,b}$. Indeed, for every $v \in \mathcal{G}_g$,

$$
\begin{aligned}
\|a_v\|_\infty &= \|\bar{f}(Y_v)\|_\infty && \text{(By Equation (84))} \\
&\leq b. && \text{(By } \bar{f}(\mathcal{D}) \subset \mathcal{C} = [-b, b]^{d \times n}\text{)}
\end{aligned}
$$

We next bound the error between $\bar{f}$ and $F_{f,g}$. Fix any $Y \in \mathcal{D}$ and write $v = \pi_g(Y)$. Then

$$
\begin{aligned}
\|F_{f,g}(Y) - \bar{f}(Y)\|_\infty &= \|\bar{f}(Y_v) - \bar{f}(Y)\|_\infty && \text{(By Equation (84) and } v = \pi_g(Y)\text{)} \\
&\leq L\|Y_v - Y\|_\infty && \text{(By } \mathrm{Lip}_\infty(\bar{f}) \leq L\text{)} \\
&\leq L\big(\|Y_v - v\|_\infty + \|v - Y\|_\infty\big) && \text{(By triangle inequality)} \\
&= L\big(\|Y_v - \pi_g(Y_v)\|_\infty + \|\pi_g(Y) - Y\|_\infty\big) && \text{(By } v = \pi_g(Y_v) = \pi_g(Y)\text{)}
\end{aligned}
$$

$$\leq 2Lg^{-1}, \tag{85}$$

where the last line is by definition of $\pi_g(\cdot)$.

Now convert the pointwise $\ell_\infty$ bound into a pointwise $\ell_p$ bound:

$$\|F_{f,g}(Y) - \bar{f}(Y)\|_p \leq N^{1/p}\|F_{f,g}(Y) - \bar{f}(Y)\|_\infty \qquad \text{(By } N = dn \text{ and } \|A\|_p \leq N^{1/p}\|A\|_\infty)$$
$$\leq 2N^{1/p}Lg^{-1}. \qquad \text{(By Equation (85))}$$

Therefore,

$$(\int_{\mathcal{D}} \|F_{f,g}(Y) - \bar{f}(Y)\|_p^p \mathrm{d}Y)^{1/p} \leq (\int_{\mathcal{D}} (2N^{1/p}Lg^{-1})^p \mathrm{d}Y)^{1/p} \qquad \text{(By the pointwise bound above)}$$
$$= 2|\mathcal{D}|^{1/p}N^{1/p}Lg^{-1} \qquad \text{(The integrand is constant in } Y)$$
$$\leq \epsilon_{\mathrm{sur}}/3, \tag{86}$$

where the last line is by the choice of $g$ in Equation (83).

We now approximate the grid-value function $F_{f,g}$ by a Transformer surrogate. Apply Proposition D.1 with $M = b$ and $\eta = 2\epsilon_{\mathrm{sur}}/3$. Since $F_{f,g} \in \mathcal{H}_{g,b}$, there exists a parameter choice $\phi_f$ in a fixed $\mathcal{T}_2$-type template

$$B_{\phi_f}(Y) = \mathrm{FF}_{2,\phi_f} \circ \mathrm{Attn}_{\phi_f}^{\mathrm{Res}} \circ \mathrm{FF}_{1,\phi_f}(Y)$$

such that

$$(\int_{\mathcal{D}} \|B_{\phi_f}(Y) - F_{f,g}(Y)\|_p^p \mathrm{d}Y)^{1/p} \leq 2\epsilon_{\mathrm{sur}}/3. \tag{87}$$

Moreover, by Proposition D.1, the dimensions of $B_{\phi_f}$, the prompt shape, and all surrogate weights are bounded by constants independent of $f$.

We now return to the original input variable $X$. Define

$$A_{\phi_f}(X) := B_{\phi_f}(X + E_p).$$

Then

$$d_p(A_{\phi_f}, f) = (\int_{\mathcal{C}} \|B_{\phi_f}(X + E_p) - f(X)\|_p^p \mathrm{d}X)^{1/p} \qquad \text{(By the definition of } A_{\phi_f} \text{ and } d_p)$$
$$= (\int_{\mathcal{D}} \|B_{\phi_f}(Y) - \bar{f}(Y)\|_p^p \mathrm{d}Y)^{1/p} \qquad \text{(By the change of variables } Y = X + E_p)$$
$$\leq (\int_{\mathcal{D}} \|B_{\phi_f}(Y) - F_{f,g}(Y)\|_p^p \mathrm{d}Y)^{1/p} + (\int_{\mathcal{D}} \|F_{f,g}(Y) - \bar{f}(Y)\|_p^p \mathrm{d}Y)^{1/p} \qquad \text{(By Minkowski inequality)}$$
$$\leq 2\epsilon_{\mathrm{sur}}/3 + \epsilon_{\mathrm{sur}}/3 \qquad \text{(By Equations (86) and (87))}$$
$$= \epsilon_{\mathrm{sur}}.$$

This proves the surrogate approximation claim.

It remains to verify the uniform boundedness claims. We check the required bounds one by one.

- By Proposition D.1, all weights in $\mathrm{FF}_{1,\phi_f}$, $\mathrm{Attn}_{\phi_f}^{\mathrm{Res}}$, and $\mathrm{FF}_{2,\phi_f}$ are bounded uniformly over $f \in \mathcal{F}_L$. The prompt blocks are built by stacking these weights and their token-indexed multiples $0, 1, \ldots, n-1$. Therefore, there exists $B_P < \infty$ such that

$$\|W_{\mathrm{FF}_1}\|_\infty, \quad \|W_{\mathrm{FF}_2}\|_\infty, \quad \|W_{\mathrm{Attn}^{\mathrm{Res}}}\|_\infty \leq B_P.$$

- Next, since $\mathcal{D} = \mathcal{C} + E_p$ is compact, there exists $B_X < \infty$ such that

$$\|X + E_p\|_\infty \leq B_X, \quad \text{for all} \quad X \in \mathcal{C}.$$

- The map $\mathrm{FF}_{1,\phi_f}$ has fixed dimensions and uniformly bounded weights. Its input $X + E_p$ is also uniformly bounded. Hence there exists $B_1 < \infty$ such that

$$\|\mathrm{FF}_{1,\phi_f}(X + E_p)\|_\infty \le B_1, \quad \text{for all} \quad f \in \mathcal{F}_L, \ X \in \mathcal{C}. \tag{88}$$

- Similarly, the matrices $W_K, W_Q, W_V$ in $\mathrm{Attn}_{\phi_f}^{\mathrm{Res}}$ have fixed dimensions and uniformly bounded entries. Combining this with Equation (88), there exists $B_{KQV} < \infty$ such that

$$\|W_K \mathrm{FF}_{1,\phi_f}(X + E_p)\|_\infty \le B_{KQV},$$
$$\|W_Q \mathrm{FF}_{1,\phi_f}(X + E_p)\|_\infty \le B_{KQV},$$
$$\|W_V \mathrm{FF}_{1,\phi_f}(X + E_p)\|_\infty \le B_{KQV}.$$

- Finally, let

$$Z_f(X) := \mathrm{FF}_{1,\phi_f}(X + E_p).$$

Every entry of a softmax matrix lies in $[0, 1]$. Therefore,

$$
\begin{aligned}
\|\mathrm{Attn}_{\phi_f}^{\mathrm{Res}}(Z_f(X))\|_\infty &= \|Z_f(X) + W_V Z_f(X)\sigma_\beta((W_K Z_f(X))^\top W_Q Z_f(X))\|_\infty \\
&\le \|Z_f(X)\|_\infty + \|W_V Z_f(X)\sigma_\beta((W_K Z_f(X))^\top W_Q Z_f(X))\|_\infty \quad \text{(By triangle inequality)} \\
&\le B_1 + n B_{KQV} \\
&=: B_{\mathrm{Attn}},
\end{aligned}
$$

where the third line is by Equation (88), $\|W_V Z_f(X)\|_\infty \le B_{KQV}$, and softmax entries lie in $[0, 1]$.

Thus

$$\|\mathrm{Attn}_{\phi_f}^{\mathrm{Res}} \circ \mathrm{FF}_{1,\phi_f}(X + E_p)\|_\infty \le B_{\mathrm{Attn}}$$

uniformly over $f \in \mathcal{F}_L$ and $X \in \mathcal{C}$.

Combining the bounds above, the encoded prompt blocks and all intermediate quantities required by Lemmas 3.1 to 3.3 are uniformly bounded over $f \in \mathcal{F}_L$ and $X \in \mathcal{C}$. All constants depend only on

$$\mathcal{C}, L, p, d, n, E_p, \rho, \epsilon_{\mathrm{sur}},$$

and not on $f$. This completes the proof. □

Now we are ready for our main proof.

**Theorem E.1** (Theorem 3.1 Restated: In-Context Universality via Transformer-Surrogate Emulation)**.** *Let $\mathcal{C} = [-b, b]^{d \times n} \subset \mathbb{R}^{d \times n}$ be compact, and let $\mathcal{F}_L := \{f : \mathcal{C} \to \mathcal{C} \mid \mathrm{Lip}_\infty(f) \le L\}$ be a set of $L$-Lipschitz continuous sequence-to-sequence functions. Fix a deterministic $\rho$-separating positional encoding $E_p \in \mathbb{R}^{d \times n}$ (Definition D.3). Then, for any $\epsilon > 0$, there exists a fixed six-layer softmax-attention emulator $T_{\theta^\star} = \mathrm{Attn}_6$ such that, for every $f \in \mathcal{F}_L$, there exists a prompt $P_f = [W_{\mathrm{FF}_1}; W_{\mathrm{AttnRes}}; I_n; W_{\mathrm{FF}_2}]$ satisfying*

$$\mathrm{d}_p(T_{\theta^\star, P_f}, f) \le \epsilon, \quad \text{where} \quad T_{\theta^\star, P_f}(X) := T_{\theta^\star}([X + E_p; P_f]).$$

*The emulator weights $\theta^\star$ and the prompt shape depend only on $\mathcal{C}, L, p, d, n, E_p, \rho, \epsilon,$, not on $f$ or $X$. Only the prompt entries depend on $f$.*

*Proof.* Let $N := dn$. Choose $\epsilon_{\mathrm{sur}}, \epsilon_{\mathrm{emu}} > 0$ such that

$$\epsilon_{\mathrm{sur}} + \epsilon_{\mathrm{emu}} \le \epsilon. \tag{89}$$

The emulation guarantees used in this proof are pointwise $\ell_\infty$ bounds, while the theorem is stated in the integrated $d_p$ metric. We therefore choose the pointwise tolerance so that its $d_p$ contribution is at most $\epsilon_{\mathrm{emu}}$. Define

$$\epsilon_{\mathrm{emu\text{-}pt}} := |\mathcal{C}|^{-1/p} N^{-1/p} \epsilon_{\mathrm{emu}}. \tag{90}$$

Apply Lemma E.4 with accuracy $\epsilon_{\mathrm{sur}}$. Then there exists a fixed surrogate template

$$A_\phi(X) = \mathrm{FF}_{2,\phi} \circ \mathrm{Attn}_\phi^{\mathrm{Res}} \circ \mathrm{FF}_{1,\phi}(X + E_p),$$

with fixed dimensions and fixed prompt shape such that, for every $f \in \mathcal{F}_L$, there exists $\phi_f$ satisfying

$$d_p(A_{\phi_f}, f) \le \epsilon_{\mathrm{sur}}.$$

Moreover, Lemma E.4 gives common bounds for the encoded prompt blocks and all intermediate quantities required by Lemmas 3.1 to 3.3, uniformly over $f \in \mathcal{F}_L$ and $X \in \mathcal{C}$.

For each $f \in \mathcal{F}_L$, let

$$P_f := [W_{\mathrm{FF}_1}; W_{\mathrm{Attn}^{\mathrm{Res}}}; I_n; W_{\mathrm{FF}_2}]$$

be the prompt associated with $\phi_f$. The fixed emulator executes the three surrogate modules in the order

$$\mathrm{FF}_1, \quad \mathrm{Attn}^{\mathrm{Res}}, \quad \mathrm{FF}_2.$$

We first define the exact intermediate states of the surrogate computations. These are the states obtained by executing the surrogate modules exactly. They serve as reference states for tracking the error of the stacked emulators:

$$Y_0^f(X) := \begin{bmatrix} X + E_p \\ W_{\mathrm{FF}_1} \\ W_{\mathrm{Attn}^{\mathrm{Res}}} \\ I_n \\ W_{\mathrm{FF}_2} \end{bmatrix}, \quad Y_1^f(X) := \begin{bmatrix} \mathrm{FF}_{1,\phi_f}(X + E_p) \\ W_{\mathrm{Attn}^{\mathrm{Res}}} \\ I_n \\ W_{\mathrm{FF}_2} \end{bmatrix},$$

and

$$Y_2^f(X) := \begin{bmatrix} \mathrm{Attn}_{\phi_f}^{\mathrm{Res}} \circ \mathrm{FF}_{1,\phi_f}(X + E_p) \\ W_{\mathrm{FF}_2} \end{bmatrix}, \quad Y_3^f(X) := A_{\phi_f}(X).$$

We now prepare the bookkeeping step for error propagation. The emulator Lemmas 3.1 to 3.3 control each module when its input is the exact state produced by the previous surrogate module. In the stacked emulator, however, the next module receives an approximate state. We therefore need a fixed region where these approximate states are still close to the exact states.

First collect all exact intermediate states:

$$\mathcal{S}_1 := \{Y_1^f(X) : f \in \mathcal{F}_L,\ X \in \mathcal{C}\}, \quad \mathcal{S}_2 := \{Y_2^f(X) : f \in \mathcal{F}_L,\ X \in \mathcal{C}\}.$$

By Lemma E.4, both sets are bounded. Thus all exact states that can appear in the proof lie inside a common bounded region.

For any set $\mathcal{S}$, define the distance from $Y$ to $\mathcal{S}$ by

$$\mathrm{dist}_\infty(Y, \mathcal{S}) := \inf_{S \in \mathcal{S}} \|Y - S\|_\infty.$$

This is the smallest $\ell_\infty$ distance from $Y$ to any point in $\mathcal{S}$.

Define the safety regions

$$\mathcal{K}_1 := \{Y : \mathrm{dist}_\infty(Y, \mathcal{S}_1) \le 1\}, \quad \mathcal{K}_2 := \{Y : \mathrm{dist}_\infty(Y, \mathcal{S}_2) \le 1\}.$$

Thus $\mathcal{K}_i$ contains all points within distance 1 of the exact state set $\mathcal{S}_i$. In particular, if an approximate state is within $\ell_\infty$ error at most 1 of its exact state, then it lies in $\mathcal{K}_i$.

We next note that each $\mathcal{K}_i$ is compact. Indeed, $\mathrm{dist}_\infty(\cdot, \mathcal{S}_i)$ is continuous, so $\mathcal{K}_i$ is closed. Also, since $\mathcal{S}_i$ is bounded, every point within distance 1 of $\mathcal{S}_i$ is bounded. Hence $\mathcal{K}_i$ is closed and bounded, and therefore compact in finite dimension.

These compact sets are used only to control later error amplification. After the later emulator maps are fixed, they are finite softmax-attention networks with fixed finite weights. Hence they are continuously differentiable, and so they have finite Lipschitz constants on the compact sets $\mathcal{K}_1$ and $\mathcal{K}_2$.

Finally, define the local tolerance

$$\bar{\epsilon}_{\text{emu-pt}} := \min\{1, \epsilon_{\text{emu-pt}}\}.$$

The cap by 1 has only one purpose: it ensures that the approximate intermediate states remain inside the safety regions $\mathcal{K}_1$ and $\mathcal{K}_2$. Since

$$\bar{\epsilon}_{\text{emu-pt}} \leq \epsilon_{\text{emu-pt}},$$

proving a pointwise emulator error bound with $\bar{\epsilon}_{\text{emu-pt}}$ is stronger than the bound needed for the final $d_p$ estimate.

We choose the three two-layer emulators from back to front, because errors from earlier emulators are passed through the later fixed maps.

- $\text{FF}_2$ **Emulator.** We first choose the terminal FFN emulator, which emulates the last surrogate module $\text{FF}_2$. Set

$$\epsilon_2 := \frac{\bar{\epsilon}_{\text{emu-pt}}}{3}. \tag{91}$$

By Lemma 3.2 and the common bounds from Lemma E.4, there exists a fixed two-layer attention $\text{Attn}_{2,3}$ such that, for every $f \in \mathcal{F}_L$ and $X \in \mathcal{C}$,

$$\left\|\text{Attn}_{2,3}\left(\begin{bmatrix} \text{Attn}_{\phi_f}^{\text{Res}} \circ \text{FF}_{1,\phi_f}(X + E_p) \\ W_{\text{FF}_2} \end{bmatrix}\right) - A_{\phi_f}(X)\right\|_\infty \leq \epsilon_2.$$

Let $L_{\text{FF}_2} \geq 1$ be a Lipschitz constant of this fixed map on $\mathcal{K}_2$.

- $\text{Attn}^{\text{Res}}$ **Emulator.** Next we choose the middle emulator, which emulates the residual-attention module and forwards the prompt block needed by $\text{FF}_2$. Set

$$\epsilon_3 := \frac{\bar{\epsilon}_{\text{emu-pt}}}{3 L_{\text{FF}_2}}. \tag{92}$$

By Lemma 3.3 and the common bounds from Lemma E.4, there exists a fixed two-layer attention $\text{Attn}_{2,2}$ such that, for every $f \in \mathcal{F}_L$ and $X \in \mathcal{C}$,

$$\left\|\text{Attn}_{2,2}\left(\begin{bmatrix} \text{FF}_{1,\phi_f}(X + E_p) \\ W_{\text{Attn}^{\text{Res}}} \\ I_n \\ W_{\text{FF}_2} \end{bmatrix}\right) - \begin{bmatrix} \text{Attn}_{\phi_f}^{\text{Res}} \circ \text{FF}_{1,\phi_f}(X + E_p) \\ W_{\text{FF}_2} \end{bmatrix}\right\|_\infty \leq \epsilon_3.$$

Let $L_{\text{Attn}} \geq 1$ be a Lipschitz constant of this fixed map on $\mathcal{K}_1$.

- $\text{FF}_1$ **Emulator.** Finally we choose the first emulator, which emulates $\text{FF}_1$ and forwards the prompt blocks needed by the later modules. Set

$$\epsilon_1 := \frac{\bar{\epsilon}_{\text{emu-pt}}}{3 L_{\text{Attn}} L_{\text{FF}_2}}. \tag{93}$$

By Lemma 3.1 and the common bounds from Lemma E.4, there exists a fixed two-layer attention $\text{Attn}_{2,1}$ such that, for every $f \in \mathcal{F}_L$ and $X \in \mathcal{C}$,

$$\left\|\text{Attn}_{2,1}\left(\begin{bmatrix} X + E_p \\ W_{\text{FF}_1} \\ W_{\text{Attn}^{\text{Res}}} \\ I_n \\ W_{\text{FF}_2} \end{bmatrix}\right) - \begin{bmatrix} \text{FF}_{1,\phi_f}(X + E_p) \\ W_{\text{Attn}^{\text{Res}}} \\ I_n \\ W_{\text{FF}_2} \end{bmatrix}\right\|_\infty \leq \epsilon_1.$$

We stack the three emulator modules as

$$T_{\theta^\star} := \underbrace{\text{Attn}_{2,3}}_{\text{FF}_2 \text{ Emulator}} \circ \underbrace{\text{Attn}_{2,2}}_{\text{Attn}^{\text{Res}} \text{ Emulator}} \circ \underbrace{\text{Attn}_{2,1}}_{\text{FF}_1 \text{ Emulator}} .$$

The weights of $T_{\theta^\star}$ are fixed because each two-layer emulator is chosen using only the common dimensions, common bounds, and error tolerances.

We now track how errors propagate through the stack. The un-hatted variables are the exact intermediate states defined above. The hatted variables are the actual states produced by the stacked emulators. This distinction is needed because each emulator lemma is stated for an exact input, while in the stack the next emulator receives the previous emulator's approximate output.

Fix $f \in \mathcal{F}_L$ and $X \in \mathcal{C}$. For readability, write

$$Y_i(X) := Y_i^f(X), \quad i = 0, 1, 2, 3.$$

Define the actual stacked states

$$\widehat{Y}_0(X) := Y_0(X), \quad \widehat{Y}_1(X) := \text{Attn}_{2,1}(\widehat{Y}_0(X)),$$

and

$$\widehat{Y}_2(X) := \text{Attn}_{2,2}(\widehat{Y}_1(X)), \quad \widehat{Y}_3(X) := \text{Attn}_{2,3}(\widehat{Y}_2(X)).$$

The goal is to bound $\|\widehat{Y}_3(X) - Y_3(X)\|_\infty$, because $\widehat{Y}_3(X) = T_{\theta^\star, P_f}(X)$ and $Y_3(X) = A_{\phi_f}(X)$.

First,

$$\|\widehat{Y}_1(X) - Y_1(X)\|_\infty \le \epsilon_1. \tag{By Lemma 3.1}$$

Since $\epsilon_1 \le \bar{\epsilon}_{\text{emu-pt}} \le 1$, we have $\widehat{Y}_1(X) \in \mathcal{K}_1$. Hence,

$$
\begin{aligned}
\|\widehat{Y}_2(X) - Y_2(X)\|_\infty &= \|\text{Attn}_{2,2}(\widehat{Y}_1(X)) - Y_2(X)\|_\infty && \left(\text{By definition of } \widehat{Y}_2\right) \\
&\le \|\text{Attn}_{2,2}(\widehat{Y}_1(X)) - \text{Attn}_{2,2}(Y_1(X))\|_\infty + \|\text{Attn}_{2,2}(Y_1(X)) - Y_2(X)\|_\infty \\
&\le L_{\text{Attn}}\|\widehat{Y}_1(X) - Y_1(X)\|_\infty + \epsilon_3 && (\text{By Lipschitzness on } \mathcal{K}_1 \text{ and Lemma 3.3}) \\
&\le L_{\text{Attn}}\epsilon_1 + \epsilon_3. && \left(\text{By the bound on } \widehat{Y}_1(X) - Y_1(X)\right)
\end{aligned}
$$

By the choices of $\epsilon_1$ and $\epsilon_3$,

$$
\begin{aligned}
L_{\text{Attn}}\epsilon_1 + \epsilon_3 &= L_{\text{Attn}}\frac{\bar{\epsilon}_{\text{emu-pt}}}{3L_{\text{Attn}}L_{\text{FF}_2}} + \frac{\bar{\epsilon}_{\text{emu-pt}}}{3L_{\text{FF}_2}} && (\text{By Equations (92) and (93)}) \\
&= \frac{2\bar{\epsilon}_{\text{emu-pt}}}{3L_{\text{FF}_2}} \\
&\le 1. && (\text{By } \bar{\epsilon}_{\text{emu-pt}} \le 1 \text{ and } L_{\text{FF}_2} \ge 1)
\end{aligned}
$$

Thus $\widehat{Y}_2(X) \in \mathcal{K}_2$. Finally,

$$
\begin{aligned}
\|\widehat{Y}_3(X) - Y_3(X)\|_\infty &= \|\text{Attn}_{2,3}(\widehat{Y}_2(X)) - Y_3(X)\|_\infty && \left(\text{By definition of } \widehat{Y}_3\right) \\
&\le \|\text{Attn}_{2,3}(\widehat{Y}_2(X)) - \text{Attn}_{2,3}(Y_2(X))\|_\infty + \|\text{Attn}_{2,3}(Y_2(X)) - Y_3(X)\|_\infty \\
&\le L_{\text{FF}_2}\|\widehat{Y}_2(X) - Y_2(X)\|_\infty + \epsilon_2 && (\text{By Lipschitzness on } \mathcal{K}_2 \text{ and Lemma 3.2}) \\
&\le L_{\text{FF}_2}(L_{\text{Attn}}\epsilon_1 + \epsilon_3) + \epsilon_2 && \left(\text{By the bound on } \widehat{Y}_2(X) - Y_2(X)\right) \\
&= L_{\text{FF}_2}(L_{\text{Attn}}\frac{\bar{\epsilon}_{\text{emu-pt}}}{3L_{\text{Attn}}L_{\text{FF}_2}} + \frac{\bar{\epsilon}_{\text{emu-pt}}}{3L_{\text{FF}_2}}) + \frac{\bar{\epsilon}_{\text{emu-pt}}}{3} && (\text{By Equations (91) to (93)}) \\
&= \bar{\epsilon}_{\text{emu-pt}}
\end{aligned}
$$

$$\le \epsilon_{\text{emu-pt}}. \qquad \qquad \text{(By definition of } \bar{\epsilon}_{\text{emu-pt}})$$

Therefore,

$$\sup_{X \in \mathcal{C}} \|T_{\theta^\star, P_f}(X) - A_{\phi_f}(X)\|_\infty \le \epsilon_{\text{emu-pt}}, \qquad (94)$$

by $\widehat{Y}_3(X) = T_{\theta^\star, P_f}(X)$ and $Y_3(X) = A_{\phi_f}(X)$

We now convert the pointwise emulator error into the $d_p$ metric:

$$
\begin{aligned}
d_p(T_{\theta^\star, P_f}, A_{\phi_f}) &= \left( \int_{\mathcal{C}} \|T_{\theta^\star, P_f}(X) - A_{\phi_f}(X)\|_p^p \mathrm{d}X \right)^{1/p} \\
&\le \left( \int_{\mathcal{C}} (N^{1/p} \|T_{\theta^\star, P_f}(X) - A_{\phi_f}(X)\|_\infty)^p \mathrm{d}X \right)^{1/p} \qquad \text{(By } N = dn \text{ and } \|A\|_p \le N^{1/p}\|A\|_\infty) \\
&\le |\mathcal{C}|^{1/p} N^{1/p} \epsilon_{\text{emu-pt}} \qquad\qquad\qquad\qquad\qquad \text{(By Equation (94))} \\
&= \epsilon_{\text{emu}}, \qquad\qquad\qquad\qquad\qquad\qquad\qquad\qquad\quad (95)
\end{aligned}
$$

where the last line is by Equation (90).

By triangle inequality,

$$
\begin{aligned}
d_p(T_{\theta^\star, P_f}, f) &\le d_p(T_{\theta^\star, P_f}, A_{\phi_f}) + d_p(A_{\phi_f}, f) \\
&\le \epsilon_{\text{emu}} + \epsilon_{\text{sur}} \qquad\qquad \text{(By Equation (95) and Lemma E.4)} \\
&\le \epsilon. \qquad\qquad\qquad\qquad\qquad\quad \text{(By Equation (89))}
\end{aligned}
$$

Since $f \in \mathcal{F}_L$ was arbitrary, the same fixed emulator $T_{\theta^\star}$ works for every target function through its corresponding prompt $P_f$. The emulator weights and prompt shape depend only on the fixed surrogate dimensions, common bounds, and error budget. By Lemma E.4, these depend only on

$$\mathcal{C}, L, p, d, n, E_p, \rho, \epsilon,$$

not on $f$, $X$, or $\phi_f$. This completes the proof. $\qquad\square$

### E.5 Proof of Lemma 3.4

We state a helper lemma from (Hu et al., 2026b).

**Lemma E.5** (Attention Preceded by a Linear Transformation Is Still an Attention, Lemma B.3 of (Hu et al., 2026b))**.** *Let $W_K, W_Q \in \mathbb{R}^{d_h \times \widetilde{d}}$, $W_V \in \mathbb{R}^{d_V \times \widetilde{d}}$, and $W_O \in \mathbb{R}^{d_O \times d_V}$ be the weight matrices of a single-head attention* Attn*. Let $A : \mathbb{R}^{d \times n} \to \mathbb{R}^{\widetilde{d} \times n}$ be a linear transformation. Then,* Attn $\circ A$ *is still an attention.*

For completeness, we restate the proof.

*Proof.* Let $D \in \mathbb{R}^{d \times n}$ denote any input. Then, we have

$$\text{Attn} \circ A(D) = W_O \cdot W_V A D \cdot \sigma_\beta((W_K A D)^\top (W_Q A D)),$$

and this is a new attention mechanism with parameters $W_K A, W_Q A$ and $W_V A$. $\qquad\square$

**Lemma E.6** (Lemma 3.4 Restated: In-Context Emulation of Multi-Head Attention)**.** *Let $X \in \mathbb{R}^{d \times n}$ be the input sequence. For any $\epsilon_4 > 0$, there exists a two-layer attention* Attn$_2$ *such that, for any $H$-head target attention* Attn *satisfying Definition 2.1, with the prompt $X_p$ defined as in Definition 3.4, and for each head $h \in [H]$, with assumptions $\|W_K^h X\|_\infty, \|W_Q^h X\|_\infty, \|W_V^h X\|_\infty \le B_{KQV}$ for some $B_{KQV} > 0$, we have*

$$\|\text{Attn}_2(X_p) - \text{Attn}(X)\|_\infty \le \epsilon_4.$$

*Proof.* We modify the proof of the single-head emulation theorem from Hu et al. (2026b).

We use Attn$^{(h)}$ to label the $h$-th head from the target Attn, and we use $W_K^h, W_Q^h, W_V^h$ for the weight matrices in Attn$^{(h)}$.

The proof consists of $4$ steps. We first write out the details of the weight encoding of target Attn for later use. Then we use the first layer of Attn$_2$ to reconstruct the key, query, and value matrices of Attn from the input weight encoding. Thirdly, we use the second layer of Attn$_2$ to assemble the reconstructed matrices into the attention form. Finally, we compute the error.

For the convenience of presentation, we denote Attn$_2$ as Attn$_{2\text{-}2} \circ$ Attn$_{2\text{-}1}$.

**Step 1: Weight Encoding.** For the convenience of presentation, we define

$$k_i^h := (W_K^h)_{:,i}^\top \in \mathbb{R}^d, \tag{96}$$

$$q_i^h := (W_Q^h)_{:,i}^\top \in \mathbb{R}^d, \tag{97}$$

$$v_i^h := (W_V^h)_{:,i}^\top \in \mathbb{R}^d, \tag{98}$$

and we define the vectorizations of the weight matrices by Definition 3.3

$$\underline{W}_K^h := \underbrace{\begin{bmatrix} k_1^h \\ k_2^h \\ \vdots \\ k_{d_h}^h \end{bmatrix}}_{dd_h \times 1}, \quad \underline{W}_Q^h := \underbrace{\begin{bmatrix} q_1^h \\ q_2^h \\ \vdots \\ q_{d_h}^h \end{bmatrix}}_{dd_h \times 1}, \quad \underline{W}_V^h := \underbrace{\begin{bmatrix} v_1^h \\ v_2^h \\ \vdots \\ v_{d_V}^h \end{bmatrix}}_{dd_V \times 1}.$$

Also, we define the concatenation

$$w^h := \begin{bmatrix} \underline{W}_K^h \\ \underline{W}_Q^h \\ \underline{W}_V^h \end{bmatrix} = \underbrace{\begin{bmatrix} k_1^h \\ \vdots \\ k_{d_h}^h \\ q_1^h \\ \vdots \\ q_{d_h}^h \\ v_1^h \\ \vdots \\ v_{d_V}^h \end{bmatrix}}_{d(2d_h + d_V) \times 1}. \tag{99}$$

To keep our proof clean, we define $d_{w^h} := d(2d_h + d_V)$.

We define $W^h$, $W_{\text{Attn}}$ to be

$$W^h := \underbrace{\begin{bmatrix} 0 \cdot w^h & 1 \cdot w^h & 2 \cdot w^h & \cdots & (n-1) \cdot w^h \\ w^h & w^h & w^h & \cdots & w^h \end{bmatrix}}_{2d_{w^h} \times n}, \tag{100}$$

$$W_{\text{Attn}} := \underbrace{\begin{bmatrix} W^1 \\ W^2 \\ \vdots \\ W^H \end{bmatrix}}_{(\sum_{h=1}^H 2d_{w^h}) \times n}. \tag{101}$$

Thus, our input prompt is

$$X_p := \underbrace{\begin{bmatrix} X \\ W_{\text{Attn}} \\ I_n \end{bmatrix}}_{(d + \sum_{h=1}^H 2d_{w^h} + n) \times n}. \tag{102}$$

**Step 2: Reconstruction of Target** $K^h, Q^h, V^h$. Again, for the convenience of presentation, we define

$$K^h := \underbrace{W_K^h}_{d_h \times d} \cdot \underbrace{X}_{d \times n}, \quad Q^h := \underbrace{W_Q^h}_{d_h \times d} \cdot X, \quad V^h := \underbrace{W_V^h}_{d_V \times d} \cdot X.$$

In this part, our goal is to build approximators for the target $K^h, Q^h, V^h$ from the input prompt via $\mathrm{Attn}_{2\text{-}1}$.

We note that each row of $K^h, Q^h, V^h$ has the inner product form: $(k_i^h)^\top X, (q_i^h)^\top X, (v_i^h)^\top X$. Therefore, we construct $\mathrm{Attn}_{2\text{-}1}$ by applying Theorem D.4 to each row separately, and $\mathrm{Attn}_{2\text{-}1}$ consists of the heads from Theorem D.4.

We state our parameter choices for Theorem D.4 to approximate each row. Firstly, we note that the inner products $(k_i^h)^\top x_r, (q_i^h)^\top x_r$ and $(v_m^h)^\top x_r$, over all $i \in [d_h], m \in [d_V]$ and $r \in [n]$, are bounded by $\pm B_{KQV}$, so we define $a_h$ and $b_h$ to be

$$a_h := -B_{KQV} \leq \min_{i,m,r}\{(k_i^h)^\top x_r, (q_i^h)^\top x_r, (v_m^h)^\top x_r\},$$

$$b_h := B_{KQV} \geq \max_{i,m,r}\{(k_i^h)^\top x_r, (q_i^h)^\top x_r, (v_m^h)^\top x_r\}.$$

That is, $a_h$ and $b_h$ denote the lower/upper bound of the inner products in the $h$-th head of $\mathrm{Attn}$.

Next, for any $\epsilon_0 > 0$, we choose the number of heads $\widetilde{H}_h$ in Theorem D.4 to be

$$\widetilde{H}_h := \lceil \frac{2(b_h - a_h)}{\epsilon_0(n-2)} \rceil,$$

such that the interpolation error in Theorem D.4 is no more than $\frac{\epsilon_0}{2}$. Also, the precision parameter in Theorem D.4 is $p_h := (n-2)H_h$, and $p_h > n$ for sufficiently small $\epsilon_0$.

Here $\widetilde{H}_h$ denotes the number of heads we use for each row of $K^h, Q^h$ and $V^h$. Since $K^h, Q^h$ each has $d_h$ rows and $V^h$ has $d_V$ rows, we need a total of $\widetilde{H}_h(2d_h + d_V)$ heads to approximate the $h$-th head of the target $\mathrm{Attn}$. As a result, we use $\sum_{h=1}^H \widetilde{H}_h(2d_h + d_V)$ heads in total to approximate all the key, query, and value matrices from $\mathrm{Attn}$.

Thus, we view $\mathrm{Attn}_{2\text{-}1}$ as groups of $\widetilde{H}_h$-head attentions, and we label each group as $\mathrm{Attn}_{2\text{-}1}^{h,j}$. Here $h \in [H]$ still identifies the $h$-th head of our target $\mathrm{Attn}$, and $j \in [2d_h + d_V]$ identifies the rows in $K^h, Q^h$ and $V^h$.

Finally, all $\widetilde{H}_h$ heads in Theorem D.4 require a shared linear mapping $A : \mathbb{R}^{d \times n} \to \mathbb{R}^{(3d+n) \times n}$. Here, we construct this common mapping with head-specific ones $A_{\widetilde{h}} : \mathbb{R}^{(d + \sum_{h=1}^H 2d_{wh} + n) \times n} \to \mathbb{R}^{(3d+n) \times n}$. $\widetilde{h}$ labels the heads in $\mathrm{Attn}_{2\text{-}1}$, and thus $\widetilde{h} \in [\sum_{h=1}^H \widetilde{H}_h(2d_h + d_V)]$. We use $A_{\widetilde{h}}$ to project the input to the required dimension $\mathbb{R}^{(3d+n) \times n}$ and to select the target $k_i^h, q_i^h$ or $v_i^h$ to approximate the desired linear transformation $(k_i^h)^\top X, (q_i^h)^\top X$ or $(v_i^h)^\top X$.

To that end, we construct $A_{\widetilde{h}}$ to be

$$A_{\widetilde{h}} := \underbrace{\begin{bmatrix} I_d & 0_{d \times d_{w\ell}} & 0_{d \times d_{w\ell}} & 0_{d \times n} \\ 0_{d \times d} & S_{\widetilde{h}} & 0_{d \times d_{w\ell}} & 0_{d \times n} \\ 0_{d \times d} & 0_{d \times d_{w\ell}} & S_{\widetilde{h}} & 0_{d \times n} \\ 0_{n \times d} & 0_{n \times d_{w\ell}} & 0_{n \times d_{w\ell}} & I_n \end{bmatrix}}_{(3d+n) \times (d + 2d_{w\ell} + n)} \cdot \underbrace{\begin{bmatrix} I_d & 0_{d \times \sum_{h=1}^H 2d_{wh}} & 0_{d \times n} \\ 0_{2d_{w\ell} \times d} & F_{\widetilde{h}} & 0_{2d_{w\ell} \times n} \\ 0_{n \times d} & 0_{n \times \sum_{h=1}^H 2d_{wh}} & I_n \end{bmatrix}}_{(d + 2d_{w\ell} + n) \times (d + \sum_{h=1}^H 2d_{wh} + n)}, \quad (103)$$

where

$$\ell(\widetilde{h}) := \sum_{h=1}^H h \cdot \mathbb{1}_{\{\lambda_{h-1} < \widetilde{h} \leq \lambda_h\}},$$

with

$$\lambda_h := \sum_{i=1}^h \widetilde{H}_i(2d_i + d_V), \quad \lambda_0 := 0.$$

Thus, $\ell(\widetilde{h}) \in [H]$ identifies the target head in $\mathrm{Attn}$.

We define $F_{\widetilde{h}}$ as

$$F_{\widetilde{h}} := \underbrace{\begin{bmatrix} 0_{2d_{w\ell} \times \zeta_{\ell-1}} & I_{2d_{w\ell}} & 0_{2d_{w\ell} \times (\zeta_H - \zeta_\ell)} \end{bmatrix}}_{2d_{w\ell} \times \zeta_H}, \quad (104)$$

with

$$\zeta_\ell := \sum_{h=1}^{\ell} 2d_{w^h}, \quad \zeta_0 := 0.$$

Thus, $F_{\widetilde{h}}$ picks out the $W_\ell$ from $W_{\mathrm{Attn}}$.

We define $S_{\widetilde{h}}$ as

$$S_{\widetilde{h}} := \underbrace{\begin{bmatrix} 0_{d \times d(j-1)} & I_d & 0_{d \times (d_{w_\ell} - d \cdot j)} \end{bmatrix}}_{d \times d_{w_\ell}}, \tag{105}$$

where

$$j(\widetilde{h}) := \lceil \frac{\widetilde{h} - \lambda_{\ell-1}}{\widetilde{H}_\ell} \rceil \in [2d_\ell + d_V].$$

Thus, $S_{\widetilde{h}}$ picks out $k_j^\ell, q_j^\ell$ or $v_j^\ell$ from $W_\ell$.

Then, we have

$$A_{\widetilde{h}} \cdot \begin{bmatrix} X \\ W_{\mathrm{Attn}} \\ I_n \end{bmatrix}$$

$$= \underbrace{\begin{bmatrix} I_d & 0_{d \times d_{w_\ell}} & 0_{d \times d_{w_\ell}} & 0_{d \times n} \\ 0_{d \times d} & S_{\widetilde{h}} & 0_{d \times d_{w_\ell}} & 0_{d \times n} \\ 0_{d \times d} & 0_{d \times d_{w_\ell}} & S_{\widetilde{h}} & 0_{d \times n} \\ 0_{n \times d} & 0_{n \times d_{w_\ell}} & 0_{n \times d_{w_\ell}} & I_n \end{bmatrix}}_{(3d+n) \times (d + 2d_{w_\ell} + n)} \cdot \underbrace{\begin{bmatrix} I_d & 0_{d \times \sum_{h=1}^{H} 2d_{w^h}} & 0_{d \times n} \\ 0_{2d_{w_\ell} \times d} & F_{\widetilde{h}} & 0_{2d_{w_\ell} \times n} \\ 0_{n \times d} & 0_{n \times \sum_{h=1}^{H} 2d_{w^h}} & I_n \end{bmatrix}}_{(d + 2d_{w_\ell} + n) \times (d + \sum_{h=1}^{H} 2d_{w^h} + n)} \cdot \underbrace{\begin{bmatrix} X \\ W_{\mathrm{Attn}} \\ I_n \end{bmatrix}}_{(d + \sum_{h=1}^{H} 2d_{w^h} + n) \times n}$$

$$\text{(By (103) and (102))}$$

$$= \underbrace{\begin{bmatrix} I_d & 0_{d \times d_{w_\ell}} & 0_{d \times d_{w_\ell}} & 0_{d \times n} \\ 0_{d \times d} & S_{\widetilde{h}} & 0_{d \times d_{w_\ell}} & 0_{d \times n} \\ 0_{d \times d} & 0_{d \times d_{w_\ell}} & S_{\widetilde{h}} & 0_{d \times n} \\ 0_{n \times d} & 0_{n \times d_{w_\ell}} & 0_{n \times d_{w_\ell}} & I_n \end{bmatrix}}_{(3d+n) \times (d + 2d_{w_\ell} + n)} \cdot \underbrace{\begin{bmatrix} X \\ F_{\widetilde{h}} \cdot W_{\mathrm{Attn}} \\ I_n \end{bmatrix}}_{(d + 2d_{w_\ell} + n) \times n} \qquad \text{(By matrix multiplication)}$$

$$= \underbrace{\begin{bmatrix} X \\ \begin{bmatrix} S_{\widetilde{h}} & 0_{d \times d_{w_\ell}} \end{bmatrix} \cdot F_{\widetilde{h}} \cdot W_{\mathrm{Attn}} \\ \begin{bmatrix} 0_{d \times d_{w_\ell}} & S_{\widetilde{h}} \end{bmatrix} \cdot F_{\widetilde{h}} \cdot W_{\mathrm{Attn}} \\ I_n \end{bmatrix}}_{(3d+n) \times n}, \tag{106}$$

and (106) follows from matrix multiplication.

$F_{\widetilde{h}} \cdot W_{\mathrm{Attn}}$ in (106) expands as

$$F_{\widetilde{h}} \cdot W_{\mathrm{Attn}} = \underbrace{\begin{bmatrix} 0_{2d_{w_\ell} \times \zeta_{\ell-1}} & I_{2d_{w_\ell}} & 0_{2d_{w_\ell} \times (\zeta_H - \zeta_\ell)} \end{bmatrix}}_{2d_{w_\ell} \times \zeta_H} \cdot \underbrace{\begin{bmatrix} W_1 \\ W_2 \\ \vdots \\ W_H \end{bmatrix}}_{\zeta_H \times n} \qquad \text{(By (104) and (101))}$$

$$= \underbrace{W_\ell}_{2d_{w_\ell} \times n} . \qquad \left( F_{\widetilde{h}} \text{ selects } W_\ell \text{ from } W_{\mathrm{Attn}} \right)$$

Then, $\begin{bmatrix} S_{\widetilde{h}} & 0_{d \times d_{w_\ell}} \end{bmatrix} \cdot F_{\widetilde{h}} \cdot W_{\text{Attn}}$ in (106) expands as

$$
\begin{aligned}
& \begin{bmatrix} S_{\widetilde{h}} & 0_{d \times d_{w_\ell}} \end{bmatrix} \cdot W_\ell \\
&= \underbrace{\begin{bmatrix} S_{\widetilde{h}} & 0_{d \times d_{w_\ell}} \end{bmatrix}}_{d \times 2d_{w_\ell}} \cdot \underbrace{\begin{bmatrix} 0 \cdot w_\ell & 1 \cdot w_\ell & 2 \cdot w_\ell & \cdots & (n-1) \cdot w_\ell \\ w_\ell & w_\ell & w_\ell & \cdots & w_\ell \end{bmatrix}}_{2d_{w_\ell} \times n} && \text{(By (100))} \\
&= \underbrace{\begin{bmatrix} 0 \cdot S_{\widetilde{h}} w_\ell & 1 \cdot S_{\widetilde{h}} w_\ell & 2 \cdot S_{\widetilde{h}} w_\ell & \cdots & (n-1) \cdot S_{\widetilde{h}} w_\ell \end{bmatrix}}_{d \times n}. && \text{(By matrix multiplication)}
\end{aligned}
$$

$S_{\widetilde{h}} w_\ell$ expands as

$$
S_{\widetilde{h}} w_\ell = \underbrace{\begin{bmatrix} 0_{d \times d(j-1)} & I_d & 0_{d \times (d_{w_\ell} - d \cdot j)} \end{bmatrix}}_{d \times d_{w_\ell}} \cdot \underbrace{\begin{bmatrix} k_1^\ell \\ \vdots \\ k_{d_\ell}^\ell \\ q_1^\ell \\ \vdots \\ q_{d_\ell}^\ell \\ v_1^\ell \\ \vdots \\ v_{d_V}^\ell \end{bmatrix}}_{d_{w_\ell} \times 1} \qquad \text{(By (105) and (99))}
$$

$$
= \begin{cases} k_j^\ell, & \text{when} \quad j \in [d_\ell] \\ q_{j-d_\ell}^\ell, & \text{when} \quad j \in [2d_\ell] \setminus [d_\ell] \\ v_{j-2d_\ell}^\ell, & \text{when} \quad j \in [2d_\ell + d_V] \setminus [2d_\ell] \end{cases} \qquad \left( S_{\widetilde{h}} \text{ selects } k_j^\ell, q_{j-d_\ell}^\ell \text{ or } v_{j-2d_\ell}^\ell \text{ from } w_\ell \right)
$$

Thus, acting $A_{\widetilde{h}}$ on our input yields

$$
A_{\widetilde{h}} \cdot \begin{bmatrix} X \\ W_{\text{Attn}} \\ I_n \end{bmatrix}
$$

$$
= \begin{cases} \underbrace{\begin{bmatrix} X \\ \begin{matrix} 0 \cdot k_j^\ell & 1 \cdot k_j^\ell & \cdots & (n-1) \cdot k_j^\ell \\ k_j^\ell & k_j^\ell & \cdots & k_j^\ell \end{matrix} \\ I_n \end{bmatrix}}_{(3d+n) \times n}, & \ell \in [H], \quad j \in [d_\ell] \\[3em]
\underbrace{\begin{bmatrix} X \\ \begin{matrix} 0 \cdot q_{j-d_\ell}^\ell & 1 \cdot q_{j-d_\ell}^\ell & \cdots & (n-1) \cdot q_{j-d_\ell}^\ell \\ q_{j-d_\ell}^\ell & q_{j-d_\ell}^\ell & \cdots & q_{j-d_\ell}^\ell \end{matrix} \\ I_n \end{bmatrix}}_{(3d+n) \times n}, & \ell \in [H], \quad j \in [2d_\ell] \setminus [d_\ell] \\[3em]
\underbrace{\begin{bmatrix} X \\ \begin{matrix} 0 \cdot v_{j-2d_\ell}^\ell & 1 \cdot v_{j-2d_\ell}^\ell & \cdots & (n-1) \cdot v_{j-2d_\ell}^\ell \\ v_{j-2d_\ell}^\ell & v_{j-2d_\ell}^\ell & \cdots & v_{j-2d_\ell}^\ell \end{matrix} \\ I_n \end{bmatrix}}_{(3d+n) \times n}, & \ell \in [H], \quad j \in [2d_\ell + d_V] \setminus [2d_\ell] \end{cases}.
$$

To keep our following proof clean, we define

$$\xi_\ell := \sum_{h=1}^{\ell} (2d_h + d_V), \quad \xi_0 := 0.$$

Then, for each $\ell \in [H]$ and $j \in [d_\ell]$, by Theorem D.4, there exists an $\widetilde{H}_\ell$-head attention $\mathrm{Attn}_{\ell,j}$ such that

$$\| \underbrace{\mathrm{Attn}_{\ell,j}(A_{\widetilde{h}} \cdot \overbrace{\begin{bmatrix} X \\ W_{\mathrm{Attn}} \\ I_n \end{bmatrix}}^{(3d+n)\times n} )_{:,i}}_{\xi_H \times 1} - \underbrace{e^{(\xi_H)}_{\xi_{\ell-1}+j} \cdot (k_j^\ell)^\top x_i}_{\xi_H \times 1} \|_\infty \le \max\{|a_\ell|, |b_\ell|\} \cdot \epsilon_0 + \frac{\epsilon_0}{2}.$$

Thus, we have

$$\| \underbrace{\mathrm{Attn}_{\ell,j}(A_{\widetilde{h}} \cdot \begin{bmatrix} X \\ W_{\mathrm{Attn}} \\ I_n \end{bmatrix})}_{\xi_H \times n} - \underbrace{e^{(\xi_H)}_{\xi_{\ell-1}+j}}_{\xi_H \times 1} \cdot \underbrace{(k_j^\ell)^\top X}_{1 \times n} \|_\infty \le \max\{|a_\ell|, |b_\ell|\} \cdot \epsilon_0 + \frac{\epsilon_0}{2},$$

and

$$\| \underbrace{\sum_{j=1}^{d_\ell} \mathrm{Attn}_{\ell,j}(A_{\widetilde{h}} \cdot \begin{bmatrix} X \\ W_{\mathrm{Attn}} \\ I_n \end{bmatrix})}_{\xi_H \times n} - \underbrace{\begin{bmatrix} 0_{d_1 \times n} \\ \vdots \\ K^\ell \\ \vdots \\ 0_{d_V \times n} \end{bmatrix}}_{\xi_H \times n} \|_\infty \le \max\{|a_\ell|, |b_\ell|\} \cdot \epsilon_0 + \frac{\epsilon_0}{2}.$$

Similarly, together with $Q^\ell$ and $V^\ell$, we have

$$\| \underbrace{\sum_{j=1}^{2d_\ell+d_V} \mathrm{Attn}_{\ell,j}(A_{\widetilde{h}} \cdot \begin{bmatrix} X \\ W_{\mathrm{Attn}} \\ I_n \end{bmatrix})}_{\xi_H \times n} - \underbrace{\begin{bmatrix} 0_{d_1 \times n} \\ \vdots \\ K^\ell \\ Q^\ell \\ V^\ell \\ \vdots \\ 0_{d_V \times n} \end{bmatrix}}_{\xi_H \times n} \|_\infty \le \max\{|a_\ell|, |b_\ell|\} \cdot \epsilon_0 + \frac{\epsilon_0}{2}$$

Finally, we have

$$\| \underbrace{\sum_{\ell=1}^{H} \sum_{j=1}^{2d_\ell+d_V} \mathrm{Attn}_{\ell,j}(A_{\widetilde{h}} \cdot \begin{bmatrix} X \\ W_{\mathrm{Attn}} \\ I_n \end{bmatrix})}_{\xi_H \times n} - \underbrace{\begin{bmatrix} K^1 \\ Q^1 \\ V^1 \\ \vdots \\ K^h \\ Q^h \\ V^h \end{bmatrix}}_{\xi_H \times n} \|_\infty \le \epsilon_0 \sum_{\ell=1}^{H} (\max\{|a_\ell|, |b_\ell|\} + \frac{1}{2}).$$

By Lemma E.5, $\text{Attn}_{\ell,j} \circ A_{\widetilde{h}}$ is still an attention, so we define

$$\text{Attn}_{2\text{-}1}^{\ell,j} := \text{Attn}_{\ell,j} \circ A_{\widetilde{h}}.$$

Then, we have

$$\| \underbrace{\sum_{\ell=1}^{H} \sum_{j=1}^{2d_\ell+d_V} \text{Attn}_{2\text{-}1}^{\ell,j} \left( \begin{bmatrix} X \\ W_{\text{Attn}} \\ I_n \end{bmatrix} \right)}_{\xi_H \times n} - \underbrace{\begin{bmatrix} K^1 \\ Q^1 \\ V^1 \\ \vdots \\ K^h \\ Q^h \\ V^h \end{bmatrix}}_{\xi_H \times n} \|_\infty \le \epsilon_0 \sum_{\ell=1}^{H} (\max\{|a_\ell|, |b_\ell|\} + \frac{1}{2}).$$

For the convenience of the following presentation, we define

$$\begin{bmatrix} (K^1)' \\ (Q^1)' \\ (V^1)' \\ \vdots \\ (K^H)' \\ (Q^H)' \\ (V^H)' \end{bmatrix} := \sum_{\ell=1}^{H} \sum_{j=1}^{2d_\ell+d_V} \text{Attn}_{2\text{-}1}^{\ell,j} \left( \begin{bmatrix} X \\ W_{\text{Attn}} \\ I_n \end{bmatrix} \right),$$

$$\widetilde{\epsilon} := \epsilon_0 \sum_{\ell=1}^{H} (\max\{|a_\ell|, |b_\ell|\} + \frac{1}{2}).$$

**Step 3: Assemble the Approximated Maps.** In this part of our proof, our goal is to reconstruct the attention mechanism $(V^h)' \cdot \sigma_{\beta_h}(((K^h)')^\top (Q^h)')$ from the approximated maps via $\text{Attn}_{2\text{-}2}$.

To that end, we construct $\text{Attn}_{2\text{-}2}$ as an $H$-head attention, with each head recovering one of the heads $(V^h)' \cdot \sigma_{\beta_h}(((K^h)')^\top (Q^h)')$. We label the $h$-th head in $\text{Attn}_{2\text{-}2}$ as $\text{Attn}_{2\text{-}2}^{(h)}$, and we use $W_{2,K}^h, W_{2,Q}^h, W_{2,V}^h$ for the weight matrices in $\text{Attn}_{2\text{-}2}^{(h)}$.

Specifically, we construct the $h$-th head in $\text{Attn}_{2\text{-}2}$ as

$$W_{2,K}^h := \underbrace{\begin{bmatrix} 0_{d_h \times d_1} & \cdots & I_{d_h} & 0_{d_h \times d_h} & 0_{d_h \times d_V} & \cdots & 0_{d_h \times d_V} \end{bmatrix}}_{d_h \times \xi_H},$$

$$W_{2,Q}^h := \underbrace{\begin{bmatrix} 0_{d_h \times d_1} & \cdots & 0_{d_h \times d_h} & I_{d_h} & 0_{d_h \times d_V} & \cdots & 0_{d_h \times d_V} \end{bmatrix}}_{d_h \times \xi_H},$$

$$W_{2,V}^h := \underbrace{\begin{bmatrix} 0_{d_V \times d_1} & \cdots & 0_{d_V \times d_h} & 0_{d_V \times d_h} & I_{d_V} & \cdots & 0_{d_V \times d_V} \end{bmatrix}}_{d_V \times \xi_H}.$$

The construction provides us with

$$W_{2,K}^h \cdot \begin{bmatrix} (K^1)' \\ \vdots \\ (K^h)' \\ (Q^h)' \\ (V^h)' \\ \vdots \\ (V^H)' \end{bmatrix} = \underbrace{\begin{bmatrix} 0_{d_h \times d_1} & \cdots & I_{d_h} & 0_{d_h \times d_h} & 0_{d_h \times d_V} & \cdots & 0_{d_h \times d_V} \end{bmatrix}}_{d_h \times \xi_H} \cdot \underbrace{\begin{bmatrix} (K^1)' \\ \vdots \\ (K^h)' \\ (Q^h)' \\ (V^h)' \\ \vdots \\ (V^H)' \end{bmatrix}}_{\xi_H \times n} = \underbrace{(K^h)'}_{d_h \times n},$$

$$W_{2,Q}^h \cdot \begin{bmatrix} (K^1)' \\ \vdots \\ (K^h)' \\ (Q^h)' \\ (V^h)' \\ \vdots \\ (V^H)' \end{bmatrix} = \underbrace{\begin{bmatrix} 0_{d_h \times d_1} & \cdots & 0_{d_h \times d_h} & I_{d_h} & 0_{d_h \times d_V} & \cdots & 0_{d_h \times d_V} \end{bmatrix}}_{d_h \times \xi_H} \cdot \underbrace{\begin{bmatrix} (K^1)' \\ \vdots \\ (K^h)' \\ (Q^h)' \\ (V^h)' \\ \vdots \\ (V^H)' \end{bmatrix}}_{\xi_H \times n} = \underbrace{(Q^h)'}_{d_h \times n},$$

$$W_{2,V}^h \cdot \begin{bmatrix} (K^1)' \\ \vdots \\ (K^h)' \\ (Q^h)' \\ (V^h)' \\ \vdots \\ (V^H)' \end{bmatrix} = \underbrace{\begin{bmatrix} 0_{d_V \times d_1} & \cdots & 0_{d_V \times d_h} & 0_{d_V \times d_h} & I_{d_V} & \cdots & 0_{d_V \times d_V} \end{bmatrix}}_{d_V \times \xi_H} \cdot \underbrace{\begin{bmatrix} (K^1)' \\ \vdots \\ (K^h)' \\ (Q^h)' \\ (V^h)' \\ \vdots \\ (V^H)' \end{bmatrix}}_{\xi_H \times n} = \underbrace{(V^h)'}_{d_V \times n}.$$

Thus, our $h$-th head of $\mathrm{Attn}_{2\text{-}2}$ is

$$\mathrm{Attn}_{2\text{-}2}^{(h)} := \underbrace{(V^h)'}_{d_V \times n} \cdot \underbrace{\sigma_{\beta_h}(((K^h)')^\top (Q^h)')}_{n \times n}.$$

Then, all of the heads from $\mathrm{Attn}_{2\text{-}2}$ give us

$$\sum_{h=1}^H \mathrm{Attn}_{2\text{-}2}^{(h)} = \sum_{h=1}^H (V^h)' \cdot \sigma_{\beta_h}(((K^h)')^\top (Q^h)').$$

**Step 4: Error Bound.** To bound the error, we compute

$$\| \sum_{h=1}^H (V^h)' \cdot \sigma_{\beta_h}(((K^h)')^\top (Q^h)') - \sum_{h=1}^H V^h \cdot \sigma_{\beta_h}((K^h)^\top Q^h) \|_\infty$$

$$\leq \sum_{h=1}^H \|(V^h)' \cdot \sigma_{\beta_h}(((K^h)')^\top (Q^h)') - V^h \cdot \sigma_{\beta_h}((K^h)^\top Q^h)\|_\infty \qquad \text{(By triangle inequality)}$$

For each $\|(V^h)' \cdot \sigma_{\beta_h}(((K^h)')^\top (Q^h)') - V^h \cdot \sigma_{\beta_h}((K^h)^\top Q^h)\|_\infty$, we utilize the error analysis from the proof of Hu et al. (2026b, Theorem 4.1). That is, for $0 < \widehat{\epsilon} < 2$, we have

$$\|(V^h)' \cdot \sigma_{\beta_h}(((K^h)')^\top (Q^h)') - V^h \cdot \sigma_{\beta_h}((K^h)^\top Q^h)\|_\infty < \widetilde{\epsilon} + n B_{KQV}\widehat{\epsilon},$$

where

$$0 < \widetilde{\epsilon} \leq \min\{1, \frac{-\ln(1 - \frac{\widehat{\epsilon}}{2})}{\beta_h d_h (2B_{KQV} + 1)}\}.$$

Thus, we have

$$\sum_{h=1}^H \|(V^h)' \cdot \sigma_{\beta_h}(((K^h)')^\top (Q^h)') - V^h \cdot \sigma_{\beta_h}((K^h)^\top Q^h)\|_\infty < \sum_{h=1}^H (\widetilde{\epsilon} + n B_{KQV}\widehat{\epsilon}).$$

Since we are able to make $\widetilde{\epsilon}$ and $\widehat{\epsilon}$ arbitrarily small, we complete the proof. $\qquad\square$

### E.6 Proof of Lemma 3.5

**Corollary E.1.1** (Lemma 3.5 Restated: In-Context Emulation of Multi-Head Attention with Flow-Through Component).
*Let $Z \in \mathbb{R}^{d_Z \times n}$ be any matrix satisfying $\|Z\|_\infty \leq B_Z$, for some $B_Z > 0$. Then, with the same assumption as Lemma 3.4, for any $\epsilon_5 > 0$, there exists a two-layer self-attention $\mathrm{Attn}_2$ such that*

$$\|\mathrm{Attn}_2(\begin{bmatrix} X_p \\ Z \end{bmatrix}) - \begin{bmatrix} \mathrm{Attn}(X) \\ Z \end{bmatrix}\|_\infty \leq \epsilon_5.$$

*Proof.* For the convenience of presentation, we denote the two-layer attention $\mathrm{Attn}_2$ as $\mathrm{Attn}_{2\text{-}2} \circ \mathrm{Attn}_{2\text{-}1}$ to distinguish the layers.

Then, our proof consists of three major steps.

**Step 1.** In this part of our proof, we construct the first layer $\mathrm{Attn}_{2\text{-}1}$. We follow the proof strategy of Lemma 3.3. That is, the heads in $\mathrm{Attn}_{2\text{-}1}$ consist of three parts. In the first part, we incorporate the first-layer emulator from Lemma 3.4 into $\mathrm{Attn}_{2\text{-}1}$. Such a technique can be found from (32) to (36). Secondly, we construct an auxiliary head to flow through the identity $I_n$ from the input. Such a technique can be found from (37) to (44). Thirdly, we construct another auxiliary head to flow through the $Z$ from the input. Such a technique can be found from (45) to (48).

With the above construction, we can have the following output from the first layer $\mathrm{Attn}_{2\text{-}1}$

$$\mathrm{Attn}_{2\text{-}1}(\begin{bmatrix} X_p \\ Z \end{bmatrix}) = \begin{bmatrix} \mathrm{Attn}'_{2\text{-}1}(X_p) \\ I_n \\ Z\sigma_{\beta_2}(I_n) \end{bmatrix}, \tag{107}$$

where $\mathrm{Attn}'_{2\text{-}1}$ denotes the first layer of the emulator from Lemma 3.4. $\beta_2$ is the Softmax temperature in the $Z$-copying auxiliary head (This temperature is up to our choice).

**Step 2.** In this part of our proof, we construct the second layer $\mathrm{Attn}_{2\text{-}2}$. Again, the heads in this layer consist of two parts. In the first part, we incorporate the second layer of the emulator from Lemma 3.4 into $\mathrm{Attn}_{2\text{-}2}$. The technique can be found from (53) to (57). Secondly, we build an auxiliary head to flow-through the $Z$ component by utilizing the intermediate identity. Such a technique can be found from (58) to (62). With the above construction, we can have the following output from the second layer

$$\mathrm{Attn}_2(\begin{bmatrix} X_p \\ Z \end{bmatrix}) = \begin{bmatrix} \mathrm{Attn}'_2(X_p) \\ Z\sigma_{\beta_2}(I_n)\sigma_{\beta_4}(I_n) \end{bmatrix},$$

where $\mathrm{Attn}'_{2\text{-}1}$ denotes the full emulator from Lemma 3.4. Again, $\beta_4$ is the Softmax temperature in the $Z$-copying auxiliary head (This temperature is up to our choice).

**Step 3.** In this step, we perform the error analysis between the model output and our target

$$\|\mathrm{Attn}_2(\begin{bmatrix} X_p \\ Z \end{bmatrix}) - \begin{bmatrix} \mathrm{Attn}(X) \\ Z \end{bmatrix}\|_\infty$$

$$\leq \|\mathrm{Attn}_2(\begin{bmatrix} X_p \\ Z \end{bmatrix}) - \begin{bmatrix} \mathrm{Attn}'_2(X_p) \\ Z \end{bmatrix}\|_\infty + \|\begin{bmatrix} \mathrm{Attn}'_2(X_p) \\ Z \end{bmatrix} - \begin{bmatrix} \mathrm{Attn}(X) \\ Z \end{bmatrix}\|_\infty \quad \text{(By triangle inequality)}$$

$$\leq \|\begin{bmatrix} \mathrm{Attn}'_2(X_p) \\ Z\sigma_{\beta_2}(I_n)\sigma_{\beta_4}(I_n) \end{bmatrix} - \begin{bmatrix} \mathrm{Attn}'_2(X_p) \\ Z \end{bmatrix}\|_\infty + \epsilon_4 \quad \text{(By (107) and Lemma 3.4)}$$

$$\leq B_Z \cdot \frac{n(n-1)}{e^{\beta_2} + (n-1)} \cdot \frac{n(n-1)}{e^{\beta_4} + (n-1)} + \frac{n(n-1)B_Z}{e^{\beta_2} + (n-1)} + \frac{n(n-1)B_Z}{e^{\beta_4} + (n-1)} + \epsilon_4. \quad \text{(By (78))}$$

For any $\widehat{\epsilon}_1 > 0$, by choosing $\beta_2, \beta_4$ as in (79), we have

$$\|\mathrm{Attn}_2(\begin{bmatrix} X_p \\ Z \end{bmatrix}) - \begin{bmatrix} \mathrm{Attn}(X) \\ Z \end{bmatrix}\|_\infty \leq 2\widehat{\epsilon}_1 + \frac{\widehat{\epsilon}_1^2}{B_Z} + \epsilon_4.$$

Since we are able to make each $\epsilon$ arbitrarily small, we complete the proof. $\square$

### E.7 Proof of Theorem 3.2

**Lemma E.7** (Uniform Bounded Attention-Only Surrogate Template). *Let $1 \le p < \infty$ and $L > 0$. Let $\mathcal{C} = [-b, b]^{d \times n} \subset \mathbb{R}^{d \times n}$, and let $\mathcal{F}_L := \{f : \mathcal{C} \to \mathcal{C} \mid \mathrm{Lip}(f) \le L\}$. Fix $\epsilon_{\mathrm{sur}} > 0$. Then there exists a fixed four-layer attention-only surrogate template*

$$A_\phi := \mathrm{Attn}_{\phi_4} \circ \mathrm{Attn}_{\phi_3} \circ \mathrm{Attn}_{\phi_2} \circ \mathrm{Attn}_{\phi_1}, \quad \phi = (\phi_1, \phi_2, \phi_3, \phi_4),$$

*with fixed layer dimensions, fixed sequence length $n + 1$, fixed head numbers, fixed head dimensions, fixed inverse temperatures, and fixed prompt shape such that, for every $f \in \mathcal{F}_L$, there exists $\phi_f$ satisfying*

$$\mathrm{d}_p((A_{\phi_f})_{:,1:n}, f) \le \epsilon_{\mathrm{sur}},$$

*Moreover, the encoded prompt blocks $W_{\phi_f,1}, \ldots, W_{\phi_f,4}$ from Definition 3.5, and all intermediate quantities required by Lemmas 3.4 and 3.5, are uniformly bounded over $f \in \mathcal{F}_L$ and $X \in \mathcal{C}$. All constants in these bounds, as well as the template dimensions and prompt shape, depend only on $\mathcal{C}, L, p, d, n, \epsilon_{\mathrm{sur}}$, not on $f$ or $X$.*

*Proof of Lemma E.7.* Let $N := dn$. Choose an integer $g \ge 2$ large enough so that

$$2|\mathcal{C}|^{1/p} N^{1/p} Lb/g \le \epsilon_{\mathrm{sur}}/3.$$

Let $\mathcal{G}_g$ and $\pi_g : \mathcal{C} \to \mathcal{G}_g$ be the standard finite-grid quantization map from Section D.3. Then

$$\|X - \pi_g(X)\|_\infty \le b/g, \quad X \in \mathcal{C}. \tag{108}$$

For each $v \in \mathcal{G}_g$, choose $X_v \in \mathcal{C}$ such that

$$\pi_g(X_v) = v.$$

This is possible because $\mathcal{G}_g = \pi_g(\mathcal{C})$.

Fix $f \in \mathcal{F}_L$. Define

$$a_v := f(X_v), \quad v \in \mathcal{G}_g,$$
$$F_{f,g}(X) := a_{\pi_g(X)}.$$

Recall

$$\mathcal{H}_{g,M} := \{F : \mathcal{C} \to \mathbb{R}^{d \times n} \mid F(X) = a_{\pi_g(X)}, \|a_v\|_\infty \le M \text{ for all } v \in \mathcal{G}_g\}.$$

Then $F_{f,g} \in \mathcal{H}_{g,b}$ from Proposition D.2, since

$$\|a_v\|_\infty = \|f(X_v)\|_\infty \le b.$$

For any $X \in \mathcal{C}$, write $v = \pi_g(X)$. Then

$$
\begin{aligned}
\|F_{f,g}(X) - f(X)\|_\infty &= \|f(X_v) - f(X)\|_\infty \\
&\le L\|X_v - X\|_\infty && \text{(By } \mathrm{Lip}(f) \le L) \\
&\le L(\|X_v - v\|_\infty + \|v - X\|_\infty) && \text{(By triangle inequality)} \\
&= L(\|X_v - \pi_g(X_v)\|_\infty + \|\pi_g(X) - X\|_\infty) && \text{(By } v = \pi_g(X_v) = \pi_g(X)) \\
&\le 2Lb/g, && (109)
\end{aligned}
$$

where the last line is by Equation (108).

Hence

$$\mathrm{d}_p(F_{f,g}, f) = \left( \int_{\mathcal{C}} \|F_{f,g}(X) - f(X)\|_p^p \mathrm{d}X \right)^{1/p}$$

$$\leq (\int_{\mathcal{C}} (N^{1/p}\|F_{f,g}(X) - f(X)\|_\infty)^p \mathrm{d}X)^{1/p} \qquad \text{(By } \|A\|_p \leq N^{1/p}\|A\|_\infty)$$

$$\leq |\mathcal{C}|^{1/p} N^{1/p} \cdot 2Lb/g \qquad \text{(By Equation (109) and } \int_{\mathcal{C}} \mathrm{d}X = |\mathcal{C}|)$$

$$\leq \epsilon_{\mathrm{sur}}/3. \qquad \text{(By the choice of } g)$$

Apply Proposition D.2 with $M = b$ and $\eta = 2\epsilon_{\mathrm{sur}}/3$. Since $F_{f,g} \in \mathcal{H}_{g,b}$, there exists a parameter choice $\phi_f$ in a fixed four-layer attention-only template $A_\phi$ such that

$$\mathrm{d}_p((A_{\phi_f})_{:,1:n}, F_{f,g}) \leq 2\epsilon_{\mathrm{sur}}/3,$$

Therefore,

$$\mathrm{d}_p((A_{\phi_f})_{:,1:n}, f) \leq \mathrm{d}_p((A_{\phi_f})_{:,1:n}, F_{f,g}) + \mathrm{d}_p(F_{f,g}, f) \qquad \text{(By triangle inequality)}$$

$$\leq 2\epsilon_{\mathrm{sur}}/3 + \epsilon_{\mathrm{sur}}/3$$

$$= \epsilon_{\mathrm{sur}}.$$

It remains to check the uniform boundedness claims.

- By Proposition D.2, the four-layer attention-only template has fixed dimensions, fixed head numbers, fixed head dimensions, fixed inverse temperatures, fixed prompt shape, and uniformly bounded weights over $f \in \mathcal{F}_L$. Hence the encoded prompt blocks from Definition 3.5 are uniformly bounded, because each block only stacks these weights and their token-indexed multiples.

- Also, $X_a$ is uniformly bounded because $X \in [-b, b]^{d \times n}$ and $X_a$ only appends zeros and an identity block. Since the template has fixed dimensions and uniformly bounded weights, each surrogate attention layer maps uniformly bounded inputs to uniformly bounded outputs. Indeed, every softmax matrix has entries in $[0, 1]$, and each layer has only finitely many heads. Iterating this argument over the four fixed layers shows that all intermediate surrogate states are uniformly bounded over $f \in \mathcal{F}_L$ and $X \in \mathcal{C}$.

- Finally, the key, query, and value matrices in the surrogate layers have fixed dimensions and uniformly bounded entries. Since the corresponding layer inputs are uniformly bounded, all K/Q/V products required by Lemmas 3.4 and 3.5 are uniformly bounded. The flow-through blocks are concatenations of later prompt blocks, so they are uniformly bounded as well.

All constants depend only on

$$\mathcal{C}, L, p, d, n, \epsilon_{\mathrm{sur}},$$

because $g$ is chosen only from these quantities and Proposition D.2 gives uniform bounds after $b, g, M \equiv b, p, d, n, \epsilon_{\mathrm{sur}}$ are fixed. They do not depend on $f$ or $X$. This completes the proof. $\square$

**Theorem E.2** (Theorem 3.2 Restated: In-Context Universality via Attention-Only Surrogate Emulation). *Let* $\mathcal{C} = [-b, b]^{d \times n} \subset \mathbb{R}^{d \times n}$ *be compact, and let* $\mathcal{F}_L := \{f : \mathcal{C} \to \mathcal{C} \mid \mathrm{Lip}(f) \leq L\}$ *be a set of L-Lipschitz continuous sequence-to-sequence functions. Then, for any* $\epsilon > 0$*, there exists a fixed eight-layer softmax-attention emulator* $T_{\theta^\star} = \mathrm{Attn}^8$ *such that, for every* $f \in \mathcal{F}_L$*, there exists a prompt* $P_f := [W_{\phi_f,1}; W_{\phi_f,2}; W_{\phi_f,3}; W_{\phi_f,4}]$ *(Definition 3.5) satisfying*

$$d_p(\widetilde{T}_{\theta^\star, P_f}, f) \leq \epsilon, \quad \widetilde{T}_{\theta^\star, P_f}(X) := T_{\theta^\star}([X_a; P_f])_{:,1:n},$$

*where* $X_a$ *follows Definition 3.5. The emulator weights* $\theta^\star$ *and the prompt shape depend only on* $\mathcal{C}, L, p, d, n, \epsilon$*, not on* $f$ *or* $X$*. Only the prompt entries depend on* $f$*.*

*Proof of Theorem 3.2.* Let $N := dn$. Choose $\epsilon_{\mathrm{sur}}, \epsilon_{\mathrm{emu}} > 0$ such that

$$\epsilon_{\mathrm{sur}} + \epsilon_{\mathrm{emu}} \leq \epsilon.$$

The emulation Lemmas 3.4 and 3.5 give pointwise $\ell_\infty$ bounds, while the theorem uses the $\mathrm{d}_p$ metric. We therefore choose the pointwise tolerance so that its contribution to $\mathrm{d}_p$ is at most $\epsilon_{\mathrm{emu}}$:

$$\epsilon_{\mathrm{emu\text{-}pt}} := |\mathcal{C}|^{-1/p} N^{-1/p} \epsilon_{\mathrm{emu}}.$$

Apply Lemma E.7 with accuracy $\epsilon_{\text{sur}}$. Then there exists a fixed four-layer attention-only surrogate template

$$A_\phi := \text{Attn}_{\phi_4} \circ \text{Attn}_{\phi_3} \circ \text{Attn}_{\phi_2} \circ \text{Attn}_{\phi_1}, \quad \phi = (\phi_1, \phi_2, \phi_3, \phi_4),$$

with fixed dimensions and fixed prompt shape such that, for every $f \in \mathcal{F}_L$, there exists $\phi_f$ satisfying

$$d_p(\widetilde{A}_{\phi_f}, f) \leq \epsilon_{\text{sur}}, \quad \widetilde{A}_{\phi_f}(X) := A_{\phi_f}(X_a)_{:,1:n}. \tag{110}$$

Moreover, Lemma E.7 gives the common bounds required by Lemmas 3.4 and 3.5, uniformly over $f \in \mathcal{F}_L$ and $X \in \mathcal{C}$.

For each $f \in \mathcal{F}_L$, let

$$P_f := [W_{\phi_f,1}; W_{\phi_f,2}; W_{\phi_f,3}; W_{\phi_f,4}]$$

be the prompt from Definition 3.5. Write

$$A_i := \text{Attn}_{i,\phi_f,i}, \quad i \in [4].$$

We first define the exact intermediate states of the four-layer surrogate. These are the states that would appear if we executed the encoded surrogate exactly:

$$\begin{aligned}
Y_0^f(X) &:= [X_a; W_{\phi_f,1}; W_{\phi_f,2}; W_{\phi_f,3}; W_{\phi_f,4}], \\
Y_1^f(X) &:= [A_1(X_a); W_{\phi_f,2}; W_{\phi_f,3}; W_{\phi_f,4}], \\
Y_2^f(X) &:= [A_2 \circ A_1(X_a); W_{\phi_f,3}; W_{\phi_f,4}], \\
Y_3^f(X) &:= [A_3 \circ A_2 \circ A_1(X_a); W_{\phi_f,4}], \\
Y_4^f(X) &:= A_4 \circ A_3 \circ A_2 \circ A_1(X_a).
\end{aligned}$$

The first three states still contain the prompt blocks needed by later surrogate layers. The last state is the surrogate output.

We next prepare the bookkeeping step for error propagation. Each emulator lemma controls one module on the exact input to that module. In the stacked emulator, however, later modules receive approximate states. We therefore define small safety regions around the exact states.

For $i = 1, 2, 3$, define

$$\begin{aligned}
\mathcal{S}_i &:= \{Y_i^f(X) : f \in \mathcal{F}_L, \ X \in \mathcal{C}\}, \\
\mathcal{K}_i &:= \{Y : \inf_{S \in \mathcal{S}_i} \|Y - S\|_\infty \leq \epsilon_{\text{emu-pt}}\}.
\end{aligned}$$

By Lemma E.7, the sets $\mathcal{S}_i$ are bounded. Hence each $\mathcal{K}_i$ is bounded. Since the distance-to-a-set map is continuous, each $\mathcal{K}_i$ is also closed. Therefore, each $\mathcal{K}_i$ is compact.

We choose the four two-layer emulators from back to front. This order matters: an error made by an early emulator passes through all later emulators. Therefore, we first fix the later emulators and then scale the earlier tolerances by their Lipschitz constants.

- **Step 1: Choose the Fourth-Layer Emulator.** Apply Lemma 3.4 to the fourth surrogate layer with tolerance

$$\epsilon_4 := \epsilon_{\text{emu-pt}}/4.$$

This gives a fixed two-layer attention emulator $E_4$ such that, for every $f \in \mathcal{F}_L$ and $X \in \mathcal{C}$,

$$\|E_4(Y_3^f(X)) - Y_4^f(X)\|_\infty \leq \epsilon_4. \tag{111}$$

After fixing $E_4$, choose a Lipschitz constant $L_4 \geq 1$ of $E_4$ on $\mathcal{K}_3$.

- **Step 2: Choose the Third-Layer Emulator.** Apply Lemma 3.5 to the third surrogate layer with tolerance

$$\eta_3 := \frac{\epsilon_{\text{emu-pt}}}{4L_4}.$$

This gives a fixed two-layer attention emulator $E_3$ such that, for every $f \in \mathcal{F}_L$ and $X \in \mathcal{C}$,

$$\|E_3(Y_2^f(X)) - Y_3^f(X)\|_\infty \leq \eta_3. \tag{112}$$

After fixing $E_3$, choose a Lipschitz constant $L_3 \geq 1$ of $E_3$ on $\mathcal{K}_2$.

- **Step 3: Choose the Second-Layer Emulator.** Apply Lemma 3.5 to the second surrogate layer with tolerance

$$\eta_2 := \frac{\epsilon_{\text{emu-pt}}}{4L_3L_4}.$$

This gives a fixed two-layer attention emulator $E_2$ such that, for every $f \in \mathcal{F}_L$ and $X \in \mathcal{C}$,

$$\|E_2(Y_1^f(X)) - Y_2^f(X)\|_\infty \le \eta_2. \tag{113}$$

After fixing $E_2$, choose a Lipschitz constant $L_2 \ge 1$ of $E_2$ on $\mathcal{K}_1$.

- **Step 4: Choose the First-Layer Emulator.** Apply Lemma 3.5 to the first surrogate layer with tolerance

$$\eta_1 := \frac{\epsilon_{\text{emu-pt}}}{4L_2L_3L_4}.$$

This gives a fixed two-layer attention emulator $E_1$ such that, for every $f \in \mathcal{F}_L$ and $X \in \mathcal{C}$,

$$\|E_1(Y_0^f(X)) - Y_1^f(X)\|_\infty \le \eta_1. \tag{114}$$

Stack the four two-layer emulators:

$$T_{\theta^\star} := E_4 \circ E_3 \circ E_2 \circ E_1.$$

This is an eight-layer attention emulator. Its weights are fixed because each $E_i$ uses only the fixed dimensions, common bounds, and error budgets above. These quantities do not depend on $f$ or $X$.

We now track the stacked error. Fix arbitrary $f \in \mathcal{F}_L$ and $X \in \mathcal{C}$. For readability, write

$$Y_i(X) := Y_i^f(X), \quad i = 0, 1, 2, 3, 4.$$

Define the actual emulator states by

$$\begin{aligned}
\widehat{Y}_0(X) &:= Y_0(X), \\
\widehat{Y}_1(X) &:= E_1(\widehat{Y}_0(X)), \\
\widehat{Y}_2(X) &:= E_2(\widehat{Y}_1(X)), \\
\widehat{Y}_3(X) &:= E_3(\widehat{Y}_2(X)), \\
\widehat{Y}_4(X) &:= E_4(\widehat{Y}_3(X)).
\end{aligned}$$

The goal is to bound $\|\widehat{Y}_4(X) - Y_4(X)\|_\infty$. This compares the stacked emulator output with the exact surrogate output.

- First, the first emulator gives

$$\begin{aligned}
\|\widehat{Y}_1(X) - Y_1(X)\|_\infty &\le \eta_1 && \text{(By Equation (114))} \\
&\le \epsilon_{\text{emu-pt}}. && \text{(By } L_2, L_3, L_4 \ge 1\text{)}
\end{aligned}$$

Hence $\widehat{Y}_1(X) \in \mathcal{K}_1$.

- Next,

$$\begin{aligned}
\|\widehat{Y}_2(X) - Y_2(X)\|_\infty &= \|E_2(\widehat{Y}_1(X)) - Y_2(X)\|_\infty \\
&\le \|E_2(\widehat{Y}_1(X)) - E_2(Y_1(X))\|_\infty + \|E_2(Y_1(X)) - Y_2(X)\|_\infty && \text{(By triangle inequality)} \\
&\le L_2\|\widehat{Y}_1(X) - Y_1(X)\|_\infty + \eta_2 && \text{(By Lipschitzness on } \mathcal{K}_1 \text{ and Equation (113))} \\
&\le L_2\eta_1 + \eta_2 \\
&= \frac{\epsilon_{\text{emu-pt}}}{4L_3L_4} + \frac{\epsilon_{\text{emu-pt}}}{4L_3L_4} && \text{(By the definitions of } \eta_1 \text{ and } \eta_2\text{)} \\
&= \frac{\epsilon_{\text{emu-pt}}}{2L_3L_4} && (115) \\
&\le \epsilon_{\text{emu-pt}}. && \text{(By } L_3, L_4 \ge 1\text{)}
\end{aligned}$$

Hence $\widehat{Y}_2(X) \in \mathcal{K}_2$.

- Similarly,

$$
\begin{aligned}
\|\widehat{Y}_3(X) - Y_3(X)\|_\infty &= \|E_3(\widehat{Y}_2(X)) - Y_3(X)\|_\infty \\
&\leq \|E_3(\widehat{Y}_2(X)) - E_3(Y_2(X))\|_\infty + \|E_3(Y_2(X)) - Y_3(X)\|_\infty && \text{(By triangle inequality)} \\
&\leq L_3\|\widehat{Y}_2(X) - Y_2(X)\|_\infty + \eta_3 && \text{(By Lipschitzness on } \mathcal{K}_2 \text{ and Equation (112))} \\
&\leq L_3 \cdot \frac{\epsilon_{\text{emu-pt}}}{2L_3 L_4} + \frac{\epsilon_{\text{emu-pt}}}{4L_4} && \text{(By Equation (115) and the definition of } \eta_3) \\
&= \frac{3\epsilon_{\text{emu-pt}}}{4L_4} && (116) \\
&\leq \epsilon_{\text{emu-pt}}. && \text{(By } L_4 \geq 1)
\end{aligned}
$$

Hence $\widehat{Y}_3(X) \in \mathcal{K}_3$.

- Finally,

$$
\begin{aligned}
\|\widehat{Y}_4(X) - Y_4(X)\|_\infty &= \|E_4(\widehat{Y}_3(X)) - Y_4(X)\|_\infty \\
&\leq \|E_4(\widehat{Y}_3(X)) - E_4(Y_3(X))\|_\infty + \|E_4(Y_3(X)) - Y_4(X)\|_\infty && \text{(By triangle inequality)} \\
&\leq L_4\|\widehat{Y}_3(X) - Y_3(X)\|_\infty + \epsilon_4 && \text{(By Lipschitzness on } \mathcal{K}_3 \text{ and the choice of } E_4) \\
&\leq L_4 \cdot \frac{3\epsilon_{\text{emu-pt}}}{4L_4} + \frac{\epsilon_{\text{emu-pt}}}{4} && \text{(By Equation (116) and } \epsilon_4 = \epsilon_{\text{emu-pt}}/4) \\
&= \epsilon_{\text{emu-pt}}.
\end{aligned}
$$

Since slicing the first $n$ columns cannot increase the entrywise $\ell_\infty$ norm, we obtain

$$
\sup_{X \in \mathcal{C}} \|\widetilde{T}_{\theta^\star, P_f}(X) - \widetilde{A}_{\phi_f}(X)\|_\infty \leq \epsilon_{\text{emu-pt}}. \tag{117}
$$

We now convert this pointwise $\ell_\infty$ emulator error into the $d_p$ metric:

$$
\begin{aligned}
d_p(\widetilde{T}_{\theta^\star, P_f}, \widetilde{A}_{\phi_f}) &= \left(\int_{\mathcal{C}} \|\widetilde{T}_{\theta^\star, P_f}(X) - \widetilde{A}_{\phi_f}(X)\|_p^p \, dX\right)^{1/p} && (118) \\
&\leq \left(\int_{\mathcal{C}} (N^{1/p}\|\widetilde{T}_{\theta^\star, P_f}(X) - \widetilde{A}_{\phi_f}(X)\|_\infty)^p \, dX\right)^{1/p} && \text{(By } \|A\|_p \leq N^{1/p}\|A\|_\infty) \\
&\leq |\mathcal{C}|^{1/p} N^{1/p} \epsilon_{\text{emu-pt}} && \text{(By (117))} \\
&= \epsilon_{\text{emu}}. && \text{(By the definition of } \epsilon_{\text{emu-pt}})
\end{aligned}
$$

By triangle inequality,

$$
\begin{aligned}
d_p(\widetilde{T}_{\theta^\star, P_f}, f) &\leq d_p(\widetilde{T}_{\theta^\star, P_f}, \widetilde{A}_{\phi_f}) + d_p(\widetilde{A}_{\phi_f}, f) \\
&\leq \epsilon_{\text{emu}} + \epsilon_{\text{sur}} && \text{(By Equations (110) and (118))} \\
&\leq \epsilon. && \text{(By the choice of } \epsilon_{\text{sur}} \text{ and } \epsilon_{\text{emu}})
\end{aligned}
$$

Since $f \in \mathcal{F}_L$ was arbitrary, the same fixed emulator $T_{\theta^\star}$ works for every target function through its corresponding prompt $P_f$. By Lemma E.7, the emulator weights and prompt shape depend only on

$$
\mathcal{C}, L, p, d, n, \epsilon,
$$

not on $f$ or $X$. This completes the proof. $\qquad \square$

### E.8 Proof of Corollary 3.2.1

We first need a helper lemma, a pointwise version of Lemma E.7.

**Lemma E.8** (Uniform Bounded Pointwise Attention-Only Surrogate Template). *Let $1 \leq p < \infty$ and $L > 0$. Let $\mathcal{C} = [-b, b]^{d \times n} \subset \mathbb{R}^{d \times n}$, and let $\mathcal{F}_L := \{f : \mathcal{C} \to \mathcal{C} \mid \mathrm{Lip}(f) \leq L\}$. Fix $\epsilon_{\mathrm{sur}} > 0$. Then there exists a fixed four-layer attention-only surrogate template*

$$A_\phi := \mathrm{Attn}_{4,\phi_4} \circ \mathrm{Attn}_{3,\phi_3} \circ \mathrm{Attn}_{2,\phi_2} \circ \mathrm{Attn}_{1,\phi_1},$$

*with fixed layer dimensions, fixed sequence length $n + 1$, fixed head numbers, fixed head dimensions, fixed inverse temperatures, and fixed prompt shape such that, for every $f \in \mathcal{F}_L$, there exists $\phi_f$ satisfying*

$$\sup_{X \in \mathcal{C}} \|A_{\phi_f}(X_a)_{:,1:n} - f(X)\|_p \leq \epsilon_{\mathrm{sur}}.$$

*Moreover, the encoded prompt blocks $W_{\phi_f,1}, \ldots, W_{\phi_f,4}$ from Definition 3.5, and all intermediate quantities required by Lemmas 3.4 and 3.5, are uniformly bounded over $f \in \mathcal{F}_L$ and $X \in \mathcal{C}$. All constants in these bounds, as well as the template dimensions and prompt shape, depend only on $\mathcal{C}, L, p, d, n, \epsilon_{\mathrm{sur}}$, not on $f$ or $X$.*

*Proof of Lemma E.8.* The proof follows the proof of Lemma E.7. The only change is the approximation guarantee used for the attention-only surrogate.

By (Hu et al., 2026a, Theorem G.1 and its proof), the four-layer attention-only surrogate can be chosen to satisfy the pointwise bound

$$\sup_{X \in \mathcal{C}} \|A_{\phi_f}(X_a)_{:,1:n} - f(X)\|_p \leq \epsilon_{\mathrm{sur}}.$$

Also, by the same uniformity argument as in Lemma E.7, the template and bounds can be chosen uniformly over $f \in \mathcal{F}_L$. Thus the surrogate dimensions, head numbers, head dimensions, inverse temperatures, and prompt shape are fixed for the whole class $\mathcal{F}_L$. All other bounds' and constants' properties follow straightforwardly. $\square$

Now we are ready for the main proof.

**Corollary E.2.1** (Corollary 3.2.1 Restated: Point-Wise In-Context Universal Approximation). *Under the same fixed architecture $T_{\theta^\star}$, function class $\mathcal{F}_L$, and boundedness conditions as in Theorem 3.2, let $P_f$ be the prompt encoding constructed from $f$ in Theorem 3.2. Then, for any $\epsilon > 0$, and every $f \in \mathcal{F}_L$, we have*

$$\sup_{X \in \mathcal{C}} \|T_{\theta^\star}(\begin{bmatrix} X_a \\ P_f \end{bmatrix})_{:,1:n} - f(X)\|_p \leq \epsilon.$$

*Proof.* Let $N := dn$. Choose

$$\epsilon_{\mathrm{sur}} := \frac{\epsilon}{2}, \quad \epsilon_{\mathrm{emu\text{-}pt}} := \frac{\epsilon}{2N^{1/p}}.$$

**Step 1: Use Pointwise Surrogate and Emulate It In-Context.** By Lemma E.8, there exists a fixed four-layer attention-only surrogate template $A_\phi$ such that, for every $f \in \mathcal{F}_L$, there exists $\phi_f$ satisfying

$$\sup_{X \in \mathcal{C}} \|A_{\phi_f}(X_a)_{:,1:n} - f(X)\|_p \leq \epsilon_{\mathrm{sur}}. \tag{119}$$

Define

$$\widetilde{A}_{\phi_f}(X) := A_{\phi_f}(X_a)_{:,1:n}.$$

Encode $\phi_f$ into

$$P_f := [W_{\phi_f,1}; W_{\phi_f,2}; W_{\phi_f,3}; W_{\phi_f,4}]$$

as in Definition 3.5.

The uniform boundedness part of Lemma E.8 gives the common bounds needed by Lemmas 3.4 and 3.5. Hence the pointwise emulator estimate from the proof of Theorem 3.2 applies with tolerance $\epsilon_{\text{emu-pt}}$. Therefore, there exists a fixed eight-layer softmax-attention emulator $T_{\theta^\star} = \text{Attn}_8$ such that, for every $f \in \mathcal{F}_L$,

$$\sup_{X \in \mathcal{C}} \|T_{\theta^\star}(\begin{bmatrix} X_a \\ P_f \end{bmatrix})_{:,1:n} - \widetilde{A}_{\phi_f}(X)\|_\infty \leq \epsilon_{\text{emu-pt}}. \tag{120}$$

**Step 2: Combine Errors.** Fix any $f \in \mathcal{F}_L$ and $X \in \mathcal{C}$. Then

$$\|T_{\theta^\star}(\begin{bmatrix} X_a \\ P_f \end{bmatrix})_{:,1:n} - f(X)\|_p$$

$$\leq \|T_{\theta^\star}(\begin{bmatrix} X_a \\ P_f \end{bmatrix})_{:,1:n} - \widetilde{A}_{\phi_f}(X)\|_p + \|\widetilde{A}_{\phi_f}(X) - f(X)\|_p \qquad \text{(By triangle inequality)}$$

$$\leq N^{1/p}\|T_{\theta^\star}(\begin{bmatrix} X_a \\ P_f \end{bmatrix})_{:,1:n} - \widetilde{A}_{\phi_f}(X)\|_\infty + \|\widetilde{A}_{\phi_f}(X) - f(X)\|_p \qquad \text{(By } \|M\|_p \leq N^{1/p}\|M\|_\infty\text{)}$$

$$\leq N^{1/p}\epsilon_{\text{emu-pt}} + \epsilon_{\text{sur}} \qquad \text{(By (120) and (119))}$$

$$= N^{1/p} \cdot \frac{\epsilon}{2N^{1/p}} + \frac{\epsilon}{2} \qquad \text{(By the choices of } \epsilon_{\text{emu-pt}} \text{ and } \epsilon_{\text{sur}}\text{)}$$

$$= \epsilon.$$

Since this bound holds for every $X \in \mathcal{C}$, we obtain

$$\sup_{X \in \mathcal{C}} \|T_{\theta^\star}(\begin{bmatrix} X_a \\ P_f \end{bmatrix})_{:,1:n} - f(X)\|_p \leq \epsilon.$$

Since $f \in \mathcal{F}_L$ was arbitrary, the same fixed emulator $T_{\theta^\star}$ works for every target function through its corresponding prompt $P_f$. The dependence of $T_{\theta^\star}$ and the prompt shape follows from Lemma E.8 and Theorem 3.2: they depend only on $\mathcal{C}, L, p, d, n, \epsilon$, not on $f$ or $X$. This completes the proof. $\square$

### E.9 Proof of Theorem 4.1

We first introduce a helper lemma.

**Lemma E.9** (Pointwise In-Context Universal Approximation with Flow-Through). *Let $\mathcal{C} = [-b, b]^{d \times n} \subset \mathbb{R}^{d \times n}$, and let $\mathcal{F}_L := \{f : \mathcal{C} \to \mathcal{C} \mid \text{Lip}(f) \leq L\}$. Fix $d_Z \geq 0$ and $B_Z > 0$. Then, for any $\epsilon > 0$, there exists a fixed-weight Transformer $T_{\theta^\star}^{\text{flow}} = \text{Attn}_8$ such that the following holds. For every $f \in \mathcal{F}_L$, there exists a prompt $P_f$ such that, for every $X \in \mathcal{C}$ and every $Z \in \mathbb{R}^{d_Z \times (n+1)}$ satisfying $\|Z\|_\infty \leq B_Z$,*

$$\|T_{\theta^\star}^{\text{flow}}(\begin{bmatrix} X_a \\ P_f \\ Z \end{bmatrix}) - \begin{bmatrix} f(X) & 0_{d \times 1} \\ I_n & 0_{n \times 1} \\ 0_{1 \times n} & 1 \\ & Z \end{bmatrix}\|_p \leq \epsilon,$$

*where $[X_a; P_f]$ is the input format from Definition 3.5. Moreover, the prompt shape and a uniform bound on $\|P_f\|_\infty$ depend only on $\mathcal{C}, L, p, d, n, d_Z, B_Z, \epsilon$, not on $f$, $X$, or $Z$.*

*Proof of Lemma E.9.* Let

$$m_a := n + 1,$$

$$D_a := d + n + 1,$$

$$I_{m_a} := \begin{bmatrix} I_n & 0_{n \times 1} \\ 0_{1 \times n} & 1 \end{bmatrix},$$

$$N_{\text{out}} := (D_a + d_Z)m_a.$$

Choose $\epsilon_{\text{sur}}, \epsilon_{\text{emu}} > 0$ such that

$$\epsilon_{\text{sur}} + \epsilon_{\text{emu}} \leq \epsilon.$$

Define

$$\epsilon_{\mathrm{pt}} := N_{\mathrm{out}}^{-1/p}\epsilon_{\mathrm{emu}},$$
$$\epsilon_{\mathrm{safe}} := \min\{1, \epsilon_{\mathrm{pt}}\}.$$

Define the lifted target class

$$\bar{\mathcal{F}}_L := \{\bar{f} : \mathcal{C} \to [-b, b]^{d \times m_a} \mid \bar{f}(X) = [f(X), 0_{d \times 1}], \ f \in \mathcal{F}_L\}.$$

For any $f \in \mathcal{F}_L$ and any $X, X' \in \mathcal{C}$, the lifted target satisfies

$$\|\bar{f}(X)\|_\infty \le b,$$
$$\|\bar{f}(X) - \bar{f}(X')\|_\infty = \|f(X) - f(X')\|_\infty$$
$$\le L\|X - X'\|_\infty.$$

Thus $\bar{\mathcal{F}}_L$ is uniformly bounded and $L$-Lipschitz.

Apply the pointwise selector-route construction used in Corollary 3.2.1 to the lifted class $\bar{\mathcal{F}}_L$. Equivalently, apply the proof of Lemma E.8 with output shape $d \times m_a$ instead of $d \times n$. The construction only changes the output coefficient blocks, and these coefficients remain bounded by $b$. Hence there exists a fixed four-layer attention-only surrogate template such that, for every $f \in \mathcal{F}_L$, there exists a parameter choice $\phi_f = (\phi_{f,1}, \phi_{f,2}, \phi_{f,3}, \phi_{f,4})$ with

$$A_{\phi_f} := A_4 \circ A_3 \circ A_2 \circ A_1, \quad A_i := \mathrm{Attn}_{\phi_{f,i}}, \quad i \in [4],$$

and

$$\sup_{X \in \mathcal{C}} \|A_{\phi_f}(X_a) - \bar{f}(X)\|_p \le \epsilon_{\mathrm{sur}}. \tag{121}$$

Moreover, the same uniform construction fixes the prompt shape and all bounds required by Lemmas 3.4 and 3.5. These fixed quantities depend only on $\mathcal{C}, L, p, d, n, \epsilon_{\mathrm{sur}}$, not on $f$ or $X$.

Fix $f \in \mathcal{F}_L$ and define its selector-route prompt as

$$P_f := [W_{\phi_f,1}; W_{\phi_f,2}; W_{\phi_f,3}; W_{\phi_f,4}].$$

Now fix $X \in \mathcal{C}$ and $Z \in \mathbb{R}^{d_Z \times m_a}$ with $\|Z\|_\infty \le B_Z$.

Define the exact flow states

$$Y_0^{\mathrm{flow}}(X, Z) := [X_a; W_{\phi_f,1}; W_{\phi_f,2}; W_{\phi_f,3}; W_{\phi_f,4}; Z],$$
$$Y_1^{\mathrm{flow}}(X, Z) := [A_1(X_a); W_{\phi_f,2}; W_{\phi_f,3}; W_{\phi_f,4}; Z],$$
$$Y_2^{\mathrm{flow}}(X, Z) := [A_2 \circ A_1(X_a); W_{\phi_f,3}; W_{\phi_f,4}; Z],$$
$$Y_3^{\mathrm{flow}}(X, Z) := [A_3 \circ A_2 \circ A_1(X_a); W_{\phi_f,4}; Z],$$
$$Y_4^{\mathrm{flow}}(X, Z) := [A_{\phi_f}(X_a); I_{m_a}; Z].$$

The first three transitions emulate $A_1, A_2, A_3$ and flow the later prompt blocks together with $Z$. For these transitions, use Lemma 3.5. For the fourth transition, use Lemma 3.5 to emulate $A_4$ and flow $Z$. In addition, add fixed identity-copy heads that read the $I_{m_a}$ subblock inside $W_{\phi_f,4}$ and write it to the output[10]. The row location of this $I_{m_a}$ subblock is fixed by the prompt format, so these heads do not depend on $f$, $X$, or $Z$. All added heads use disjoint output rows, so they do not interfere with the surrogate-emulation heads.

For $i \in [3]$, define the exact-state sets

$$\mathcal{S}_i := \{Y_i^{\mathrm{flow}}(X, Z) \mid f \in \mathcal{F}_L, \ X \in \mathcal{C}, \ \|Z\|_\infty \le B_Z\}.$$

---

[10]This is a harmless extension of Lemma 3.5: the same two-layer construction already copies the identity subblock internally to route the flow-through block. We add output heads with disjoint output rows to expose this copied identity block in the final output.

The uniform surrogate bounds, the uniform prompt bounds, and $\|Z\|_\infty \le B_Z$ imply that $\mathcal{S}_1, \mathcal{S}_2, \mathcal{S}_3$ are bounded. Define the safety regions

$$\mathcal{K}_i := \{Y \mid \inf_{S \in \mathcal{S}_i} \|Y - S\|_\infty \le \epsilon_{\text{safe}}\}, \quad i \in [3].$$

Each $\mathcal{K}_i$ is compact. After a fixed emulator block is chosen, it has a finite Lipschitz constant on the corresponding compact safety region.

We choose the four two-layer emulator blocks from back to front.

- **Fourth Layer.** Choose a two-layer emulator $E_4$ for the fourth transition with tolerance

$$\eta_4 := \frac{\epsilon_{\text{safe}}}{4}.$$

  Then, for every $f, X, Z$ as above,

$$\|E_4(Y_3^{\text{flow}}(X, Z)) - Y_4^{\text{flow}}(X, Z)\|_\infty \le \eta_4. \tag{122}$$

  Let $L_4 \ge 1$ be a Lipschitz constant of $E_4$ on $\mathcal{K}_3$.

- **Third Layer.** Choose a two-layer emulator $E_3$ for the third transition with tolerance

$$\eta_3 := \frac{\epsilon_{\text{safe}}}{4L_4}.$$

  Then, for every $f, X, Z$ as above,

$$\|E_3(Y_2^{\text{flow}}(X, Z)) - Y_3^{\text{flow}}(X, Z)\|_\infty \le \eta_3. \tag{123}$$

  Let $L_3 \ge 1$ be a Lipschitz constant of $E_3$ on $\mathcal{K}_2$.

- **Second Layer.** Choose a two-layer emulator $E_2$ for the second transition with tolerance

$$\eta_2 := \frac{\epsilon_{\text{safe}}}{4L_3L_4}.$$

  Then, for every $f, X, Z$ as above,

$$\|E_2(Y_1^{\text{flow}}(X, Z)) - Y_2^{\text{flow}}(X, Z)\|_\infty \le \eta_2. \tag{124}$$

  Let $L_2 \ge 1$ be a Lipschitz constant of $E_2$ on $\mathcal{K}_1$.

- **First Layer.** Choose a two-layer emulator $E_1$ for the first transition with tolerance

$$\eta_1 := \frac{\epsilon_{\text{safe}}}{4L_2L_3L_4}.$$

  Then, for every $f, X, Z$ as above,

$$\|E_1(Y_0^{\text{flow}}(X, Z)) - Y_1^{\text{flow}}(X, Z)\|_\infty \le \eta_1. \tag{125}$$

Stack the four emulator blocks as

$$T_{\theta^\star}^{\text{flow}} := E_4 \circ E_3 \circ E_2 \circ E_1.$$

This is an eight-layer attention emulator. Its weights depend only on the fixed dimensions, common bounds, $B_Z$, and the error budget.

We now track the stacked error. Write

$$Y_i := Y_i^{\text{flow}}(X, Z), \quad i = 0, 1, 2, 3, 4.$$

Define the actual emulator states

$$\widehat{Y}_0 := Y_0,$$
$$\widehat{Y}_1 := E_1(\widehat{Y}_0),$$
$$\widehat{Y}_2 := E_2(\widehat{Y}_1),$$
$$\widehat{Y}_3 := E_3(\widehat{Y}_2),$$
$$\widehat{Y}_4 := E_4(\widehat{Y}_3).$$

First,

$$\|\widehat{Y}_1 - Y_1\|_\infty \le \eta_1 \qquad \text{(By Equation (125))}$$
$$\le \epsilon_{\text{safe}}.$$

Hence $\widehat{Y}_1 \in \mathcal{K}_1$. Next,

$$
\begin{aligned}
\|\widehat{Y}_2 - Y_2\|_\infty &= \|E_2(\widehat{Y}_1) - Y_2\|_\infty \\
&\le \|E_2(\widehat{Y}_1) - E_2(Y_1)\|_\infty + \|E_2(Y_1) - Y_2\|_\infty && \text{(By triangle inequality)} \\
&\le L_2\|\widehat{Y}_1 - Y_1\|_\infty + \eta_2 && \text{(By Lipschitzness on } \mathcal{K}_1 \text{ and Equation (124))} \\
&\le L_2\eta_1 + \eta_2 \\
&= \frac{\epsilon_{\text{safe}}}{2L_3L_4} && \text{(By the definitions of } \eta_1, \eta_2) \\
&\le \epsilon_{\text{safe}}.
\end{aligned}
$$

Hence $\widehat{Y}_2 \in \mathcal{K}_2$. Similarly,

$$
\begin{aligned}
\|\widehat{Y}_3 - Y_3\|_\infty &= \|E_3(\widehat{Y}_2) - Y_3\|_\infty \\
&\le \|E_3(\widehat{Y}_2) - E_3(Y_2)\|_\infty + \|E_3(Y_2) - Y_3\|_\infty && \text{(By triangle inequality)} \\
&\le L_3\|\widehat{Y}_2 - Y_2\|_\infty + \eta_3 && \text{(By Lipschitzness on } \mathcal{K}_2 \text{ and Equation (123))} \\
&\le L_3 \cdot \frac{\epsilon_{\text{safe}}}{2L_3L_4} + \frac{\epsilon_{\text{safe}}}{4L_4} \\
&= \frac{3\epsilon_{\text{safe}}}{4L_4} \\
&\le \epsilon_{\text{safe}}.
\end{aligned}
$$

Hence $\widehat{Y}_3 \in \mathcal{K}_3$. Finally,

$$
\begin{aligned}
\|\widehat{Y}_4 - Y_4\|_\infty &= \|E_4(\widehat{Y}_3) - Y_4\|_\infty \\
&\le \|E_4(\widehat{Y}_3) - E_4(Y_3)\|_\infty + \|E_4(Y_3) - Y_4\|_\infty && \text{(By triangle inequality)} \\
&\le L_4\|\widehat{Y}_3 - Y_3\|_\infty + \eta_4 && \text{(By Lipschitzness on } \mathcal{K}_3 \text{ and Equation (122))} \\
&\le L_4 \cdot \frac{3\epsilon_{\text{safe}}}{4L_4} + \frac{\epsilon_{\text{safe}}}{4} \\
&= \epsilon_{\text{safe}}. && (126)
\end{aligned}
$$

Since

$$\widehat{Y}_4 = T_{\theta^\star}^{\text{flow}}\left(\begin{bmatrix} X_a \\ P_f \\ Z \end{bmatrix}\right), \quad Y_4 = \begin{bmatrix} A_{\phi_f}(X_a) \\ I_{m_a} \\ Z \end{bmatrix},$$

Equation (126) gives

$$\left\|T_{\theta^\star}^{\text{flow}}\left(\begin{bmatrix} X_a \\ P_f \\ Z \end{bmatrix}\right) - \begin{bmatrix} A_{\phi_f}(X_a) \\ I_{m_a} \\ Z \end{bmatrix}\right\|_\infty \le \epsilon_{\text{safe}} \le \epsilon_{\text{pt}}. \qquad (127)$$

Therefore,

$$
\left\| T_{\theta^\star}^{\text{flow}}\left(\begin{bmatrix} X_a \\ P_f \\ Z \end{bmatrix}\right) - \begin{bmatrix} A_{\phi_f}(X_a) \\ I_{m_a} \\ Z \end{bmatrix} \right\|_p
$$

$$
\leq N_{\text{out}}^{1/p} \left\| T_{\theta^\star}^{\text{flow}}\left(\begin{bmatrix} X_a \\ P_f \\ Z \end{bmatrix}\right) - \begin{bmatrix} A_{\phi_f}(X_a) \\ I_{m_a} \\ Z \end{bmatrix} \right\|_\infty \qquad \left(\text{By } \|M\|_p \leq N_{\text{out}}^{1/p}\|M\|_\infty\right)
$$

$$
\leq N_{\text{out}}^{1/p}\epsilon_{\text{pt}} \qquad\qquad\qquad \left(\text{By Equation (127)}\right)
$$

$$
= \epsilon_{\text{emu}}. \qquad\qquad\qquad \left(\text{By the definition of } \epsilon_{\text{pt}}\right)
$$

Also,

$$
\left\| \begin{bmatrix} A_{\phi_f}(X_a) \\ I_{m_a} \\ Z \end{bmatrix} - \begin{bmatrix} \bar{f}(X) \\ I_{m_a} \\ Z \end{bmatrix} \right\|_p
$$

$$
= \|A_{\phi_f}(X_a) - \bar{f}(X)\|_p \qquad\qquad \left(\text{The } I_{m_a} \text{ and } Z \text{ blocks cancel}\right)
$$

$$
\leq \epsilon_{\text{sur}}. \qquad\qquad\qquad \left(\text{By Equation (121)}\right)
$$

Combining the two bounds gives

$$
\left\| T_{\theta^\star}^{\text{flow}}\left(\begin{bmatrix} X_a \\ P_f \\ Z \end{bmatrix}\right) - \begin{bmatrix} \bar{f}(X) \\ I_{m_a} \\ Z \end{bmatrix} \right\|_p
$$

$$
\leq \epsilon_{\text{emu}} + \epsilon_{\text{sur}} \qquad\qquad \left(\text{By triangle inequality}\right)
$$

$$
\leq \epsilon.
$$

Finally,

$$
\begin{bmatrix} \bar{f}(X) \\ I_{m_a} \\ Z \end{bmatrix} = \begin{bmatrix} f(X) & 0_{d\times 1} \\ I_n & 0_{n\times 1} \\ 0_{1\times n} & 1 \\ & Z \end{bmatrix}.
$$

Since $f$, $X$, and $Z$ are arbitrary, this proves the uniform claim. The construction uses only the fixed quantities $\mathcal{C}, L, p, d, n, d_Z, B_Z, \epsilon$. This completes the proof. $\qquad\square$

Now we are ready for the main proof.

**Theorem E.3** (Theorem 4.1 Restated: One-Pass In-Context Composition). *Let $\mathcal{C} = [-b, b]^{d\times n} \subset \mathbb{R}^{d\times n}$, and let $\mathcal{F}_L := \{f : \mathcal{C} \to \mathcal{C} \mid \text{Lip}(f) \leq L\}$. Fix $m \in \mathbb{N}^+$. Then, for any $\epsilon > 0$, there exists a fixed-weight Transformer $T_{\theta^\star} = \text{Attn}_{8m}$ such that the following holds. For every $f_1, \ldots, f_m \in \mathcal{F}_L$, there exist prompts $P_{f_1}, \ldots, P_{f_m}$ in the format of Definition 3.5. Define the target composition function and the composite prompt as $g := f_m \circ f_{m-1} \circ \cdots \circ f_1$ and $P_g := [P_{f_1}; P_{f_2}; \cdots ; P_{f_m}]$. Then, the induced map $\widetilde{T}_{\theta^\star, P_g}(X) := T_{\theta^\star}([\begin{smallmatrix} X_a \\ P_g \end{smallmatrix}])_{1:d, 1:n}$ satisfies*

$$
\mathrm{d}_p(\widetilde{T}_{\theta^\star, P_g}, g) \leq \epsilon.
$$

*The emulator weights $\theta^\star$ and the composite prompt shape depend only on $\mathcal{C}, L, p, d, n, m, \epsilon$, not on $f_1, \ldots, f_m$ or $X$. Only the prompt entries in $P_g$ depend on the subroutines $f_1, \ldots, f_m$.*

*Proof of Theorem 4.1.* Let $N := dn$. For any $U \in \mathcal{C}$, write $U_a$ for the augmented input in Definition 3.5 with $U$ in place of $X$.

**Step 1: Choose the Final Pointwise Error Budget.** We first choose a pointwise $\ell_\infty$ error that implies the desired $\mathrm{d}_p$ error:

$$
\epsilon_{\text{pt}} := |\mathcal{C}|^{-1/p} N^{-1/p}\epsilon.
$$

Indeed, if the final output has pointwise $\ell_\infty$ error at most $\epsilon_{\text{pt}}$, then its $\mathrm{d}_p$ error is at most $\epsilon$.

**Step 2: Construct the Stage Emulators from Back to Front.** We choose the emulators in reverse order. This accounts for how later stages amplify errors made by earlier stages.

- **Last Stage.** Define

$$\eta_m := \epsilon_{\mathrm{pt}}/m.$$

Apply Lemma E.9 with $d_Z = 0$, $B_Z = 1$, and tolerance $\eta_m$. This gives a fixed eight-layer emulator $E_m$ such that, for every $f_m \in \mathcal{F}_L$, there exists a prompt $P_{f_m}$ satisfying, for every $U \in \mathcal{C}$,

$$\|E_m([U_a; P_{f_m}]) - f_m(U)_a\|_p \le \eta_m.$$

Thus, for every $U \in \mathcal{C}$,

$$\|E_m([U_a; P_{f_m}]) - f_m(U)_a\|_\infty \le \eta_m. \qquad\qquad (\text{By } \|A\|_\infty \le \|A\|_p)$$

If $m \ge 2$, define

$$\begin{aligned}
\mathcal{S}_{m-1} &:= \{[U_a; P_{f_m}] : U \in \mathcal{C}, \ f_m \in \mathcal{F}_L\}, \\
\mathcal{K}_{m-1} &:= \{Y : \inf_{S \in \mathcal{S}_{m-1}} \|Y - S\|_\infty \le \epsilon_{\mathrm{pt}}\}.
\end{aligned}$$

By Lemma E.9, the prompts $P_{f_m}$ have fixed shape and a uniform entrywise bound. Hence $\mathcal{K}_{m-1}$ is compact. After fixing $E_m$, choose a Lipschitz constant $L_m \ge 1$ of $E_m$ on $\mathcal{K}_{m-1}$.

- **Earlier Stages.** For $i = m - 1, \ldots, 1$, suppose that we have already fixed

$$E_{i+1}, \ldots, E_m, \quad L_{i+1}, \ldots, L_m,$$

and the prompt maps

$$f_j \mapsto P_{f_j}, \quad j = i + 1, \ldots, m.$$

Define the later-stage amplification factor

$$R_i := \prod_{j=i+1}^{m} L_j,$$

and set

$$\eta_i := \frac{\epsilon_{\mathrm{pt}}}{mR_i}.$$

For any $f_{i+1}, \ldots, f_m \in \mathcal{F}_L$, define the remaining-prompt block

$$Z_i := [P_{f_{i+1}}; \cdots ; P_{f_m}].$$

The prompt bounds from the later stages give a constant $B_{Z,i} < \infty$ such that

$$\|Z_i\|_\infty \le B_{Z,i}.$$

Apply Lemma E.9 with flow-through dimension equal to the row dimension of $Z_i$, bound $B_{Z,i}$, and tolerance $\eta_i$. This gives a fixed eight-layer emulator $E_i$ such that, for every $f_i \in \mathcal{F}_L$, there exists a prompt $P_{f_i}$ satisfying, for every $U \in \mathcal{C}$ and every such $Z_i$,

$$\left\| E_i\left( \begin{bmatrix} U_a \\ P_{f_i} \\ Z_i \end{bmatrix} \right) - \begin{bmatrix} f_i(U)_a \\ Z_i \end{bmatrix} \right\|_p \le \eta_i.$$

Thus,

$$\left\| E_i\left(\begin{bmatrix} U_a \\ P_{f_i} \\ Z_i \end{bmatrix}\right) - \begin{bmatrix} f_i(U)_a \\ Z_i \end{bmatrix} \right\|_\infty \le \eta_i. \qquad (\text{By } \|A\|_\infty \le \|A\|_p)$$

If $i \ge 2$, define

$$\mathcal{S}_{i-1} := \{[U_a; P_{f_i}; \cdots; P_{f_m}] : U \in \mathcal{C}, \ f_i, \ldots, f_m \in \mathcal{F}_L\},$$
$$\mathcal{K}_{i-1} := \{Y : \inf_{S \in \mathcal{S}_{i-1}} \|Y - S\|_\infty \le \epsilon_{\mathrm{pt}}\}.$$

The distance-to-a-set map is continuous, so $K_{i-1}$ is closed. The uniform bounds imply that $K_{i-1}$ is bounded. Since the ambient space is finite-dimensional, $K_{i-1}$ is compact. After fixing $E_i$, choose a Lipschitz constant $L_i \ge 1$ of $E_i$ on $\mathcal{K}_{i-1}$.[11]

Repeating this construction gives fixed emulators

$$E_1, \ldots, E_m$$

and prompt maps

$$f_i \mapsto P_{f_i}, \quad i \in [m].$$

Define the full one-pass emulator

$$T_{\theta^\star} := E_m \circ E_{m-1} \circ \cdots \circ E_1.$$

Then $T_{\theta^\star} = \mathrm{Attn}_{8m}$.

**Step 3: Track the Exact and Approximate Trajectories.** Fix arbitrary $f_1, \ldots, f_m \in \mathcal{F}_L$ and $X \in \mathcal{C}$. Define the exact composition trajectory

$$c_0(X) := X,$$
$$c_i(X) := f_i(c_{i-1}(X)), \quad i \in [m].$$

Since each $f_i$ maps $\mathcal{C}$ to $\mathcal{C}$, we have

$$c_i(X) \in \mathcal{C}, \quad i \in [m].$$

Define

$$g := c_m = f_m \circ f_{m-1} \circ \cdots \circ f_1,$$
$$P_g := [P_{f_1}; P_{f_2}; \cdots; P_{f_m}].$$

The exact state after stage $i$ contains the exact current data and the remaining prompts:

$$Y_i(X) := [c_i(X)_a; P_{f_{i+1}}; \cdots; P_{f_m}], \quad i = 0, 1, \ldots, m,$$

where no prompt block appears when $i = m$. The actual state comes from running the fixed emulators:

$$\widehat{Y}_0(X) := Y_0(X),$$
$$\widehat{Y}_i(X) := E_i(\widehat{Y}_{i-1}(X)), \quad i \in [m].$$

Define the stage error

$$e_i(X) := \|\widehat{Y}_i(X) - Y_i(X)\|_\infty.$$

---

[11]Since $E_i$ is a fixed finite softmax-attention map, it is smooth. Thus it has a finite Lipschitz constant on the compact set $K_{i-1}$. Choose such a constant and enlarge it if necessary so that $L_i \ge 1$.

**Step 4: Prove the Stage Error Bound by Induction.** We prove that the final state satisfies

$$e_m(X) \leq \epsilon_{\text{pt}}.$$

- **Single-Stage Case.** If $m = 1$, then

$$\begin{aligned} e_1(X) &= \|\widehat{Y}_1(X) - Y_1(X)\|_\infty \\ &\leq \eta_1 && \left(\text{By the choice of } E_1\right) \\ &= \epsilon_{\text{pt}}. && \left(\text{By } \eta_1 = \epsilon_{\text{pt}}/m \text{ and } m = 1\right) \end{aligned}$$

Hence the claim holds.

- **Base Case.** Now assume $m \geq 2$. For $i = 1, \ldots, m-1$, recall

$$R_i := \prod_{j=i+1}^{m} L_j.$$

The quantity $R_i$ records how much the later stages can amplify an error made at stage $i$. We first prove that, for every $i = 1, \ldots, m-1$,

$$e_i(X) \leq \frac{i\epsilon_{\text{pt}}}{mR_i}.$$

For $i = 1$,

$$\begin{aligned} e_1(X) &= \|\widehat{Y}_1(X) - Y_1(X)\|_\infty \\ &\leq \eta_1 && \left(\text{By the choice of } E_1\right) \\ &= \frac{\epsilon_{\text{pt}}}{mR_1}. && \left(\text{By the definition of } \eta_1\right) \end{aligned}$$

Thus the claim holds for $i = 1$. Moreover,

$$e_1(X) \leq \epsilon_{\text{pt}},$$

because $R_1 \geq 1$. Hence $\widehat{Y}_1(X) \in \mathcal{K}_1$.

- **Induction Step.** Suppose the claim holds for $i - 1$, where $2 \leq i \leq m - 1$. Then

$$e_{i-1}(X) \leq \epsilon_{\text{pt}},$$

because $i - 1 \leq m - 1$ and $R_{i-1} \geq 1$. Hence $\widehat{Y}_{i-1}(X) \in \mathcal{K}_{i-1}$. Therefore,

$$\begin{aligned} e_i(X) &= \|E_i(\widehat{Y}_{i-1}(X)) - Y_i(X)\|_\infty \\ &\leq \|E_i(\widehat{Y}_{i-1}(X)) - E_i(Y_{i-1}(X))\|_\infty + \|E_i(Y_{i-1}(X)) - Y_i(X)\|_\infty && \left(\text{By triangle inequality}\right) \\ &\leq L_i e_{i-1}(X) + \eta_i && \left(\text{By Lipschitzness on } \mathcal{K}_{i-1} \text{ and the choice of } E_i\right) \\ &\leq L_i \cdot \frac{(i-1)\epsilon_{\text{pt}}}{mR_{i-1}} + \eta_i && \left(\text{By the induction hypothesis}\right) \\ &= L_i \cdot \frac{(i-1)\epsilon_{\text{pt}}}{mR_{i-1}} + \frac{\epsilon_{\text{pt}}}{mR_i} && \left(\text{By the definition of } \eta_i\right) \\ &= L_i \cdot \frac{(i-1)\epsilon_{\text{pt}}}{mL_iR_i} + \frac{\epsilon_{\text{pt}}}{mR_i} && \left(\text{By } R_{i-1} = L_iR_i\right) \\ &= \frac{(i-1)\epsilon_{\text{pt}}}{mR_i} + \frac{\epsilon_{\text{pt}}}{mR_i} \\ &= \frac{i\epsilon_{\text{pt}}}{mR_i}. \end{aligned}$$

Thus the claim holds for $i$. Also,

$$e_i(X) \leq \epsilon_{\text{pt}},$$

because $i \leq m - 1$ and $R_i \geq 1$. Hence $\widehat{Y}_i(X) \in \mathcal{K}_i$, and the induction continues.

- **Last Stage.** It remains to handle the last stage. Since $\widehat{Y}_{m-1}(X) \in \mathcal{K}_{m-1}$, we have

$$
\begin{aligned}
e_m(X) &= \|E_m(\widehat{Y}_{m-1}(X)) - Y_m(X)\|_\infty \\
&\leq \|E_m(\widehat{Y}_{m-1}(X)) - E_m(Y_{m-1}(X))\|_\infty + \|E_m(Y_{m-1}(X)) - Y_m(X)\|_\infty && \text{(By triangle inequality)} \\
&\leq L_m e_{m-1}(X) + \eta_m && \text{(By Lipschitzness on } \mathcal{K}_{m-1} \text{ and the choice of } E_m) \\
&\leq L_m \cdot \frac{(m-1)\epsilon_{\text{pt}}}{mL_m} + \frac{\epsilon_{\text{pt}}}{m} && \text{(By the bound for } e_{m-1} \text{ and } \eta_m = \epsilon_{\text{pt}}/m) \\
&= \epsilon_{\text{pt}}.
\end{aligned}
$$

Since $Y_m(X) = c_m(X)_a = g(X)_a$, slicing the first $d$ rows and first $n$ columns gives

$$
\|\widetilde{T}_{\theta^\star, P_g}(X) - g(X)\|_\infty \leq \epsilon_{\text{pt}}.
$$

**Step 5: Convert the Pointwise Bound to the** $\text{d}_p$ **Metric.** We now integrate the pointwise error:

$$
\begin{aligned}
\text{d}_p(\widetilde{T}_{\theta^\star, P_g}, g) &= \left(\int_{\mathcal{C}} \|\widetilde{T}_{\theta^\star, P_g}(X) - g(X)\|_p^p \, \text{d}X\right)^{1/p} \\
&\leq \left(\int_{\mathcal{C}} (N^{1/p}\|\widetilde{T}_{\theta^\star, P_g}(X) - g(X)\|_\infty)^p \, \text{d}X\right)^{1/p} && \left(\text{By } \|A\|_p \leq N^{1/p}\|A\|_\infty\right) \\
&\leq |\mathcal{C}|^{1/p} N^{1/p} \epsilon_{\text{pt}} \\
&= \epsilon. && \left(\text{By the definition of } \epsilon_{\text{pt}}\right)
\end{aligned}
$$

**Step 6: Check Fixedness.** All emulators $E_1, \ldots, E_m$ become fixed after we fix

$$
\mathcal{C}, L, p, d, n, m, \epsilon.
$$

Hence $T_{\theta^\star}$ also becomes fixed after these quantities are fixed. It does not depend on $f_1, \ldots, f_m$ or $X$. Only the prompt entries in $P_g$ depend on the subroutines. This completes the proof. $\qquad\square$

### E.10 Proof of Theorem 4.2

**Theorem E.4** (Theorem 4.2 Restated: Re-Prompting In-Context Composition). *Let* $\mathcal{C} = [-b, b]^{d \times n} \subset \mathbb{R}^{d \times n}$, *and let* $\mathcal{F}_L := \{f : \mathcal{C} \to \mathcal{C} \mid \text{Lip}(f) \leq L\}$. *Fix* $m \in \mathbb{N}^+$. *Then, for any* $\epsilon > 0$, *there exists a fixed-weight Transformer* $T_{\theta^\star} = \text{Attn}_8$ *such that the following holds. For every* $f_1, \ldots, f_m \in \mathcal{F}_L$, *there exist prompts* $P_{f_1}, \ldots, P_{f_m}$ *in the format of* Definition 3.5. *For every* $X \in \mathcal{C}$, *define For every* $X \in \mathcal{C}$, *define the exact trajectory* $c_i$ *and the re-prompted trajectory* $\widehat{c}_i$ *recursively by:*

$$
c_0(X) := X, \quad c_i(X) := f_i(c_{i-1}(X)), \quad i \in [m],
$$

$$
\widehat{c}_0(X) := X, \quad \widehat{c}_i(X) := T_{\theta^\star}\left(\begin{bmatrix} \widehat{c}_{i-1}(X) & 0_{d \times 1} \\ I_n & 0_{n \times 1} \\ 0_{1 \times n} & 1 \\ P_{f_i} \end{bmatrix}\right)_{:,1:n}, \quad i \in [m].
$$

*Then, for every* $i \in [m]$,

$$
d_p(\widehat{c}_i, c_i) \leq \epsilon.
$$

*The emulator weights* $\theta^\star$ *and the prompt shape depend only on* $\mathcal{C}, L, p, d, n, m, \epsilon$, *not on* $f_1, \ldots, f_m$ *or* $X$. *Only the prompt entries* $\{P_{f_i}\}_{i \in [m]}$ *depend on the subroutines* $f_1, \ldots, f_m$.

*Proof of Theorem 4.2.* Let $N := dn$. For any $U \in \mathbb{R}^{d \times n}$, write $U_a$ for the augmented input in Definition 3.5 with $U$ in place of $X$.

**Step 1: Choose Error Budget and Safety Domain.** We first choose a pointwise $\ell_\infty$ error that implies the desired $\mathrm{d}_p$ error:

$$\epsilon_{\mathrm{pt}} := |\mathcal{C}|^{-1/p} N^{-1/p} \epsilon.$$

The exact states stay in $\mathcal{C}$, because each $f_i$ maps $\mathcal{C}$ to $\mathcal{C}$. The approximate states may leave $\mathcal{C}$ by a small amount. Thus, we introduce the auxiliary box

$$\mathcal{C}_+ := [-b - \epsilon_{\mathrm{pt}}, b + \epsilon_{\mathrm{pt}}]^{d \times n}.$$

Let $\Pi_{\mathcal{C}} : \mathcal{C}_+ \to \mathcal{C}$ denote the entrywise clipping map onto $\mathcal{C}$. Then, for every $U, V \in \mathcal{C}_+$,

$$\|\Pi_{\mathcal{C}}(U) - \Pi_{\mathcal{C}}(V)\|_\infty \leq \|U - V\|_\infty.$$

Also, for every $V \in \mathcal{C}$,

$$\Pi_{\mathcal{C}}(V) = V.$$

**Step 2: Extend the Subroutines to Safety Domain.** For each $f_i \in \mathcal{F}_L$, define

$$\bar{f}_i(U) := f_i(\Pi_{\mathcal{C}}(U)), \qquad U \in \mathcal{C}_+.$$

Then $\bar{f}_i : \mathcal{C}_+ \to \mathcal{C} \subset \mathcal{C}_+$. Moreover, for every $U, V \in \mathcal{C}_+$,

$$\begin{aligned}
\|\bar{f}_i(U) - \bar{f}_i(V)\|_\infty &= \|f_i(\Pi_{\mathcal{C}}(U)) - f_i(\Pi_{\mathcal{C}}(V))\|_\infty \\
&\leq L\|\Pi_{\mathcal{C}}(U) - \Pi_{\mathcal{C}}(V)\|_\infty && (\text{By } \mathrm{Lip}(f_i) \leq L) \\
&\leq L\|U - V\|_\infty. && (\text{By nonexpansiveness of } \Pi_{\mathcal{C}})
\end{aligned}$$

Thus each $\bar{f}_i$ belongs to the $L$-Lipschitz class on $\mathcal{C}_+$.

**Step 3: Choose One Fixed Re-Prompting Emulator.** Define

$$S_m := \sum_{k=0}^{m-1} L^k.$$

Choose the one-step tolerance

$$\eta := \epsilon_{\mathrm{pt}}/S_m.$$

Apply Corollary 3.2.1 on the domain $\mathcal{C}_+$ with tolerance $\eta$. This gives one fixed eight-layer Transformer $T_{\theta^\star} = \mathrm{Attn}_8$ such that, for every $\bar{f}_i$, there exists a prompt $P_{f_i}$ satisfying, for every $U \in \mathcal{C}_+$,

$$\left\|T_{\theta^\star}\left(\begin{bmatrix} U_a \\ P_{f_i} \end{bmatrix}\right)_{:,1:n} - \bar{f}_i(U)\right\|_p \leq \eta. \tag{128}$$

Thus, for every $U \in \mathcal{C}_+$,

$$\left\|T_{\theta^\star}\left(\begin{bmatrix} U_a \\ P_{f_i} \end{bmatrix}\right)_{:,1:n} - \bar{f}_i(U)\right\|_\infty \leq \eta. \qquad (\text{By } \|A\|_\infty \leq \|A\|_p)$$

**Step 4: Track Re-Prompting Error.** Fix arbitrary $f_1, \ldots, f_m \in \mathcal{F}_L$ and $X \in \mathcal{C}$. Define

$$\begin{aligned}
c_0(X) &:= X, \\
c_i(X) &:= f_i(c_{i-1}(X)), \qquad i \in [m],
\end{aligned}$$

and

$$\widehat{c}_0(X) := X,$$

$$\widehat{c}_i(X) := T_{\theta^\star}\left(\begin{bmatrix} \widehat{c}_{i-1}(X)_a \\ P_{f_i} \end{bmatrix}\right)_{:,1:n}, \qquad i \in [m].$$

Define

$$e_i(X) := \|\widehat{c}_i(X) - c_i(X)\|_\infty, \qquad i = 0, 1, \ldots, m.$$

For $i \in [m]$, define

$$S_i := \sum_{k=0}^{i-1} L^k,$$

and set $S_0 := 0$. Then

$$S_i = 1 + LS_{i-1}, \qquad i \in [m].$$

We prove that, for every $i = 0, 1, \ldots, m$,

$$e_i(X) \leq \eta S_i, \qquad \widehat{c}_i(X) \in \mathcal{C}_+.$$

- **Base Case.** For $i = 0$,

$$e_0(X) = \|\widehat{c}_0(X) - c_0(X)\|_\infty = \eta S_0 = 0.$$

  Also,

$$\widehat{c}_0(X) = X \in \mathcal{C} \subset \mathcal{C}_+.$$

- **Induction Step.** Suppose the claim holds for $i - 1$. Then $\widehat{c}_{i-1}(X) \in \mathcal{C}_+$, so the one-step emulator guarantee applies to $\widehat{c}_{i-1}(X)$. Therefore,

$$
\begin{aligned}
e_i(X) &= \|\widehat{c}_i(X) - c_i(X)\|_\infty \\
&= \left\|T_{\theta^\star}\left(\begin{bmatrix} \widehat{c}_{i-1}(X)_a \\ P_{f_i} \end{bmatrix}\right)_{:,1:n} - f_i(c_{i-1}(X))\right\|_\infty \\
&\leq \left\|T_{\theta^\star}\left(\begin{bmatrix} \widehat{c}_{i-1}(X)_a \\ P_{f_i} \end{bmatrix}\right)_{:,1:n} - \bar{f}_i(\widehat{c}_{i-1}(X))\right\|_\infty + \|\bar{f}_i(\widehat{c}_{i-1}(X)) - f_i(c_{i-1}(X))\|_\infty && \text{(By triangle inequality)} \\
&\leq \eta + \|f_i(\Pi_\mathcal{C}(\widehat{c}_{i-1}(X))) - f_i(c_{i-1}(X))\|_\infty && \text{(By Equation (128) and the definition of } \bar{f}_i) \\
&\leq \eta + L\|\Pi_\mathcal{C}(\widehat{c}_{i-1}(X)) - c_{i-1}(X)\|_\infty && \text{(By } \mathrm{Lip}(f_i) \leq L) \\
&= \eta + L\|\Pi_\mathcal{C}(\widehat{c}_{i-1}(X)) - \Pi_\mathcal{C}(c_{i-1}(X))\|_\infty && \text{(By } c_{i-1}(X) \in \mathcal{C}) \\
&\leq \eta + L\|\widehat{c}_{i-1}(X) - c_{i-1}(X)\|_\infty && \text{(By nonexpansiveness of } \Pi_\mathcal{C}) \\
&= \eta + Le_{i-1}(X) \\
&\leq \eta + L\eta S_{i-1} && \text{(By the induction hypothesis)} \\
&= \eta S_i. && \text{(By } S_i = 1 + LS_{i-1})
\end{aligned}
$$

Since $c_i(X) \in \mathcal{C}$ and

$$
\begin{aligned}
e_i(X) &\leq \eta S_i \\
&\leq \eta S_m \\
&= \epsilon_{\mathrm{pt}}, && (\text{By } \eta = \epsilon_{\mathrm{pt}}/S_m)
\end{aligned}
$$

we have

$$
\begin{aligned}
\|\widehat{c}_i(X)\|_\infty &\leq \|c_i(X)\|_\infty + \|\widehat{c}_i(X) - c_i(X)\|_\infty && (\text{By triangle inequality}) \\
&\leq b + \epsilon_{\mathrm{pt}}. && (\text{By } c_i(X) \in \mathcal{C} \text{ and } e_i(X) \leq \epsilon_{\mathrm{pt}})
\end{aligned}
$$

Hence

$$\widehat{c}_i(X) \in \mathcal{C}_+.$$

Thus the induction continues.

**Step 5: Convert Pointwise Bound to $d_p$ Metric.** For every $i \in [m]$, Step 4 gives

$$
\begin{aligned}
e_i(X) &\leq \eta S_i \\
&\leq \eta S_m \\
&= \epsilon_{\mathrm{pt}}.
\end{aligned}
$$

Hence, for every $i \in [m]$,

$$
\begin{aligned}
\mathrm{d}_p(\widehat{c}_i, c_i) &= \left( \int_{\mathcal{C}} \|\widehat{c}_i(X) - c_i(X)\|_p^p \, \mathrm{d}X \right)^{1/p} \\
&\leq \left( \int_{\mathcal{C}} (N^{1/p} \|\widehat{c}_i(X) - c_i(X)\|_\infty)^p \, \mathrm{d}X \right)^{1/p} && \left( \text{By } \|A\|_p \leq N^{1/p} \|A\|_\infty \right) \\
&\leq |\mathcal{C}|^{1/p} N^{1/p} \epsilon_{\mathrm{pt}} \\
&= \epsilon. && \left( \text{By the definition of } \epsilon_{\mathrm{pt}} \right)
\end{aligned}
$$

**Step 6: Check Fixedness.** The auxiliary box $\mathcal{C}_+$ depends only on $\mathcal{C}, p, d, n, m, \epsilon$. The one-step tolerance $\eta$ depends only on $L, m, \epsilon_{\mathrm{pt}}$. Therefore, the fixed emulator $T_{\theta^\star}$ depends only on

$$
\mathcal{C}, L, p, d, n, m, \epsilon.
$$

It does not depend on $f_1, \ldots, f_m$ or $X$. Only the prompt entries $\{P_{f_i}\}_{i \in [m]}$ depend on the subroutines. This completes the proof. $\qquad\square$

### E.11 Lipschitzness of Self-Attention

We state a helper lemma from Edelman et al. (2022). This lemma provides the Lipschitzness of a single-head self-attention.

**Lemma E.10** (Lemma A.14 of Edelman et al. (2022)). *For any single-head self-attention* $\mathrm{Attn}$ *specified by* $W_K, W_Q, W_V \in \mathbb{R}^{k \times d}$, *for all* $X, \widehat{X} \in \mathbb{R}^{d \times n}$ *such that* $\|X\|_{2,\infty}, \|\widehat{X}\|_{2,\infty} \leq 1$,

$$\|\mathrm{Attn}(X) - \mathrm{Attn}(\widehat{X})\|_{2,\infty} \leq \|W_V\|_2 (1 + 4\|W_K^\top W_Q\|_2) \|X - \widehat{X}\|_{2,\infty}.$$

In the following, we utilize the above lemma to derive the Lipschitzness of multi-head self-attention with an inverse temperature $\beta$.

**Lemma E.11** (Lipschitzness of One-Layer Self-Attention). *Let* $\mathrm{Attn}$ *be an* $H$-head self-attention. *Let* $\mathcal{C} \in \mathbb{R}^{d \times n}$ *be a compact domain, and let* $X, \widehat{X} \in \mathcal{C}$ *denote two input sequences. We assume* $\|X\|_\infty, \|\widehat{X}\|_\infty \leq B_X$. *Then, we have*

$$\|\mathrm{Attn}(X) - \mathrm{Attn}(\widehat{X})\|_\infty \leq L\|X - \widehat{X}\|_\infty,$$

*where*

$$L = \sum_{h=1}^{H} \sqrt{d} \cdot \|W_O^h W_V^h\|_2 (1 + 4\beta d B_X^2 \|(W_K^h)^\top W_Q^h\|_2).$$

*Proof.* We start with the norm we want to bound

$$\|\mathrm{Attn}(X) - \mathrm{Attn}(\widehat{X})\|_\infty$$

$$= \|\sum_{h=1}^{H} W_O^h \cdot W_V^h X \cdot \sigma_\beta((W_K^h X)^\top W_Q^h X) - \sum_{h=1}^{H} W_O^h \cdot W_V^h \widehat{X} \cdot \sigma_\beta((W_K^h \widehat{X})^\top W_Q^h \widehat{X})\|_\infty \quad \text{(By Definition 2.1)}$$

$$\leq \sum_{h=1}^{H} \|W_O^h \cdot W_V^h X \cdot \sigma_\beta((W_K^h X)^\top W_Q^h X) - W_O^h \cdot W_V^h \widehat{X} \cdot \sigma_\beta((W_K^h \widehat{X})^\top W_Q^h \widehat{X})\|_\infty \quad \text{(By triangle inequality)}$$

$$\leq \sum_{h=1}^{H} \|W_O^h \cdot W_V^h X \cdot \sigma_\beta((W_K^h X)^\top W_Q^h X) - W_O^h \cdot W_V^h \widehat{X} \cdot \sigma_\beta((W_K^h \widehat{X})^\top W_Q^h \widehat{X})\|_{2,\infty}, \quad (129)$$

and (129) holds since $\|\cdot\|_\infty \leq \|\cdot\|_{2,\infty}$.

Then, our goal is to bound each term in (129) with Lemma E.10. For that purpose, we define $R := \sqrt{d} B_X$, and

$$X' := \frac{1}{R} X, \quad \widehat{X}' := \frac{1}{R} \widehat{X}, \quad (W_K^h)' := \beta R W_K^h, \quad (W_Q^h)' := R W_Q^h, \quad (W_V^h)' := R W_V^h,$$

such that $\|X'\|_{2,\infty}, \|\widehat{X}'\|_{2,\infty} \leq 1$. Also, (129) becomes

$$\sum_{h=1}^{H} \|W_O^h \cdot W_V^h X \sigma_\beta((W_K^h X)^\top W_Q^h X) - W_O^h \cdot W_V^h \widehat{X} \sigma((W_K^h \widehat{X})^\top W_Q^h \widehat{X})\|_{2,\infty}$$

$$= \sum_{h=1}^{H} \|W_O^h \cdot (W_V^h)' X' \sigma(((W_K^h)' X')^\top (W_Q^h)' X') - W_O^h \cdot (W_V^h)' \widehat{X}' \sigma(((W_K^h)' \widehat{X}')^\top (W_Q^h)' \widehat{X}')\|_{2,\infty}$$

$$\leq \sum_{h=1}^{H} \|W_O^h (W_V^h)'\|_2 (1 + 4\|((W_K^h)')^\top (W_Q^h)'\|_2) \|X' - \widehat{X}'\|_{2,\infty} \quad \text{(By Lemma E.10)}$$

$$= \sum_{h=1}^{H} \|W_O^h W_V^h\|_2 (1 + 4\beta R^2 \|(W_K^h)^\top W_Q^h\|_2) \|X - \widehat{X}\|_{2,\infty}$$

$$\leq \sum_{h=1}^{H} \sqrt{d} \|W_O^h W_V^h\|_2 (1 + 4\beta d B_X^2 \|(W_K^h)^\top W_Q^h\|_2) \|X - \widehat{X}\|_\infty. \quad \text{(By } \|X - \widehat{X}\|_{2,\infty} \leq \sqrt{d} \cdot \|X - \widehat{X}\|_\infty)$$

This completes the proof. $\square$

**Lemma E.12** (Bounded Output Range of Self-Attention). *Let* $\mathrm{Attn}$ *be an $H$-head self-attention. Let $\mathcal{C} \in \mathbb{R}^{d \times n}$ be a compact domain, and let $X \in \mathcal{C}$ denote the input sequence. We assume $\|X\|_\infty \leq B_X$. Then, we have*

$$\|\mathrm{Attn}(X)\|_\infty \leq ndB_X \sum_{h=1}^{H} \|W_O^h W_V^h\|_\infty.$$

*Proof.* For the convenience of our presentation, we write the $h$-th head of $\mathrm{Attn}$ as

$$\mathrm{Attn}^{(h)}(X) = W_O^h \cdot W_V^h X \cdot S^h(X), \quad S^h(X) := \sigma_\beta((W_K^h X)^\top W_Q^h X).$$

Then, we start with the norm we want to bound

$$\|\mathrm{Attn}(X)\|_\infty = \|\sum_{h=1}^{H} \mathrm{Attn}^{(h)}(X)\|_\infty$$

$$\leq \sum_{h=1}^{H} \|\mathrm{Attn}^{(h)}(X)\|_\infty. \qquad \text{(By traingle inequality)}$$

For each head $h \in [H]$, we have

$$\|\mathrm{Attn}^{(h)}(X)\|_\infty = \|W_O^h W_V^h X \cdot S^h(X)\|_\infty$$

$$\leq n\|W_O^h W_V^h X\|_\infty \|S^h(X)\|_\infty. \qquad (130)$$

Since every entry of a softmax output lies in $[0, 1]$, it follows that

$$\|S^h(X)\|_\infty \leq 1.$$

Also, we have

$$\|W_O^h W_V^h X\|_\infty \leq d\|W_O^h W_V^h\|_\infty \|X\|_\infty$$

$$\leq d\|W_O^h W_V^h\|_\infty \cdot B_X. \qquad \text{(By } \|X\|_\infty \leq B_X)$$

Therefore,

$$\|\mathrm{Attn}^{(h)}(X)\|_\infty \leq n\|W_O^h W_V^h X\|_\infty \|S^h(X)\|_\infty \qquad \text{(By (130))}$$

$$\leq n \cdot d\|W_O^h W_V^h\|_\infty B_X \cdot 1$$

$$= ndB_X \|W_O^h W_V^h\|_\infty.$$

Summing over all heads yields

$$\|\mathrm{Attn}(X)\|_\infty \leq \sum_{h=1}^{H} \|\mathrm{Attn}^{(h)}(X)\|_\infty$$

$$\leq \sum_{h=1}^{H} ndB_X \|W_O^h W_V^h\|_\infty.$$

This completes the proof. $\qquad \square$

**Lemma E.13** (Lipschitzness of $\ell$-Layer Self-Attention). *Let* $\mathrm{Attn}_\ell$ *be an $\ell$-layer self-attention. Let $\mathcal{C} \in \mathbb{R}^{d \times n}$ be a compact domain, and let $X, \widehat{X} \in \mathcal{C}$ denote two input sequences. We assume $\|X\|_\infty, \|\widehat{X}\|_\infty \leq B_X$. Then, we have*

$$\|\mathrm{Attn}_\ell(X) - \mathrm{Attn}_\ell(\widehat{X})\|_\infty \leq (\prod_{r=1}^{\ell} L_r)\|X - \widehat{X}\|_\infty,$$

*where*

$$L_r := \sum_{h=1}^{H_r} \sqrt{d} \|W_{r,O}^h W_{r,V}^h\|_2 (1 + 4\beta d B_{r-1}^2 \|(W_{r,K}^h)^\top W_{r,Q}^h\|_2).$$

*and*

$$B_{r-1} := (\prod_{i=1}^{r-1} c_i) B_X, \quad c_i := nd \sum_{h=1}^{H_i} \|W_{i,O}^h W_{i,V}^h\|_\infty.$$

*Proof.* We set up our notation first. For the convenience of presentation, we denote the $r$-th layer of attention as

$$\mathrm{Attn}_{\ell\text{-}r}(X) = \sum_{h=1}^{H_r} \mathrm{Attn}_{\ell\text{-}r}^{(h)}(X),$$

and

$$\mathrm{Attn}_{\ell\text{-}r}^{(h)}(X) = W_{r,O}^h \cdot W_{r,V}^h X \cdot S_r^h(X), \quad S_r^h(X) := \sigma_\beta(X^\top M_r^h X), \quad M_r^h := (W_{r,K}^h)^\top W_{r,Q}^h.$$

Then, our proof consists of three major steps. We prove the bounded output range of each layer, the Lipschitz constant of each layer, and the final $\ell$-layer Lipschitz constant.

**Step 1. Bounded Output Range of Each Layer.** By Lemma E.12, we know that the first layer has a bound on its output

$$\|\mathrm{Attn}_{\ell\text{-}1}(X)\|_\infty \le nd B_X \sum_{h=1}^{H_1} \|W_{1,O}^h W_{1,V}^h\|_\infty.$$

This makes sure the input to the second layer satisfies the assumption of Lemma E.12 again. Then, by Lemma E.12, we have

$$\|\mathrm{Attn}_{\ell\text{-}2} \circ \mathrm{Attn}_{\ell\text{-}1}(X)\|_\infty \le nd \cdot (nd B_X \sum_{h=1}^{H_1} \|W_{1,O}^h W_{1,V}^h\|_\infty) \cdot (\sum_{h=1}^{H_2} \|W_{2,O}^h W_{2,V}^h\|_\infty)$$

$$= (nd \sum_{h=1}^{H_2} \|W_{2,O}^h W_{2,V}^h\|_\infty) \cdot (nd \sum_{h=1}^{H_1} \|W_{1,O}^h W_{1,V}^h\|_\infty) \cdot B_X.$$

To keep our proof clean, for $r \in [\ell]$, we define

$$c_r := nd \sum_{h=1}^{H_r} \|W_{r,O}^h W_{r,V}^h\|_\infty, \quad B_r := (\prod_{i=1}^r c_i) \cdot B_X$$

Then, by induction, for $r \in [\ell]$, we have

$$\|\mathrm{Attn}_{\ell\text{-}r} \circ \mathrm{Attn}_{\ell\text{-}(r-1)} \circ \cdots \circ \mathrm{Attn}_{\ell\text{-}1}(X)\|_\infty \le B_r. \tag{131}$$

This proves the bounded output range of each layer.

**Step 2. Lipschitz Constant of Each Layer.** Next, we derive the Lipschitz constant of each layer on the set it actually sees.

For $r \in [\ell]$, let $U_r, \widehat{U}_r$ denote two input sequences to the $r$-th layer. Then, if

$$\|U_r\|_\infty, \|\widehat{U}_r\|_\infty \le B_{r-1}, \quad (B_0 := B_X),$$

by Lemma E.11, we have

$$\|\mathrm{Attn}_{\ell\text{-}r}(U_r) - \mathrm{Attn}_{\ell\text{-}r}(\widehat{U}_r)\|_\infty \le L_r \|U_r - \widehat{U}_r\|_\infty, \tag{132}$$

where

$$L_r := \sum_{h=1}^{H_r} \sqrt{d} \|W_{r,O}^h W_{r,V}^h\|_2 (1 + 4\beta d B_{r-1}^2 \|(W_{r,K}^h)^\top W_{r,Q}^h\|_2).$$

This gives the Lipschitz constant of each layer on the bounded set it actually receives.

**Step 3. $\ell$-Layer Lipschitz Constant.** Finally, we derive the Lipschitz constant for the $\ell$-layer attention.

We start with peeling the $\ell$-layer difference from the top

$$\|\text{Attn}_\ell(X) - \text{Attn}_\ell(\widehat{X})\|_\infty$$
$$= \|\text{Attn}_{\ell\text{-}\ell} \circ \text{Attn}_{\ell\text{-}(\ell-1)} \circ \cdots \circ \text{Attn}_{\ell\text{-}1}(X) - \text{Attn}_{\ell\text{-}\ell} \circ \text{Attn}_{\ell\text{-}(\ell-1)} \circ \cdots \circ \text{Attn}_{\ell\text{-}1}(\widehat{X})\|_\infty$$
$$\leq L_\ell \|\text{Attn}_{\ell\text{-}(\ell-1)} \circ \cdots \circ \text{Attn}_{\ell\text{-}1}(X) - \text{Attn}_{\ell\text{-}(\ell-1)} \circ \cdots \circ \text{Attn}_{\ell\text{-}1}(\widehat{X})\|_\infty. \qquad \text{(By (132))}$$

Now peel the next layer

$$\|\text{Attn}_{\ell\text{-}(\ell-1)} \circ \cdots \circ \text{Attn}_{\ell\text{-}1}(X) - \text{Attn}_{\ell\text{-}(\ell-1)} \circ \cdots \circ \text{Attn}_{\ell\text{-}1}(\widehat{X})\|_\infty$$
$$\leq L_{\ell-1} \|\text{Attn}_{\ell\text{-}(\ell-2)} \circ \cdots \circ \text{Attn}_{\ell\text{-}1}(X) - \text{Attn}_{\ell\text{-}(\ell-2)} \circ \cdots \circ \text{Attn}_{\ell\text{-}1}(\widehat{X})\|_\infty. \qquad \text{(By (132))}$$

Substituting back gives

$$\|\text{Attn}_\ell(X) - \text{Attn}_\ell(\widehat{X})\|_\infty \leq L_\ell L_{\ell-1} \|\text{Attn}_{\ell\text{-}(\ell-2)} \circ \cdots \circ \text{Attn}_{\ell\text{-}1}(X) - \text{Attn}_{\ell\text{-}(\ell-2)} \circ \cdots \circ \text{Attn}_{\ell\text{-}1}(\widehat{X})\|_\infty.$$

Continuing this peeling all the way down to the input yields

$$\|\text{Attn}_\ell(X) - \text{Attn}_\ell(\widehat{X})\|_\infty \leq \left(\prod_{r=1}^{\ell} L_r\right)\|X - \widehat{X}\|_\infty.$$

This completes the proof. $\qquad\qquad\square$

# F  Extension to Chain-of-Thought Setting

This section discusses a possible extension from re-prompting to CoT-style execution. The only new step is to internalize the user action in Theorem 4.2. Instead of externally supplying $P_{f_t}$ at round $t$, a selector head retrieves $P_{f_t}$ from stored subroutine prompts. After this selection step, the executor from Theorem 4.2 runs unchanged.

Let $P_f \in \mathbb{R}^{p \times n}$ denote the prompt format used by the executor. For each subroutine $f_i$, write

$$P_{f_i} = \begin{bmatrix} p_{i,1} & \cdots & p_{i,n} \end{bmatrix}, \quad p_{i,j} \in \mathbb{R}^p.$$

Let $s_i \in \mathbb{R}^m$ be the $i$-th stage tag, and let $c_j \in \mathbb{R}^n$ be the $j$-th column tag. The CoT input increases the hidden dimension to store these tags and increases the context length to store all subprompt columns.

At step $t$, the full input has the form

$$U_t = \begin{bmatrix} V_t & M & r & Z_t \end{bmatrix}.$$

Here $V_t$ is the active block, $M$ stores all subprompt columns, $r$ is a dummy token, and $Z_t$ is the scratchpad block.

The active block is

$$V_t = \begin{bmatrix} v_{t,1} & \cdots & v_{t,n} \end{bmatrix},$$

where

$$v_{t,j} = \begin{bmatrix} x_{t,j} \\ a_{t,j} \\ 0_p \\ q_t \\ c_j \\ 0_m \\ 0_n \\ 1 \\ 0 \\ 0 \end{bmatrix}, \quad q_t = s_t, \quad j \in [n].$$

Here $x_{t,j}$ is the $j$-th column of the current state $X_t$. The vector $a_{t,j} \in \mathbb{R}^p$ is the $j$-th column of the active prompt slot

$$A_t = \begin{bmatrix} a_{t,1} & \cdots & a_{t,n} \end{bmatrix} \in \mathbb{R}^{p \times n}.$$

The selector will fill $A_t$ with $P_{f_t}$. The stage tag $q_t = s_t$ records the current subroutine index. The column tag $c_j$ records the active column index.

The memory block stores all subprompt columns:

$$M = \begin{bmatrix} u_{1,1} & \cdots & u_{m,n} \end{bmatrix},$$

where

$$u_{i,k} = \begin{bmatrix} 0_d \\ 0_p \\ p_{i,k} \\ 0_m \\ 0_n \\ s_i \\ c_k \\ 0 \\ 1 \\ 0 \end{bmatrix}, \quad i \in [m], \quad k \in [n].$$

The tag $s_i$ identifies the stored subroutine. The tag $c_k$ identifies the stored column.

The dummy token is

$$r = \begin{bmatrix} 0_d \\ 0_p \\ 0_p \\ 0_m \\ 0_n \\ 0_m \\ 0_n \\ 0 \\ 0 \\ 1 \end{bmatrix}.$$

It gives non-active tokens a zero-value target for the selector head.

We now define the selector head. The selector writes only into the active prompt-slot rows. All unlisted projections are zero. Choose fixed projections such that

$$W_Q^{\text{sel}} v_{t,j} = \begin{bmatrix} q_t \\ c_j \\ 0 \end{bmatrix}, \quad W_K^{\text{sel}} u_{i,k} = \begin{bmatrix} s_i \\ c_k \\ 0 \end{bmatrix}, \quad W_V^{\text{sel}} u_{i,k} = p_{i,k}.$$

For the dummy token, choose

$$W_K^{\text{sel}} r = \begin{bmatrix} 0_m \\ 0_n \\ 1 \end{bmatrix}, \quad W_V^{\text{sel}} r = 0.$$

For memory tokens, choose

$$W_Q^{\text{sel}} u_{i,k} = \begin{bmatrix} 0_m \\ 0_n \\ C \end{bmatrix},$$

where $C > 2$ is fixed. Thus memory tokens attend to the dummy token and receive zero from this selector head.

For an active token $v_{t,j}$, the score to memory token $u_{i,k}$ is

$$\left(W_Q^{\mathrm{sel}} v_{t,j}\right)^\top \left(W_K^{\mathrm{sel}} u_{i,k}\right) = \langle q_t, s_i \rangle + \langle c_j, c_k \rangle.$$

Since $q_t = s_t$, this score is uniquely maximized at $(i, k) = (t, j)$. Hence, for large enough softmax inverse temperature $\beta$, the selector retrieves the value $p_{t,j}$ and writes it into the active prompt slot $a'_{t,j} \approx p_{t,j}$. Running this for all $j \in [n]$ gives

$$A'_t \approx P_{f_t}.$$

After selection, the active rows contain

$$\begin{bmatrix} X_t \\ A'_t \end{bmatrix} \approx \begin{bmatrix} X_t \\ P_{f_t} \end{bmatrix}.$$

Thus the executor from Theorem 4.2 runs unchanged on the active block. It reads only the $X_t$ rows and the active prompt-slot rows. All other rows and all non-active tokens form the flow-through block. Therefore,

$$X_{t+1} \approx f_t(X_t).$$

It remains to update the stage tag. Let $S_m \in \mathbb{R}^{m \times m}$ be the fixed shift matrix satisfying

$$S_m s_t = s_{t+1}.$$

A second fixed head writes only into the active stage-tag rows. Choose projections so that active tokens attend to themselves through their column tags:

$$W_Q^{\mathrm{ctr}} v_{t,j} = c_j, \quad W_K^{\mathrm{ctr}} v_{t,k} = c_k, \quad W_V^{\mathrm{ctr}} v_{t,k} = S_m q_t.$$

The score is maximized when $k = j$. Thus this head writes $S_m q_t$ into the stage-tag rows of each active token. For memory tokens, use the same dummy-token trick as above, so this head writes zero to them. Hence the next active block contains

$$q_{t+1} = S_m q_t = s_{t+1}.$$

The selector weights stay fixed. Only the stage tag changes, so the same selector retrieves $P_{f_{t+1}}$ at the next step.

Overall, one step maps

$$U_t = \begin{bmatrix} V_t & M & r & Z_t \end{bmatrix} \longmapsto U_{t+1} = \begin{bmatrix} V_{t+1} & M & r & Z_{t+1} \end{bmatrix}.$$

The next active block contains $X_{t+1}$, the updated stage tag $q_{t+1}$, and an active prompt slot for the next round.

This gives a block-by-block CoT-style execution. Each CoT unit is an $n$-token block, so the executor remains a sequence-to-sequence map on $\mathbb{R}^{d \times n}$.

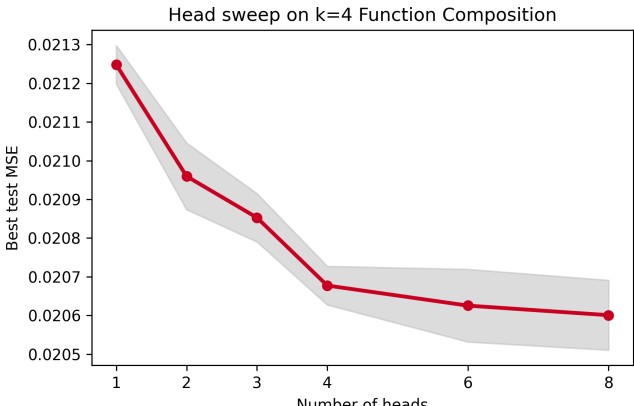

*Figure 5.* **Head Sweep on Prompted Function Composition.** Each function is a tokenwise 2-layer ReLU MLP $f_i : \mathbb{R}^{16 \times 16} \to \mathbb{R}^{16 \times 16}$ with hidden width 8. We sample a fixed library of 50 functions, and encode each function's true parameters as a prompt block stacked below $X$. We train a shared executor on 4-function compositions. The full order space has $50^4 = 6{,}250{,}000$ compositions. We split it into disjoint train/test pools with 4,375,000 and 1,875,000 possible orders, and sample 20,000 training and 5,000 test examples. The executor uses depth 2 with hidden dimension 96. For training, we set dropout 0.1, AdamW with learning rate $5 \times 10^{-4}$ and weight decay $10^{-3}$, batch size 128, and train for 30 epochs. We sweep the number of heads over $\{1, 2, 3, 4, 6, 8\}$ and report the best test MSE over 5 random seeds. The shaded region shows one standard deviation. We observe trained model achieve low mse error, and the error decrease as number head increase.

## G  Experimental Studies

We provide a proof-of-concept experiment below. The goal is to verify: (i) Transformer reads the weight encoding to approximate the target function; (ii) When multiple prompts provided, Transformer performs function composition; and (iii) The error dependency on head complexity.

**Model.** The executor is a shared Transformer that maps the prompted input $X_{\text{aug}}$ to the predicted output. In the head-sweep experiment, we use a 2-layer Transformer, with hidden dimension=96, and FFN hidden size 192. We sweep the number of heads over $\{1, 2, 3, 4, 6, 8\}$.

**Data.** Each primitive function is a tokenwise 2-layer ReLU MLP $f_i : \mathbb{R}^{16 \times 16} \to \mathbb{R}^{16 \times 16}$ with hidden width 8. We sample a fixed library of 50 such functions. For each primitive $f_i$, we serialize its true parameters into a prompt block and stack this block below the input $X$ in the feature dimension. The target task is 4-function composition, $f_{i_4} \circ f_{i_3} \circ f_{i_2} \circ f_{i_1}(X)$. With repetition allowed, the full composition space contains $50^4 = 6{,}250{,}000$ possible orders. We split this space into disjoint train and test pools with 4,375,000 and 1,875,000 possible orders, respectively, and then sample 20,000 training examples and 5,000 test examples from these pools. Thus, evaluation is performed on held-out composition orders.

**Training and Evaluation.** We train the Transformer executor with AdamW, learning rate $5 \times 10^{-4}$, weight decay $10^{-3}$, dropout 0.1, batch size 128, and 30 epochs, and report the best test MSE averaged over 5 random seeds $\{0, 1, 2, 3, 4\}$. We use mean squared error (MSE) as the training loss and evaluation metric:

$$\text{MSE} = \frac{1}{Bdn} \sum_{b=1}^{B} \sum_{i=1}^{d} \sum_{t=1}^{n} (\widehat{Y}_{b,i,t} - Y_{b,i,t})^2,$$

where $d = n = 16$ in our setting.

**Results.** Figure 5 shows that the executor achieves low test MSE on held-out composition orders. This verifies (i) that the Transformer reads the prompt-encoded weights to approximate the target function, and (ii) that when multiple prompt blocks are provided, it performs function composition rather than memorizing a small set of seen orders. Moreover, test MSE decreases as the number of heads increases and then saturates. This verifies (iii) that approximation error improves with model complexity.

