# OpenReview forum: "In-Context Universal Approximation, Compositional Generalization, and Algorithm Emulation"
_ICML.cc/2026/Conference — ICML 2026 regular_

### Official Review · Reviewer_DbaQ · 2026-02-18

**Soundness:** 3
**Presentation:** 2
**Significance:** 3
**Originality:** 3
**Overall Recommendation:** 5
**Confidence:** 2

**Summary:**

The paper gives a construction showing that a single fixed transformer can emulate certain broad classes of smooth functions, somewhat analogous to the result that there exists a single fixed universal Turing machine which can emulate any other Turing machine. Their construction is relevant to practice because it roughly mimics the way in which LLMs are "programmed" through prompts, and so provides an existence proof that such programming is universal (although ofc real LLMs use learned weights and not their constructed "universal" weights).

They then extend this to function composition by showing that multiple functions can be chained together by "looping" their universal construction back onto itself, similar to (old) equilibrium networks or (new) tiny recursion nets, among others.

The paper is theoretical and presents no experiments.

**Compliance With Llm Reviewing Policy:**

Affirmed.

**Final Justification:**

I remain positive about the paper and appreciate their revision to include extra illustrations and a working demonstration experiment

**Key Questions For Authors:**

How would you align your work better with the manner in which LLMs are used in practice to emulate algorithms and chain together functions, i.e. autoregressively and using chain of thought?

**Limitations:**

yes

**Strengths And Weaknesses:**

Strengths:
- Rigorous theoretical work conceptually relevant to practice
- Super job laying out the relation to past theory in this space. As someone not in this area, the related work positioning was very helpful!
- Just cool and clever. Fun idea to encode the literal weights of another network into the prompt; as someone not in the area, I can't comment on the originality of that, though

Weaknesses:
- I just don't think this very analogous to how function composition / algorithm emulation in LLMs works. In practice, LLMs are autoregressive (not "looped"), and use chain-of-thought tokens as a scratchpad, not intermediate activations. Still, this work could be a stepping stone toward theory which more closely aligns with practice along those dimensions
- No experiments!
- It would really help to have a Fig 1 explaining your constructions. I didn't get that it was "looped" until much later, or that parameters of another net were being put into the prompt until well into the paper.

---

> ### Author Rebuttal · Authors · 2026-03-27
>
> > ### W1. I just don't think ... LLMs work. In practice, LLMs are autoregressive (not "looped"), and use chain-of-thought tokens as a scratchpad... Still, this work could be a stepping stone toward theory which more closely aligns with practice along those dimensions
>
> Thank you for your kind words and for bringing up the gap!  To clarify, **in practice, complex tasks like function composition do not always require multiple-step inference/reasoning like CoT. While our results use one-forward pass inference of the emulator, they are in fact very easy to generalize to CoT settings**, please see the response for Q1 for the details.
>
> We also emphasize that a single forward pass is still a standard transformer. It is also a common setting in Transformer theory [1-4].
>
> [1] "Learning compositional functions with transformers from easy-to-hard data. COLT (2025)
>
> [2] "Transformers as algorithms: Generalization and stability in in-context learning." NeurIPS (2023)
>
> [3] "Transformers as statisticians: Provable in-context learning with in-context algorithm selection." NeurIPS (2023)
>
> [4] "Transformers, parallel computation, and logarithmic depth." ICML (2024)
>
> > ### W2. No experiments!
>
> Thanks for pointing this out. We provide a proof-of-concept experiment below. The goal is to verify:
> 1. transformer reads the weight encoding to approximate the target function;
> 2. when multiple prompts provided, transformer performs function composition
> 3. the error dependency on head complexity.
>
> In the experiment, we first sample 50 functions and encode each function’s parameters as prompt block below $X$. We train a executor on prompted compositions and evaluate it on held-out composition orders. Concretely:
> - Target function: 2-layer ReLU MLP $f_i : \mathbb{R}^{16 \times 16} \to \mathbb{R}^{16 \times 16}$ with hidden width $8$.
> - 4-function composition.
> - Split it into train and test pools, with $\sim4000$k possible train orders and $\sim1800$k possible held-out test orders
> - Sample a fixed number of training (20000) and test (5000) examples from these pools.
>
> Results: https://imgur.com/a/Lgsift5. The model achieves low mse error, and thus verifies (1-3) and aligns well with our theory.
>
>
> > ### W3. It would really help to have a Fig 1 explaining your constructions. I didn't get that it was "looped" until much later, or that parameters of another net were being put into the prompt ...
>
> We apologize for the confusion here. We include new figures to clarify our construction. **For Sec 3 and weight encoding, see https://imgur.com/a/Sx2S93b. For Thm 4.2 and looped behaviour, see https://imgur.com/a/2uIgVwD.**
>
> We also want to clarify 2 points:
>
> - **Thm 4.2 is the only result that is looped. All other results don't rely on looping operation.**
>
> - For the “parameters of another net”, we believe there might be some oversight. Such a relation is stated in the contribution of the intro section. Section 3 also mentions this in the first paragraph and lemma/theorem statement. However, we agree current figure 1 doesn’t plot $P_f$ as weight encoding explicitly. We will update the figure to make the connection clear in the final revision.
>
>
> > ### Q1. How would you align your work better with the manner in which LLMs are used in practice to emulate algorithms and chain together functions, i.e. autoregressively and using chain of thought?
>
> Thanks for the question. To clarify, **in practice, complex tasks like  function composition do not always require multiple-step inference/reasoning like CoT.** Also, **while our results use one-forward pass inference of the emulator, they are in fact very easy to generalize to CoT settings.** It only complicate our proofs. See sketch below.
>
> Section 4 already includes 2 CoT ingredients:
> - a fixed scratchpad state via the flow-through component (lemma 4.1).
> - iterative reuse of the same executor construction (proof of Thm 4.1/4.2).
>
> With token-by-token generation, this corresponds to the CoT mechanism. The only missing part is to select the correct subprompt at each step. We achieve by:
>
> - Let $q_t = e_t$ be a **stage token** that encodes the current step $t$.
> - Attach to each stored subprompt $P_{f_i}$ a tag $h_i = e_i$ to indicate which subprompt is this
> - Use one fixed attention head with $q_t$ as query, the tags $h_i$ as keys, and the $P_{f_i}$ as values.
> - Because $\langle q_t, h_i \rangle = 1$ only when $i=t$, the head selects $P_{f_t}$ for sufficiently large $\beta$.
> - Run the existing six-layer emulator on the selected subprompt and the current scratchpad.
>
> With this component one can generalize section 4 to CoT setting. We will include detailed proof in the final revision.
>
> Finally, we emphasize that **our current setup aligns well with LLM practice.** Thm 4.1 corresponds to prompt all the subtasks and step at once when the task is simple. Thm 4.2 corresponds to back-and-forward discussion with LLM. For clarity, we further provide figures to demonstrate our idea: https://imgur.com/a/xbwG6K5.

---

> > ### Author Rebuttal · Reviewer_DbaQ · 2026-04-03
> >
> > Thank you for adding the experiment and illustration. I'll raise my score accordingly.

---

> > > ### Author Response · Authors · 2026-04-03
> > >
> > > We are glad that our clarifications meet your expectations! Thank you again for your time and feedback to help improve the clarity of our work.
> > >
> > > We will include the new figures, experiment and how to generalize to CoT setting into our final version.

---

### Official Review · Reviewer_HuxQ · 2026-03-09

**Soundness:** 3
**Presentation:** 2
**Significance:** 2
**Originality:** 2
**Overall Recommendation:** 4
**Confidence:** 4

**Summary:**

This paper studies the “prompt as program” view for standard softmax Transformers with frozen weights. The core claim is that a single fixed model can be steered, via a structured prompt that encodes a procedure, to approximate arbitrary continuous sequence to sequence functions on a compact domain. Building on that, the paper proves a compositional result where prompts corresponding to subroutines can be assembled so the same frozen model executes their composition.

Concretely, the paper presents two in-context universality routes. One route shows how a fixed attention stack can emulate a Transformer block that includes feed forward networks and attention, when the prompt contains a weight encoding of those components. The other route shows universality with an attention only construction based on an interpolation or selector style mechanism. On top of the universality result, the paper proves two composition modes. One mode performs a length m composition in a single forward pass with depth scaling linearly in m. The other mode uses repeated prompting across turns, keeping model depth fixed while executing arbitrarily long compositions iteratively.

**Compliance With Llm Reviewing Policy:**

Affirmed.

**Final Justification:**

The rebuttal addressed most of the main concerns that I raised.

**Key Questions For Authors:**

1. The route that goes from universal approximation by Transformers to in context universality via weight encoding feels like a natural combination of known ingredients. Can you clarify what is genuinely new beyond the composition result. A short “what would fail without our new lemmas” paragraph would help?

2. The prompt is effectively an analog encoding of weights. How sensitive is the construction to perturbations in the encoded values? Is there any margin argument that would tolerate quantization at realistic precision?

3. Do you see a path to extend the framework to common discrete tasks, or to piecewise continuous mappings that arise from algorithmic decisions? Even a discussion of what breaks would be useful.

**Limitations:**

The composition theorems and the constructive prompt mechanism are fair conceptual contributions in the theory of in context computation for softmax Transformers. The crucial downside is that the results remain far from an empirically grounded explanation of real LLM prompting, and the resource scaling is likely enormous.

**Strengths And Weaknesses:**

**Strengths**

1. The paper is explicit about what it wants from an in context theory, namely standard softmax attention, one frozen model across tasks, and a constructive prompt mechanism. That framing is helpful and makes the contribution easier to place.
2. The weight encoding prompt mechanism and the supporting emulation lemmas are a meaningful step beyond statements of representational capacity. The paper tries to spell out how prompts can function like “machine code” for a fixed interpreter.
3. The two composition regimes match real usage patterns. The single pass result captures one shot chaining via deeper computation. The multi turn result captures step by step prompting workflows. Having a formal statement that connects modular prompt assembly to function composition is valuable.

**Weaknesses**

1. The constructions are highly idealized. They treat prompts as real valued encodings of weights and wiring. That is a reasonable move for theory, but the paper should be careful about how much this explains real LLM behavior under discrete tokenization, limited precision, and finite context.
2. The results are qualitative and existential in the sense of resource blowup. The needed context length, embedding dimension, head counts, and softmax sharpness likely scale very poorly with dimension and target error. Without explicit scaling summaries, it is hard to judge what is “explained” versus what is “possible in principle.”
3. Composition across m steps is where small details matter. The theorems state controlled error, but the reader would benefit from an explicit dependence on m and on continuity or Lipschitz constants of the composed functions. Right now it is easy to miss which parts depend on m and how the per step approximation budget is allocated.
4. Much of the work is in the appendix and in reductions to prior universality constructions.

---

> ### Author Rebuttal · Authors · 2026-03-28
>
> We thank the reviewer for the detailed review. In response, we have addressed all concerns and questions in the following replies.
>
> > ### W1: The constructions are highly idealized ...under discrete tokenization, limited precision, and finite context.
>
> Thanks for the comments and sorry for any confusion caused. We believe there might be some oversight.
>
> **Our setup is very generic. In which the Transformer inputs are all numerical vectors. This is the same in real LLMs.** This setting is general and includes the mentioned discrete tokens since tokens are numerical vectors in LLMs.
>
> Furthermore, prompt as real valued encodings is also common in real-world prompt/prefix tuning. Also, it’s the default setup for majority ICL theory. People encode the weight of the simulated model into the input sequence all the time (e.g., [Bai et al., NeurIPS 2023, "Transformer as statisticians"]).
>
> Third, our proofs do not require particular float-point precision. Therefore we believe there should be no concern about limited precision here.
>
> Lastly, our analysis considers finite context. So there should be no concern about finite context here.
>
> Thanks for pointing these out. We hope our clarifications above suffice!
>
> ---
>
> > ### W2. The results are qualitative and...The needed context length, ...
>
> Thanks for bringing this up. We discuss the model complexity of (1) target attention/FFN (2) attention emulator separately as follows.
>
> * **Target Attention/FFN**. **We emphasize that universality over arbitrary continuous targets carries a worst-case exponential price in model size. This is unavoidable and not specific to our construction** (see, e.g., Hu et al. 2025; Yun et al. 2019; Kajitsuka & Sato 2023; and classical FFN universality results). Improving these worst-case rates without additional regularity assumptions is widely viewed as an open problem, not a paper-specific flaw.
>
>     For the softmax sharpness, same logic applies. Continuous functions may have sharp transitions. This requires a large $\beta$. In contrast, if the target is smoother, small $\beta$ achieve same error. Our theoretical result aims for arbitrary continuous function, hence large $\beta$ is unavoidable for generality.
>
> * **Attention Emulator**. Here we provide a complexity table of our main results: https://imgur.com/a/xBUN7D6.
>
> ---
>
> > ### W3: Composition across m steps ...explicit dependence on $m$ and on continuity or Lipschitz constants of the composed functions...
>
> Sorry for the confusion caused. Due to the pages limit, we defer the error dependency of $m$ and Lipschitz constants into appendix D.9, `line 3287` and D.10, `line 3362`. We will link them to the main text in final revision.
>
> ---
>
> > ### W4. Much of the work ... in reductions to prior universality constructions.
> > ### Q1. The route ... A short “what would fail without our new lemmas” paragraph would help?
>
> Thanks for the suggestions. We’d like to clarify our contributions as follows.
>
> * **The genuinely new part is a constructive prompt-programmed theory, not just another universality proof.** Specifically, we show (1) frozen transformers realize many target functions by changing only the prompt, and (2) give an explicit rule that maps a function $f$ to a prompt $P_f$ (Remark 3.1, 3.2, D.1, D.2). Existing universal approximation and attention emulation do not give these results.
>
> * Achieving the above results is non-trivial. Prior work only emulates single-head, single-layer attention. Our results emulates more complex attention: (1) multi-head setting, (2) with residual connection, (3) flow-through component and (4) arbitrary layer. We also provide new theory on emulate FFN.
>
> * Without these lemmas, one can’t emulate transformers (multi-head attention + FFN) and deep models. We also can’t get in-context composition guarantees.
>
> ---
>
> > ### Q2: The prompt is ... How sensitive ...
>
> Thanks for the question. The sensitivity is controlled by the Lipschitzness $L$ of the emulator. If input prompt is perturbed by $\Delta P$, the output change is bounded by $L \cdot || \Delta P||_\infty$. Hence the input quantization error is bounded by this.
> We include explicit calculation of single and multi-layer attention Lipschitzness in the revision.
>
> ---
>
> > ### Q3: Do you see a path to extend the framework to common discrete tasks, ...
>
> Thanks for the question. [Hu et. al, 2025] show attention approximates ReLU NN. Since ReLU is a piecewise linear function, it is natural to fit piecewise continuous mappings. We will add this discussion in our concluding remark.
>
> Discrete tasks are harder because there are no existing UAP results on discrete function classes. More work will need to be done there. But we deem them as exciting future works.
>
> > ### Limitations.
>
> Please see above responses to W1 and W2.
>
> ---
>
> Thanks again for your time and effort! We have addressed all your concerns and questions with utmost care. Please let us know if there is anything else we can clarify better!

---

> > ### Author Rebuttal · Reviewer_HuxQ · 2026-04-03
> >
> > Thanks. My concerns have been addressed.

---

> > > ### Author Response · Authors · 2026-04-03
> > >
> > > Thank you again for the time and consideration you put into our work. We are glad our responses resolved your concerns!  We will incorporate the discussed changes into the final revision.

---

### Official Review · Reviewer_nYwJ · 2026-03-12

**Soundness:** 3
**Presentation:** 2
**Significance:** 4
**Originality:** 4
**Overall Recommendation:** 5
**Confidence:** 2

**Summary:**

This paper proves that a fixed-weight softmax transformer can approximate a broad class of sequence-to-sequence functions with proper prompts. In addition, it can also execute the composition of a sequence of such functions.

**Compliance With Llm Reviewing Policy:**

Affirmed.

**Final Justification:**

My final recommendation remains positive. My concerns were mostly about the writing and positioning. My confidence is limited because most of the proofs are in the appendix. The rebuttal answered some questions I had, but it did not increase my confidence or change my assessment.

**Key Questions For Authors:**

Does sequential function composition increase expressive power? In other words, is the function class here, namely any continuous function defined on a compact domain, closed under composition? If not, what is the resulting class?

**Limitations:**

yes

**Strengths And Weaknesses:**

The paper proves interesting and important results on prompts being able to induce a transformer to perform transformer-implementable functions, arbitrary functions, and even sequential subroutines.
My criticism is mostly about the writing.
1. It is a bit weird that the authors pitch this as a study of in-context learning, which typically involves in-context examples in the prompt, whereas this paper does not.
2. Most of the proofs are buried in the appendices, without proof sketches in the main text, which makes the paper hard to follow. I understand the space limitations, and I do not think this should be a major weakness, but it does lower my confidence, as I did not have the capacity to go through the appendices in detail. I would suggest moving some of the lemmas to the appendix and giving more high-level intuition for the constructions in the main text.
3. The distinction for functions, algorithms, and programs is not very intuitive to me. It would help if these were defined formally or illustrated with examples.
4. The introduction spans three pages, yet it still fails to situate the paper in the literature. As someone who is not familiar with the field, I had to do a bit of research myself to properly understand the contribution of this work compared to prior work.

---

> ### Author Rebuttal · Authors · 2026-03-26
>
> We thank the reviewer for the positive review and encouraging words. We have addressed all remaining concerns as follows with utmost care:
>
> > ### W1. It is a bit weird that the authors pitch this as a study of in-context learning, which typically involves in-context examples in the prompt, whereas this paper does not.
>
> Sorry for any confusion caused. But we believe there might be some overlook. We did not pitch this paper as merely ICL. What’s established here is beyond ICL, as the title suggested. Our paper studies a broader setting. **We focus on prompts that specify not only context data, but also the procedure to apply to that data.** In this sense, **a standard example-based ICL is one special case of our setting.** Please see `line118-135` for our problem definition.
>
> To be concrete, yes, in standard ICL, the model may see input-output pair examples and then predict on a new input. In our setting, the prompt can also tell the model which procedure to execute on the in-context data. Our theorem therefore goes beyond showing learning from examples. It shows that a fixed Transformer can execute arbitrary procedures given in context, and can even compose such procedures in context.
>
> We appreciate the opportunity to clarify and hope this clarification suffices.
>
> ---
>
> > ### W2. Most of the proofs are buried in the appendices, without proof sketches in the main text, which makes the paper hard to follow. I understand the space limitations, and I do not think this should be a major weakness, but it does lower my confidence, as I did not have the capacity to go through the appendices in detail. I would suggest moving some of the lemmas to the appendix and giving more high-level intuition for the constructions in the main text.
>
> Thank you for this helpful suggestion. We agree that the main text should include more intuition. In the revision, we will add proof sketches of our main results to provide intuition if space permits.
>
> ---
>
> > ### W3. The distinction for functions, algorithms, and programs is not very intuitive to me. It would help if these were defined formally or illustrated with examples.
>
> Thanks for bringing this up.
>
> Use linear regression as an example. Algorithm: GD of linear regression. function: either the closed-form solution (as a function of data), or estimator of GD weight update (please see Corollary 3.1.2 of [Hu et al., ICLR 2026 "In-Context Algorithm Emulation"] for example). Program: data + algorithm, which is prompt in our setup.
>
> We understand the submission clarity is not optimal. We have revise the wording in preliminary and intro accordingly. Thank you!
>
> ---
>
> > ### W4. The introduction spans three pages, yet it still fails to situate the paper in the literature. As someone who is not familiar with the field, I had to do a bit of research myself to properly understand the contribution of this work compared to prior work.
>
> In response, we'd like to clarify two points.
>
> - First, **our paper studies a broader setting than standard example-based ICL.** We call this setting *prompt-programmed execution* (see intro and fig 1.) In particular, the prompt specifies both the context data (i.e, program input $X$) and the procedure to apply (via program text $P_f$ for algorithm $f$). We show the transformer is capable of doing prompt-programmed execution. In our setting,  ICL is just one special case of our setup with $f$ being a learning algorithm.
>
> - Second, we agree that the clarity is not optimal in the submission. We will improve the presentation and make the connection to prior work clearer in the final version.
>
> We also remind the reviewer, the positioning against literature is integrated in the intro from `line L098-L117` of the submission. We now see this is not highlighted enough. We will strengthen this part in our final version. Thank you for pointing this out.
>
> ---
>
> > ### Q1: Does sequential function composition increase expressive power? In other words, is the function class here, namely any continuous function defined on a compact domain, closed under composition? If not, what is the resulting class?
>
>
> Thank you for the question. Under the standard function-class notion of expressiveness, the answer is NO to the 1st question. Our function class is the class of continuous self-maps on a compact domain $\mathcal{C}$. So YES to the second question, this class is closed under composition.
>
> Our intended contribution of composition results is different. It shows that a fixed Transformer can execute multiple prompt-specified functions sequentially, while passing intermediate outputs from one stage to the next. Thus, the novelty is not a larger function class, but a stronger in-context composition capability. We will clarify this distinction in the revision.
>
> ---
>
> Thanks again for your time and effort! We have taken all comments into careful consideration and have made corresponding revisions to address the concerns raised. We look forward to further feedback and discussion.

---

> > ### Author Rebuttal · Reviewer_nYwJ · 2026-04-03
> >
> > Thank you for the rebuttal. I will maintain my positive review.

---

> > > ### Author Response · Authors · 2026-04-04
> > >
> > > Thank you again for your time and detailed review of our paper. We are glad our response solves your concerns! We will incorporate your comments and clarified points in our final version.

---

### Official Review · Reviewer_AgtT · 2026-04-03

**Soundness:** 4
**Presentation:** 3
**Significance:** 4
**Originality:** 3
**Overall Recommendation:** 5
**Confidence:** 1

**Summary:**

The paper provides a theoretical framework for in-context universal approximation and compositional generalization of softmax Transformers, providing a viewpoint on how prompts serve as programs and fixed-weight Transformers as interpreters.

**Compliance With Llm Reviewing Policy:**

Affirmed.

**Final Justification:**

I believe this is a strong submission to my knowledge, with clear technical contributions, and I do not have any major concerns.

**Key Questions For Authors:**

1. Could you please provide a small empirical demonstration to better validate the theoretical results?
2. Could you please address the aforementioned typos to improve the clarity of the paper?

**Limitations:**

Yes

**Strengths And Weaknesses:**

1. Sound theoretical results and insightful connections to single completion and re-prompting.
2. Clear problem framing, definitions, and proofs, and comparison to prior work, which enhances the significance and originality.
3. Numerous minor typos/formatting (e.g., Turning complete) and a lack of empirical results, which would improve practical results.

---

### Decision · Program_Chairs · 2026-04-30

**Decision:**

Accept (regular)

**Comment:**

This work studies the in-context universal approximation and compositional generalization of softmax Transformers. The authors show two important properties of a fixed-weight softmax Transformer: It approximates a broad class of continuous sequence-to-sequence functions encoding in the prompts; and also executes their compositions.

All reviewers reached a positive consensus about this work, regarding the theoretical results on prompt composition that formally capture common LLM usage patterns. Although reviewers raised concerns about the verification of the theoretical results in realistic cases and the presentation clarity, the authors did a good job in addressing the concerns of reviewers. In addition, while one reviewer submitted the review after the rebuttal deadline, the authors also addressed the concerns satisfactorily.

Therefore, I believe this work could be a valuable contribution to the community by integrating all the discussions with reviewers into the revision. It'd could be further strengthened to discuss more with modern architectures of transformers.